# Inter-comparison and evaluation of Arctic sea ice type products

Yufang Ye[1], Yanbing Luo[1], Yan Sun[1], Mohammed Shokr[2], Signe Aaboe[3], Fanny Girard-Ardhuin[4], Fengming Hui[1], Xiao Cheng[1] and Zhuoqi Chen[1]

[1]School of Geospatial Engineering and Science, Sun Yat-sen University, and Southern Marine Science and Engineering Guangdong Laboratory (Zhuhai), Zhuhai 519082, China
[2]Meteorological Research Division, Environment and Climate Change Canada, Toronto M3H5T4, Canada
[3]Department of Remote Sensing and Data Management, Norwegian Meteorological Institute, Tromso, Norway
[4]Laboratoire d'Océanographie Physique et Spatiale (LOPS), Ifremer-Univ. Brest-CNRS-IRD, IUEM, F-29280, Plouzané, France

*Correspondence to*: Zhuoqi Chen (chenzhq67@mail.sysu.edu.cn)

**Abstract.** Arctic sea ice type (SITY) variation is a sensitive indicator of climate change. However, systematic inter-comparison and analysis for SITY products are lacking. This study analyzed eight daily SITY products from five retrieval approaches covering the winters of 1999–2019, including purely radiometer-based (C3S-SITY), scatterometer-based (KNMI-SITY and IFREMER-SITY) and combined ones (OSISAF-SITY and Zhang-SITY). These SITY products were inter-compared against a weekly sea ice age product (i.e. NSIDC-SIA) and evaluated with five Synthetic Aperture Radar images. The average Arctic multiyear ice (MYI) extent difference between the SITY products and NSIDC-SIA varies from $-1.32 \times 10^6 \ km^2$ to $0.49 \times 10^6 \ km^2$. Among all, KNMI-SITY and Zhang-SITY in the QSCAT period agree best with NSIDC-SIA and perform the best. In the ASCAT period, KNMI-SITY tends to overestimate MYI (especially in early winter), whereas Zhang-SITY and IFREMER-SITY tend to underestimate MYI. C3S-SITY performs well in some early winter cases however exhibits large temporal variabilities as OSISAF-SITY. Factors that could impact their performances are analyzed and summarized: (1) Ku-band scatterometer generally performs better than C-band scatterometer on SITY discrimination, while the latter sometimes identifies first-year ice (FYI) more accurately, especially when surface scattering dominants the backscatter signature. (2) Simple combination of scatterometer and radiometer data is not always beneficial without further rules of priority. (3) The representativeness of training data and efficiency of classification are crucial for SITY classification. Spatial and temporal variation of characteristic training dataset should be well accounted in the SITY method. (4) Post-processing corrections play important roles and should be considered with caution.

## 1 Introduction

Sea ice is an important component of the earth system. Sea ice influences climate change through two primary processes: the ice-albedo feedback and the insulating effect. Sea ice reflects more solar radiation than the ocean due to its high albedo. In addition, sea ice hinders the heat exchange between the ocean and the atmosphere because of its low thermal conductivity. Through global warming, the loss of sea ice leads to increasing absorption of solar radiation and heat flux from the ocean to the atmosphere, which further enhances the loss of sea ice and global warming. Arctic sea ice has been declining dramatically

over the past four decades (Onarheim et al., 2018; Comiso et al., 2008). Its extent has reduced by 40%–50% of its average in the 1980s (Perovich et al., 2020), whereas the average ice thickness has decreased by about 1.75 m in winter in the central Arctic Ocean (Rothrock et al., 2008; Kwok and Cunningham, 2015), which eventually leads to a volume loss of roughly 66% since 1980 (Petty et al., 2020; Kwok, 2018). Meanwhile, the ice drifting and deformation rates are increasing (Kwok et al., 2013; Hakkinen et al., 2008). The Arctic sea ice has been increasingly dominated by thinner and younger first-year ice (FYI) instead of thicker and older multiyear ice (MYI), the ice that has survived at least one summer melt (Maslanik et al., 2007). FYI comprised 35–50% of the ice cover in the mid-1980s. In comparison, this proportion increased to ~70% in 2019, while MYI covered less than one-third of the Arctic Ocean (Perovich et al., 2019; Kwok, 2018). The change of sea ice type (SITY) distribution impacts the climate of the Arctic and mid-high latitude regions through changes in water vapor, cloud properties, as well as large-scale atmospheric circulations such as the Atlantic Meridional Overturning (Liu et al., 2012; Screen et al., 2013; Boisvert et al., 2016; Belter et al., 2021). In addition, it influences the Arctic ecosystems by changing the habitat conditions for various Arctic species and is crucial for human activities such as shipping, tourism and resource extraction (Emmerson and Lahn, 2012; Meier et al., 2014). Studies found that the MYI area anomalies can largely explain (about 85%) the variance in Arctic sea ice volume anomalies (Kwok, 2018). Understanding the distribution and transition of Arctic SITY (especially MYI) is therefore of great scientific, as well as practical importance. SITY is a key parameter for sea ice thickness and total ice volume estimation (Alexandrov et al., 2010). Wrong assignment of SITY of a grid cell can distort the corresponding calculated ice thickness by more than 25% (Kwok and Cunningham, 2015). Accurate estimation of SITY is needed in many other areas of intertest, e.g. ice navigation, off-shore engineering and construction (Imarest, 2015) and weather forecasting(Jung et al., 2014).

To monitor Arctic sea ice type distribution changes at the hemispheric scale, various algorithms have been developed using microwave satellite data. Among them, most algorithms focus on the discrimination of MYI and FYI. These algorithms identify SITY (i.e. the discrimination of MYI and FYI in this study) based on the distinct radiometric and scattering characteristics of different ice types. On one hand, brightness temperatures (Tbs) of MYI tend to be lower than that of FYI because of its low-loss, low-salinity properties (Vant et al., 1978; Weeks and Ackley, 1986). Such difference is generally larger at higher frequencies (i.e. smaller penetration depth), which reflects the distinguished physical properties of MYI and FYI at the sub-surface layer (Sinha and Shokr, 2015). On the other hand, due to the high volume scattering and low scattering loss, MYI has a relatively higher backscatter than FYI at the same frequency (Onstott, 1992). There exist different algorithms which either provide a fractional MYI/FYI coverage or assignment of one or the other ice type (e.g. MYI and FYI) to a grid cell. The former referred to as sea ice type concentration (SITC) algorithms, includes algorithms such as the NASA Team algorithm and ECICE algorithm (Shokr et al., 2008; Cavalieri et al., 1984; Gloersen and Cavalieri, 1986), which are commonly used for sea ice concentration retrieval, as well as those particularly for MYI concentration estimation (Lomax et al., 1995; Kwok, 2004). The latter referred to as SITY algorithms, includes many algorithms, which differ from each other on input microwave observations, classification approaches, training datasets and post-processing (Ezraty and Cavanié, 1999; Belchansky and Douglas, 2000;

Anderson and Long, 2005; Walker et al., 2006; Hughes, 2009; Xu et al., 2022; Brath et al., 2013; Zhang et al., 2022). The passive microwave-based SITY algorithm was firstly adopted to derive Arctic SITY distribution from the Special Sensor Microwave/Imager (SSM/I) data (Andersen, 2000). This algorithm was later adapted to the follow-on passive microwave sensors, which consequently gives a long-term SITY product, available at the Copernicus Climate Change Service (C3S). For active microwave data, a long-term SITY distribution record since 1992 was derived based on geophysical module functions and dual-thresholds from inter-calibrated scatterometer data (Belmonte Rivas et al., 2018). Time-dependent dynamic thresholds were applied for ice type classification from 2002 to 2009 using QuikSCAT (QSCAT) data (Swan and Long, 2012), which was extended to 2014 with Oceansat-2 Ku-band Scatterometer (OSCAT) (Lindell and Long, 2016b). It is found the classifier accuracy can be improved by combining radiometer and scatterometer data (Yu et al., 2009). Multi-sensor approaches have been applied to derive SITY products (Shokr et al., 2008; Zhang et al., 2019; Lindell and Long, 2016a). Although the performances of passive and active microwave data on ice classification under various conditions have been compared in several studies (Zhang et al., 2021; Rivas et al., 2018; Yu et al., 2009), there is rarely analysis for the combined use of both data.

Comparison and evaluation of SITY products are needed for error estimation, error source control and improvement of SITY retrieval methods. Lacking in-situ data, evaluations of most SITY algorithms and products are limited to inter-comparisons. Consistency with other sea ice products is regarded as one of the best approaches (Belmonte Rivas et al., 2018). Operational SITY maps, ice charts, buoy measurements and ship observations are commonly used (Lee et al., 2017; Zhang et al., 2019). While the ice chart is used as "ground truth" in some validation (Aaboe et al., 2021a), some areas of MYI in the ice charts correspond to areas with MYI concentration of approximately 50% or greater (Lindell and Long, 2016a), indicating the overestimation of MYI in ice charts. Synthetic aperture radar (SAR) images are also used to evaluate ice type classification accuracy. The inconsistencies between products are attributed to the usage of different thresholds and satellite observation inputs (Ezraty and Cavanié, 1999; Belmonte Rivas et al., 2012). To date, systematic inter-comparison and method analysis for SITY products are still lacking. The questions remain as to how the SITY products perform and what factors we should consider to improve the SITY products.

This study aims to investigate differences among the SITY products and give comprehensive evaluations on the identification of MYI and FYI. We inter-compared eight SITY products from five SITY retrieval approaches for winters from 1999 to 2019 in this paper. Spatio-temporal variations and retrieval methods of the SITY products are investigated in detail. This paper is organized as follows. Section 2 introduces the data, whereas Section 3 describes the methods for the inter-comparison and evaluation. Section 4 starts with temporal and spatial analysis of the SITY products, and proceeds with regional validation with SAR images. Factors that influence the performance of SITY products are discussed in Section 5. Finally, conclusions are highlighted in Section 5.

## 2 Data

### 2.1 Microwave remote sensing data

#### 2.1.1 Microwave radiometer data

Passive and active microwave remote sensing data are commonly used in SITY estimation. The passive microwave data (i.e. microwave radiometer) used in the eight SITY products (to be introduced in Section 2.2) includes that from the Scanning Multichannel Microwave Radiometer (SMMR), SSM/I, the Special Sensor Microwave Image/Sounder (SSMIS), the Advanced Microwave Scanning Radiometer for EOS (AMSR-E) and the Advanced Microwave Scanning Radiometer 2 (AMSR2). Specifications of the different sensors are shown in **Table A1**.

The SMMR on Nimbus-7 was operating from October 1978 to August 1987. It provides five-frequency, dual-polarized (ten-channel) Tb observations with conical scanning at the zenith angle of 50.2°. The SSM/I aboard the Defence Meteorological Satellite Program operated from September 1987 to December 2008, providing four-frequency, seven-channel Tb measurements. Its successor, SSMIS (24 channels at 21 frequencies), has been operating since October 2003 to present. SSM/I and SSMIS are conically scanning radiometers with a constant incidence angle of around 53.1°.

The AMSR-E aboard the Aqua satellite is a twelve-channel, six-frequency radiometer, operating between 2002 and 2011. Its successor, AMSR-2 on the GCOM-W1, has been operating since 2012. Both AMSR-E and AMSR2 have a conical scan mechanism and maintain a constant incidence angle of 55°. In comparison, AMSR-E and AMSR2 provide Tb measurements with higher spatial resolution than SSMR/SSMI/SSMIS (**Table A1**).

#### 2.1.2 Microwave scatterometer data

The active microwave data (i.e. scatterometer) used in the SITY products includes that from the Active Microwave Instrument on European Remote-sensing Satellite (ERS), the SeaWinds scatterometer on QuikSCAT (QSCAT), the OceanSat Scatterometer (OSCAT) and the Advanced Scatterometer (ASCAT) onboard EUMETSAT's Metop-A and Metop-B, with specifications shown in **Table A1**.

ERS operated a C-band scatterometer (5.3GHz, VV polarization) from August 1991 to July 2011. It measures backscatter from

a broad range of incidence angles (18° to 47°). QSCAT is a Ku-band (13.4 GHz) conically scanning pencil-beam scatterometer. The inner beam is horizontally polarized (HH) at an incidence angle of 46°, whereas the outer beam is vertically polarized (VV) at an incidence angle of 54.1°. OSCAT is similar to QSCAT, operating at the frequency of 13.5 GHz with incidence angles for inner HH beam at 48.9° and outer VV beam at 57.6° from September 2009 to February 2014. ASCAT is a C-band (5.255 GHz) scatterometer with three vertically polarized (VV) antennas, whose incidence angle varies between 25° and 65°.

### 125 2.2 Sea ice type products

FYI and MYI can be discriminated from microwave satellite observations based on their distinctive radiometric and scattering signatures. The microwave radiometer measures the emitted radiation from the Earth in terms of brightness temperature (Tb),

which is linearly proportional to emissivity of the object. The microwave scatterometer measures the backscattered radar signal reflected off the Earth surface in terms of backscatter coefficient ($\sigma_0$), which is determined by the scattering properties.

Depending on the ambient conditions, sea ice at different stages of development undergoes different thermodynamic and dynamic processes, resulting in disparate microwave radiometric and scattering properties of different sea ice types (especially FYI and MYI). FYI is the sea ice of no more than one winter's growth. Brine is entrapped in ice during ice formation, leading to a relatively high salinity of FYI. The brine is expelled from sea ice during the melting and growing processes, leading to a near-zero level of salinity and high air inclusion in MYI. Due to the high dielectric constant of the brine, FYI has relatively

low radiation loss and thus high emissivity. On contrary, MYI has lower emissivity because of the desalinated property and the presence of air bubbles. Such differences in the physical properties are at the same time dependent on both frequencies and polarizations of the radiation. The distinct properties of FYI and MYI in the sub-surface layer are found to be better demonstrated at high frequencies (Vant et al., 1978).

In addition to Tbs at different channels, parameters of their combinations are also used in sea ice type discrimination, such as

the Polarization Ratio (PR) and Gradient Ratio (GR). PR is the normalized difference between the horizontally and vertically polarized Tbs, whereas GR is the normalized difference between Tbs at two frequencies, defined as:

$$PR_f = \frac{Tb_{fv} - Tb_{fh}}{Tb_{fv} + Tb_{fh}} \qquad\qquad 2.1$$

$$GR_{f_1 p f_2 p} = \frac{Tb_{f_1 p} - Tb_{f_2 p}}{Tb_{f_1 p} + Tb_{f_2 p}} \qquad\qquad 2.2$$

Where $Tb_{fv}$ and $Tb_{fh}$ means the vertically and horizontally polarized Tb at the frequency of $f$ and other parameters are presented in the same manner.

Microwave scattering of sea ice is determined by the surface and volume scattering, which is influenced by factors such as surface roughness, salinity, thickness and microstructure. Undeformed FYI is generally characterized by a level surface, while MYI has a much rougher surface, with hummocks and refrozen melt ponds. Meanwhile surface scattering of FYI could be much higher under deformation, resulting in generally higher surface scattering of MYI than undeformed FYI however comparable with deformed FYI. Air pockets within the subsurface layer of sea ice contribute to the volume scattering. The

above effect eventually leads to a relatively low backscatter for FYI and high backscatter for MYI. Such difference meanwhile depends on the frequency, polarization and observation angle of scatterometer, which could further influence the accuracy of SITY product.

During most of the winter months, MYI and FYI can be discriminated based on the above signatures. However, it becomes indistinguishable when it comes to the melting season, when microwave radiation can only penetrate the top layer of melting

snow (Hallikainen and Winebrenner, 1992; Carsey, 1985; Kern et al., 2016). Therefore, most SITY products only provide data from the winter months (mostly from October to April, some even from November to April).

This study inter-compares eight daily SITY products from five SITY retrieval approaches, including those obtained from the C3S (referred to as C3S-SITY) (Aaboe et al., 2020), Ocean and Sea Ice Satellite Application Facility (referred to as OSISAF-

SITY) (Breivik et al., 2012), Royal Netherlands Meteorological Institute (KNMI) (referred to as KNMI-SITY) (Rivas et al.,

2018), the Satellite Data Processing and Distribution Centre of French Research Institute for Exploitation of the Sea (CERSAT/Ifremer) (referred to as IFREMER-SITY) (Girard-Ardhuin, 2016), and Beijing Normal University (referred to as Zhang-SITY) (Zhang et al., 2019). Basic information of the SITY products is shown in **Table 1**, with the time line of satellite inputs visualized in **Figure 1**. Among them, OSISAF-SITY before 2010 and C3S-SITY solely use radiometer data, while KNMI-SITY and IFREMER-SITY only use scatterometer data. For OSISAF-SITY after 2009 and Zhang-SITY, both

radiometer and scatterometer measurements are utilized. Retrieval methods of these SITY products are summarized from the aspects of input parameters, classification methods and correction methods (**Table 2**), with detailed descriptions in the sub-sections below.

### 2.2.1 C3S-SITY

C3S-SITY is a purely radiometer-based product, provided in the Equal-Area Scalable Earth (EASE2) grid of 25 km spacing.

C3S-SITY has been released in two versions. The first version, C3S-1, was released in 2017 and the climate record covered the period 1979-2020. In 2021, the second version, C3S-2, was released and fully replaced C3S-1 with data available from late 1978 to present. An upgraded third version is ready to be released by the end of 2022 however is not be included in this study. SMMR, SSM/I and SSMIS data from the Fundamental Climate Data Record (FCDR) are the primary input data in the C3S-SITY products.

The retrieval of C3S-SITY entails three processing stages: pre-processing, core classification and post-processing.

In the pre-processing, the Tbs are collated and corrected for land spill-over due to the influence from land (Maaß and Kaleschke, 2010; Wentz, 1997), and hereafter corrected for atmospheric noise by using a Radiative Transfer Model (RTM) function with numerical weather prediction data (Maaß and Kaleschke, 2010; Wentz, 1997). In the latter process, C3S-1 and C3S-2 differ slightly by using different versions of atmospheric reanalysis from the European Centre for Medium-Range Weather Forecasts

integrated Forecast System (ECMWFs), ERA-Interim and ERA-5, respectively. The swath data of Tbs are later gridded into daily fields with 25 km spacing.

In the second processing stage, the core of classification is based on a Bayesian approach using the classification parameter $GR_{37v19v}$. This approach computes the probability of each surface class and selects the most likely class in each pixel. The algorithm is tuned by daily updated training dataset of Tb observations collected within the nearest 15 days over pre-defined

areas. The daily updated probability density functions (PDFs) of the collected training data are dynamic in time and capture the seasonal and interannual variabilities. The pre-defined areas over which the data are collected are the climatological MYI and FYI regions, which are north of Greenland and Canada with longitude between 30°W and 120°W for MYI, and the Kara Sea, Baffin Bay, Laptev Sea and the Bay of Bothnia for FYI.

To exclude outliers from the training dataset, ice type classification is conducted for pixels from the pre-defined areas with the

initial PDF. Observations from the misclassified pixels are then excluded from the training dataset for the second round of classification. Note that C3S-SITY defines an ambiguous ice type class (referred to as Amb) in addition to the pure MYI and

FYI classes. The Amb class represents sea ice with a low classification probability. It may be both pure MYI, FYI or a mixture of FYI and MYI (Aaboe et al., 2021c).

In the last stage, several filters and correction schemes are applied to correct misclassified classes. Open Water filters are applied to remove spurious sea ice in the open ocean; one filter is based on a threshold of $GR_{37v19v}$ to remove erroneously classified ice pixels caused by atmospheric influence, and another filter utilizes 2-m air temperature to exclude the warm water pixels. In addition, misclassified MYI is reassigned to FYI partly based on a geographical mask and partly on a statistical threshold filter caused by the overfitted Gaussian distribution of MYI at $GR_{37v19v}$, which gives rise to erroneous classification in some extreme cases and can be restored by thresholding. Besides, a geographic mask is used in C3S-SITY to remove unphysical MYI pixels within the FYI climatological area. Finally, an additional correction scheme based on air temperature is implemented in C3S-2 algorithm and reassign misclassified FYI, which is induced by warm air intrusions (Ye et al., 2016a).

### 2.2.2 OSISAF-SITY

The retrieval behind the OSISAF-SITY product is very similar to the C3S-SITY. It differs in being a near-real-time product, and it is provided in the polar stereographic projection with 10 km grid spacing. OSISAF-SITY has been available since 2005, however, with regular updates in both the input data and methodology. Therefore, the existing archive of data is not consistent in time and the quality of the product is expected to be higher towards the present time (Aaboe et al., 2021a). In the period of 2005-2009, OSISAF-SITY is a purely radiometer-based product only using SSM/I as input data. Since 2009, it has been a multi-sensor product when scatterometer data from ASCAT was introduced to supplement the radiometer data. In 2016, the radiometer is switched to AMSR-2 data (**Figure 1**).

Unlike C3S-SITY, both the pre-processing of the Tb data and the core Bayesian computation in the OSISAF-SITY are performed on the swath data instead of on gridded data. The computation of PDFs changes in 2015. Before 2015, static PDFs are used in the classifier, which are derived from a fixed training dataset based on observations of the pre-defined areas (same as that in C3S-SITY) during specific years. Since 2015, dynamic PDFs, based on daily updated training dataset as in C3S-SITY, was introduced and used ever since. Note that the classification uses $GR_{19v37v}$ data solely during 2005–2009, however introduces additionally backscatter from ASCAT ($\sigma_0$) since 2009. Ice types and their probabilities are derived using classifiers based on the respective observational parameters ($GR_{19v37v}$ and $\sigma_0$), where swath data of different sensors are used. The probabilities are then gridded based on the distance between each footprint and the polar stereographic grid. The final ice type in each grid is determined by the class with the highest probability. Same as that in C3S-SITY, a category of Amb is defined additionally to MYI and FYI in OSISAF-SITY, where the highest ice type probability is less than 75% (Aaboe et al., 2021b).

In the post-processing stage, OSISAF-SITY uses the same geographical mask as those in C3S-SITY (Aaboe et al., 2021b).

### 2.2.3 KNMI-SITY

KNMI-SITY is a series of purely scatterometer-based products with grid spacing of 12.5 km in a polar stereographic projection. The scatterometer data used includes ERS, QSCAT, OSCAT and ASCAT, which results in four respective SITY products, referred to as KNMI-E, KNMI-Q, KNMI-O and KNMI-A respectively, available during the periods of 1992–2001, 1999–2009,
2010–2013 and 2007–2016. In this study, KNMI-Q and KNMI-A are included in the comparison and evaluation considering the comparable input data as other products.

In the pre-processing stage, the ASCAT measurements are normalized to a standard incidence angle of 52.8°, which is close to that of the VV-polarization channel of QSCAT. The normalization is performed according to the dependency of C-band sea ice backscatter on incidence angle.

In the stage of classification. KNMI-SITY uses a refined Bayesian algorithm for ice/water discrimination, based on the probabilistic distances to the Geophysical Model Functions of ocean wind and sea ice. The sea ice pixels are then classified into FYI, second-year ice (SYI) and older MYI using VV polarized backscatter with two thresholds, which are determined from the data of March of each year (Belmonte Rivas et al., 2018).

In the last stage, a geographic mask is used to remove unphysical MYI signatures in Greenland, Kara Sea, Barents Sea and
Chukchi Sea.

### 2.2.4 IFREMER-SITY

IFREMER-SITY is another series of purely scatterometer-based products, with grid spacing of 12.5 km in a polar stereographic projection. There are two SITY products in IFREMER-SITY, which use QSCAT and ASCAT data for the respective years of 1999–2009 and 2010–2015, referred to as IFREMER-Q and IFREMER-A, respectively.

In the first stage, the backscatter coefficients at different incidence angles (e.g., ASCAT backscatter) are normalized to the value at a constant incidence angle of 40° to account for the influence of varying incidence angles. In the core classification, a set of day-to-day-varying thresholds are then used for the discrimination between MYI and FYI. These thresholds are derived from the backscatter data of several winters and are found to be inter-annually consistent (Girard-Ardhuin, 2016). Unlike other SITY products, no post-processing has been applied in IFREMER-SITY.

### 2.2.5 Zhang-SITY

Zhang-SITY is a combined SITY product with grid spacing of 4.45 km at polar stereographic projection from 2002 to 2020. The radiometer data is utilized in a way that AMSR-E/2 data comes prioritized whenever available and is supplemented with SSMIS whenever AMSR-E/2 is missing. The AMSR-E data is obtained from the NASA Scatterometer Climate Pathfinder (SCP) with grid spacing of 8.9 km, whereas the AMSR2 and SSMIS data is from GCOM-W1 and NSIDC with grid spacing
of 10 km and 25 km, respectively. Scatterometer data from QSCAT and ASCAT is used successively in Zhang-SITY, with

QSCAT data until November 23, 2009. All the scatterometer data is obtained from SCP with enhanced spatial resolution of 4.45 km, as a result of the scatterometer image reconstruction technique (Early and Long, 2001; Long et al., 1993).

In the pre-processing, the ASCAT data is normalized to the value at incidence angle of 40° as that in IFREMER-SITY. All the radiometer and scatterometer data are then re-gridded to the same spacing of 4.45 km.

Before ice type classification, open water and low sea ice concentration area are flagged out based on a threshold method using Tbs at 6.9 GHz V channel. For the ice pixels, an adaptive classification method based on K-means clustering is applied to the observation vectors consisting of Tbs at 36 GHz H channel and VV polarization backscatter $\sigma_0$. It is an unsupervised classification approach thus does not require the selection of training dataset. In addition, the results from different sensors are generally consistent thus no further processing is conducted for the satellite data (Zhang et al., 2019).

In the last stage, a correction scheme based on sea ice motion and a median filter considering the spatial consistency are used in the post-processing. The former is introduced to eliminate anomalous MYI overestimation, shown as the sudden presence of MYI pixels far away from the estimated MYI pack, based on the MYI temporal record and ice motion. The latter is used to remove large unusual spatial variations of ice types (Zhang et al., 2019).

## 2.3 Sea ice age product

In this study, the sea ice age (SIA) product from NSIDC is used for inter-comparison, referred to as NSIDC-SIA (Tschudi et al., 2020). NSIDC-SIA is a weekly product available all year round at 12.5 km spacing in the EASE grid from 1984 to 2021. It is derived by tracking trajectories of virtual Lagrangian ice parcels of each grid cell. The ice motion data used in the tracking process is based on passive and active microwave observations as well as auxiliary data such as drifting buoys (Fowler et al., 2004; Maslanik et al., 2011; Tschudi et al., 2020).

NSIDC-SIA has been widely used to assess the Arctic sea ice cover changes because of its high consistency in long time series (Liu et al., 2016; Meier et al., 2014; Perovich et al., 2020). Due to its middle-of-the-road scheme between directly satellite-derived and buoy-derived data, NSIDC-SIA supplies a comparable and independent reference for sea ice parameters that are entirely based on remote sensing data, e.g. sea ice type and thickness (Tschudi et al., 2016; Lee et al., 2017).

The accuracy of NSIDC-SIA largely depends on the ice trajectories tracking technique and ice motion data. There are mainly
two sources of error in NSIDC-SIA: the tracking errors related to the coarse resolution of microwave satellite data and those induced by ice motion data vacancy near the coast. The under-sampling of ice motion along with the scheme of oldest ice age assignment lead to an overall discontinuous sea ice age distribution and overestimation of old ice (Korosov et al., 2018). Besides, ice motion velocities from buoys are generally higher than those from satellite data (Schwegmann et al., 2011). Improper interpolation approach could lead to artificial divergence in ice motion when the buoy estimation differs significantly
from the satellite-based data. It could result in approximately 20% less MYI in the buoy-affected region according to a

numerical experiment (Szanyi et al., 2016). Such impact is mainly found in the years 1983–2005 and has been largely mitigated by tuning the interpolation approach in the current version (Tschudi et al., 2020).

## 2.4 other data

Three Radarsat-1 (referred to as RS-1) and Two Sentinel-1 (referred to S-1) SAR images are visually interpreted in terms of
sea ice type classification and used for validation. RS-1 operated from 1995 to 2013, providing C-band (5.3 GHz) SAR images at HH polarization. The incidence angle ranges from 20° to 49°. S-1 has been operating since 2014, providing C-band (5.4 GHz) SAR images at co- and cross-polarized with incidence angle between 18.9° to 47.0°. The three RS-1 images are in ScanSAR wide beam mode, whereas those from S-1 are in extra-wide swath mode. The geolocations and acquiring dates are shown in **Figure 2**.

Auxiliary data from atmospheric reanalysis is used in addition to the SAR images in the validation. The reanalysis data includes 2 m air temperature and 10 m wind from the ERA5 hourly dataset, produced using 4D-Var data assimilation and model forecasts in CY41R2 of the European Centre for Medium-Range Weather Forecasts integrated Forecast System (ECMWFs) (Hersbach et al., 2018).

## 3 Methodology

**3.1 Estimation of MYI extent**

For the inter-comparison, the Arctic MYI extent is calculated from the respective SITY and SIA products. The calculations are performed on the area within the Arctic Basin and limited by the polar hole of 87°N (used in (Belmonte Rivas et al., 2018) see in **Figure 2**). Note that data deficiency area of SITY products around the North Pole is excluded from the extent calculation and analysis. For the SITY products, the Arctic MYI extent is estimated as the integral extent of pixels specified as MYI within
the above-defined area. Both SYI and MYI (ice that is older than two years here) classes in KNMI-SITY are included in MYI extent calculation. The Amb class in C3S-SITY and OSISAF-SITY could be regarded as either MYI or FYI thus the MYI extent is calculated under both circumstances. This results in two values for the respective SITY products, one for the pixels of MYI class and the other for the pixels of MYI and Amb classes. For NSIDC-SIA, the Arctic extent is calculated by the integral extent of pixels with an ice age of two years at least.

As described above, C3S-SITY and NSIDC-SIA are in the EASE grid, while other products are in the polar stereographic grid, with the projection plane tangent to the Earth's surface at 70°N. The EASE grid is an equal areal projection, whereas the polar stereographic grid translates to a 6% distortion at the North Pole. To account for the areal distortion, all the SITY products in polar stereographic grids (namely OSISAF-SITY, KNMI-SITY, IFREMER-SITY and Zhang-SITY) are re-projected to the EASE grid before the calculation of MYI extent. Besides, the MYI extents from all the daily SITY products are averaged

weekly to inter-compare with that from the weekly NSIDC-SIA product to account for potential uncertainties introduced by the temporal inconsistency.

## 3.2 Visual interpretation of SAR imagery

SAR images have been widely used for SITY classification due to the distinct scattering properties between major ice types. As described in Section 2.1, backscattering from sea ice is predominantly a function of surface scattering for FYI, and the

combination of surface and volume scattering for MYI. Such difference is determined by sea ice properties such as salinity, porosity, snow grain size and crystalline structure as well as the sensor specifications such as frequency, polarization and observation angle (Gray et al., 1982; Kim et al., 1985). Because of the high spatial resolution, there are additionally texture and shape information from SAR imagery available for ice type discrimination compared to scatterometer data (Holmes et al., 1984). FYI can be formed under calm conditions, resulting in a smooth and level surface, while ridged, rubble or brash ice are

formed under turbulent conditions. In contrast, bubble-rich hummocks and much less bubbly refrozen melt ponds are significant features of MYI. Particularly, the MYI floes could develop a clear round shape during the collisions against one another (Onstott, 1992).

Visual interpretation of SAR images is performed based on the following principles: (1) FYI with level surface exhibits low backscatter signals and smooth textures (**Figure 3**a). Ridged FYI presents bright linear structures over the dark background in

SAR images (**Figure 3**b), while brash ice has high backscatter and is usually found between ice floes (**Figure 3**c). (2) Backscatter of newly formed ice is usually low. However, it could be high when frost flowers are formed on the refrozen leads or the ice is rough due to deformation (bright features over the darker strips in **Figure 3**d). (3) MYI presents a relative high backscatter and coarse texture (**Figure 3**e). The round floe structures could be used for the identification of MYI (**Figure 3**f). In addition, the sea ice extent record and the minimum ice extent of the previous summer could be both used as additional

information for the ice type interpretation from SAR imagery.

Before visual interpretation, all the SAR images are radiometrically calibrated and projected to the UTM projection with pixel size of 50 m for RS-1 data and 40 m for S-1. A refined denoising method is applied to S-1 images to reduce the extensive thermal noise at HV-polarized channel (Sun and Li, 2021). Images at HV polarization are prioritized for the visual interpretation if provided, since the cross-polarized backscattering signals have been shown to increase the separability

between MYI and FYI (Gray et al., 1982; Onstott et al., 1979). After the above pre-processing, ice type classification is manually conducted following the afore-mentioned principles. The classification results are then compared to those from the SITY products for accuracy estimation, when the respective Kappa coefficient and overall accuracy are calculated.

## 4 Results

This section starts with a temporal and spatial comparison of the SITY products, with NSIDC-SIA as a reference dataset. It then proceeds with validation against SAR images. The temporal and spatial comparison provides clues about the overall performance, while the evaluation against SAR images provides more concrete evidence in a case study of five representative cases. For analysis of spatial patterns, the defined area is divided into three regions: the central Arctic Ocean (CAO), the East Siberian and Laptev Seas (ELS), along with the Beaufort and Chukchi Seas (BCS).

### 4.1 Temporal analysis

#### 4.1.1 Weekly MYI extent variation

The Arctic MYI extent from the eight SITY products is compared with the NSIDC-SIA product for the winters from 1999 to 2019 (**Figure 4**). The dashed and solid lines represent the respective daily and weekly MYI extent of each SITY product, with the shaded area indicating the ambiguous extent from Amb class (in C3S-1, C3S-2 and OSISAF-SITY), whereas the stacked block represents the extent for the corresponding age of ice in NSIDC-SIA. Theoretically, since FYI can only turn to MYI when surviving a melting season, the overall Arctic MYI extent cannot increase over the winter – it can only decrease through ice advection out of the Arctic. However, it can temporarily or regionally increase due to ice divergence or advection from neighbouring regions (Kwok et al., 1999).

The SITY products show overall negative trends of the MYI extent within most of the winters as expected. Exceptions occur in some winters for almost all the SITY products. For instance, all the SITY products show increasing MYI extent in March/April 2017 except Zhang-SITY. This could be caused by the enhanced melting during this period (Raphael and Handcock, 2022; Ye et al., 2016a), which leads to the radiometric and scattering signatures of FYI similar as that of MYI therefore unsatisfactory performances of the SITY algorithms. The ice motion refined post-processing technique in Zhang-SITY may help to mitigate such overestimation problem of MYI (Zhang et al., 2019). Similar increasing patterns are found in October/November of different years for the respective SITY product, e.g., 2001 and 2003 for C3S-SITY, 2009 and 2017 for OSISAF-SITY, and all the years after 2007 for KNMI-A. For C3S-SITY and OSISAF-SITY, such pattern is caused by underestimation of MYI in October, while for KNMI-A it is mainly due to the overestimation of MYI in November in the peripheral seas of the Arctic and will be further discussed later in Section 5. Note that, the other two SITY products (i.e. IFREMER-SITY and Zhang-SITY) do not provide data in October therefore do not show such pattern.

Among all the SITY products, KNMI-SITY, especially KNMI-A, has overall the highest Arctic MYI extent, with bias of $0.49\times 10^6$ $km^2$ compared to that from NSIDC-SIA (**Table 3**). On the contrary, OSISAF-SITY in the SSMIS period (S, 2006–2009 in **Table 4**) and IFREMER-A (2012–2015) show the lowest values, with bias of $-1.32$– $-0.86\times 10^6$ $km^2$ and $-0.99\times 10^6$ $km^2$, respectively. All other SITY products exhibit negative bias in MYI extent compared to NSIDC-SIA. Among them, Zhang-SITY during the QSCAT period (2002–2009) agrees best with NSIDC-SIA on estimating MYI extent, the average bias and mean absolute deviation (MAD) of which is $-0.02\times 10^6$ $km^2$ and $0.10\times 10^6$ $km^2$, respectively. Similar as the comparison

of MYI extent, we calculate the Arctic FYI extent for the respective SITY and SIA product. All the SITY products exhibit overestimation of FYI extent (positive bias) than NSIDC-SIA except KNMI-SITY (**Table 3**). KNMI-Q has the best agreement with NSIDC-SIA on FYI extent estimation, with bias and mean absolute deviation (MAD) of -0.001$\times 10^6$ $km^2$ and 0.15$\times 10^6$ $km^2$, respectively. Overall, the scatteromter-combined SITY product agrees better with NSIDC-SIA than the solely radiometer-based product, e.g. OSISAF-SITY during the ASCAT (2009–2019) and SSMIS period (2006–2009). The QSCAT-

based SITY product are more consistent with NDISC-SIA than the ASCAT-based product, e.g. KNMI-Q and KNMI-A.

For the SITY products with the Amb class, the average extent of this class is $0.21 \times 10^6$ $km^2$, $0.26 \times 10^6$ $km^2$ and $0.26 \times 10^6$ $km^2$, respectively, for C3S-1, C3S-2 and OSISAF-SITY. As described in Section 2.2, these Amb pixels have atypical microwave signatures of MYI/FYI thus high uncertainties on ice type discrimination. Compared with the average Arctic MYI extent difference against NSIDC-SIA ($0.42\times 10^6$ $km^2$, $0.45\times 10^6$ $km^2$, $0.79 \times 10^6$ $km^2$ for C3S-1, C3S-2 and OSISAF-

SITY, respectively), the contribution of these pixels to the comparison is overall considerable. In addition, it could be large under situations that trigger the atypical microwave signatures, which will be further discussed in section 4.1.2.

In terms of temporal stabilities, OSISAF-SITY and C3S-SITY (especially C3S-1) show larger day-to-day variabilities in MYI extent than other SITY and SIA products (**Figure** ). Considering the scatterometer data used in the SITY products (**Figure 1**), we can find that KNMI-SITY, IFREMER-SITY and Zhang-SITY exhibit larger day-to-day variabilities during the ASCAT

period (2009–2019) than the QSCAT period (2002–2009), especially in early winter months such as October and November. In comparison, OSISAF-SITY shows smaller temporal variabilities when backscatter data is used in addition to radiometer data (2009–2019).

Between any two SITY products, the difference in weekly MYI extent is up to $4.5 \times 10^6$ $km^2$, which occurs between OSISAF-SITY and KNMI-A in late October 2008. Considering the size of the study region (about $6.5 \times 10^6$ $km^2$), such discrepancy

is significant. This is caused by the relatively low MYI extent from OSISAF-SITY and the exceptional high value from KNMI-A in late October, the reason for which will be discussed in Section 5. On the other hand, different SITY products could have consistent MYI extent with nearly negligible difference, which occurs mostly in mid-winter months. Among all, KNMI-Q is most consistent with IFREMER-Q (1999–2008), with daily MYI extent difference varying between $0.01 \times 10^e$ $km^2$ and $0.8 \times 10^6$ $km^2$.

**4.1.2 Monthly MYI extent variation**

The monthly average MYI extent of all the SITY and SIA products is presented in **Figure** , with monthly differences between the respective SITY product and NSIDC-SIA varying from $1.2\times 10^3$ $km^2$ to $2.3\times 10^6$ $km^2$. The comparison is demonstrated in three months—November, January and April, on behalf of early, mid- and late winter, respectively. Overall, the deviation of MYI extent from all the SITY products is the smallest in January. The cold temperatures and relatively stable sea ice physical

properties in mid-winter lead to small uncertainties of ice type discrimination. Among the three stages of winter, the deviation

of the various SITY products is the largest in early winter, while the extent of the Amb class in C3S-SITY and OSISAF-SITY (shaded area in **Figure** ) is the largest in late winter. Both indicate the difficulties and large discrepancies of SITY products in the transition between summer and winter.

For the inter-annual evolution of MYI extent, C3S-SITY and OSISAF-SITY differ most from other SITY products. The latter exhibit mild negative trend during 2000–2007 and rapid negative trend from 2007 to 2013, while the former show larger inter-annual variabilities. This is mainly attributed to the large discrepancies in the winters of 2001–2003, 2006–2008 and 2016–2018. KNMI-Q, IFREMER-Q, IFREMER-A and Zhang-SITY agree well with NSIDC-SIA, with modest discrepancies in all stages of winter. Although the MYI extent from KNMI-A shows the largest discrepancy in early winter, it demonstrates high consistency with NSIDC-SIA in mid- and late winter.

## 4.2 Spatial analysis

### 4.2.1 Regional MYI extent evolution

To further explain the classification discrepancies between products, we divided the Arctic into three regions (**Figure 2**) and analysed the regional evolution pattern (**Figure** ). Overall, the MYI extent in the CAO and ESL regions shows a consistently negative trend, while that in the BCS region remains constant or is increasing. The former mainly results from the outflow of MYI to more southern areas. On one hand, MYI is extensively exported through the Fram Strait (Kuang et al., 2022) and, by small fractions, into the Barents Sea and through the Nares Strait following the Transpolar Drift Stream. In the ESL region, the MYI extent even decreases to zero in some winters (e.g. 2007–2009, 2012–2013), which is in line with the record low Arctic minimum sea ice extent in the previous Septembers. On the other hand, MYI is advected towards south along the Canadian Arctic Archipelago driven by the Beaufort Gyre. In the BCS region, large quantities of MYI are pushed out of this region following the anticyclonic current, the Beaufort Gyre, meanwhile replaced by the MYI from the CAO region. This eventually leads to the nearly constant or increasing MYI extent in the BCS region. In the ESL and BCS regions, it is found that the NSIDC-SIA MYI extent is usually considerably larger than the MYI extent from the SITY products, whereas it is less pronounced in the CAO region. This indicates that the mixture of MYI and FYI, which leads to the "overestimated" NSIDC-SIA MYI extent, occurs more frequently in the ESL and BCS regions than the CAO region, which could be explained by the more dynamic ice characteristics in the two regions.

In the winters of 1999–2019, most SITY products show similar intra-seasonal variation in the CAO region, while exhibiting disparate intra-seasonal evolutions in the BCS and ESL regions (especially in early and late winter). For instance, the anomalously large MYI extent from KNMI-SITY in October and November as mentioned before is mainly attributed to the BCS and ESL regions. The large underestimations of MYI extent for OSISAF-SITY in CAO and BCS before 2010 is for the early period of the product before the inclusion of scatterometer data and algorithm upgrades. C3S-SITY shows striking MYI extent fluctuations in 2001–2004 in BCS and ESL which show that the distinct inter-annual pattern seen in **Figure** is mainly

originating from these regions. For both C3S-SITY and OSISAF-SITY, the late-winter positive trend in 2016–2017 (**Figure** )
is seen for all three regions, however mostly pronounced in BCS and ESL regions.

### 4.2.2 SITY distribution maps

The classification results of SITY products are directly mapped on the perspective of the Arctic to make the inter-comparison
of SITY spatial distribution more intuitive. **Figure 7** and   show the available SITY and SIA distribution maps in the winters
of 2001–2002 and 2007–2008, 2011–2012 and 2016–2017, respectively. Maps of these dates are selected to present typical
discrepancies of SITY products as mentioned in previous sections (see **Figure** and **Figure** ).

In **Figure** a–e, the SITY distribution maps of four SITY products and NSIDC-SIA on October 18, 2001 are shown for visual
analysis. C3S-SITY shows obviously less MYI than KNMI-Q, IFREMER-Q and NSIDC-SIA, while the latter two exhibit a
quite consistent SITY distribution pattern. The discrepancy of MYI extent between C3S-SITY and NSIDC-SIA is up
to $0.29\times 10^6$ $km^2$ during the winters of 2002–2019. In **Figure** a and b (along with **Figure A1** a–d, f–i in Appendix and   a–b,
h–i), the discontinuous FYI delineation in the inner part of MYI pack is well demonstrated, which occurs in all winter months
and could partly explain the MYI extent fluctuations in C3S-SITY. On the other hand, IFREMER-Q (e.g. **Figure** c) shows
constantly less MYI than KNMI-Q (e.g. **Figure** d) in the transition zone of MYI and FYI in BCS, in good agreement with
their difference as shown in **Figure 错误!未找到引用源。**.

**Figure 7**f–m shows the classification maps of seven SITY products and NSIDC-SIA on November 15, 2007. As presented in
the previous section, the MYI extent of KNMI-A is much larger than other products in early winter, with exceptionally
extensive MYI distributed in the peripheral seas of the Arctic basin (**Figure** j). In comparison, KNMI-Q has the second largest
MYI coverage among the seven SITY products, with a slightly more finger-like structure of MYI extending through the
Chukchi Sea into the ESL region. The other five SITY products show generally consistent SITY distribution patterns with
NSIDC-SIA. Minor differences are found in the BCS area. Additionally, C3S-SITY and OSISAF-SITY show notably less
MYI in the Fram Strait.

The classification maps in   a–g demonstrate a typical scenario with small MYI extent. In the maps of March 28, 2012, the
SITY distribution from SITY products is not so consistent with that from NSIDC-SIA. The difference between NSIDC-SIA
and C3S-SITY is the smallest, which could also be reflected in the MYI extent. The weekly MYI extent from NSIDC-SIA is
about $1.99\times 10^6$ $km^2$, whereas it is $1.99\times 10^6$ $km^2$ and $1.70\times 10^6$ $km^2$ for C3S-1 and C3S-2 (Amb class not included),
respectively. OSISAF-SITY and Zhang-SITY show very similar distribution patterns (**Figure** e–f), with the Arctic MYI extent
of about $1.55\times 10^6$ $km^2$ and $1.30\times 10^6$ $km^2$, respectively. IFREMER-A shows the smallest MYI extent ($1.05\times 10^6$ $km^2$).
KNMI-A differs substantially from other SITY products as that in other cases (e.g., **Figure** f–m). However, the difference is
mainly from the Barents and Kara Seas in this case, not the central Arctic as in other cases. Overall, large discrepancies are
found among the SITY types, mainly in the BCS region.

**Figure 8**h–l shows the classification of C3S-SITY, OSISAF-SITY, Zhang-SITY and NSIDC-SIA on March 29, 2017. In this month, C3S-SITY and OSISAF-SITY show consistent SITY distribution with NSIDC-SIA except in BCS where the MYI is overestimated compared to NSIDC-SIA. This overestimation of MYI leads to the abnormal positive trend of MYI extent in BCS and the Arctic during the winter of 2016–2017 in C3S-SITY and OSISAF-SITY (**Figure** and **Figure** ). Furthermore, the thin tongue-shape MYI distribution extending across ESL and BCS is not well preserved in Zhang-SITY.

## 4.3 Validation based on SAR

In this section, SITY products are evaluated using ice type classification results interpreted from RS-1 and S-1 SAR images. Visual interpretation of the SAR images is based on the principles introduced in Section 3.2. Five cases are addressed in this study to present SITY distributions under different conditions based on the availability of data and feasibility of visual interpretation. The cases in early and late winter are selected to demonstrate situations with notable discrepancies of SITY products, whereas the cases in mid-winter are included to explore the performances of SITY products under relatively steady circumstances. In each case, the SAR image and its interpretation results are presented along with the SITY and SIA products (**Figs. Figure 9**–**Figure 13**). The Kappa coefficient and overall accuracy (OA) of the respective SITY product for each case are calculated and presented in **Table 4**.

### 4.3.1 Cases in early winter

In Case 1, a typical scene of early winter (November 2017) in MIZ is shown in **Figure** . Compacted ice edge with relatively high backscatter could be observed across the SAR image. In area D, open water manifests high backscatter because of the high wind speed (over 15m/s). Sea ice in the west part (area C) with coarse texture appears to be MYI. In the upper part of the image (represented by area A), the coarse texture and darker backscatter signature than area C make it more likely to be MYI, which drifts from the central Arctic. At the margin of sea ice and the northeast corner (area B), the quasi-smooth texture, dark backscatter of leads and bright signature of frost flower in between could be interpreted as newly generated FYI.

The SITY distribution from Zhang-SITY agrees generally well with the SAR image in this case, with the largest OA (0.88) and Kappa coefficient (0.80), although it partly misclassifies FYI as open water or MYI (e.g. area B and the block between areas A and B). Compared with the SAR image, IFREMER-Q shows an underestimation of MYI in area A. C3S-SITY (C3S-1 and C3S-2) and OSISAF-SITY underestimate MYI in areas A and C (note that scatterometer data is not used in OSISAF-SITY in 2007), with slightly less MYI compared to IFREMER-Q. On this day, the wind field was dominated by strong (~15 m/s) southerly wind which may explain some of the disagreements seen in daily averaged products in regions close to a border between classes. KNMI-SITY products overestimate MYI generally. The overestimation is more extensive in KNMI-A (when ASCAT is used), leading to a Kappa coefficient of 0.58 and OA of 0.74 (**Table 4**). NSIDC-SIA overestimates MYI generally thus yields a median Kappa coefficient and OA (0.56 and 0.73, respectively).

Case 2 is located in the East Siberian Sea on November 2015 (**Figure 10**). The air temperature was below −10℃. The wind speed in the western part was higher than in the eastern part. A bright longitudinal feature is clearly shown in the SAR image. It could be identified as MYI with bright backscatter and coarse texture (area A). In area D, rounded MYI floes can be identified. The east and west part shows low backscatter and smooth texture (enlarged in areas B and C, respectively), which are typical features of FYI. The backscatter signature in area B is brighter than that in C, influenced by the incidence angle.

SITY distribution patterns of C3S-SITY (C3S-1 and C3S-2) agree best with the SAR image. As shown in **Table 3 Bias and mean absolute deviation (MAD) between the SITY products and NSIDC-SIA in MYI and FYI extent**

| SITY product | MYI extent | | FYI extent | |
|---|---|---|---|---|
| | bias [$10^6 \ km^2$] | MAD [$10^6 \ km^2$] | bias [$10^6 \ km^2$] | MAD [$10^6 \ km^2$] |
| C3S-1 | -0.29 – -0.08 | 0.37 – 0.41 | 0.28 – 0.48 | 0.43 – 0.54 |
| C3S-2 | -0.40 – -0.06 | 0.39 – 0.45 | 0.36 – 0.60 | 0.44 – 0.62 |
| OSISAF-SITY | -0.77 – -0.50 | 0.56 – 0.79 | 0.55 – 0.81 | 0.59 – 0.83 |
| OSISAF-SITY* (S, 2006–2009) | -1.32 – -0.86 | 0.86 – 1.32 | 0.86 – 1.33 | 0.86 – 1.33 |
| OSISAF-SITY* (A, 2009–2019) | -0.54 – -0.35 | 0.44 – 0.57 | 0.42 – 0.60 | 0.48 – 0.62 |
| KNMI-Q | 0.29 | 0.29 | **-0.001** | **0.15** |
| KNMI-A | 0.49 | 0.54 | -0.25 | 0.51 |
| IFREMER-Q | -0.36 | 0.36 | 0.64 | 0.64 |
| IFREMER-A | -0.99 | 0.99 | 1.27 | 1.27 |
| Zhang-SITY | -0.29 | 0.32 | 0.52 | 0.52 |
| Zhang-SITY* (Q, 2002–2009) | **-0.02** | **0.10** | 0.26 | 0.26 |
| Zhang-SITY* (A, 2009–2019) | -0.47 | 0.47 | 0.68 | 0.68 |

*: S, Q and A represents the SSMIS, QSCAT and ASCAT period of the SITY product, respectively.

, the C3S-SITY products have the best performances in this case, with slightly higher Kappa coefficient in C3S-2. A slight underestimation of MYI can be found in OSISAF-SITY in areas A and D (scatterometer data is used in this case). KNMI-A largely overestimates MYI, especially in the western part of SAR image. Zhang-SITY totally ignores the MYI pack (narrow MYI tongue across the ESL area, similar to the case in   h–l), which lasts for the whole winter (maps not shown). MYI is slightly underestimated in NSIDC-SIA, with the Kappa coefficient of 0.57 and OA of 0.80. Yet such difference is nearly negligible considering their different temporal resolutions and the mobility features of sea ice.

### 4.3.2 Cases in mid-winter

To investigate the constant discrepancies among SITY products, two cases in mid-winter are selected with focus on the transition zones between MYI and FYI. Case 3 shows the comparison of seven SITY products in **Figure 11**, with SAR image

located in the region across BCS and ESL and obtained on February 14, 2007. A large area of MYI with high backscatter, ice floe structure and coarse texture could be observed in the centre of the SAR image (Area B). Areas A and C present low backscatter and smooth texture, which are typical characteristics of FYI. The backscatter in Area D is slightly higher, however its smooth texture makes it more likely to be FYI.

The general SITY distribution patterns of KNMI-SITY (KNMI-Q and KNMI-A) and Zhang-SITY are basically consistent with the SAR image, with Kappa coefficient of around 0.7 (**Table 4**). KNMI-Q and Zhang-SITY slightly underestimate MYI in the southwest corner. IFREMER-Q, C3S-SITY (C3S-1 and C3S-2) and OSISAF-SITY (radiometer-only period) ignore the MYI pack in this area, and this regional scale misclassification of MYI holds through the whole winter (maps not shown). Compared to the SAR image, the SITY distribution in NSIDC-SIA has a distinct pattern, with overestimation of MYI in the northwest part of the image (area A) meanwhile underestimation in the northern part (east of area A). As mentioned previously, such discrepancies are mainly attributed to the mobility features of sea ice and the different temporal resolutions between NSDIC and the SAR image.

The 4th case was acquired on February 16, 2008 and shown in **Figure 12**. The bright MYI floe feature is clear in the northeast part of the SAR image, so as the dark FYI feature in the southwest part. Areas A and D exhibit high backscatter of round MYI floe, and areas B and C present typical characteristics of FYI with smooth texture and low backscatter.

The high resolution of SAR images can clearly show diverse MYI floes within the FYI area (e.g. **Figure 12**) and vice versa, which is however not well been reflected in SITY products. Taking this into consideration, all the SITY products agree generally well with the SAR image except OSISAF-SITY, which fails to identify the MYI floes in the northeast part. Due to the finer resolution, a more detailed SITY distribution is preserved in Zhang-SITY, leading to the largest Kappa coefficient and OA (0.57 and 0.82, respectively). A slight underestimation of MYI can be found in IFREMER-Q (area A). In addition, IFREMER-Q fails to identify FYI in this case (misclassified as OW), which may be caused by the day-to-day varying thresholds and leads to the lowest Kappa coefficient and OA. KNMI-A manages to identify FYI better than KNMI-Q in area B however overestimate the MYI floes in area D, otherwise the two KNMI-SITY products are very similar. The C3S-SITY products (C3S-1 and C3S-2) are generally consistent with the SAR image however show slight misclassifications in different areas (areas A and C), which may be due to the highly mixed distribution of ice types and coarse resolution. Despite a westward shift, the SITY distribution pattern from NSIDC-SIA is overall similar to the SAR image and indicates a generally older type of MYI (> 3 years).

### 4.3.3 Case in late winter

In Case 5, a S-1 SAR image covering the southern part of ESL near the coast, acquired on April 27, 2015, is shown in **Figure 13**. The air temperature was around −10℃. The wind speed and sea ice drift speed were relatively low. The elongated bright feature across the central part of the SAR image appears to be MYI, with a clear floe structure observed in area B. The coarse texture and bright backscatter signature can be found south of the island in the SAR image (area C). As the ice in area C is

close/attached to the coast meanwhile far away from the minimum sea ice extent of the previous summer, it is more likely to
be land-fast ice rather than MYI. Area A is identified as deformed FYI because of the low-backscatter background and
numerous bright linear features of ridges. Area D is interpreted as FYI based on the typical smooth texture and overall dark
backscatter signature.

The MYI distribution pattern of KNMI-A resembles the SAR image except for a slight overestimation of MYI in the northern
part of the image (area A) and nearly the island, which may be caused by ice deformation. The Kappa coefficient and OA is
the largest for KNMI-A in this case. IFREMER-A and Zhang-SITY both completely ignore the MYI pack, and this error lasts
for the whole winter (maps not shown). C3S-SITY (C3S-1 and C3S-2) and OSISAF-SITY manage to identify FYI in area A,
and sporadically capture an elongated MYI feature northeast of the image (partly classified as Amb). However, they
underestimate MYI in area B and overestimate MYI in the southern part (areas C and D), which leads to a near-zero level
Kappa coefficient. NSIDC-SIA clearly captures the elongated MYI feature in this case though has slight underestimation of
MYI in area B.

### 4.3.4 Performances of sea ice type and age products

Performances of SITY and SIA products in the above five cases are summarized in **Table 4**, including the general pattern,
Kappa coefficient and OA. In all the five cases, NSIDC-SIA can generally capture the SITY distribution pattern meanwhile
exhibits slight over- or underestimation of MYI, which can be explained by the ice age assignment of the oldest ice and distinct
temporal resolution of NSIDC-SIA compared to SAR. These results agree with previous studies (Korosov et al., 2018; Ye et
al., 2019) and once again confirm the use of SIA product as a cross-validation dataset.

In the two cases of early winter (Cases 1 and 2, **Figure** and **Figure 10**), C3S-SITY (C3S-1 and C3S-2) has overall the best
performances with slight underestimation of MYI in Case 1 due to a northward shift of the MYI edge, which can be explained
by the persistent southerly wind. On contrary, C3S-SITY totally ignores the identification o of MYI in Case 3, leading to the
Kappa coefficient of 0. In Cases 4 and 5, C3S-SITY captures the SITY distribution pattern to some extent but do not come out
best under different circumstances. Between the two products of C3S-SITY, C3S-2 performs slightly better than C3S-1 with
more alike SITY distributions with the SAR images in cases 4 and 5 (**Figure 12** and **Figure 13**), also reflected in the Kappa
coefficient and OA. However, the improvement is insignificant in these five cases

OSISAF-SITY tends to underestimate MYI in almost all the five cases (**Table 4**), which is especially obvious for the period
before the inclusion of scatterometer data and dynamically updated PDFs (2005–2009, Case 1, 3 and 4). It shows generally
better performance with more recent upgrades of the algorithm, which can also be found in the MYI extent time series (**Figure**
and **Figure** ), where the MYI extent from OSISAF-SITY are more consistent with other SITY and SIA products after 2010.

In contrast to OSISAF-SITY, KNMI-SITY products (KNMI-Q and KNMI-A) tend to overestimate MYI in the two cases of
early winter (Cases 1 and 2) (**Table 4**). Such overestimation is especially obvious in KNMI-A and can be found in almost all

the winter months. This is well reflected in the extraordinarily large MYI extent of KNMI-A in November (**Figure** , upper panel), which is attributed to the misclassified MYI in the peripheral seas of the Arctic Basin (**Figure** ). In other three cases, especially Cases 3 and 5, KNMI-SITY has one of the best performances. It manages to preserve the SITY distribution pattern in the cases of mid- and late winter. This is in line with the good agreement of MYI extent between KNMI-SITY and NSIDC-SIA in January and April (mid- and lower panels in **Figure** ).

The IFREMER-SITY products (IFREMER-Q and IFREMER-A) tend to underestimate MYI as seen in the time series of MYI extent and case studies. On the other hand, the performance of IFREMER-SITY varies with the cases, which may be caused by the day-to-day varying thresholds and no post-processing to account for the spatio-temporal variations. In Case 1 (**Figure** ), the MYI distribution from IFREMER-Q agrees generally well with the SAR images, with slight underestimation of MYI. On the contrary, it fails to identify the FYI in Case 4 (**Fig. 12**).

Zhang-SITY performs generally well in the QSCAT period (Cases 1, 3 and 4) with slight underestimation of FYI and MYI in Cases 1 and 3, respectively. It however fails to identify the MYI pack of thin tongue-shape in the ASCAT period (Cases 2 and 5). Such pattern is also reflected in the monthly MYI extent time series (**Figure** ), where the difference between Zhang-SITY and NSIDC-SIA is minimal before 2009 and increases after 2009 (i.e., the ASCAT period).

## 5 Discussion

Performances of SITY products could be attributed to the following factors: (1) input parameters, (2) classification methods and (3) correction schemes in the post-processing procedure. For further discussion, we analysed the five SITY products from the above three perspectives (**Table 4**).

### 5.1 Input parameters

The efficacy of input parameters depends on their capability to separate and physical properties of the sea ice types in question.
For instance, the contrast between MYI and FYI is high in the $GR_{37v19v}$ (and $GR_{19v37v}$) fields. However, this parameter can be impacted by surface features (e.g., snow properties) during the winter season (Rostosky et al., 2018; Ye et al., 2019; Comiso, 1983). In the beginning and ending stages of winter, the variability of $GR_{37v19v}$ can be significant when air temperature exhibits warm-cold cycles, which trigger wet-dry cycles or melt-refreeze cycles of snow (Voss et al., 2003; Ye et al., 2016b; Ye et al., 2016a), or when wet/thick precipitation suddenly appears (Voss et al., 2003; Rostosky et al., 2018). This can partly explain the
extensive MYI underestimation in the CAO region from C3S-SITY in October (**Figure** and **Figure** ), and the MYI overestimation in BCS and ESL in the second half of winter ( ). Such misclassification in C3S-1 is mitigated in C3S-2 due to the upgraded processing in C3S-2, which includes the temperature-based correction in the post-processing and the use of reanalysis data from ERA-5 instead of ERA-Interim in the atmospheric correction for Tb (see section 2.2). Another example is the backscatter coefficient ($\sigma^0$), which is commonly used in ice type discrimination due to the disparate scattering features

of MYI and FYI. Backscatter is highly impacted by surface roughness. As a result, deformed FYI, the backscatter of which is relatively high, can be misclassified as MYI when scatterometer data is used. In comparison, the backscatter of MYI and FYI is more disparate at Ku-band than C-band (Rivas et al., 2018; Bi et al., 2020). Products using Ku-band backscatter generally perform better on identifying MYI, e.g. KNMI-Q, IFREMER-Q, and Zhang-SITY before 2009. This could be due to the fact that Ku-band scatterometer is more sensitive to the crystal structure of MYI since its wavelength (about 1.7 cm ~ 2.5 cm) is more consistent with the characteristic dimension of air bubbles in MYI (Ezraty and Cavanie, 1999). On the other hand, the dominant effect of surface scattering and the higher dependence on incidence angle makes C-band backscatter more suitable to distinguish the ice types with disparate surface roughness features, e.g. Cases 3 and 4 in **Figure 11**, **Figure 12**.

It has been shown that the combination of radiometer and scatterometer data helps to identify ice types due to their complementary information (Yu et al., 2009). This statement holds under most conditions in this study (Zhang-SITY in Cases 3 and 4, **Figure 11** and **Figure 12**). However, when passive and active microwave signatures both behave anomalously, such combination does not help to mitigate the misclassification problems without regulating rules of priority between the two. In the peripheral sea that is ice-free during summer, introducing backscatter does not help to improve ice type identification in OSISAF-SITY and Zhang-SITY (Case 2, **Figure 10**). In the Beaufort and East Siberian Seas in late winter, employing Tb and backscatter measurements even leads to the worst SITY classification in OSISAF-SITY and Zhang-SITY (Case 5, **Figure 13**). This indicates that simple data combination does not necessarily imply better classification results.

## 5.2 Classification methods

The representativeness of training datasets and the efficiency of classification methods are crucial for ice type classification. Most SITY products are based on a priori training dataset, which are used to determine the threshold for ice type discrimination. Some algorithms use the thresholds derived from a training dataset that does not vary with time, region or satellite sensors, namely fixed thresholds, while others employ dynamic thresholds to account for the variability of training dataset. The former algorithms work relatively well under conditions similar to the training dataset, however it gives anomalous SITY distribution results in other conditions. For instance, KNMI-SITY uses the threshold extracted from the mid-winter of each year. Extensive anomalous SITY misclassification is found in the beginning of winter, when the backscatter characteristics of MYI and FYI differ largely from those in the mid-winter, especially for C-band backscatter. On the other hand, the dynamic threshold approach considers the spatio-temporal variability of the microwave radiometric and scattering characteristics. However, it may introduce additional temporal instability to the SITY products. The MYI extent from IFREMER-SITY shows high-frequency temporal oscillations in some winters, e.g. in 2008 April (see **Figure** ), which may be caused by the day-to-day-varying thresholds used in IFREMER-SITY (see section 2.2.4) and no post-processing to account for the spatio-temporal variations. C3S-SITY and OSISAF-SITY derive PDFs of FYI and MYI from daily training data of fixed target areas. The daily PDFs of the parameter $GR_{37v19v}$ for MYI are highly variable (Aaboe et al., 2021b). The possible explanations could be that the sample area of MYI is susceptible to the change of surface features such as snow properties. Microwave characteristics

of the ice samples from a fixed region may not be representative of the whole Arctic Basin, leading to occasionally extensive misclassifications (see Cases 3, 4 and 5, **Figure 11**, **Figure 12** and **Figure 13**). This leads to SITY distributions with high-frequency oscillations and large inter-annual variabilities in the C3S-SITY and OSISAF-SITY products.

An adaptive clustering algorithm is used in Zhang-SITY, without prior training data. The classification depends on the clustering pattern of the two-dimensional scatter of Tb and backscatter. Compared to the QSCAT period (2002–2009), Zhang-SITY shows more anomalous fluctuations and fails to identify narrow MYI tongue in peripheral seas in the ASCAT period (2009–2020). On one hand, the characteristic microwave signatures of FYI and MYI have more overlaps thus become more difficult to separate due to the ice loss in the winters over 2007–2009 (Belmonte Rivas et al., 2018). The large loss of old ice

(e.g. older than four years) in the Arctic Ocean leads to a younger MYI regime in the Arctic, thus smaller microwave signature differences between MYI and FYI (Tschudi et al., 2020). On the other hand, because of the lower sensitivity of C-band scatterometer on MYI identification (as explained in section 5.1), the separation between FYI and MYI becomes more difficult, especially from ASCAT data (Belmonte Rivas et al., 2018; Zhang et al., 2019).

**5.3 Correction schemes**

Post-processing correction plays an important role in the SITY products. For more accurate SITY distribution, various correction schemes are implemented in the SITY products. These correction schemes can be summarized as follows: (1) corrections based on geographic mask, (2) corrections based on statistical threshold, (3) corrections based on temperature records and the temporal variabilities of SITY distribution, (4) corrections based on fixed tolerance of ice motion and preceding results, and (5) corrections based on spatial filtering.

The first kind of correction scheme, a mask of the Arctic basin, has been used in C3S-SITY, OSISAF-SITY and KNMI-SITY to remove the unphysical MYI signature in areas such as the Greenland, Kara, Barents and Chukchi Seas. This is restricted to these areas and could not modify classification results within the central Arctic as delineated in this study. The thresholding filter in C3S-SITY and OSISAF-SITY exclude extreme values that are likely to cause misclassification, e.g., values beyond the simulated FYI PDF however within the wide simulated MYI PDF, which usually occurred in ice edge areas (Aaboe et al.,

2021b; Aaboe et al., 2021c). These two kinds of corrections exclude misclassification cases in regions outside the central Arctic thus have little impact on the overall SITY distributions.

The temperature-based correction in C3S-2 aims to reassign the erroneously classified FYI, which exhibits similar microwave signatures as FYI due to warm air intrusions (Ye et al., 2016a; Shokr and Agnew, 2013). As a result, the discontinuous FYI delineation in the inner part of MYI pack in C3S-2 is partly mitigated compared to C3S-1 (**Figure A1**). In Zhang-SITY, an ice

motion confining procedure is introduced to eliminate overestimated MYI. The procedure builds upon the ice motion temporal records and confines the evolution of MYI according to the tolerance of ice motion. One drawback of this post-processing is that, the wrong reassignment of MYI to FYI could lead to continuous underestimation of MYI in consecutive days. Another correction used in Zhang-SITY is the median filter correction, which considers spatial consistency and is employed to remove

large unusual SITY spatial variations. These two correction schemes in Zhang-SITY help to mitigate the afore-mentioned
problems. However, inappropriate thresholds in them may lead to over-correction, making Zhang-SITY incapable to identify
the narrow MYI tongue in peripheral seas (Cases 2 and 5, **Figure 10** and **Figure 13**).

Apart from the above three aspects (input parameters, classification methods and correction schemes), factors such as the
covering period and spatial resolution make the five series SITY products different from each other. The seasonal length of
classification differs from the "all year" KNMI-SITY products to a limited winter period for the other products (see **Table 1**).
In early and late winter larger uncertainties are likely to occur due to surface processes such as wet snow attenuation and
changes in brine salinity (Barber and Thomas, 1998; Voss et al., 2003; Ye et al., 2016a; Ye et al., 2016b). Some SITY products
do not provide data in these months (e.g. Zhang-SITY in October), the inter-comparison and evaluation in such conditions
thus cannot be done.

In this study, the grid resolution of the SITY products ranges between 4.45 km and 25 km. These different resolutions are
reflected in the SITY distribution and how well the products capture smaller-scale features such as ice floes and ice edges. For
instance, more detailed information can be found in Zhang-SITY in case 4 (**Figure 12**), whereas C3S-SITY and OSISAF-
SITY fail to resolve the floe distribution pattern. On the other hand, finer resolution does not necessarily mean higher accuracy.
Higher frequency Tb channels (e.g. 89 GHz) come along with higher resolution however are more likely to be influenced by
atmospheric conditions.

## 6 Conclusion

Arctic sea ice cover has decreased dramatically over the past few decades, especially MYI. The change of SITY distribution
impacts the Arctic and global climate. However, systematic inter-comparison and analysis for SITY products are still lacking.
In this paper, eight SITY products based on five retrieval approaches were inter-compared through temporal and spatial
analysis, with the NSIDC-SIA product as a comparative reference. Performances of them are evaluated qualitatively and
quantitatively using five SAR images.

The eight SITY products show overall negative trends of MYI extent as expected within most winters. Exceptions occur mainly
in early and late winter months such as October/November and March/April. Compared to NSIDC-SIA, all the SITY products
show smaller MYI extent and larger FYI extent except KNMI-SITY (KNMI-Q and KNMI-A). The bias of MYI extent between
the SITY products and NSIDC-SIA varies from $-1.32\times 10^6\ km^2$ (OSISAF-SITY, 2006–2009) to $0.49\times 10^6\ km^2$ (KNMI-A,
2009–2019). Among all the SITY products, Zhang-SITY in the QSCAT period and KNMI-Q agree best with NSIDC-SIA on
the estimation of MYI and FYI extent, respectively.

Between any two SITY products, the difference in weekly MYI extent spans from $0.01 \times 10^6\ km^2$ to $4.5 \times 10^6\ km^2$. The
largest discrepancy occurs between OSISAF-SITY and KNMI-A in late October 2008, while the smallest difference is found

between KNMI-Q and IFREMER-Q in mid-winter months. It is in line with the spread of the SITY products, which is largest in early winter months such as November and smallest in mid-winter months like January.

Performances of the SITY products can be summarized as follows:

1) C3S-SITY is a pure radiometer-based product. It has the longest temporal record however large temporal variability and anomalous intra-seasonal trends in MYI extent. It performs generally well in the early winter cases however yields unsatisfactory results in the rest. The fluctuation and misclassification are likely attributed to the single classification parameter and day-to-day varying training datasets from the pre-defined region, which are vulnerable to weather and ambient conditions and may not be representative for the entire Arctic. C3S-2 performs slightly better than C3S-1 with less misclassification and smaller temporal variability, which could be resulted from the temperature-based correction in post-processing and the upgrades of reanalysis data in the atmospheric correction for Tbs;

2) OSISAF-SITY has an overall underestimation of MYI. Such underestimation is more obvious during the radiometer-only period (2005–2009) while mitigated due to the upgrades in different periods. The use of additional scatterometer data and finer spatial resolution radiometer data, along with the dynamic PDFs lead to overall better performance of OSISAF-SITY after 2009 however still large temporal fluctuations in SITY distribution;

3) For the two pure scatterometer-based products, KNMI-SITY tends to overestimate MYI (especially in early winter), while IFREMER-SITY is prone to underestimate MYI. The thresholds used in the classification algorithms play an important role in these two SITY products. KNMI-SITY performs generally well in mid-winter months. The overestimation of MYI occurs mainly in the Arctic peripheral seas in October and November, especially during the C-band scatterometer period (KNMI-A). IFREMER exhibits high-frequency temporal variations in MYI extent, which could be caused by the day-to-day varying thresholds and improved by including appropriate post-processing;

4) Zhang-SITY exhibits disparate performances in the two scatterometer periods, with good performance in 2002–2009 (Ku-band scatterometer) while an underestimation of MYI and more anomalous fluctuations after 2009 (C-band scatterometer). During the latter period, it shows difficulties in detecting thin tongue-shape distribution of MYI in the Arctic peripheral seas, which could be caused by the excessive correction during post-processing.

Among all the SITY products, KNMI-Q and Zhang-SITY in the QSCAT period perform the best. This confirms that QSCAT is very capable of distinguishing sea ice types. KNMI-A has one of the best performances during mid- and late winter months however presents extensive overestimation of MYI in early winter months. Zhang-SITY in the ASCAT period and IFREMER-SITY (IFREMER-Q and IFREMER-A) show an overall underestimation of MYI. C3S-SITY performs well in some early winter cases, however has large daily variability as OSISAF-SITY and occasionally presents extensive misclassification.

Based on the above inter-comparisons, we further investigate the factors that may impact the SITY production. The main findings can be summarized as follows:

- Ku-band scatterometer generally performs better than C-band scatterometer on ice type discrimination (Belmonte Rivas et al., 2018). However, the latter sometimes could identify FYI more accurately, e.g. comparably small areas of FYI within a region dominated by MYI;

- The simple combination of scatterometer and radiometer data is not always beneficial without further rules of priority between the two. In peripheral seas in early winter, introducing backscatter does not help for better ice type identification. In Beaufort and East Siberian seas in late winter, the performance of the SITY products using combined data is weighed down by the radiometer data, and the MYI identification advantage of backscatter diminishes;

- The representativeness of training data and the efficiency of the classification method are crucial for ice type classification. Thresholds extracted from mid-winter may not be suitable for the entire winter, while dynamic training datasets could be susceptible to local environmental conditions and introduce temporal instability. On the other hand, the adaptive classification method that depends on the clustering pattern of the radiometric and backscattering signatures may be inefficient when the characteristic signatures of MYI and FYI have large overlaps;

- Post-processing corrections play important roles in SITY products and should be considered with caution. Excessive post-processing such as ice motion confining could lead to an over-correction problem, which becomes the basis for the subsequent corrections and eventually result in accumulative errors. Any post-processing should be flagged in a related product variable.

Accurate estimation of Arctic SITY distribution is crucial for better understanding regional and global climate change, as well as defining sea ice and snow properties for ice thickness retrievals, sea ice models and so on. This study inter-compares eight SITY products and provides hints for further improvement of SITY retrieval approaches. With the new twin-frequency scatterometer (WindRAD, Ku- and C-band) onboard Fengyun (FY)-3E satellite, the potential of scatterometer measurements for ice type discrimination can be further investigated. On the other hand, the Copernicus Imaging Microwave Radiometer with higher spatial resolution at low-frequency channels in near future opens the opportunity of using low-frequency microwave radiometer measurements for SITY classification (Kilic et al., 2018).

**Appendix A**

See **Table A1Table A1**, **Figure A1**.

**Author contribution**

Y.Y. designed the experiments and lead the manuscript writing. Y.L. and Y.S. conducted the data analysis. M.S. provided access to the SAR images and contributed to interpretation of the SAR images. Y.L., S.A. and F.G. contributed to result analysis. F.H., X.C. and Z.C. contributed to the research design and results analysis. All co-authors participated in the fruitful discussions and manuscript revision.

**Competing interests**

The authors declare that they have no conflict of interest.

**Acknowledgement**

This work is supported by the National Natural Science Foundation of China (Grant No. 42106225), the Innovation Group Project of Southern Marine Science and Engineering Guangdong Laboratory (Zhuhai) (Grant No. 311021008), the National

Key Research and Development Program of China (Grant No. 2019YFC1509104) and the Natural Science Foundation of Guangdong Province, China (Grant No. 2022A1515011545). We greatly thank C3S (https://doi.org/10.24381/cds.29c46d83, accessed on 1 April 2022), OSI SAF (https://osi-saf.eumetsat.int/products/osi-403-d,accessed on 1 April 2022), KNMI (https://dataplatform.knmi.nl/dataset/, accessed on 1 April 2022), Fanny Girard-Ardhuin and Zhilun Zhang for providing SITY products. We also thank NSIDC for providing SIA product (https://nsidc.org/data/NSIDC-0611/versions/4, accessed on 1

April 2022), ECWMF for providing ERA5 reanalysis (https://www.ecmwf.int/en/forecasts/datasets/reanalysis-datasets/era5, accessed on 1 April 2022) as well as the ASF and MDA for providing Radarsat-1 and Sentinel-1 images (https://search.asf.alaska.edu/, accessed on 1 April 2022).

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

**Table 1: Basic information of the SITY products**

| SIT product | | Covering period | | Satellite input | Frequency | Grid size |
|---|---|---|---|---|---|---|
| C3S-SITY | C3S-1 | 1979–2020 | Oct.1–Apr.30 | SMMR, SSM/I, SSMIS | daily | 25 km |
| | C3S-2 | 1978–present | Oct.15–Apr.30 | SMMR, SSMI/I, SSMIS | daily | 25 km |
| OSISAF-SITY | | 2005–present | Oct.1–Apr.30 | ASCAT, SSMIS, AMSR-2 | daily | 10 km |
| KNMI-SITY | KNMI-Q | 1999–2009 | All the year | QSCAT | daily | 12.5 km |
| | KNMI-A | 2007–2016 | | ASCAT | | |
| IFREMER-SITY | IFREMER-Q | 1999–2009 | Oct.1–Apr.30 | QSCAT | daily | 12.5 km |
| | IFREMER-A | 2010–2015 | Nov.1–Apr.30 | ASCAT | | |
| Zhang-SITY | | 2002–2020 | Nov.1–Apr.30 | QSCAT, ASCAT, ASMR-E, AMSR-2, SSM/I | daily | 4.45 km |

**Table 2 SITY retrieval methods**

| SITY retrieval method | Input parameters | | Classification method | Correction method |
|---|---|---|---|---|
| | Radiometer | Scatterometer | | |
| C3S-1 | $GR_{37v19v}$ | \ | dynamic PDF, Bayesian method | filters for open water[*], geographical mask, 'extreme' value threshold filter |
| C3S-2 | $GR_{37v19v}$ | \ | dynamic PDF, Bayesian method | filters for open water[*], geographical mask, 'extreme' value threshold filter, temperature-based correction |
| OSISAF-SITY | $GR_{19v37v}$ | $\sigma_0$[**] | dynamic PDF[***], Bayesian method | geographical mask, 'extreme' value threshold filter |
| KNMI-SITY | \ | $\sigma_0$ | Bayesian method, thresholds derived from March of each year | geographical mask |
| IFREMER-SITY | \ | $\sigma_0$ | day-to-day varying thresholds | \ |
| Zhang-SITY | $TB_{37h}$ | $\sigma_0$ | adaptive clustering | ice motion confining and spatial filtering |

[*] Open water filters based on gradient ratio and temperature values are used to verify the discriminate between sea ice open water. In this study, discussion of correction methods focuses on those for MYI and FYI.

[**] Scatterometer data from ASCAT was introduced to the OSISAF-SITY retrieval method in 2009.

[***] Dynamical PDF based on daily training data was introduced to the OSISAF-SITY retrieval method in 2015.

**Table 3 Bias and mean absolute deviation (MAD) between the SITY products and NSIDC-SIA in MYI and FYI extent**

| SITY product | MYI extent | | FYI extent | |
|---|---|---|---|---|
| | bias [$10^6\ km^2$] | MAD [$10^6\ km^2$] | bias [$10^6\ km^2$] | MAD [$10^6\ km^2$] |
| C3S-1 | -0.29 – -0.08 | 0.37 – 0.41 | 0.28 – 0.48 | 0.43 – 0.54 |
| C3S-2 | -0.40 – -0.06 | 0.39 – 0.45 | 0.36 – 0.60 | 0.44 – 0.62 |
| OSISAF-SITY | -0.77 – -0.50 | 0.56 – 0.79 | 0.55 – 0.81 | 0.59 – 0.83 |
| OSISAF-SITY[*] (S, 2006–2009) | -1.32 – -0.86 | 0.86 – 1.32 | 0.86 – 1.33 | 0.86 – 1.33 |
| OSISAF-SITY[*] (A, 2009–2019) | -0.54 – -0.35 | 0.44 – 0.57 | 0.42 – 0.60 | 0.48 – 0.62 |
| KNMI-Q | 0.29 | 0.29 | **-0.001** | **0.15** |
| KNMI-A | 0.49 | 0.54 | -0.25 | 0.51 |
| IFREMER-Q | -0.36 | 0.36 | 0.64 | 0.64 |
| IFREMER-A | -0.99 | 0.99 | 1.27 | 1.27 |
| Zhang-SITY | -0.29 | 0.32 | 0.52 | 0.52 |
| Zhang-SITY[*] (Q, 2002–2009) | **-0.02** | **0.10** | 0.26 | 0.26 |
| Zhang-SITY[*] (A, 2009–2019) | -0.47 | 0.47 | 0.68 | 0.68 |

[*]: S, Q and A represents the SSMIS, QSCAT and ASCAT period of the SITY product, respectively.

**Table 4: Performances of the SITY products compared to SAR images**

| SITY product | Case 1 (Nov. 2007) | | | Case 2 (Nov. 2015) | | | Case 3 (Feb. 2007) | | |
|---|---|---|---|---|---|---|---|---|---|
| | General pattern | Kappa coefficient | Overall Accuracy | General pattern | Kappa coefficient | Overall Accuracy | General pattern | Kappa coefficient | Overall Accuracy |
| C3S-1[*] | - | 0.72 – 0.77 | 0.81 – 0.84 | ○ | 0.69 – 0.70 | 0.85 – 0.86 | -- | 0.00 | 0.47 – 0.47 |
| C3S-2 | - | 0.74 – 0.79 | 0.82 – 0.86 | ○ | **0.71 – 0.72** | **0.86 – 0.87** | -- | 0.00 | 0.47 – 0.47 |
| OSISAF-SITY | -- | 0.57 – 0.62 | 0.70 – 0.74 | - | 0.50 – 0.54 | 0.78 – 0.79 | -- | 0.00 | 0.47 – 0.47 |
| KNMI-Q | + | 0.64 | 0.78 | / | / | / | - | 0.72 | 0.86 |
| KNMI-A | ++ | 0.57 | 0.75 | ++ | 0.37 | 0.66 | ○ | **0.77** | **0.89** |
| IFREMER-Q | - | 0.76 | 0.84 | / | / | / | -- | 0.00 | 0.47 |
| IFREMER-A | / | / | / | / | / | / | / | / | / |
| Zhang-SITY | ○ | **0.80** | **0.88** | -- | 0.00 | 0.60 | - | 0.68 | 0.84 |
| NSIDC-SIA | ++ | 0.56 | 0.73 | - | 0.57 | 0.80 | ++ | 0.23 | 0.62 |

| SITY product | Case 4 (Feb. 2008) | | | Case 5 (Apr. 2015) | | |
|---|---|---|---|---|---|---|
| | General pattern | Kappa coefficient | Overall Accuracy | General pattern | Kappa coefficient | Overall Accuracy |
| C3S-1[*] | +- | 0.40 – 0.47 | 0.73 – 0.80 | +- | 0.00 – 0.06 | 0.54 – 0.67 |
| C3S-2 | +- | 0.42 – 0.45 | 0.77 – 0.82 | + | 0.00 – 0.08 | 0.49 – 0.67 |
| OSISAF-SITY | -- | 0.16 – 0.33 | 0.79 – 0.81 | +- | 0.18 – 0.25 | 0.70 – 0.76 |

| | | | | | | |
|---|---|---|---|---|---|---|
| KNMI-Q | + | 0.50 | 0.78 | / | / | / |
| KNMI-A | + | 0.50 | 0.78 | ○ | **0.61** | **0.87** |
| IFREMER-Q | - | 0.12 | 0.18 | / | / | / |
| IFREMER-A | / | / | / | -- | 0.00 | 0.81 |
| Zhang-SITY | ○ | **0.57** | **0.82** | -- | 0.00 | 0.84 |
| NSIDC-SIA | ++ | 0.25 | 0.64 | - | 0.46 | 0.83 |

[*]: The Kappa coefficient and Overall Accuracy values of C3S-1, C3S-2 and OSISAF-SITY are represented within a lower bound and an upper bound calculated when the Amb class is regarded as FYI and MYI respectively.

○: best matches, +/-: overestimates/underestimates MYI, ++/--: overestimates/underestimates MYI in greater degree, /: no data.

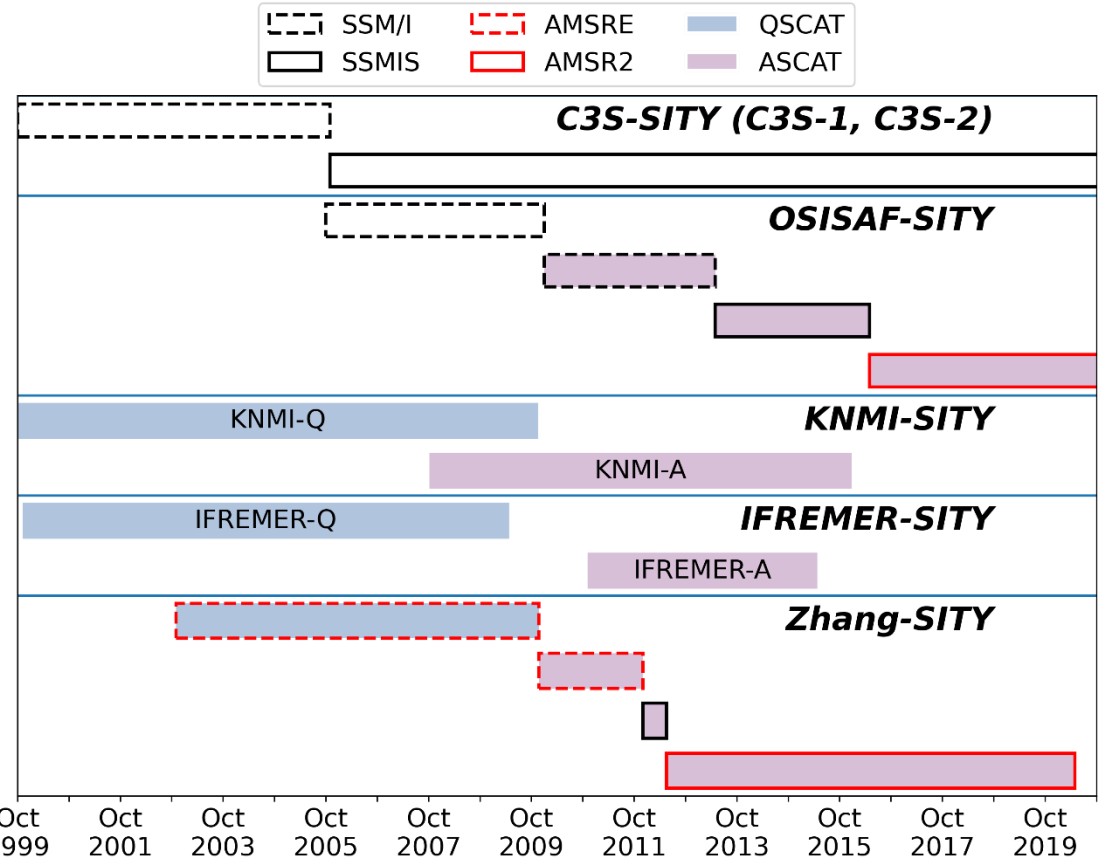

**Figure 1: The time lines and satellite data input of eight SITY products in this study based on five SITY retrieval schemes.**

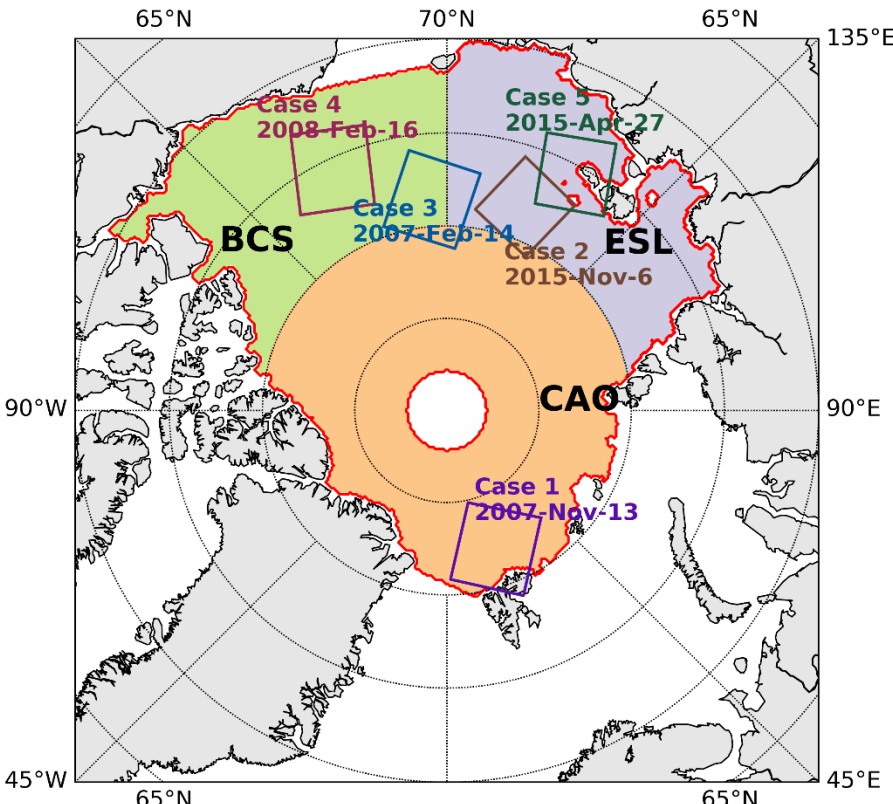

**Figure 2: Geographic locations of the SAR images for five cases and outline of the Arctic Basin (red contour, provided by (Belmonte Rivas et al., 2018)). The Arctic Basin is divided into three subregions: the central Arctic Ocean (CAO), the East Siberian and Laptev Seas (ESL) and the Beaufort and Chukchi Seas (BCS).**

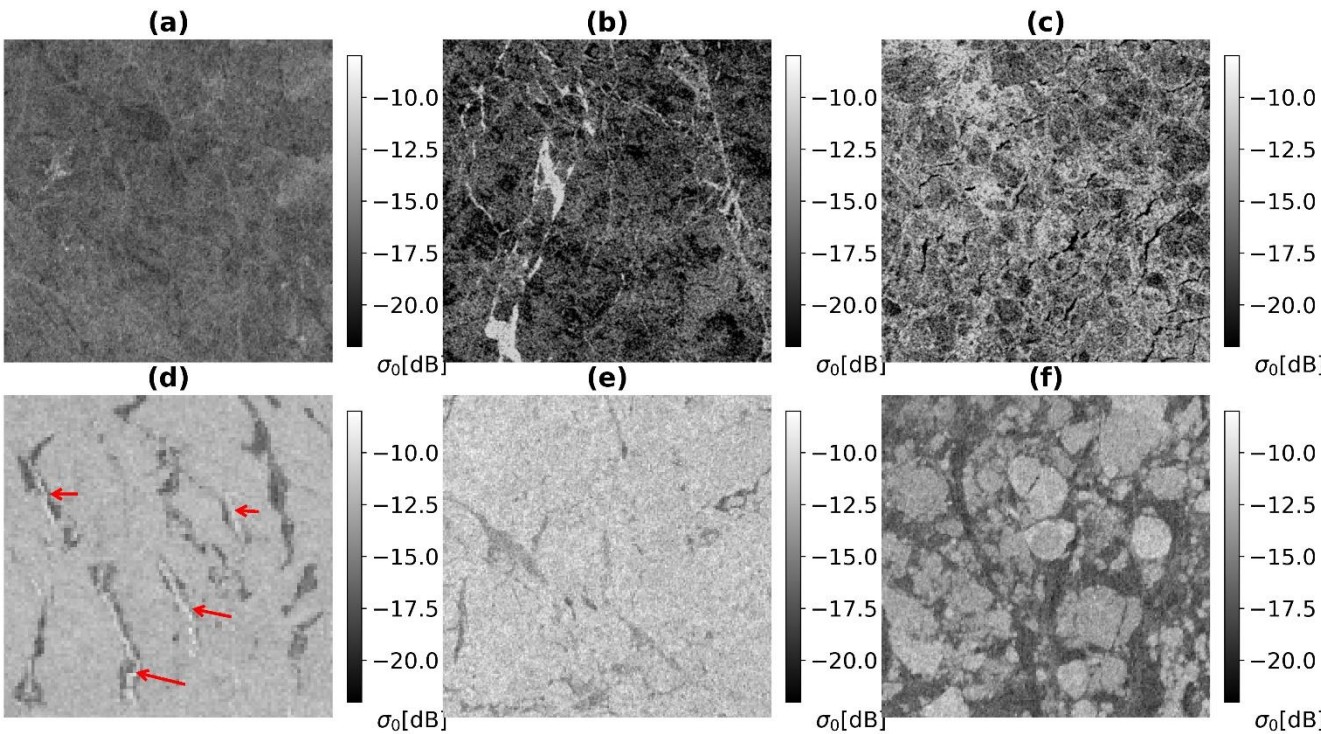

1010

**Figure 3: Scenes of SAR images (C-band, HH polarization) showing different sea ice features. (a) FYI with smooth textures, (b) FYI with ridged ice in bright linear features, (c) Brash ice between ice floes, (d) Refrozen leads with bright features, marked with red arrows, (e) MYI with bright backscatter and (f) MYI floes.**

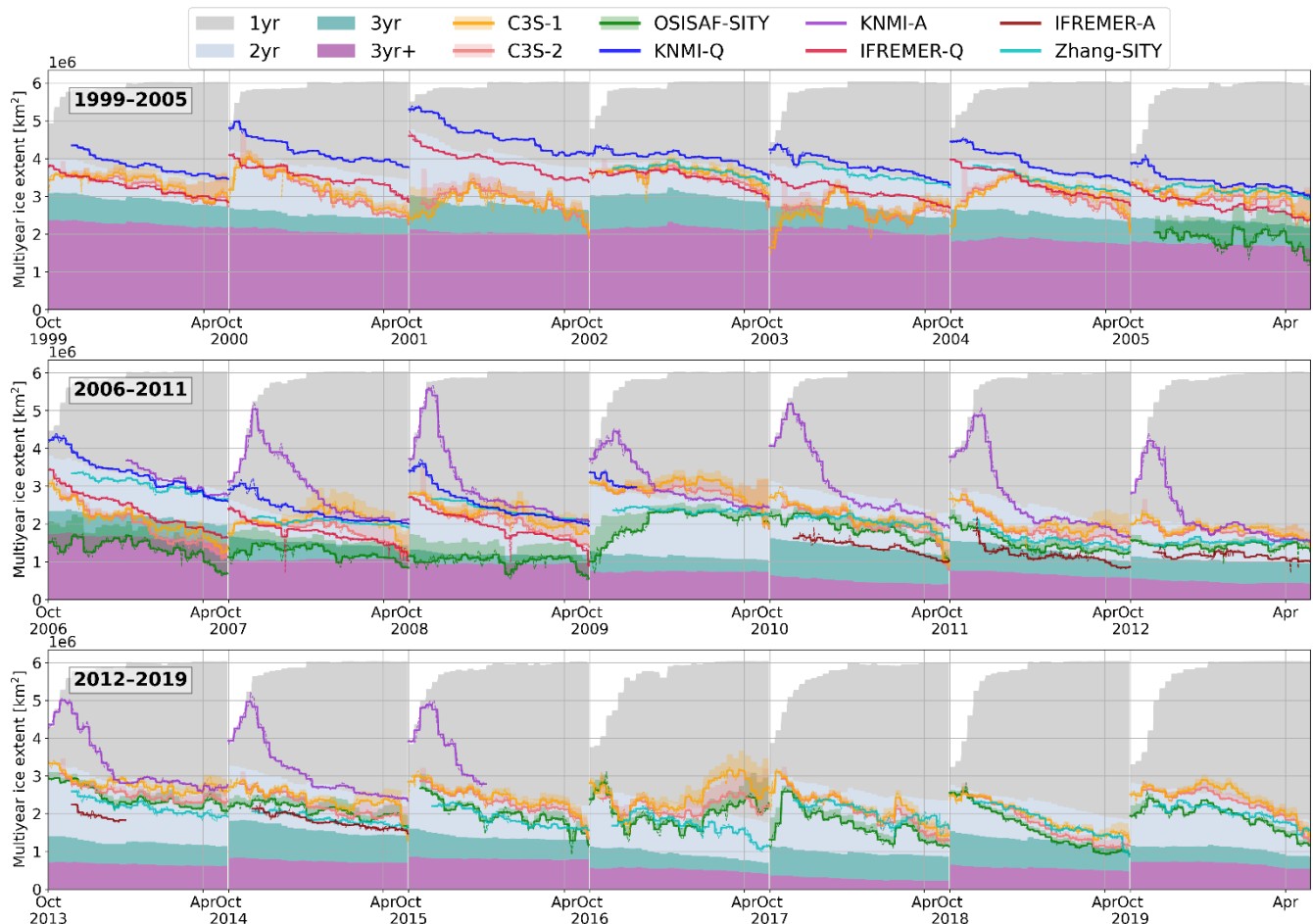

Figure 4: Arctic MYI extent variation of SITY products and NSIDC-SIA. The solid line represents weekly MYI extent of the SITY product, the dashed line represents daily MYI extent and the shaded area represents the ambiguous extent from Amb class (in C3S-1, C3S-2 and OSISAF-SITY), while the stacked block in background represents ice extent with the corresponding age of NSIDC-SIA.

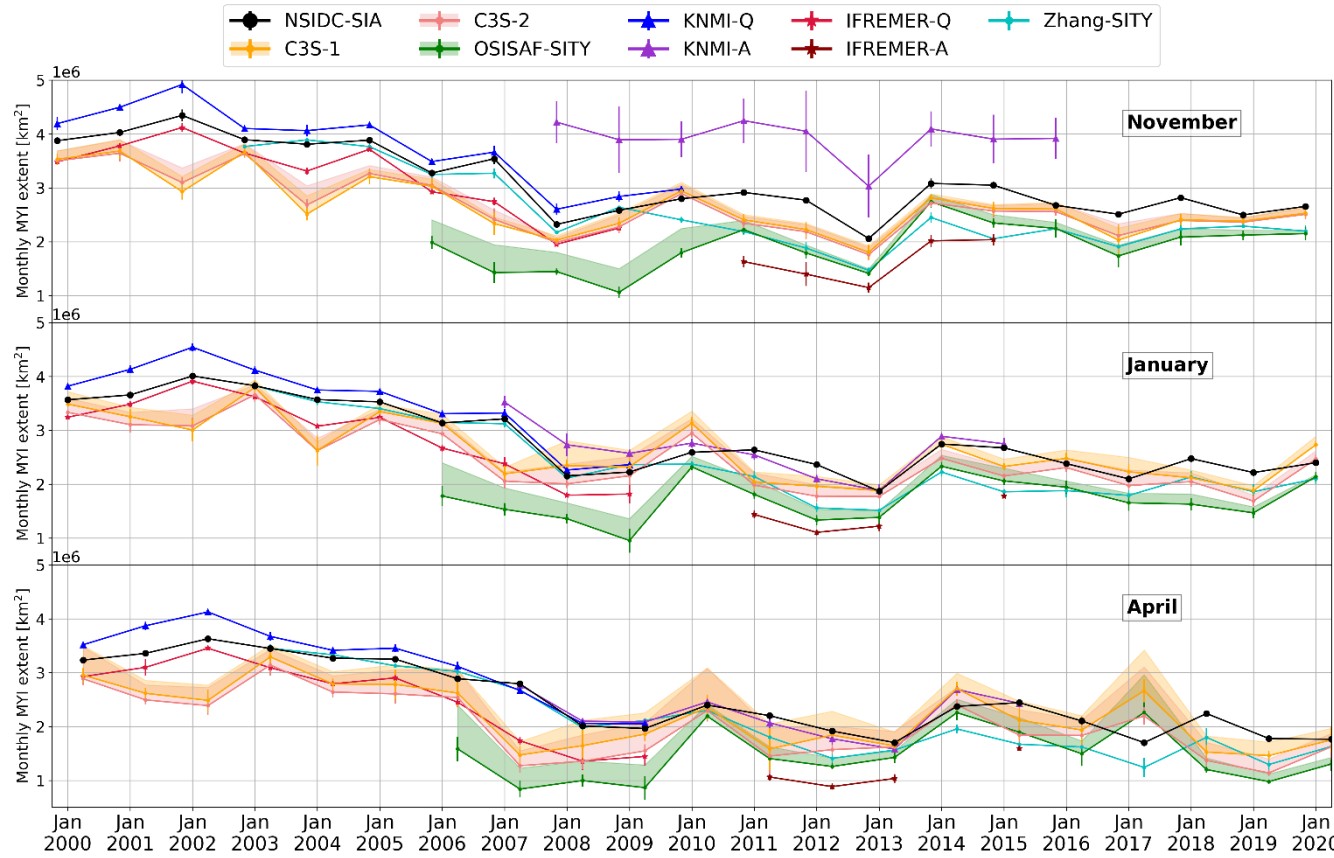

1020

**Figure 5: Monthly MYI extent of SITY products and NSIDC-SIA in November (top panel), January (middle panel) and April (bottom panel) from November 1999 to April 2020. The shaded area represents the ambiguous extent value for C3S-1, C3S-2 and OSISAF-SITY respectively. The error bar represents the range between maximum and minimum MYI extent in the month.**

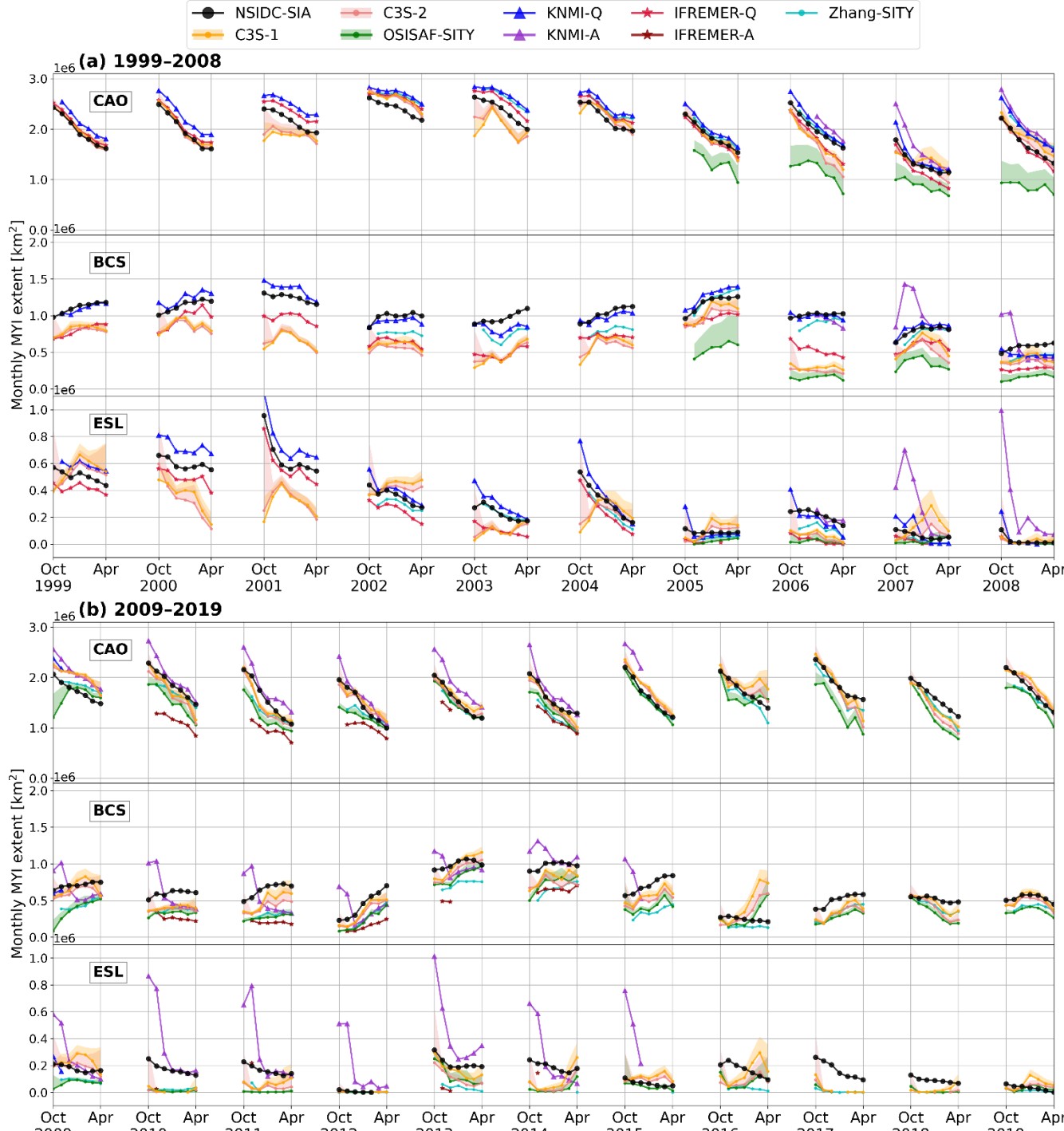

1025

**Figure 6: Monthly MYI extent of SITY products and NSIDC-SIA in the years (a) 1999–2008 and (b) 2009–2019 in the central Arctic Ocean (CAO), the Beaufort and Chukchi Seas (BCS) and the East Siberian and Laptev Seas (ESL) (see in Figure 2). The shaded areas represent the ambiguous extent values for C3S-1, C3S-2 and OSISAF-SITY respectively.**

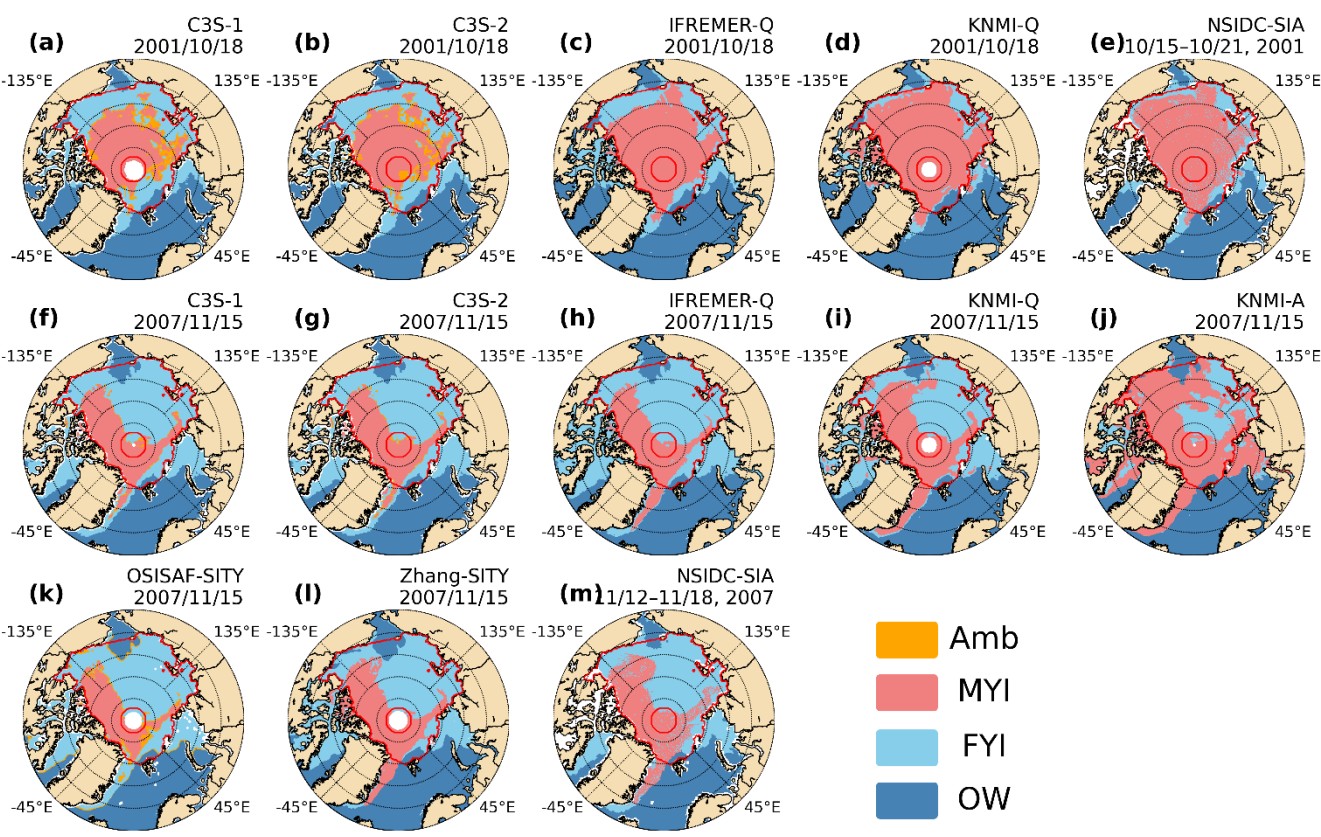

Figure 7: Arctic SITY distribution maps from daily SITY products and weekly NSIDC-SIA on October 18, 2001 (a–e) and November 15, 2007 (f–m).

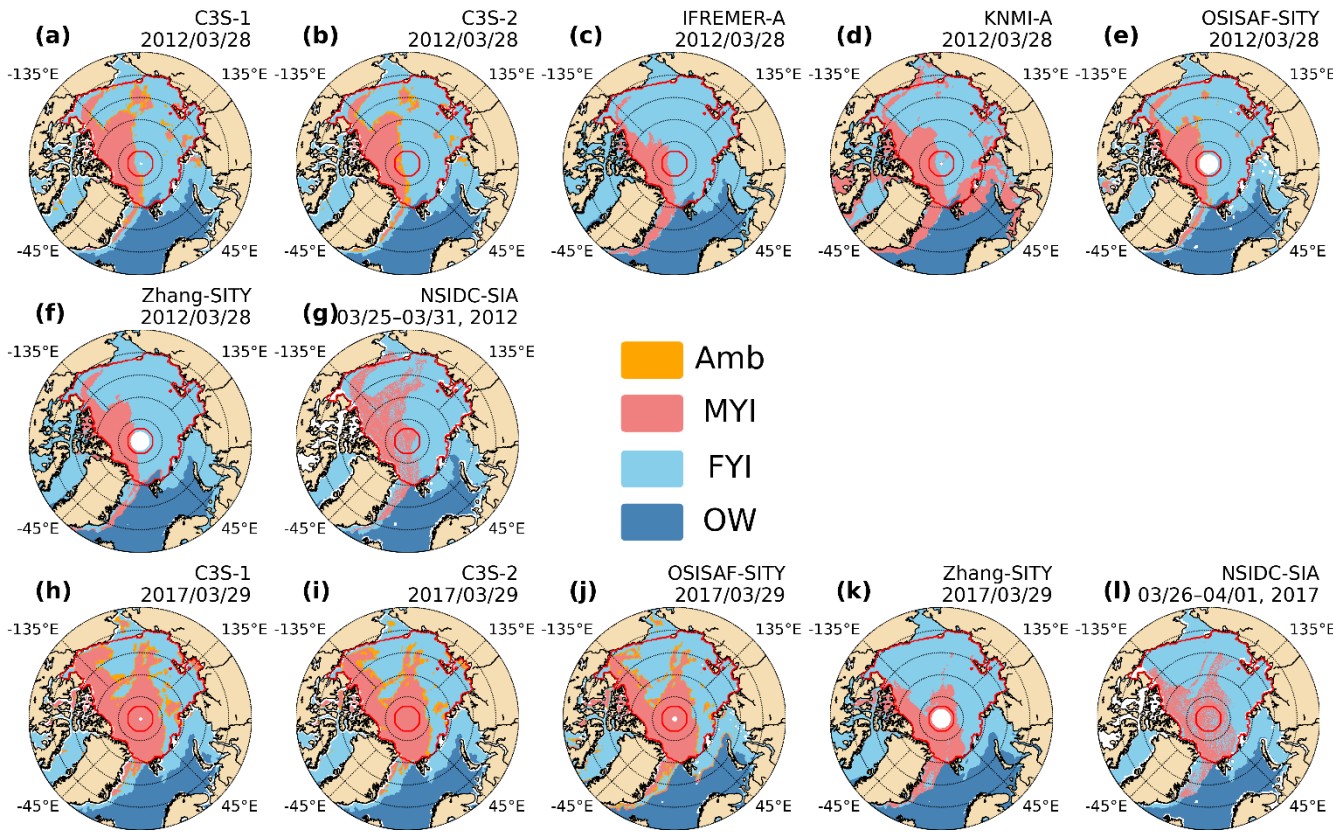

**Figure 8: Arctic SITY distribution maps from daily SITY products and weekly NSIDC-SIA on March 28, 2012 (a–g) and March 29, 2017 (i–l).**

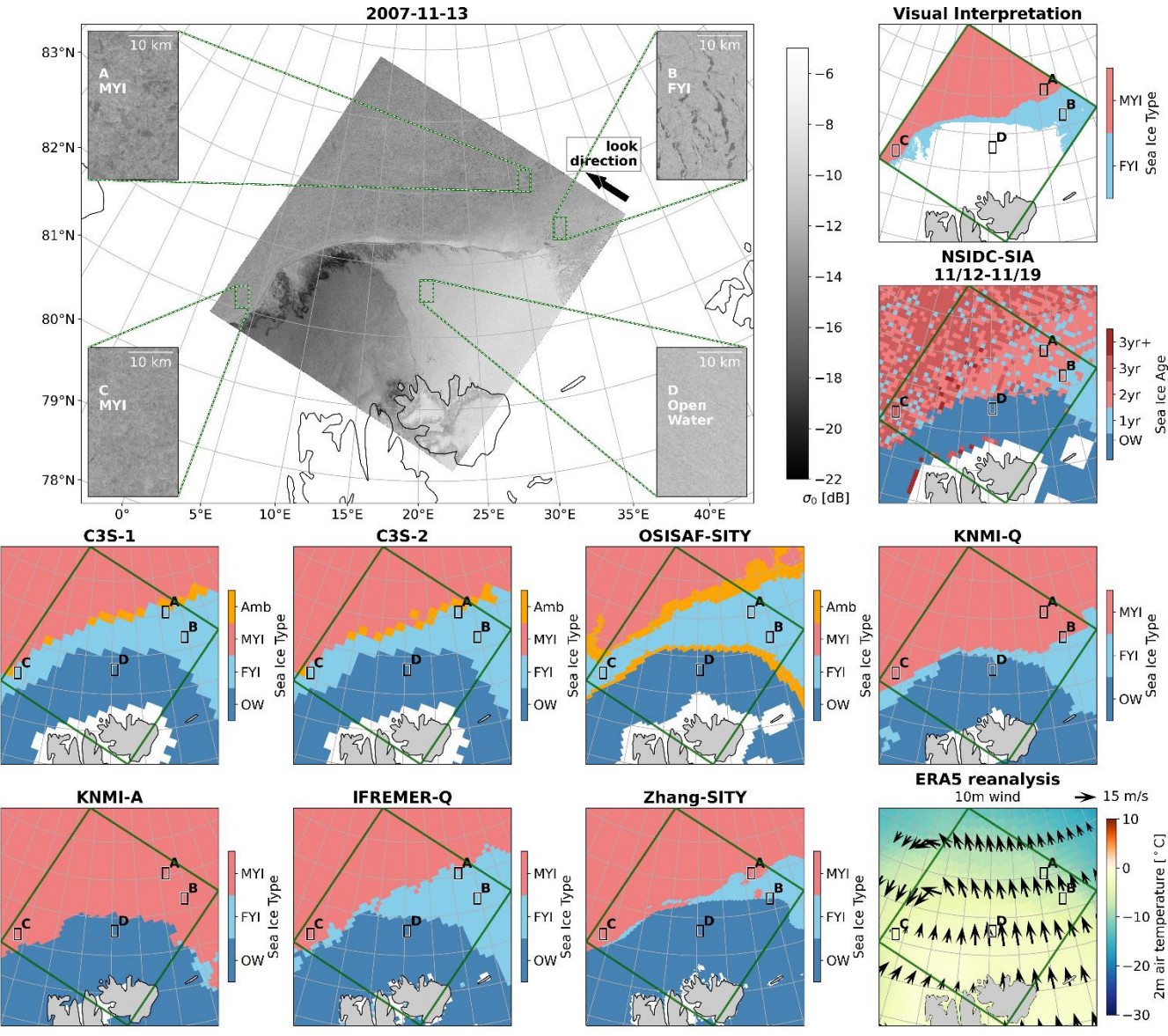

**Figure 9: RS-1 image, ice type distribution from seven SITY products (C3S-1, C3S-2, OSISAF-SITY, KNMI-Q, KNMI-A, IFREMER-Q and Zhang-SITY), weekly NSIDC-SIA product and visual interpretation result based on the SAR image, along with 2m air temperature and 10m wind from ERA5 reanalysis on November 13, 2007.**

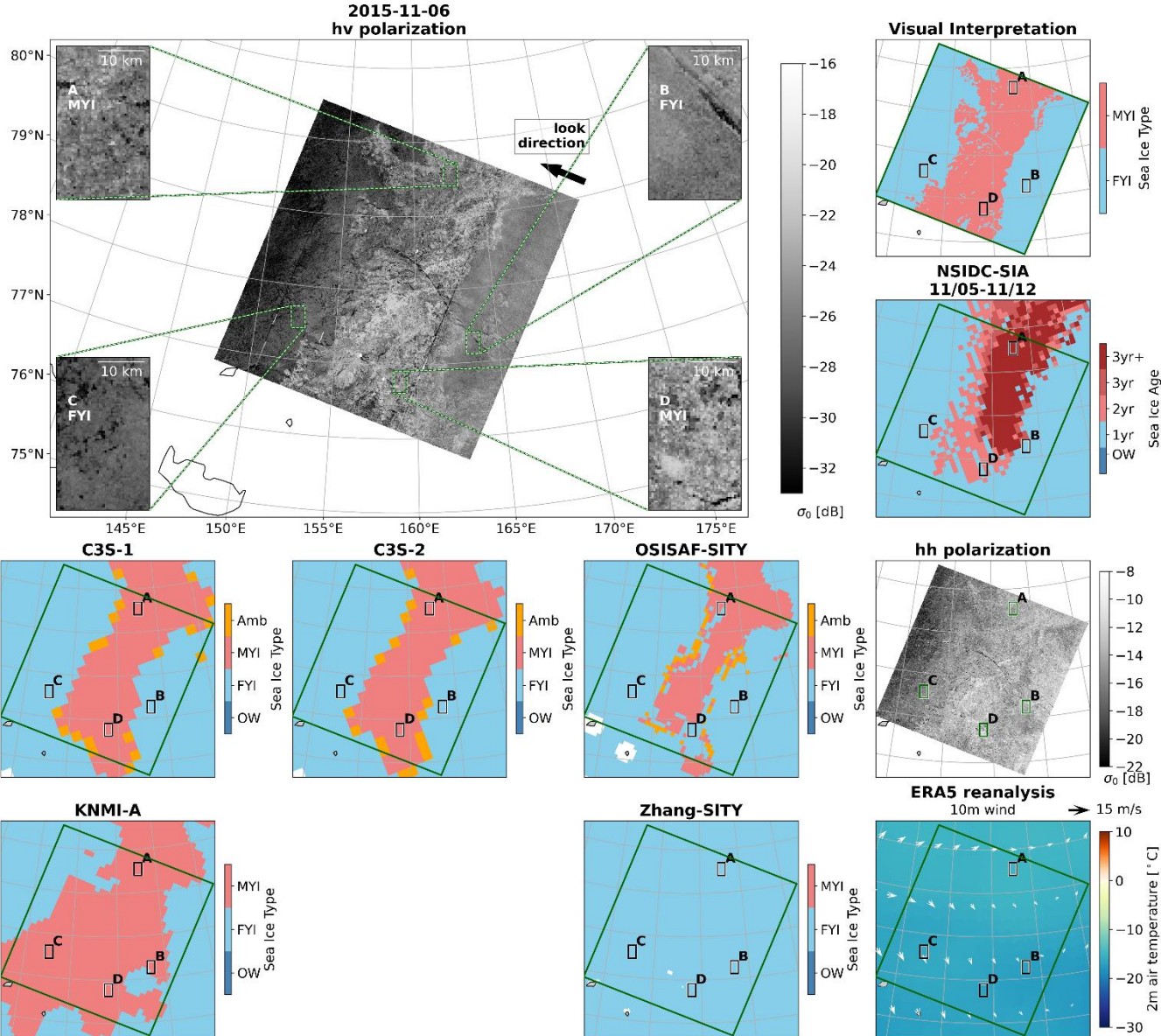

Figure 10: HV and HH polarization channels of S-1 image, ice type distribution from five SITY products (C3S-1, C3S-2, OSISAF-SITY, KNMI-A and Zhang-SITY), weekly NSIDC-SIA product and visual interpretation result based on the SAR image, along with 2m air temperature and 10m wind from ERA5 reanalysis on November 6, 2015.

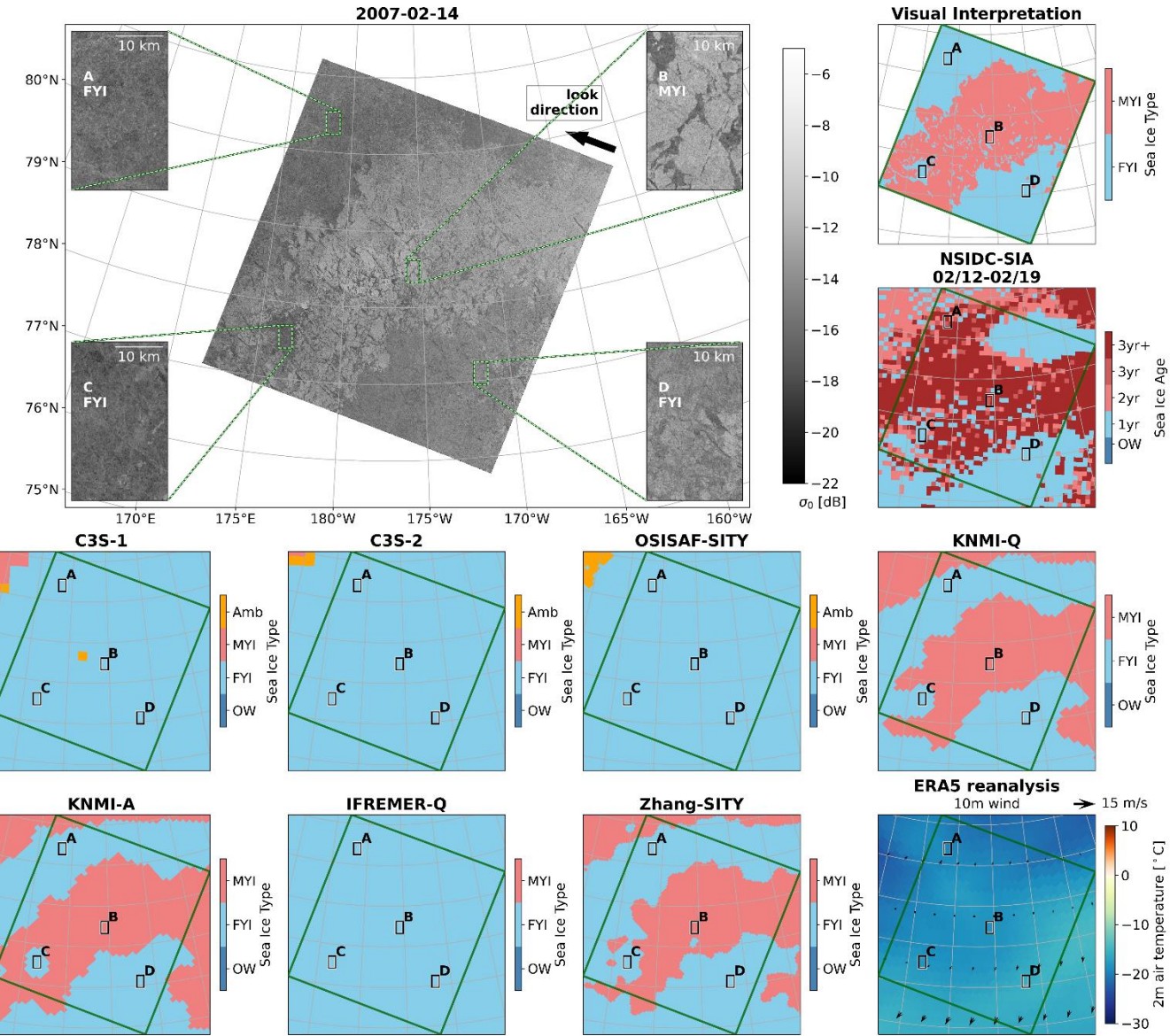

**Figure 11: RS-1 image, ice type distribution from seven SITY products (C3S-1, C3S-2, OSISAF-SITY, KNMI-Q, KNMI-A, IFREMER-Q and Zhang-SITY), weekly NSIDC-SIA product and visual interpretation result based on the SAR image, along with 2m air temperature and 10m wind from ERA5 reanalysis on February 14, 2007.**

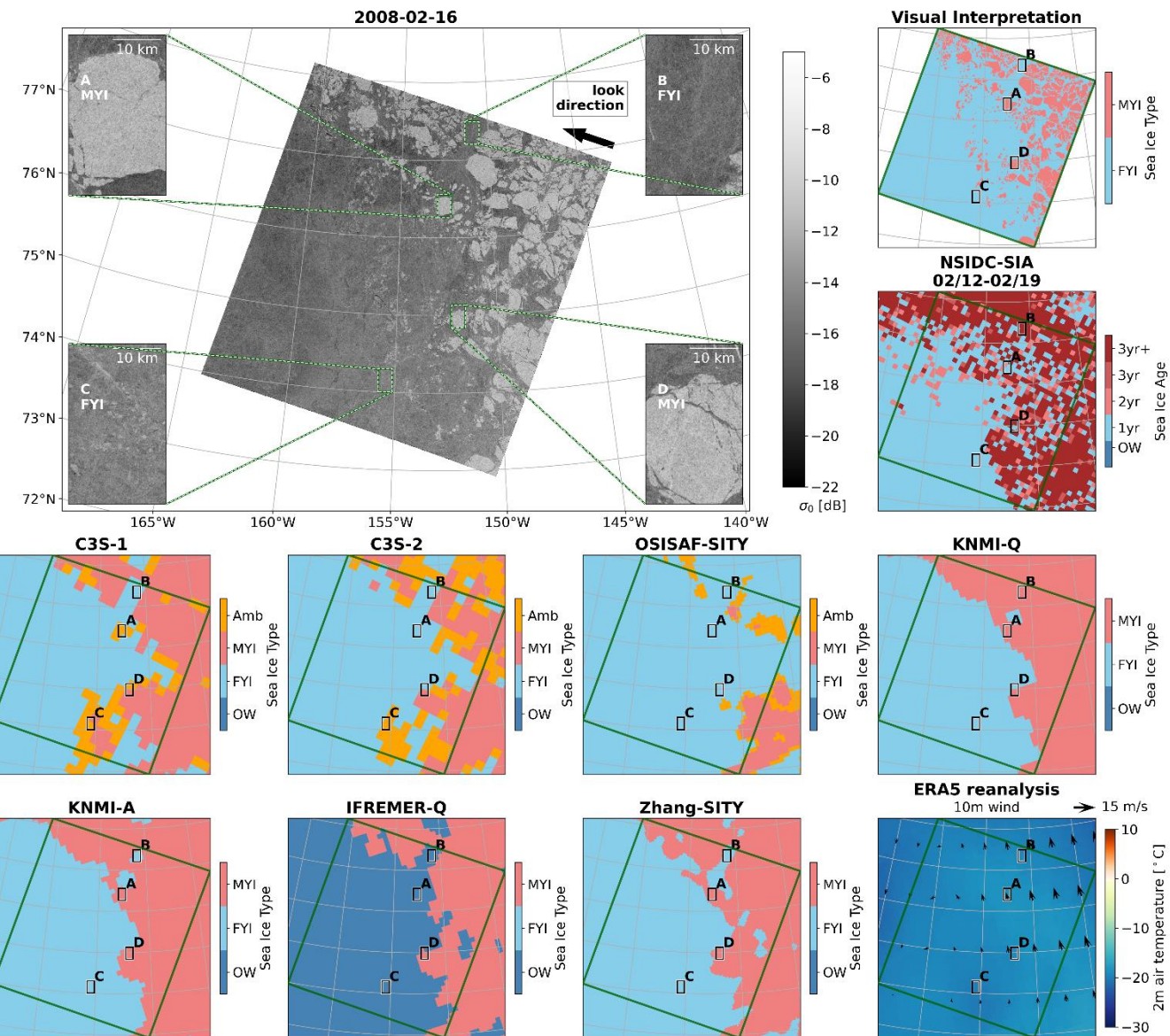

**Figure 12: RS-1 image, ice type distribution from seven SITY products (C3S-1, C3S-2, OSISAF-SITY, KNMI-Q, KNMI-A, IFREMER-Q and Zhang-SITY), weekly NSIDC-SIA product and visual interpretation result based on the SAR image, along with 2m air temperature and 10m wind from ERA5 reanalysis on February 16, 2015.**

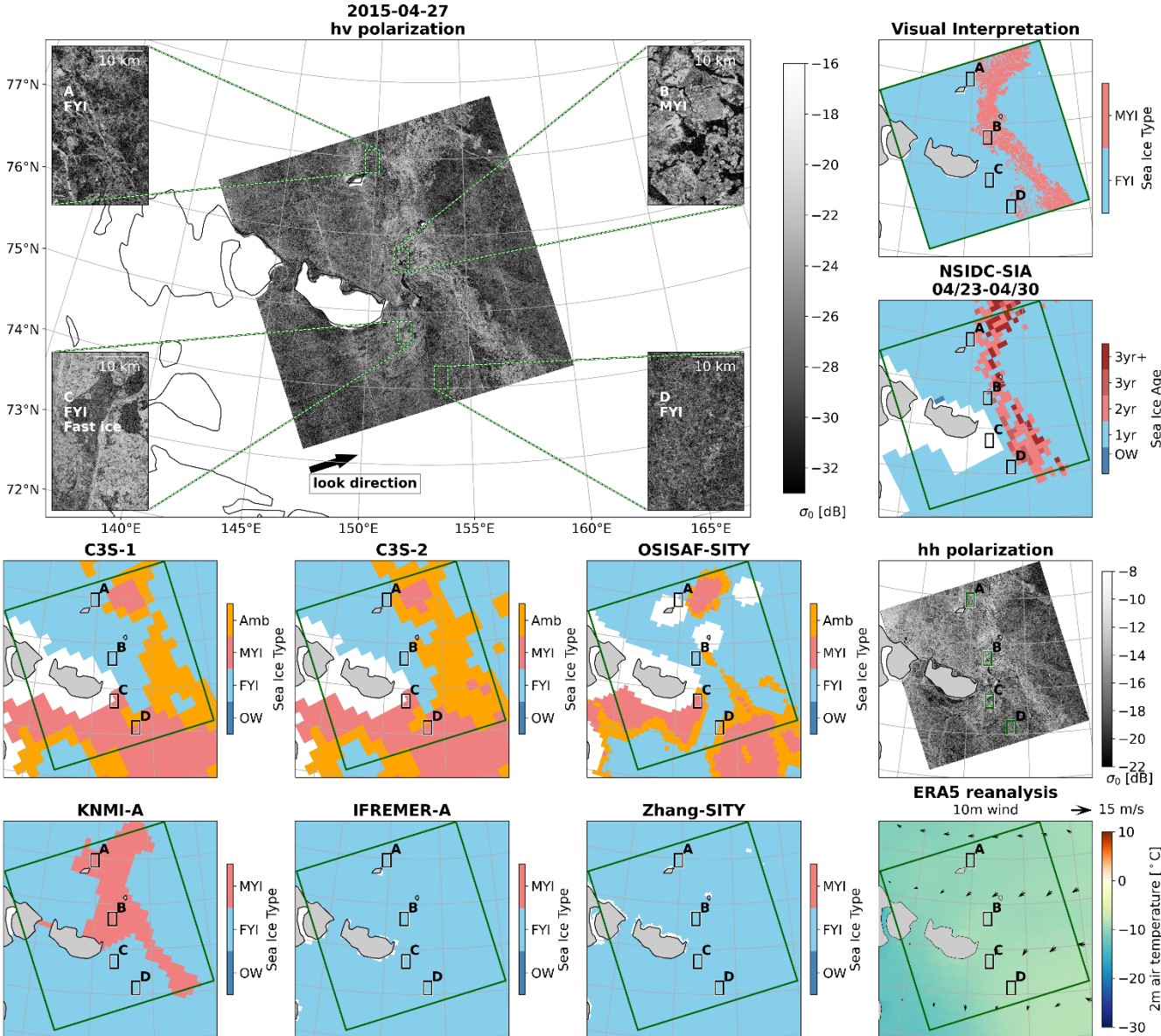

**Figure 13: HV and HH polarization channels of S-1 image, ice type distribution from six SITY products (C3S-1, C3S-2, OSISAF-SITY, KNMI-A, IFREMER-A and Zhang-SITY), weekly NSIDC-SIA product and visual interpretation result based on the SAR image, along with 2m air temperature and 10m wind from ERA5 reanalysis on April 27, 2015.**

1055

**Table A1: Specific information of the different sensors and channels used in the SITY products**

| Sensor | Temporal Coverage [YYYY/MM/DD] | Channels [GHz, pol] | | Footprint [km] | Incidence angle [degree] |
|---|---|---|---|---|---|
| SMMR | 1978/10/25–1987/08/20 | 18.0 | V, H | 41×55 | 50.2 |
| | | 37.0 | V, H | 18×27 | |
| SSM/I | 1987/09/07–2008/12/31 | 19.35 | V, H | 43×69 | 53.1 |
| | | 37.0 | V, H | 28×37 | |
| SSMIS | 2000/01/24–present | 19.35 | V, H | 42×70 | 53.1 |
| | | 37.0 | V, H | 27×44 | |
| AMSR-E | 2002/05/02–2011/12/04 | 18.7 | V, H | 14×22 | 55 |
| | | 36.5 | V, H | 7×12 | |
| AMSR2 | 2012/05/18–present | 18.7 | V, H | 14×22 | 55 |
| | | 36.5 | V, H | 7×12 | |
| ERS | 1991/08/01–2011/07/04 | 5.3 (C) | VV | 25×37 | 18–47 |
| QSCAT | 1999/06/19–2009/11/23 | 13.4 (Ku) | VV | 25×37 | 54.1 (VV), 46 (HH) |
| OSCAT | 2009/09/23–2014/02/20 | 13.5 (Ku) | VV | 25×37 | 57.6 (VV), 28.9 (HH) |
| ASCAT | 2006/10/19–present | 5.255 (C) | VV, HH` | 25×34 | 25–65 |

1060

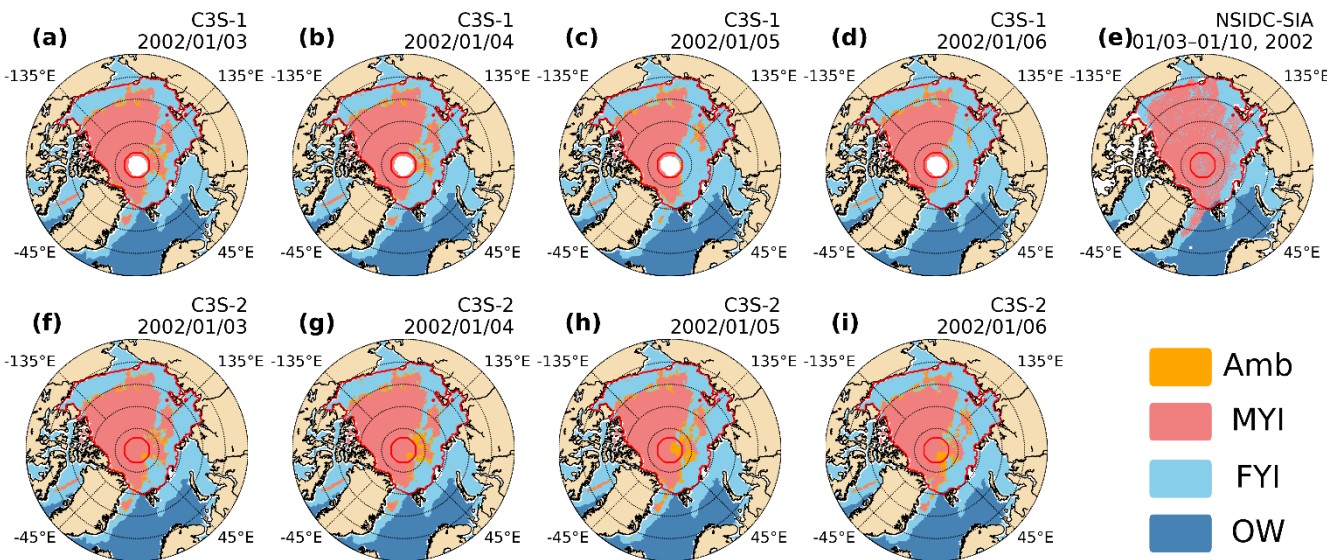

**Figure A1: Arctic SITY distribution maps from daily SITY product C3S-1 (a–d), C3S-2 (f–i) and weekly NSIDC-SIA (g) from January 3 to January 6, 2002.**