# Peer review of "Inter-comparison and evaluation of Arctic sea ice type products"

_The Cryosphere, 2022_

## Author Comment (AC1)

Response to RC 1:
(The reviewer comments appear in black, the responses are in blue and the proposed changes to manuscript are in ***bold italics***.)

**Summary**

This paper compares different sea ice type products currently available to the community. The products are based on passive microwave data, scatterometer data (C or Ku band), or a combination of both. The products have been developed empirically via training data. The type fields are inter-compared and evaluated against a widely-used sea ice age product and SAR retrievals. The products perform better in mid-winter than in early or late winter when melt/re-freeze may occur. Ku-band scatterometer generally is better at type discrimination. Combination of passive microwave and scatterometer data can yield better performance at times, but not in all situations.

**General comment**

This is a fairly comprehensive review of the primary sea ice type products available. There are notable differences in how the products are assembled, the input source data, and their performance in different conditions. Thus, this paper is a valuable contribution to the community be providing such an assessment. The paper is quite thorough and overall it does a good job in presenting the inter-comparison and evaluation of the products.

Reply:

Thank you for the thorough review. Your comments and suggestions are highly appreciated. For better comparison and evaluation of the sea ice type products, we will revise the manuscript from the following two aspects:

1) The "Data" section will be re-structured. This section will include four sub-sections: "2.1 Microwave remote sensing", "2.2 Sea ice type products" and "2.3 Other data". In section 2.1, specifications of the sensors and the satellite data will be introduced in a chronological order, with subsections of passive/active microwave remote sensing data. In section 2.2, theory of SIT classification will be introduced at the beginning, followed with the overall description of the respective SIT products in terms of grid size, projection, availability period, a summary of the satellite data used and the algorithm with necessary details. In section 2.3, sea ice age product (with evaluations from previous studies) and the SAR images will be described accordingly.

2) A new section of "Methods" will be added, which includes "3.1 Estimation of MYI extent" and "3.2 Interpretation of SAR imagery". We will modify the computation of MYI extent in the revision for consistent griding, projection among all the SIT products. In section 3.1, Information such as co-locating/re-griding the data and calculation of the MYI extent will be introduced. In section 3.2, the theory and characteristics of sea ice classification in SAR images will be introduced with references from previous studies and examples from our study. In addition, we will interpret the entire SAR images, consult with ice experts regarding the results, convert the sea ice classification results from ice types polygons to grided ice classification results, and eventually give quantitative evaluation results.

Besides, case studies will be presented in the chronological order with more discussions referring to the physical background and the algorithms of SIT products. Figures will be modified for better presentation. A thorough edit of the language style and grammar will be conducted. And all the references and citations will be double-checked and corrected accordingly.

Specific comments are below, but one overall comment is on the SAR data used for evaluation. In general, SAR is going to be the best "truth" for comparison. It is high resolution, so it can delineated even individual floes often. And it is all-sky, so retrievals of type are available anywhere the sensor collects imagery. However, the challenge with SAR is interpreting the imagery. The authors interpret the SAR imagery and classify various locations as a given ice type, but they don't give a particular rationale or provide references for their classification basis. Often, expert ice analysts interpret SAR fields for operational ice charts. They have deep experience in understanding the imagery and properly defining features. It appears the authors here classify the imagery themselves. This is okay, but I would like to see more substantial justification for their classification.

Reply: Thank you for the advice. In the revision, we will add a new section regarding the theory of SAR interpretation. The theory and characteristics of sea ice classification in SAR images will be introduced in the section "3.2 visual interpretation of SAR imagery".

Another weakness with the SAR comparison is that it is just a few scenes in selected regions and selected periods. And even within the SAR scenes, a few specific locations are picked out as "pure types" for comparison. Ideally, a full SAR image would be classified and compared. I know automated SAR classification algorithms for sea ice are troublesome, so I can understand the approach taken, but it results is a fairly ad hoc and qualitative evaluation. Since this paper is otherwise quite comprehensive, I won't request more evaluation, but ideally (perhaps in a future paper), it would be good to get classified SAR images – perhaps from an expert ice analyst at an operational ice center – and conduct a more comprehensive and quantitative evaluation of the ice type products.

Reply: Thank you for the advice. In the revision, we will interpret the entire SAR images, consult with ice experts, convert the sea ice classification results from ice types polygons to grided ice classification results, and eventually give the quantitative evaluation results.

A final note is that there is a need for a thorough copy edit for English language style and grammar. The issues are mostly minor – in particular, there are numerous missing articles ("the", "a", "an") – but they are widespread throughout the manuscript. I don't bother to point them out individually as they are too numerous, but they need to be addressed before final publication.

Reply: Thank you for the advice. We will go through the manuscript and conduct a thorough edit for the language style and grammer.

**Specific comments (by line number):**
11: The authors definite "sea ice type" as "SIT" here. This is fine and it is used consistently throughout the manuscript. However, as a sea ice scientist, "SIT" means "sea ice thickness" to me. And particularly with numerous thickness products coming out from altimeters, "SIT" is becoming quite common in the community to denote thickness. I can understand wanting to use an abbreviation and "SIT" makes sense for ice type, and the context is clear throughout the manuscript. So, I can't say it needs to be changed, but it might be something for the authors to consider. For me, every time I saw it, "thickness" popped into my mind first until I recalibrated. I can't think of another good abbreviation myself, but one could just use "type" or "Type" as a short-hand, instead of "SIT".

Reply: Thank you for the comment. We now use "SITY" instead to represent "sea ice type". All the abbreviations in the manuscript will be modified accordingly.

28-30: I'm struck by the use of more than author listed and then "et al." in the citations – i.e., "Comiso, Parkinson, et al., 2008". Generally, if there are more than two authors, just the first author is listed followed by "et al." – i.e., it would be "Comiso et al., 2008". In looking at The Cryosphere guidance for citations, I don't see anything that indicates two authors should be listed, so I'm not sure of the rationale. This seems to be done throughout the manuscript. (If there are only two authors, you list both, e.g., if it were "Comiso and Parkingson, 2008".) Not a big deal and I assume the copy editing will decide the proper citation format. I just haven't seen this before and it struck me as odd.
Reply: Thank you for the advice. We will carefully check all the references and correct the citations in the revision.

31-32: Be careful about terminology. "Thin" and "Young" ice are standard stage of development classifications. I think here you mean "thinner and younger" for FYI, and then "thicker MYI". I'm also not sure what you mean by "firm" in relation to MYI?
Reply: Agree. The sentence will be modified to:
*"The Arctic sea ice has been increasingly dominated by thinner and younger first-year ice (FYI) instead of thicker and older multiyear ice (MYI)."*

57: "ergodic" is an obscure word – I was not familiar with it. Based on my understanding after looking it up, I'm not sure it is used properly here. Regardless, I think a simpler word is appropriated here or I wonder if it is needed at all – "combined use of both data" is clear to me.
Reply: Agree. The word "ergodic" is deleted. "combination use of both data" will be replaced with "combined use of data".

62-63: "While ice chart…" is a confusing sentence – not sure what it is say. I would suggest revising.
Reply: Thanks. The sentence will be modified to
*"While ice chart is used as "ground truth" in some validation (Aaboe, Breivik, et al. 2016), some areas of MYI in the ice charts correspond to areas with MYI concentration of approximately 50% or greater"*

72: Just one example of grammar/style issues: "…are detailed investigated." – It should be "are investigated in detail."
Reply: Done.

107-109: Is AMSR-E used in the product? The description indicates only AMSR2 is used. So, why describe AMSR-E characteristics? Why not just describe AMSR2 characteristics?
Reply: AMSR-E is used in Zhang-SIT from 2002 to 2011. In the revision, we will add a new section "2.1 Microwave remote sensing data", where the satellite data will be introduced in a chronological order meanwhile with same level of details. The microwave radiometer data will be introduced as two series of dataset: one includes SSM/I and SSMIS, the other means AMSR-E and AMSR2.

109: Maybe another grammar/style issue: "working" is okay, but typically when describing sensors or satellites, "operating" or "collecting data" are more common.

"working" seems a bit colloquial here.
Reply: Agree. "working" is replaced with "operating".

147: This goes for all products, but noting here because NSIDC products have specified references that should be used. For SIA, it is:
Tschudi, M., W. N. Meier, J. S. Stewart, C. Fowler, and J. Maslanik. 2019. EASE-Grid Sea Ice Age, Version 4. [Indicate subset used]. Boulder, Colorado USA. NASA National Snow and Ice Data Center Distributed Active Archive Center. doi: https://doi.org/10.5067/UTAV7490FEPB. [Date Accessed].
This should be cited in the manuscript text and listed in the references. I see that the dataset website is noted in the Acknowledgment section, but where a reference is provided, it should be included in the manuscript proper, including the dataset DOI. I know all datasets do not provide a formal citation and/or DOI – for example for OSI-SAF, their recommended citation is simply: "The type dataset shall be referred to as the Sea ice type product of the EUMETSAT Ocean and Sea Ice Satellite Application Facility (OSI SAF, osi-saf.eumetsat.int)." If that is all that is provided, that is fine, though I would also say that the product ID (OSI-403-d) and version (if provided) should be included. The other datasets used should be cited to the extent they properly can be.
Reply: Agree. References for all the datasets used in the manuscript will be updated accordingly.

185-186: I think the potential for MYI increase could be explained better here. In practice, overall Arctic MYI cannot increase over the winter – it can only decrease via advection out of the Arctic. "Temporary" increases can happen within products due to divergence – e.g., a 100% MYI pixel diverging into two pixels with 50% ice each; if the threshold for detection is <50%, there will now be two pixels. And regionally, MYI can increase, both due to divergence or due to advection into the region from neighboring regions.
Reply: Thank you for the advice. The sentence will be modified as below:
*"However, it can temporarily or regionally increase due to ice divergence or advection from neighbouring regions (Kwok, Cunningham, et al. 1999)."*

191: This is discussed a bit more later, but this left me hanging: "why such a dramatic peak in the first half of winter?" Maybe provide a brief explanation and then say it will be discussed further later in the paper.
Reply: Thanks for your suggestion. The following sentence will be added to give a brief explanation:
*"…, which is due to misclassification of FYI as MYI in the peripheral seas of the Arctic and will be further discussed later in the paper."*

204: I would use "to" instead of a "-" because it looks like a minus sign. Or use an "em-dash" or "en-dash" with spaces on each side.
Reply: Agree. "-" is replaced with "to" in the sentence.

219: Figure 5 is mentioned quite cursorily here, but I notice the behavior of several products in BS during 2016-2017. That sticks out compared to other years and regions. Why was the performance so different?
Reply: Thanks for the comment. Explanations for the performance will be added in the

revision. This sentence will be modified as below (Note that we modify the abbreviations of each region in the revision)

*"Overall, the MYI extent in the CAO and ESL regions shows consistently negative trend, while that in the BCS region remains constant or is increasing. The former mainly results from the outflow of MYI to more southern areas…."*

224-225: Okay, the KNMI-SIT increase is mainly in the BS and ESS regions. But why? In general, this paragraph (223-229) feels like it needs to drill down a bit more and give more detail/explanation.
Reply: Thank you for the advice. We will give more explanations in the revision.

259: Kind of the same thing here. Okay, you have an overestimation of MYI, but that doesn't specifically explain the abnormal increase in MYI during 2016-2017. Why was the MYI overestimated in the one year versus others.
Reply: Thanks for the advice. We will specify the reasons in the revision.

265: How are cases selected? Were they ad hoc? Random? Was it simply availability of imagery? Or was there some physical rationale to select the scenes? I understand in general wanting different regimes and different time periods, but why those specific images on those specific days at those specific regions? In other words, what "different conditions" were you selecting for here?
Reply: Thank you for the advice. The images were selected based on the availability, time, region and overall SIT distribution. We will include sentences to explain the rationale of selecting the scenes.

268-271: Following from my general comment above, how were characteristics of the SAR images used for visual interpretation. What is the basis? There are no references here to justify the classifications.
Reply: Thank you for the comment. We will add references to justify the rationale of the classification. More specifically, in the new section "3.2 Visual interpretation of SAR imagery", the theory and characteristics of sea ice classification in SAR images will be introduced.

273-278: This paragraph illuminates the previous comment. The text is very "squishy". You say things like "appears to be MYI", "more likely to be MYI", "which could be interpreted as newly generated FYI". This is very qualitative and seemingly tentative. I think maybe you just need to say why something "appears to be MYI" – what is in the SAR image that leads to that conclusion and what is the basis for that?
Reply: Thank you for the advice. We will modify the words in the revision. In addition, as mentioned in previous replies, the theory and characteristics of sea ice classification in SAR images will be introduced in the new section 3.2.

288-293: Same here. You have "could be identified as MYI", "are a typical feature of FYI".
Reply: Thanks. Such expressions will be modified in the revision similarly as we mentioned in the above replies.

299: I guess there is a thematic reason for the order – looking at early and late winter as "edge" cases, but it seems more logical to order these subsections chronologically: early winter, mid-winter, and then late winter.

Reply: Thank you. These cases will be presented in the chronological order as suggested.

315-319: Again, very qualitative.
Reply: Thanks. Quantitative results such as overall accuracy of the respective SIT products on each case will be added based on the visually interpreted results of each image.

391-446: I can see the logic of discussing the methods here – you are linking them to the performance assessed previously. However, to a large extent, this feels like it should go with the data product descriptions in Section 2.1. I guess moving this there would make that section rather long. But I kind of feel like I get to here and I finally understand how the type products are created – after all of the evaluation and comparison. I'll leave it to the authors to decide if this fits better in 2.1 or should stay here. Or maybe, put some description in 2.1 and then the relation to the product evaluation here.
Reply: Thank you for the thoughtful comments. In the revision, we will provide more details of the algorithm used in the respective SIT products in the section "2.2 sea ice type products" and put the description of satellite data to section 2.1 "Microwave remote sensing data".

400: "[55]"? Is this a numbered reference?
Reply: Yes, We will correct the citation in the revision.

434-436: Melt affects the performance in early and late winter. But melt also basically makes the algorithms ineffective in spring and summer. That is implied, but never really stated explicitly it seems.
Reply: Thank you for the advice. We will give more explicit statement in the beginning of the new section "2.2 sea ice type products".

Figure 5: Following up from above, I'm struck by the notable increase in many products in BS during 2016-2017. That is not noticeable in another region or year. There is some discussion, in relation to OSI-SAF in Figure 7, but the text doesn't really discuss this. I think this deserves some explanation in the text.
Reply: Thanks. We will add explanations in the text regarding the notable increase in BS.

Figure 5: What is the shaded green region that accompanies OSISAF? Maybe it is in the main text and I missed it, but regardless, it should also be included in the caption for the figure.
Reply: The upper and lower lines of shaded green region represents the MYI extent calculated under the assumptions of 1) regarding the "Amb" class as MYI, and 2) regarding the "Amb" class as FYI. We will add such information in the caption.

Figure 6-7: This is a style/aesthetics thing, but the beige/brown for the OW seems odd. Such a color is more commonly used for land, and definitely not for ocean. I would suggest considering a different OW color – just swapping the land color (light gray) for the ocean color, would be more logical to me. Of course, you want to make sure that the colors contrast and are clearly delineated. But a good solution other than beige/brown for OW seems possible.
Reply: We will change the colors as suggested in the revision.

---

## Author Comment (AC2)

Response to RC 2:
(The reviewer comments appear in black, the responses are in blue and the proposed changes to manuscript are in ***bold italics***.)

General comment

This manuscript presents the inter-comparison of various SIT products from microwave remote sensing data. The performance of the SIT products was evaluated and the causes of differences in the products were analyzed. SIT has been used as an important information in research for global climate change and future prediction. Therefore, a comparative study on the performance of the operationally used SIT products is of high importance.

The manuscript is well written, and appropriate tables and figures are used to explain the results. It seems very meaningful to analyze the comparison results in time and space. However, in order for this manuscript to be published in the Cryosphere, more descriptions should be added about the data and methodology used in the study (requires a section on methodology). More discussion of the results is required.

Reply:

  Thank you for the thorough review. Your comments and suggestions are highly appreciated. For better comparison and evaluation of the sea ice type products, we will revise the manuscript from the following two aspects:

1) The "Data" section will be re-structured. This section will include four sub-sections: "2.1 Microwave remote sensing", "2.2 Sea ice type products" and "2.3 Other data". In section 2.1, specifications of the sensors and the satellite data will be introduced in a chronological order, with subsections of passive/active microwave remote sensing data. In section 2.2, theory of SIT classification will be introduced at the beginning, followed with the overall description of the respective SIT products in terms of grid size, projection, availability period, a summary of the satellite data used and the algorithm with necessary details. In section 2.3, sea ice age product (with evaluations from previous studies) and the SAR images will be described accordingly.

2) A new section of "Methods" will be added, which includes "3.1 Estimation of MYI extent" and "3.2 Interpretation of SAR imagery". We will modify the computation of MYI extent in the revision for consistent griding, projection among all the SIT products. In section 3.1, Information such as co-locating/re-griding the data and calculation of the MYI extent will be introduced. In section 3.2, the theory and characteristics of sea ice classification in SAR images will be introduced with references from previous studies and examples from our study. In addition, we will interpret the entire SAR images, consult with ice experts regarding the results, convert the sea ice classification results from ice types polygons to grided ice classification results, and eventually give quantitative evaluation results.

  Besides, case studies will be presented in the chronological order with more discussions referring to the physical background and the algorithms of SIT products. Figures will be modified for better presentation. A thorough edit of the language style and grammar will be conducted. And all the references and citations will be double-checked and corrected accordingly.

Specific comments

Abstract: Specify the names of the SIT products (algorithms) analyzed in this study.

Reply: Thanks for the advice. We will modify the sentence as follows.
*"This study analyzed nine daily SITY products from five SITY retrieval approaches covering the winters from 1999 to 2018, namely OSISAF-, C3S-, KNMI-, IFREMER- and Zhang-SITY."*
(Note that sea ice type is abbreviated as SITY in the revision)

Line 27: Please state clearly why sea ice is a sensitive indicator of climate change.
Reply: Thanks. The sentence will be modified as follows in the revision.
"*Sea ice is an important component of the earth system. Due to its high albedo, sea ice reflects more solar radiation than the ocean. Because of the ice-albedo positive feedback, sea ice loss and decreased surface albedo (increased absorption of solar radiation) enhance each other, making sea ice a sensitive indicator of climate change.*"

Line 29: It would be nice if it quantitatively indicated how much the thickness and volume of sea ice decreased.
Reply: Thanks. How much the thickness and volume of sea ice decreased will be added, using quantities from the references.

Line 35-37: Please specify how sea ice patterns affect Arctic and mid-high latitude regions and how they affect Arctic ecosystems.
Reply: Thanks. We will modify the sentence accordingly in the revision.

Line 68: The authors did three scientific questions, but the second question (how we choose SIT product for different applications) is lacking in discussion.
Reply: Thanks. How we choose SIT product for different applications will be added to the discussion in the revision.

Line 83: How are the microwave scattering and radiometric characteristics of MYI and FYI different?
Reply: Thank you for your advice. We will revise the manuscript as suggested.
    In the "Introduction" section, we will include sentences about the physical background for microwave SITY classification as mentioned above. The sentences included additionally start with:
*"On one hand, brightness temperatures (Tbs) of MYI tend to be lower than that of FYI because of high loss of radiation caused by scattering when going through the bubbly layer in the sub-surface of hummock area (Sinha and Shokr, 2015). Such difference depends on the wavelength of the radiation with respect to the typical dimensions of the bubbles (they should be comparable for the loss to be effective). On the other hand, due to the high volume scattering and low scattering loss, MYI have relatively higher backscatter than FYI at the same frequency (Carsey, 1992)."*

References:
Sinha, N. K., & Shokr, M. (2015). Sea ice: physics and remote sensing. John Wiley & Sons.
Carsey, F. D. (Ed.). (1992). Microwave remote sensing of sea ice. American Geophysical Union.

    In the "Data" section, before introducing the individual sea ice types, we will include a paragraph that describes the physical background. The paragraph starts with:

*"Microwave radiometer and scatterometer are used to discriminate MYI and FYI due to their distinctive signatures. Microwave radiometer Microwave radiometers measure the upwelling radiation emitted by the Earth in terms of brightness temperature (Tb), which is linearly proportional to the physical temperature and microwave emissivity of the observed object…"*

Line 77: Each product has a different grid size. It should be explained how it was dealt with in the comparative evaluation.

Reply: Thanks for the advice. We will modify the computations in the revision to account for the factor of different grid size. SITY products in polar-stereographic projection will be firstly regridded to the EASE grid before the computation of MYI extent. Details of such will be presented in the new Section"3 Methods" (3.1 Estimation of MYI extent).

Line 78-143: Please describe in more detail how FYI and MYI are distinguished due to which characteristics in each SIT algorithm. For example, if a SIT product is produced based on PR and GR, an explanation is required for the differences between the values of PR and GR of ice types and why the differences occur.

Reply: Thank you for the advice. In the revision, the section of "Data" will be modified as follows. The new section 2.1 will be entitled with "Microwave remote sensing data", where the specification of the sensors and the satellite data used in all the SIT products be will introduced. Definition (and equation) of PR and GR will be introduced in this section. The new section 2.2 will be "sea ice type products", where the grid resolution and details of the algorithm used in each SIT product will be introduced.

Line 150: NSIDC-SIA was used as reference data. How accurate is NSIDC-SIA?

Reply: Thanks. In the revision, references regarding the evaluation of NSIDC-SIA will be added and summarized.

Line 153: How and what information was retrieved from the SAR images for the SIT products evaluation should be described.

Reply: To answer the questions here, the sentence will be modified as below.

*"Radarsat-1 (referred to as RS-1) and Sentinel-1 SAR images were both visually interpreted in terms of sea ice type classification and used for validation."*

In the revision, we will add a new section regarding the theory of SAR interpretation. The theory and characteristics of sea ice classification in SAR images will be introduced in the section "3.2 visual interpretation of SAR imagery". In addition, we will interpret the entire SAR images, convert the sea ice classification results from ice types polygons to grided ice classification results, and eventually give the quantitative evaluation results.

Line 173: Is it the result of this study that different SIT distribution patterns were found in the regions selected by the authors?

Reply: It is a misleading sentence here. We will delete the sentence in the revision.

Line 185-186: Is 'divergent movements' the only cause of increase in the MYI extent?

Reply: "Advection from neighbouring regions" could also lead to increase in the MYI extent. The sentence will be modified as follows in the revision.

*"However, it can temporarily or regionally increase due to ice divergence or advection from neighbouring regions (Kwok, Cunningham, et al. 1999)."*

Line 263: The authors compared SIT daily products with the SAR images. It is necessary to discuss the comparison between the image captured at a specific time and the daily product.
Reply: Thank you for the advice. In the revision, the impact of such will be discussed in the comparison.

Line 263: The authors identified the distribution of MYI by visually analyzing the SAR image. It would be better if MYI could be determined by quantitatively analyzing backscattering or textures from the SAR images.
Reply: Thank you for the advice. In the revision, we will add a new section regarding the theory of SAR interpretation. The theory and characteristics of sea ice classification in SAR images will be introduced in the section "3.2 visual interpretation of SAR imagery". In addition, we will interpret the entire SAR images, convert the sea ice classification results from ice types polygons to grided ice classification results, and eventually give the quantitative evaluation results.

Line 368-371: How are the input parameters affected by atmospheric factors and surface features? More discussion is needed.
Reply: Thank you for the advice. We further investigate the magnitude of cloud liquid water values (atmospheric factors) and its impact. It turns out that variation of the cloud liquid water path has little impact on GR. We will therefore remove this sentence in the revised manuscript. Regarding the impact of surface features, more discussion will be added. The sentences in the next few lines will be modified as below:
*"In the beginning and ending stage of winter, the variability of $GR_{37v19v}$ can be significant when air temperature exhibits warm-cold cycles which triggers wet-dry cycles or melt-refreeze cycles of snow (Ye et al., 2016a 2016b; Voss et al., 2003), or when wet/thick precipitation suddenly appears (Voss et al., 2003; Rostosky et al., 2018). This can partly explain"*

Line 393: Explain clearly about the training dataset.
Reply: Thanks. To better explain the training dataset, we will rephrase the sentence as below:
*"Some use fixed threshold classification algorithm, while others employ dynamic thresholds, which may vary with time, region and the satellite sensors"*

Technical comments
Line 157: SAR Wide B → SAR Wide Beam
Reply: Done

Line 199: What does (2000) mean?
Reply: It is a typo here. The reference and citation will be corrected.

Line 402: Is [55] a reference number?
Reply: Yes. The citation will be corrected.

Figure 4: Delete 'Jan' on the horizontal axis.
Reply: Done

---

## Author Comment (AC3)

Response to RC 3:
(The reviewer comments appear in black, the responses are in blue and the proposed changes to manuscript are in **bold italics**.)

Comments are assigned to co-authors that we think could help to sort out.

Summary:

A large, albeit shrinking portion of the Arctic Ocean sea ice cover is made of multiyear ice (MYI) that has survived at least one summer melt season. In order to more accurately assess the trend in Arctic Ocean MYI cover and the coverage of first-year ice, and to more reliably use these ice type fractions in other research areas, such as sea ice thickness retrieval, it is important to evaluate the existing sea ice type products. This study is an attempt into this direction. Nine different sea ice type products based on five different algorithms are compared with the NSIDC sea ice age data set and the MYI extent derived from it as well as with a set of five qualitatively interpreted satellite synthetic aperture radar (SAR) images. Time series of the MYI extent at daily and monthly temporal resolution are shown, inter-compared and discussed qualitatively in the light of the different algorithms, their potential limitations and post-processing steps. The performance of the different products is compared for specifically selected sub-regions of the SAR images.
I have a number of concerns with this manuscript which I summarize in my general comments and detail in my specific comments.
I also would like to note that the manuscript is difficult to read because of quite a number of strange formulations and problems with English grammar.
Reply:
    Thank you for the thorough review. Your comments and suggestions are highly appreciated. For better comparison and evaluation of the sea ice type products, we will revise the manuscript from the following two aspects:
1) The "Data" section will be re-structured. This section will include four sub-sections: "2.1 Microwave remote sensing", "2.2 Sea ice type products" and "2.3 Other data". In section 2.1, specifications of the sensors and the satellite data will be introduced in a chronological order, with subsections of passive/active microwave remote sensing data. In section 2.2, theory of SIT classification will be introduced at the beginning, followed with the overall description of the respective SIT products in terms of grid size, projection, availability period, a summary of the satellite data used and the algorithm with necessary details. In section 2.3, sea ice age product (with evaluations from previous studies) and the SAR images will be described accordingly.
2) A new section of "Methods" will be added, which includes "3.1 Estimation of MYI extent" and "3.2 Interpretation of SAR imagery". We will modify the computation of MYI extent in the revision for consistent griding, projection among all the SIT products. In section 3.1, Information such as co-locating/re-griding the data and calculation of the MYI extent will be introduced. In section 3.2, the theory and characteristics of sea ice classification in SAR images will be introduced with references from previous studies and examples from our study. In addition, we will interpret the entire SAR images, consult with ice experts regarding the results, convert the sea ice classification results from ice types polygons to grided ice classification results, and eventually give quantitative evaluation results.

Besides, case studies will be presented in the chronological order with more discussions referring to the physical background and the algorithms of SIT products. Figures will be modified for better presentation. A thorough edit of the language style and grammar will be conducted. And all the references and citations will be double-checked and corrected accordingly.

General comments:

GC1: As the authors state, this is one of the first (kind of) comprehensive evaluation of sea ice type products. This calls for provision of a solid physical background of the sea ice and its snow cover as relevant for its remote sensing using active and passive microwave instruments. This element is missing and jeopardizes the usefulness of the entire manuscript.

Reply: Thank you for your advice. The physical background of sea ice type classification from passive and active microwave remote sensing will be added in the revision. Such descriptions will be presented in the following sections: 1) section "2.2 sea ice type products", where the theory of sea ice classification will be introduced before all the SIT products; 2) section "3.2 visual interpretation of SAR imagery", which is a subsection of "3 Methods", and presents the physical background and sea ice scattering characteristics in SAR image.

GC2: The description of the input satellite data and the algorithms used in the products as well as in the one major evaluation data set used is very heterogenuous and not complete for the understanding of the manuscript and its results. At least two products (NASA-Team MYI concentration and ECICE MYI concentration) are missing in addition.

Reply: Thank you for the advice. To better illustrate the input satellite data and algorithms used in the products, we restructured the Section of Data. In the revised manuscript, Section "2.1 Microwave remote sensing data" describes the satellite data, whereas Section "2.2 Sea ice type products" provides details of the algorithms used.

As for the two products you mentioned, since this study focuses on inter-comparison of sea ice type products that tells ice type classifications without information on the specific fraction. We therefore did not include NASA-Team and ECICE MYI concentration products. In order to the clarify the focus, we will modify the sentences in the section "Introduction", which reads as below:

*"There exist different algorithms which either provide a fractional MYI/FYI coverage or assignment of one or the other ice type (e.g. MYI and FYI) to a grid cell. The former, referred to as sea ice type concentration (SITC) algorithms, includes algorithms such as the NASA Team and ECICE algorithm …The latter, referred to as SITY algorithms, include …"*

GC3: The inter-comparison contains, if at all, little quantitative results. The results often appear to be quite hypothetical. As I see it, there are two main reasons for that. At first, the NSIDC sea ice age data set used as the main evaluation data set requires an evaluation that justifies its usage for the purpose of this manuscript. In addition, there is a methodological inconsistency behind comparing daily sea ice type products with weekly sea ice age data. Secondly, the SAR images used are only interpreted in a qualitative way. With that they can be used as a means for a consistency check of the general performance of the sea ice type products - but only within the error margin proposed by this manual interpretation. Both together clearly reduces the value of this

manuscript, which has the character of a pure, qualitative inter-comparison study with little in-depth recommendations resulting from it for i) which product to pick and ii) how to improve which product in which way.

Reply: Thank you for your advice. In the revision, we will include more quantitative results and analysis in the manuscript. Regarding the two points you mentioned in the comments. 1) the NSIDC sea ice age data is used for overall comparison. References regarding the evaluation of the sea ice age product will be added. In addition, weekly MYI extent will be calculated from sea ice type products for consistent comparison. 2) in the revision, we will interpret the entire SAR images, convert the sea ice classification results from ice types polygons to grided ice classification results, and eventually give the quantitative evaluation results (e.g. overall accuracy for each SAR image).

For the manual interpretation, the theory and characteristics of sea ice classification in SAR images will be introduced in the section "3.2 visual interpretation of SAR imagery". In addition, we will conduct the interpretation and consult with sea ice experts who are experienced in SAR interpretation for more reliable results.

GC4: The discussion of the results is not well linked to the existing literature.

Reply: Thank you for the advice. We will add more discussions that are linked with existing literature.

Specific comments (contain some typos / editoral comments):

Abstract:

- I recommend that you consider to find and use a different acronym for sea ice type because I find "SIT" very often used as an acronym for sea-ice thickness. A possible alternative could be SITY. Or, since "type" is not really that long compared to the words, e.g. thickness or concentration, you might also consider write the full expression all the time. But "SIT" is a bit unfortunate.

Reply: Agree. We use "SITY" as the abbreviation for "sea ice type" in the revised manuscript.

- I also recommend that you very briefly describe the various products named in the abstract. Perhaps they can be categorized into those products that rely solely on C-Band or Ku-Band data and/or products that use both active and passive microwave data? Please check the maximum allowed length of the abstract and perhaps delete details towards the end for more clarity of what types of products you did compare.

Reply: Thank you for the advice. We will categorize the sea ice type products into 1) and 2) those using solely active/passive microwave data and 3) those with combined microwave data.

- I recommend to state upfront that by "sea ice type" you merely refer to multiyear ice and first-year ice. As you know, there is a number of other sea-ice types which you, however, not appear to take into account.

Reply: Agree. On one hand, we will modify the first sentence in the last paragraph of "Introduction" Section. The sentence is modified to "… give comprehensive evaluations on the identification of MYI and FYI". On the other hand, in the new section of "Method", we will give clear statement that this paper focuses on the classification of MYI and FYI.

- L13/14: "towards sea ice ... images" --> "against a sea ice age product and compared with five Synthetic Aperture Radar images"
Reply: Done.

- While you write in Lines 14/15 about results found at daily and monthly temporal resolution it is not clear whether all products used come at daily temporal resolution. I also note that the sea ice age data set comes at weekly temporal resolution.
Reply: Agree. We will specify the temporal resolution in the abstract. In the revision, we will mention "… nine daily SITY products", "… a weekly sea ice age product".

- L14/15: Please also see my over-arching comment to the conclusions.
Reply: Thank you for the comment. We confirm that the numbers are correct. For the exceptional large difference, we will add discussions in the Results section.

- You might want to re-phrase "anomalous fluctuations" because it is not clear what you mean by that in the context of an underestimation (Line 17).
Reply: Agree. The sentence is modified to
*"…Zhang-SITY shows underestimation of MYI with relatively large fluctuations"*

- Under (3) you write about details with respect to the classification (Line 23). Is the retrieval of all products investigated based on a classification approach?
Reply: Thank you for the comment. The retrieval of all products are not based on one classification approach. In the revised manuscript, we give more detailed description about the classification algorithms used in the SITY products. This study focuses on the comparison and evaluation of current SITY products. Further investigation of classification algorithms could be done in future studies.

- I have the feeling that the "Additionally, the change of separation pattern ... SIT method" (Lines 24/25) could be deleted for the sake of having more room for the above-mentioned suggestions.
Reply: Done.

Lines 41-57: I suggest to better structure this paragraph and in addition provide more background information. Specifically I recommend to
i) Tell the reader that by sea ice type discrimination you are referring to distinguishing between FYI and MYI;
Reply: Agree. We add descriptive sentences/phrases at the beginning of this paragraph (second and third sentences). The modified sentences are:
*"…microwave satellite data. Among them, most algorithms focus on the discrimination of MYI and FYI. These algorithms identify SITY (i.e. the discrimination of MYI and FYI in this study) based on the distinct radiometric and scattering characteristics of different ice types."*

ii) Write what the fundamental differences in the physical properties of these ice types are that allow us to separate them by means of their microwave signature (be it for active or passive microwave sensors);
Reply: Agree. We add sentences with brief description of the fundamental differences for microwave SITY classification. The new sentences are:

*"On one hand, brightness temperatures (Tbs) of MYI tend to be lower than that of FYI because of high loss of radiation caused by scattering when going through the bubbly layer in the sub-surface of hummock area (Sinha and Shokr, 2015). Such difference depends on the wavelength of the radiation with respect to the typical dimensions of the bubbles (they should be comparable for the loss to be effective). On the other hand, due to the high volume scattering and low scattering loss, MYI have relatively higher backscatter than FYI at the same frequency (Carsey, 1992)."*
References:
Sinha, N. K., & Shokr, M. (2015). Sea ice: physics and remote sensing. John Wiley & Sons.
Carsey, F. D. (Ed.). (1992). Microwave remote sensing of sea ice. American Geophysical Union.

iii) Explain more clearly - but still briefly - what the different retrieval approaches are. It is for instance not clear whether the main approach used is a classification. The NASA-Team algorithm (see below) does not use a classification, neither does ECICE.
Reply: Agree. In the revised manuscript, we firstly summarize the approaches in terms of fractional and binary results, respectively. We later mention the typical algorithms accordingly. The modified sentences are:
*"There exist different algorithms which either provide a fractional MYI/FYI coverage or assignment of one or the other ice type (e.g. MYI and FYI) to a grid cell. The former, referred to as sea ice type concentration (SITC) algorithms, includes algorithms such as the NASA Team and ECICE algorithm, …. The latter, referred to as SITY algorithms, include …"*

iv) Move information about evaluation results obtained by others so far into the next paragraph (see Lines 47/48: "By comparing ... Kwok 2004)."
Reply: Agree. We move this information to the next paragraph in the revised manuscript.

v) Mention that different methods exist which either provide a fractional MYI/FYI coverage or a binary classification (or assignment of one or the other ice class to a grid cell).
Reply: Agree. In the revised manuscript, we summarize the approaches in terms of fractional and binary results, respectively. The modified sentences are:
*"There exist different algorithms which either provide a fractional MYI/FYI coverage or assignment of one or the other ice type (e.g. MYI and FYI) to a grid cell. The former, referred to as sea ice type concentration (SITC) algorithms, includes algorithms such as the NASA Team and ECICE algorithm, …. The latter, referred to as SITY algorithms, include …"*

- In addition, I recommend to delete the Lomax et al. 1995 paper and instead include literature related to the NASA-Team algorithm and to the ECICE algorithm which both permit to compute FYI and MYI fractions and are both so far missing completely in your list. I am wondering why you are not considering these products as well in your inter-comparison. I am also wondering whether it would not make sense to get hands on the MYI data sets created by Ron Kwok and used in various publications of him and his group.

Reply: Thank you for the advice. In the revised manuscript, we specify that the sea ice type concentration algorithms include both kinds of approaches. We therefore keep all the references you mentioned in the comment.

Regarding the MYI data sets created by Ron Kwok, it is not included in the inter-comparison since our manuscript focus on the inter-comparison and evaluation on SITY products, which refer to the binary products (whereas Kwok's dataset is a dataset of fractional MYI/FYI coverage). As mentioned in previous replies, we will add sentences in this paragraph to clarify the focus of our manuscript.

Lines 58/59: "Comparison ... methods" --> While an evaluation of products is per se an excellent idea and the improvement of the used retrieval methods a good motivation, I strongly suggest to provide 1-2 sentences that specify more clearly why it is important to (finally) provide a more comprehensive evaluation of these products. The first paragraph of your introduction only tells the reader that sea ice type is important. But requirements about the accuracy and a specific example where an error in the sea ice type distribution of, e.g. 50%, would have which implications is not yet given in a convincing way.

Reply: Thanks for the advice. Implications of large errors can be found in previous studies however have not been quantified to our knowledge. We will add sentences regarding the implications of large errors to highlight the need of evaluation.

Line 61: Why "limited"? There are plenty of ship observations (see e.g.: https://www.cen.uni-hamburg.de/en/icdc/data/cryosphere/seaiceparameter-shipobs.html)

Reply: Agree. "limited" is deleted in the revised manuscript.

Lines 62-64: "... some MYI ... in ice charts." --> I don't understand this sentence; please consider to re-phrase it.

Reply: Thanks for the comment. The sentence is rephrased to

*"While ice chart is used as "ground truth" in some validation (Aaboe, Breivik, et al. 2016), some areas of MYI in the ice charts correspond to areas with MYI concentration of approximately 50% or greater"*

Line 67: "operational" --> please check what you mean here by operational. Do you mean existing? Or are you really referring to all sea-ice type products that are currently operationally (aka daily) produced and provided to the users?

Reply: We mean the products that are currently operationally produced and provided to the users. To avoid unnecessarily misunderstanding, we delete this word in the revision.

Lines 83-86: "Microwave radiometer ... 2016)" --> As stated already in the context of the introduction, it would make a lot of sense to include a paragraph that clearly describes the relevant physical properties of the different sea ice types that are relevant for their discrimination in the different active and passive microwave signals. This is required to understand the algorithm details and to understand their limitations (also during the freezing and/or shoulder seasons) and would be important for the discussion section as well. Since this is the first paper of this kind it is certainly worth to dig into physics here.

Reply: Thank you for the advice. We will revise the manuscript as suggested.

In the "Introduction" section, we will include sentences about the physical background for microwave SITY classification as mentioned above. The sentences included additionally start with:

*"On one hand, brightness temperatures (Tbs) of MYI tend to be lower than that of FYI because of high loss of radiation caused by scattering when going through the bubbly layer in the sub-surface of hummock area (Sinha and Shokr, 2015). Such difference depends on the wavelength of the radiation with respect to the typical dimensions of the bubbles (they should be comparable for the loss to be effective). On the other hand, due to the high volume scattering and low scattering loss, MYI have relatively higher backscatter than FYI at the same frequency (Carsey, 1992)."*

References:
Sinha, N. K., & Shokr, M. (2015). Sea ice: physics and remote sensing. John Wiley & Sons.
Carsey, F. D. (Ed.). (1992). Microwave remote sensing of sea ice. American Geophysical Union.

In the "Data" section, before introducing the individual sea ice types, we will include a paragraph that describes the physical background. The paragraph starts with:
*"Microwave radiometer and scatterometer are used to discriminate MYI and FYI due to their distinctive signatures. Microwave radiometer Microwave radiometers measure the upwelling radiation emitted by the Earth in terms of brightness temperature (Tb), which is linearly proportional to the physical temperature and microwave emissivity of the observed object…"*

Line 91: "on coarse resolution" --> Please write in the text more clearly the grid resolution of the data and, if relevant, also the native resolution of the data used as input.
Reply: Thank you for the advice. In the revision, the "Data" section will be divided into four subsections: "2.1 Microwave remote sensing data", "2.2 sea ice type products" and "2.3 other data". In the section of "2.1", we will introduce the native resolution of the microwave remote sensing data. In the section of "2.2" we will clarify the grid resolution of the sea ice type products.

Lines 93/94: I can understand that Tb measurements are corrected for the atmospheric influence because it disturbs the sea ice signal. I cannot understand why you need to correct the Tb measurements for sea ice concentration ... What I can imagine is that you use an additional sea ice concentration product to restrict the analysis of the sea ice type on the sea-ice covered area. If this is the case then please write it accordingly. However, admittedly this would contradict a bit the next sentence about the Bayesian appraoch do discriminate open ocean and sea ice. In short: You need to rephrase these statements.
Reply: Thank you for the question. We meant to say that the Tb measurements are corrected for the atmospheric influence by using auxiliary data and radiative transfer model. One of the auxiliary data is sea ice concentration, which is used to estimate the surface emissivity in the radiative transfer model. Note that the authors do not correct the Tb measurements for sea ice concentration. To avoid misunderstanding, we modify the sentence as below:
*"The Tb measurements are corrected for the atmospheric influence by using a Radiative Transfer Model function (Wentz, 1977) and auxiliary data such as sea ice concentration and atmosphere reanalysis data."*

Line 96: I suggest to remove the "further" and to also provide an equation of how the gradient ratio is computed.
Reply: "further" is deleted. In the revised manuscript, there will be 3 subsections in the "Data" section. The equation of gradient ratio will be included in the new section "2.1 microwave remote sensing".

Line 98: Where are these "fixed target areas" located? How are these selected? How large are these? Do these change annually? And: Why are these fixed?
Reply: Thanks for the comment. The sentence is misleading thus is modified to:
*"The probability density functions (PDFs) of given ice classes used in the Bayesian classification are dynamically derived from training datasets, which are extracted from the pre-defined areas of MYI and FYI during the 15-day period centered on the specific day (Aaboe, Sørensen, et al. 2018). The pre-defined areas of MYI locate in the north of Greenland and Canadian Arctic Archipelago between 30°W and 120°W, whereas the FYI areas lie in the Kara Sea, Baffin Bay, Laptev Sea and the Bay of Bothnia.*

Lines 100-102: "In 2021 ... scheme" --> My impression is that you are not including data of this new version into your comparison. Therefore I recommend to move this announcement towards the end of your paper, e.g. into the discussion where it could fit with your outlook / description of which improvements are (already) underway. But perhaps 2021 was a typo ...?
Reply: Thanks for the comment. Data of this new version, which is referred to as C3S-2, is indeed included in the comparison. One of the purpose of including this data is to see how much does it differ from the old version, C3S-1. We therefore would like to keep it in the manuscript.

Lines 103-115: I recommend to comment on / give more details on:
i) the fact that the OSISAF-SIT is based on a very heterogeneous set of input parameters and on changes in the training data set (L111/112), which both could have an impact on the sea ice type product in terms of its consistency over time;
Reply: As mentioned above, there will be 3 subsections in the new "Data" section. In the new section "2.1 microwave remote sensing", we will introduce all the microwave remote sensing data used in the various SITY products. In the new section "2.2 sea ice type products", we will then describe each product in details. Introduction of each product will include the following information: 1) grid resolution, projection, the satellite data used in different periods; 2) the algorithm used in the product, where we could include more details.

ii) what a "sigma_nought" is (Line 112) and in which way this variable is used (is it corrected towards a common incidence angle? for instance); what is the incidence angle range used? (Compare the next paragraph where you are comparably detailed as far as it concerns the Ku-Band scatterometers.);
Reply: Thanks for the advice. We will include more details regarding the product and satellite data. As mentioned in the previous reply, details of the microwave remote sensing data will be given in the new section 2.1. It will be stated that the incidence angle of ASCAT ranges from 25° to 65°. Backscatter from ASCAT is normalized to a reference incidence angle of 40 deg and later used as an input for the classification.

iii) what the native resolution of the scatterometer data is;

Reply: the native resolution of ASCAT is around 10km x 25 km.

iv) what a "swath projection" is (Line 112).
Reply: Thanks for the comment. The sentence is modified to:
*"By using the Bayesian approach, sea ice class-probabilities are estimated for $GR_{19v37v}$ and $\sigma_0$, respectively."*

- I furthermore find the introduction of AMSR2 and AMSR-E the way done confusing. AMSR2 is available since July 2012 but it is included since 2016; whether AMSR-E data were at all used is not clear but AMSR-E is introduced.
Reply: Thanks. In the new section 2.1, satellite data will be introduced in a chronological order.

- You describe the different sensors used with different degrees of detail; for instance you do not mention that SSM/I and SSM/IS are multi-channel radiometers with a number of frequencies while you do so for AMSR-E. You refer to "coarse" (previous paragraph) and "medium-resolution" (Line 106) as well as "higher spatial resolution" (Line 109) without a specific motivation. Why is it important to know the spatial resolution? How does the product (also the other products) actually deal with input data being available at different spatial resolutions?
Reply: Thank you for the advice. In the new section 2.1 "Microwave remote sensing data", the satellite data will be introduced in a chronological order meanwhile with same level of details. The microwave radiometer data will be introduced as two series of dataset: one includes SSM/I and SSMIS, the other means AMSR-E and AMSR2. Spatial resolution of the sea ice type products will be mentioned in the new section 2.2.

- What is the spatial resolution achieved by ASCAT and what is the polarization used?
Reply: Details of ASCAT will be presented in the new section 2.1. ASCAT provides only vertically polarized backscatter. The spatial resolution achieved by ASCAT is 12.5 km.

- What are "given weights" (Line 113)? How are these defined?
Reply: The "given weights" are dependent on the distance between centers of grid and footprint. Such details is not crucial for the product. This sentence is therefore deleted in the revision.

- L113-115 you could rephrase for improved clarity along the lines: Both C3S-SIT and OSISAF-SIT provide, in addition to the pure ice type classes FYI and MYI, an ambiguous ice type class that represents an unknown mixture of both ice types, referred to as "Amb". The products are provided with ...
Reply: Done.

Lines 120/121: "ASCAT is ..." certainly belongs either to the paragraph where you introduce ASCAT data for the first time. Or, alternatively, you could think about adding a sub-section wherein which you introduce all sensors and their specifications as far as relevant for this paper. Table 1 provides not enough information.
Reply: Thank you for the advice. We will add a new subsection "2.1 Microwave remote sensing" in the revision, where the sensors and specifications will be introduced.

Lines 116-126:

- I recommend that within this paragraph you underline more clearly that KNMI-SIT is actually a synonym for three different sea ice type products of which you include two into your evaluation. I would then also avoid speaking of "the KNMI-SIT" but in general speak about KNMI sea-ice type products and then define KNMI-Q and KNMI-A as those you are referring to henceforth.

Reply: Thanks. In the revision, we will modify the sentences to underline more clearly the meaning of each acronym, and specify the two products we include in the evaluation is KNMI-Q and KNMI-A. The modified sentences are as below:

*"KNMI-SITY is a series of sea ice type products, which are purely based on scatterometer data, including ERS, QSCAT, OSCAT and ASCAT data. The corresponding four SITY products in KNMI-SITY are referred to as KNMI-E, KNMI-Q, KNMI-O, and KNMI-A, respectively. They are available during the periods of 1992-2001, 1999-2009, 2010-2013 and 2007-2016, respectively. In this study, KNMI-Q and KNMI-A are included in the comparison and evaluation."*

- While your refer to swath and grid in the previous paragraph you don't do this here. In which form are the data of the different scatterometers used within the sea-ice type retrieval? What is the grid resolution? What is the native resolution of the OSCAT and QuikSCAT data? Please refer to Table 1 / Figure 1 for clarification in terms of the time periods the different satellite data and hence sea-ice type products are available. Reading the text it is not clear which time periods the different (?) products cover.

Reply: Thanks. We will refer to the Table 1/Figure 1 when describing details such as periods. In the revision, specifications of the sensors will be introduced in the new section 2.1 with same level of details, whereas the grid resolution and algorithm will be introduced in the new section 2.2.

- L123/124: "In KNMI-SIT ..." --> Does this apply to all three products? Or is there a merged product? Is this classification done after FYI and MYI have been separated? What is the difference in the microwave signal that is exploited to separate SYI from older ice?

Reply: Yes, it applies to all the KNMI sea ice type products. The algorithm classifies surface into four categories (open water, first-year ice, second-year ice and older multiyear ice). We regard the second-year ice and the older multiyear ice as the MYI for comparable analysis. As explained in the previous sentence, "Fixed thresholds extracted from stable wintertime (March) data are adopted for sea ice type classification." The fixed thresholds mean all the thresholds used for ice type classification. Sea ice and open water is classified before the three ice types are further separated, while the three ice types are separated using the thresholds at the same time.

- L125/126: "In this study, backscatter ... SIT products." --> I don't understand this sentence; please re-phrase it.

Reply: Since KNMI-O, which uses OSCAT data, is not included in the comparison of our study. We delete this sentence to avoid misunderstanding.

Lines 127-131:
- Like for the previous paragraph it is not entirely clear whether IFREMER-SIT is again just a synonym for the two other products IFREMER-Q and IFREMER-A or whether these two are merged to form one product.

Reply: Thanks. These sentences are modified to

*"IFREMER-SITY is another series of scatterometer-only products. It includes two SITY products, which use QSCAT and ASCAT data for the years of 1999-2009 and 2010-2015, referred to as IFREMER-Q and IFREMER-A, respectively."*

- I note that you give a few details about IFREMER-A but not about IFREMER-Q.
Reply: We modify the sentence as below to avoid providing different details for one of the product.
*"To account for the varying incidence angle of ASCAT data, the backscatter coefficients are normalized to the backscatter at a constant incidence angle of 40°."*

- I am not sure I understand what you mean by "series of time-varying thresholds" ... What is this "series"? Are you referring to a time series of backscatter data for several winters as written in Line 130? What do you mean by "seasonally consistent"? That the values agree with each other through the course of the freezing season?
Reply: To answer the questions you raise here, we modify the sentences as below:
*"In IFREMER SITYs, day-to-day-varying thresholds are used for the separation between MYI and FYI. These thresholds are derived by analysing the backscatter data for several winters and are found to be inter-annually consistent (Girard-Ardhuin 2016)."*

- While we learn here that the product is gridded to a polarstereographic grid, there is no information about the grid in the previous paragraphs.
Reply: In the revision, we will add projection and grid spacing information of all the SIY products.

Lines 132-137:
- "employs adaptive" --> "employs an adaptive"
Reply: Done

- "based on the thought of clustering" could possibly be re-phrased. What kind of clustering approach is used? K-means?
Reply: it is modified to "based on K-means clustering" in the revision.

- For the other approaches listed above that utilize radiometer data you state that the gradient ratio of the 37 and 19 GHz channels with vertical polarization are used. Which channels are used here?
Reply: It uses horizontally polarized Tbs at 37 GHz and the backscatter from scatterometer. We will add such information in the revision.

- Is it correct that the approach combines coarse resolution radiometer data (what is the resolution? How is the difference in spatial resolution between SSM/I / SSMIS and AMSR-E/2 taken into account?) with fine resolution scatterometer data? What kind of radiometer data are used? Daily gridded? Swath? Which grid?
Reply: As mentioned in the afore-mentioned replies, information regarding the microwave satellite data will be introduced in the new section 2.1.

- You write that QuikSCAT and ASCAT are used successively. Does this mean that you use QuikSCAT data until the very end of its nominal time with regular data

provision in 2009 (?) and only afterwards ASCAT? How does the algorithm deal with the substantial difference in sensing geometry and coverage?

Reply: To answer the questions here, we will add the following sentences in the revision. *"Scatterometer data is obtained from QSCAT and ASCAT successively since QSCAT stopped functioning on November 23, 2009."*

The ASCAT data is normalized to backscatters at constant incidence angle. In addition, it is found that the algorithm in Zhang-SITY is not highly influenced by the use of different scatterometer data. Such details will be added in the paragraph of describing the algorithm used in Zhang-SITY.

*""*

Lines 140/141: "climate consistent data record SIT products" --> Given the heteogeneity of the products described in terms of the spatial resolution of the input data and the various combinations of frequencies and potentially also polarizations used, I doubt that any of the above-mentioned products deserves yet an assignment into the group "climate consistent data record". Therefore, personally, I would skip this whole last paragraph (Lines 138-142); I don't think it is relevant for the paper.

Reply: Agree. We delete this paragraph in the revised manuscript.

Lines 147-150:

- The description of the sea-ice age product should be revised according to the information given in the more recent paper by Tschudi et al., Cryosphere, 14, 2020. In particular statements like "tracking of ice trajectories" should be avoided as should be wrong information about how the data set is derived like "passive and active microwave observations". This ice age data set is derived from the NSIDC sea ice motion data set (which in some way is described in the same paper).

Reply: Thanks, we will modify the description of the sea ice age product accordingly.

- The paper by Korosov et al, 2018, is about the deficiencies and limitations of the NSIDC sea ice age data set but should not be cited in the context of its description. I can kind of guess that you added this information "limited by the simple drift model and the oldest ice age assignment of grids" to illustrate that the NSIDC sea ice age data set may have its limitations but this would need to be explained in far more detail than in half a sentence. In fact, it is likely that the sea ice age product overestimates the presence of old ice and therefore is biased towards old ice. Whether this already applies to the discrimination between FYI and SYI I don't know; this I leave to you to think about.

Reply: We agree that the sea ice age product tends to overestimate the presence of old ice if we regard the pixels fully covered by the ice at "the age" (given by the product). Since SYI is regarded as MYI in this manuscript, the overestimation applies to the discrimination between FYI and SYI also. We will remove the citation and rephrase the description here.

Line 157: "Images are" --> "All five SAR images are ..."

Reply: Done

- Was any filtering (speckle?) applied?

Reply: No. And it was a typo for the pixel size of Sentinel-1, which should be "40 m" not "160 m".

Lines 159/160: "the geolocations and acquiring dates of the SAR images" --> "the location of the five SAR images". There is no acquisition date given in Figure 2. Hence the acquisition dates are missing and the time difference with respect to the sea ice type products the SAR images are compared to is unknown. This needs to be included in the revised version of the manuscript.

Reply: In the revision, we will include the acquisition date of the SAR images. The modified figure will be as below:

[Figure]

Lines 161-165:
- "For better interpretation of SAR images" --> This motivation needs to be explained better. It is not at all clear why, for the interpretation of the few SAR images used, these two additional data sets are required. What is the problem with the SAR images that such data are needed?

Reply: The interpretation of the SAR images does not need these data. We will therefore delete these sentences in the revision.

- Why do you use the CERSAT/Ifremer product - which appears to be quite heterogeneous in terms of the input data when you could have used the NSIDC sea ice motion product coming at 25 km grid resolution on an EASE grid and with daily temporal resolution.

Reply: We admit that it is not appropriate to use such heterogeneous data in the manuscript. The data is also not required for the SAR interpretation. We will delete these sentences.

- What is the grid resolution of the ERA5 data and how did you co-locate these data with the sea ice type products and/or the SAR images?

Reply: As mentioned in the previous replies, we will delete these sentences in the revision.

Line 172: The naming of the regions is partly wrong and needs to be corrected. What you call ESS is actually the combined area of the East Siberian Sea and the Laptev Sea.

What you call BS is not the Barents Sea but the combined area of the Beaufort Sea and the Chukchi Sea.

Reply: Agree. Naming of the regions is modified to "the central Arctic Ocean (CAO), the East Siberian and Laptev Seas (ESL), along with the Beaufort and Chukchi Seas (BCS)" in the revised manuscript. Abbreviations will be updated accordingly.

Line 180:
- Here, in this line your write "extent is calculated by general extent of pixels", in line 176 you write "MYI extent is estimated as the integral of all pixels specified ..." . Both formulations are not to the point and not specific enough. I recommend to re-phrase in both cases along the lines "We computed the MYI extent as the sum of the area of all grid cells classified as MYI."

Reply: Agree. We delete the sentence of Line 180 and modified the sentence in Line 176 to

***"For SITY and SIA products, we computed the MYI extent as the sum of the area of all grid cells classified as multiyear sea ice."***

- In this context I have two questions. 1) Did you use a common land mask? Or is this not required because the region of interest that is delineated by the red line in Fig. 2 is clear of any land influence? 2) Did you take into account that the grid cell area is only a constant in the EASE grid projection while it changes with latitude for the products in polar-stereographic projection? If you did not take this into account yet you must correct your computations.

Reply: Thank you for the comment. Here are the replies to the two questions: 1) No, we did not use a common land mask because of the reason you mentioned. 2) No, it is not accounted for in the original manuscript. We will therefore correct the computations in the revision. SITY products in polar-stereographic projection are firstly regridded to the EASE grid before the computation of MYI extent. Details of such will be presented in the new Section"3 Methods" (3.1 Estimation of MYI extent).

Lines 183++: I have a conceptual difficulty with comparing daily sea ice type maps with weekly sea ice type maps derived from the NSIDC sea ice age. The comparison would be much more meaningful if you would average all daily products over every single week also used in the sea ice type product derived from the sea ice age. After all, this is your main data product for the inter-comparison.

Reply: Thank you for the advice. In the revised manuscript, we will compare the weekly averaged MYI extent from the SITY products with the NSIDC-SIA product. All the relevant numbers will be modified accordingly. Plots of the weekly averaged MYI extent will be presented in Fig. 3. The modified figure will be as below:

[Figure]

Line 184: "decreasing trend" --> What you possibly mean is a decrease or an increase in the MYI extent (over time) or a positive or negative trend of the MYI extent (over time). A decreasing (or increasing) trend, in contrast, is a trend that changes its value with time or, in other words, if this was a linear trend then the slope of the trend line would decrease (or increase) with time. Therefore, please correct your writing accordingly throughout the manuscript.

Reply: Thank you for the advice. This sentence will be modified as follows.

*"since FYI can only turn to MYI when surviving a melting season, overall Arctic MYI extent cannot increase over the winter – it can only decrease through ice advection out of the Arctic"*

Other problematic expressions will be modified accordingly, and all the "decreasing/increasing trend" will be modified to "negative/positive trend".

Line 186: "the divergent movements" --> Which movements? Movements of what? Are you referring to divergent sea ice motion?

Reply: Thank you for the advice. This sentence is modified to

*"it can temporarily or regionally increase due to ice divergence or advection from neighbouring regions"*

Line 194: "to the NSIDC-SIA extents ... 2-3 years" --> You stated earlier that you compute the MYI extent from the NSIDC-SIA data set by summing over all grid cells exhibiting a sea ice age of 2 years or older. Hence you can simply write "to the MYI extent derived from the NSIDC SIA."

Reply: Done.

Lines 199/200: Given the fact that the entire Arctic Ocean (i.e. approximately the region of your study) has a size of about 7 x 10^6 km2, and the fact that rarely the entire Arctic Ocean is covered with MYI, this difference is far above being reasonable and requires more explanation. It is a 100% error.

Reply: Thank you for the comment. We confirm that the largest difference is correctly calculated. Explanations will be added accordingly.

Line 200: Are these extent estimates for class "ambiguous" reasonable? How do these values relate to the entire MYI extent? Please be more critical about and more specific within your interpretations.

Reply: These are averages of the extent estimate from the "ambiguous" pixels. As explained in the first paragraph of Section "Temporal reanalysis", "Amb class in C3S-SITY and OSISAF-SITY could be regarded as MYI and FYI thus the MYI extent is calculated under both circumstances". Large extent for the "ambiguous" class means large number of pixels with high uncertainties on ice type discrimination, which have atypical microwave signatures of MYI/FYI and usually lead to large discrepancies between the SITY products. We will add a few sentences here to explain the meaning of these values and how they relate to the entire MYI extent. These sentences are

*"As described in section 2.2, the Amb pixels have atypical microwave signatures of MYI/FYI thus high uncertainties on ice type discrimination. Compared with the average MYI extent difference between C3S-SITY and OSISAF-SITY, the contribution of these pixels to the comparison is overall small, however could be large under situations that trigger the atypical microwave signatures, which will be further discussed in section 4.1.2"*

Lines 204-210: "This is expected ... summer and winter." --> These lines call for the more careful delineation of the physics behind the various retrieval methods which I asked for earlier. Without that physical background these statements all remain hypothetical and are not sufficiently backed up by existing knowledge and hence not in line with good scientific practice.

Reply: Thank you for the advice. In the section of "Introduction" and "Data", we will add descriptions about the physical background for microwave SITY classification and theory behind the classification algorithms.

Line 214: What do you mean by "most distinct variations"?

Reply: We want to say that the inter-annual evolution of MYI extent from C3S- and OSISAF-SITY differs most with NSIDC-SIA. The sentence will be modified to

*"Overall speaking, the inter-annual pattern of MYI extent from C3S-SITY and OSISAF-SITY differs most with NSIDC-SIA, with large discrepancies in the winter months of 2001-2003, 2006-2008 and 2016-2018."*

Subsection 3.2.1:
- One could have expected that you dedicate a bit more time to comment on the details such as the drop of the MYI extent to zero in some winters, when looking at the NSIDC SIA MYI extent of region ESS.
- What explains, to your opinion, the observation that especially for BS and ESS NSIDC SIA MYI extent is often considerably larger than the MYI extent offered by all other products? This is less pronounced for region CAO where we can also see numerous cases where the other MYI extent products exceed the NSIDC SIA MYI extent.

Reply: Thank you for the advice. In the revised manuscript, we will add details as you mentioned and give possible explanations for the inter-comparison of SIA and SITY MYI extent in the first paragraph of this section. The new sentences are as follows.

*"In the ESL region, the MYI extent even decreases to zero in some winters (e.g. 2007-2009, 2012-2013), which is in line with the record low Arctic minimum sea ice extent in the previous Septembers. In the ESL and BCS regions, it is found that the NSIDC-SIA MYI extent is usually considerably larger than the MYI extent from SITY*

*products, whereas it is less pronounced in the CAO regions. This indicates that the mixture of MYI and FYI, which leads to "overestimated" NSIDC-SIA MYI extent, occurs more frequently in the ESL and BCS regions than the CAO region, which could be explained by the more dynamic ice characteristics in the two regions."*

Lines 220/221:
- "The former ... Stream" --> This is not entirely correct and requires re-phrasing. You have chosen your region CAO such that the Transpolar Drift Stream goes right through it ... from the Pacific side towards the Atlantic side. Therefore, what explains the decrease in MYI extent is i) the export through Fram Strait and, by smaller fractions, into the Barents Sea and through Nares Strait, and is ii) the export driven by the Beaufort Gyre towards the South along the Canadian Arctic Archipelago.
Reply: Thank you for the advice. The sentence will be rephrased to
*"The former mainly results from the outflow of MYI to more southern areas. On one hand, MYI is extensively exported through the Fram Strait and, by small fractions, into the Barents Sea and through the Nares Strait following the Transpolar Drift Stream. On the other hand, MYI is advected towards south along the Canadian Arctic Archipelago driven by the Beaufort Gyre."*

- Note, in the sentence before "BS keeps constant or increasing" should also be re-phrased. You want to state something like the MYI extent in the BS/CS region remains constant or is increasing.
Reply: Thank you for the advice. The sentence is modified to
*"Overall, the MYI extent in the CAO and ESL regions shows consistently negative trend, while that in the BCS region remains constant or is increasing."*

- Finally, what explains the decrease in MYI extent in region ESS? This is not clear yet. If it is exported towards the CAO, then this is going to be a northward flow ... a direction not yet mentioned in your description.
In short: Please be more accurate in the description of the results of your work.
Reply: Thank you for the advice. This sentence is modified to
*"In the BCS region, large quantities of MYI is pushed out of this region following the anticyclonic current, the Beaufort Gyre, meanwhile replaced by the MYI from the CAO region. This eventually leads to nearly constant or increasing MYI extent in the BCS region."*

Line 223: What are "varying evolution trends"? Either do trends vary between the different winters of years. Then this could be termed inter-annual variation of the trends describing the evolution of the MYI extent in the respective region. Or you want to comment that within a season, the evolution of the MYI extent from month to month differs between different winters or years. Then you need to specify that you are referring to the intra-seasonal variation of the MYI extent and need to drop the word "trend". Please be more clear in your writing.
Reply: Thank you for the advice. We would like to refer to the evolution from month to month. The sentence will be modified to:
*"most SITY products show similar intra-seasonal variation in the CAO region, while exhibiting disparate intra-seasonal evolutions in the BCS and ESL regions (especially in early and late winter)"*

Lines 238/239: "the discontinuous ... C3S-SIT" --> This is a too global statement because it reads as if daily MYI extent fluctuations are always explained by this discontinuous FYI delineation. You should not forget that this is a scene at the verge of freeze-up and therefore one cannot expect that all of the MYI has a "mature" microwave signature yet which would the algorithms let define it as such. In addition, I'd say this is an issue that is possibly limited to the late October / early November cases and is not of general validity. Please correct your writing accordingly.

Reply: Thank you for the advice. In fact, this issue can be found in different winter months, e.g. Figs. 6, 7 and A1. To illustrate it more clearly, we will revise the sentence accordingly:

*"In 错误!未找到引用源。 a and b (along with 错误!未找到引用源。 a-d, f-i in Appendix and Fig. 7 a-b, i-j), the discontinuous FYI delineation in the inner part of MYI pack is well demonstrated, which occurs in different winter months and could partly explain the daily MYI extent fluctuations in C3S-SITY".*

Line 245: "with exceptional MYI distributed ... as ESS." --> I can agree on the 2nd largest MYI extent but there is only a quite small part where the finger-like structure of MYI extends through Chukchi Sea into the ESS region. The other finger-like structure at Severnaya Zemlya can be observed in basically all products and is hence not exceptional. Perhaps you want to state that this protrusion from the Chukchi Sea into ESS is not in agreement with the NSIDC SIA sea ice type?

Reply: Thank you for the advice. The expression was wrong here. We were about to use "exceptional" to describe the MYI extent from KNMI-A not KNMI-Q. The sentence will be therefore modified to:

*"KNMI-Q has the second largest MYI coverage among the seven SITY products, with slightly more finger-like structure of MYI extending through the Chukchi Sea into the ESL region"*

Line 256: Looking back at this paragraph and the top two rows of Figure 7 you could also state that this is a good example where assigning the ambiguous ice type pixels to MYI actually improves the agreement in the spatial pattern with NSIDC SIA sea ice type.

Reply: Thank you for the advice. As stated in the top two rows of Figure 7, "in the BCS region, MYI is overestimated compared to NSIDC-SIA", if we assign the ambiguous ice type pixels to MYI, this will actually worsen the agreement with NSIDC-SIA. This piece of information does not seem to be useful for SITY product improvement, we therefore did not include it in the manuscript.

Lines 268-271:
- This information should be placed in a section about methodologies of the inter-comparison, where you describe how you co-located data, and how you computed the MYI extent from the different data sets (and grids).

Reply: Thank you for the advice. In the revised manuscript, we will add a section of "Methods". Information such as co-locating data and calculation of MYI extent will be placed in the "Methods" Section, more specifically in section "3.1 Estimation of MYI extent".

- There you also should reflect upon why and how you selected the boxes as you did and why these have a different size.

Reply: It is difficult to see the details of SAR image without enlargement. The boxes are selected from the locations where there are typical backscattering characteristics and could show the agreement/discrepancies among the various SITY products. In the revision, we will adjust the boxes to ensure the similar sizes.

- It would be furthermore more than beneficial if you would elaborate on the way how you decided, based on the SAR images, which part of the ice is FYI and which MYI. Your sentence "Characteristics of brightness, texture, gemometric shape and context ..." is not sufficient for a journal such as "The Cryosphere"; it rather reads like written for a public science magazine, I am sorry. You have decibel values at hand and by digging into published literature you can get a much better, even quantitative handle on the interpretation of the SAR images.

Reply: In the revision, we will add a new section "3.2 Visual interpretation of SAR imagery", where we will provide more details about the characteristics. In addition, we will interpret the entire SAR images, convert the sea ice classification results from ice types polygons to grided ice classification results, and eventually give the quantitative evaluation results.

- I note that you used HV-pol data from Sentinel-1 SAR. Why did you use cross-polarized images instead of co-polarized images? What is the advantage using those? Can I assume that the RADARSAT-1 images were HH-pol? You could note this additional information in the respective figures.

Reply: We will add the polarization information in the respective figures. In addition, in the new section "3.2 Visual interpretation of SAR imagery", we will add sentences as below:
"Images at HV polarization channel are prioritized if available. This is because the backscatter at cross-polarization channel is usually dominated by volume scattering thus is more disparate between FYI and MYI."

Lines 273-278:
- The description of what is seen in terms of ice types in the SAR image appears to be hypothetical and descriptive. There are tables and publications from which you can learn about the typical signatures (sigma nought) of MYI and FYI at C-Band HH-polarization. You should find and use these to put your assumptions on solid ground. Otherwise also these SAR images cannot serve as an evaluation or even validation data set but rather represent a vague inter-comparison source. And with that you can by no means adequately draw conclusions about the quality of the sea ice type products you are investigating here. You then also need to change the title of the manuscript, leaving out "evaluation". Also the usage of the NSIDC SIA MYI extent does not warrant so because it is known to be biased (this is visible in your manuscript as well) and is not a good source for evaluation in the way carried out by you.

Reply: Thank you for the advice. We add a section of "Method" in the revised manuscript, where the theory of interpreting SAR images is introduced. In addition, we make visual interpretation quantitatively based on the theory and discussed the quantitative comparison in the section of "validation based on SAR". As for the NSIDC-SIA product, it is true that the NSIDC-SIA MYI extent could be biased. However, NSIDC-SIA is used as a reference dataset for large-scale comparison. Discrepancies between the SITY products and NSIDC-SIA does not mean "bad". This requires further "validation" such as that based on SAR images. To better illustrate this

logic, we will modify the sentences at the beginning of the "Results" section and the last paragraph of the "Introduction" section.

- Please add month and year of the scene to the text.
Reply: Done. Month and year of the scene is added to the first sentence of this section. It reads:
*"A typical scene of early winter (November, 2017) in MIZ is shown in 错误!未找到引用源。"*

Lines 279++: I am wondering whether it would make sense to not comment on / discuss every product here in the figures showing the comparison to the SAR images. Perhaps the most striking discrepancies would be enough to mention.
Reply: If we do not comment on each product here, it would be difficult to summarize later. We therefore keep the comment as it is in the revised manuscript.

Lines 284/285: "which might be caused ... the product" --> It is not clear how these two issues can lead to an overestimation of the MYI extent derived from the sea ice age product when the delineation relevant to state whether a grid cell contains MYI or FYI is between FYI and SYI, and is therefore only influenced by the time from the last fall until the date this example is from ... hence basically 6 weeks in this case. Only a small amount of FYI is grown until then in that region and one can be confident that the majority of the grid cells is in fact predominantly MYI as seen in the sea ice age product.
Reply: Thank you for the advice. SITY distribution in NSIDC-SIA is generally consistent with the SAR image. The sentence is therefore modified to
*"NSIDC-SIA shows generally consistent SITY distribution with the SAR image."*

Lines 289-293:
- Same comment as for Fig. 8 with respect to how to assign features and/or brightness distributions to ice types. This is a purely qualitative inter-comparison and not an evaluation.
Reply: Thank you for the advice. We will add a section of "SAR interpretation" in the revised manuscript. In addition, we will provide quantitative interpretation results in terms of statistics such overall accuracy.

- When I was working with SAR data during my PhD days I was always urged to denote the sensor flight and look direction by arrows. You could do so as well so that it is more clear where the low and where the high incidence angles are located.
Reply: Thank you for the advice. Marks of the azimuth and range direction will be added to the SAR images.

Lines 297/298: "The MYI underestimation ... weekly temporal resolution." --> Why? What is the physical process required to have large discrepancies between a weekly ice type map and a daily ice type map? Is there evidence in the additional data used by you about this physical process?
Reply: Thank you for the advice. MYI is slightly underestimated in NSIDC-SIA. This is nearly negligible considering the temporal resolution difference. We therefore modify the sentence to
*"MYI is slightly underestimated in NSIDC-SIA. Yet such difference is nearly negligible considering their different temporal resolutions and the mobility features of sea ice."*

Lines 301-305: Again the interpretation of the SAR image (and the boxes zoomed into) appears to be very hypothetical and is not well backed up by what could be taken from published literature (if the authors would have considered to use HH images instead of HV images). This would also have resulted in less processing artefacts in the image.
Reply: Thank you for the advice. The scattering characteristics are more separable in HV image than HH image. We therefore presented the HV image instead of HH image in the manuscript. In fact, both HH and HV image are used for the visual interpretation. To illustrate the procedure better, we add a section of "SAR interpretation" in the revised manuscript. The physical background and how the zoomed boxes are selected are explained accordingly in the new section. For the Sentinel-1 image, we will include additionally the HH image in the corresponding cases.

Lines 306-311: I would say this is a classical example where the NSIDC-SIA is one of the more useful data sets here. Looking carefully it is clear that regions C and D are both FYI. You can check the minimum extent end of summer 2014 please to check whether in that area close to Severnaya Zemlya sea ice survived the summer melt. I doubt so. Hence these two areas are located within the landfast sea ice (FYI) cover that develops there ususally - as is also well backed up by a sea ice drift speed of zero. While I could agree that region B is in fact MYI I doubt that region A is MYI. This is certainly an area where i) deformation and ii) deep snow plays a significant role in shaping the different microwave signals contributing to the (every) sea ice type classification.
Reply: Thank you for the comment. We agree that this case actually show typical case of landfast ice (FYI) mixed with MYI. In the original manuscript, our interpretation about areas A and B is consistent with you (A as FYI and B as MYI). You might have doubted that area C is MYI. We also agree with on it. Area C should be landfast FYI instead of MYI. Therefore, we will modify the texts in this paragraph accordingly.

Line 312: Is there any reason why you put mid-winter after late-winter?
Reply: We put mid-winter after late-winter period since we wanted to show the cases with large discrepancies first and then later those with smaller discrepancies. The purpose is not well described in the original manuscript meanwhile not that necessary. In the revised manuscript, we will therefore present the cases in the chronological order (early-, mid-, and late-winter).

Line 322: "Compared to the SAR image ... is overestimated ..." --> This is only part of the story. I would see this more differently and urge the authors to have another look to see that the agreement between NSIDC SIA and the supposedly FYI - MYI distribution in the SAR image is only acceptable in the bottom part of the SAR image whereas towards the top and top left there is both an underrepresentation of MYI and an overrepresentation of MYI, respectively.
Reply: Thank you for the advice. We agree with you on the comparison. To present the difference more clearly, we overlay the visual interpreted results (contours) on each subfigure. This sentence is modified to:
*"… the MYI pack in NSIDC-SIA is overestimated in the northeast part of the image (area A) meanwhile underestimated in the northern part (east of area A), …"*

Line 323: Please look at my comment to a similar statement made by you further above.
Reply: Thank you for the advice. We will modify the sentence to

*"As explained previously, such discrepancies are mainly attributed to the mobility features of sea ice and the different temporal resolutions between NSDIC and the SAR image."*

Lines 336-339:
- I don't see how your results underline or agree with the results of Korosov et al. and I also don't see how your results confirm that the NSIDC SIA data set is a cross-validation data set. It is at most a data set for consistency checks and inter-comparison. I will detail why below.
At first: none of the sea ice type products investigated has a finer temporal resolution than the NSIDC SIA product (and hence the MYI extent derived from it). Hence you cannot look at the sub-grid scale distribution of sea ice types (and age) in the NSIDC SIA maps and the information you claim to have at hand originates from the publication mentioned above and is not your own result.
Secondly, even though you have SAR images at hand you did not make the effort to first perform a high-level evaluation of the NSIDC SIA producte BEFORE you use it as a data set for inter-comparison. You could have carried out a dedicated pixel-wise comparison between the NSIDC SIA product and the SAR images used. But this would require i) more SAR images covering the same region over the weekly period represented by the NSIDC SIA product (i.e. ideally one at the beginning and one at the end of the 7-day period) for ... say ... 50 cases (which is a big project) and ii) using SAR images in a quantitative way, i.e. using the sigma nought values to delineate FYI from MYI, and in addition taking carefully the drift and deformation history of the respective regions into account to ensure that areas with a bright signature caused by deformation are not misinterpreted as MYI. Only with such a comparison, looking at the sub-grid scale distribution of the different ice types within single NSIDC SIA 12.5 km grid cells, you can shed more light about the "cross-validation" potential of these maps.
Reply: Thank you for the advice. We agree that the NSIDC-SIA product is used for consistency check and inter-comparison. It is not proper to regard it as "cross-validation". In the revised manuscript, we modify such expressions meanwhile use SAR images in a quantitative way. We add a "Methods" section, where the theory for the visual interpretation is described.

Line 367: "stability of the sea ice types" --> what do you mean with that? FYI will not disintegrate spontaneously and MYI will not become FYI over night. Please rewrite.
Reply: Thank you the advice. The sentence is modified to
*"The efficacy of input parameters is dependent on the capability to separate and physical properties of the sea ice types in question."*

Lines 368-370: "This parameter ... or high frequency channels" --> I agree to this statement; however, I am wondering what the magnitude of cloud liquid water values typically observed during winter in the Arctic would be and what the impact would be specifically on the GR. I am pretty sure you can dig out this information in the available literature and back up your statement adequately. There are sea ice concentration algorithms that specifically make use of the two channels that form this GR, e.g. the Comiso algorithm frequency mode; perhaps the paper by Andersen et al. from 2006 in Remote Sensing of Environment could enlighten you here. In short, unless the impact of atmospheric parameters such as cloud liquid water and water vapor on the GR at these two frequencies is really measurable I would remove this piece of information. If kept it needs to be backed up by adequate literature.

Reply: Thank you for the advice. We further investigate the magnitude of cloud liquid water values and its impact. It turns out that variation of the cloud liquid water path has little impact on GR. We will therefore remove this sentence in the revised manuscript.

Line 368: I am surprised that one of the sea ice type algorithms uses this GR ratio the other way round, i.e. 19 V minus 37V. Please check. This (again) calls also for a better and more comprehensive description of the algorithm behind the products inter-compared in this study.
Reply: Thank you for the advice. We confirm that one algorithm (OSISAF-SITY) uses this ratio the other way around. In the revised manuscript, we give detailed description of each algorithm when introducing each SITY product.

Line 371: "ice layering" is one component of the snow properties and should not be mentioned as if it is a different thing.
Reply: Thank you for the advice. We delete "ice layering" in the revised manuscript.

Lines 373-375: "when air temperatures fluctuates around freezing point and triggers snow metamorphism" --> Apart from the fact that this is another example of bad English grammar this statements needs to be formulated in a less global way.
A) What you call snow metamorphism with a likely impact on brightness temperatures particularly at 37 GHz are melt-refreeze cycles caused by elevated solar radiation during spring (April); during these cycles the air temperatures do not necessarily fluctuate around the freezing point.
B) In October solar radiation is absent, hence cannot be the trigger for snow metamorphism. Melt-refreeze cycles are also absent. What can happen in October is advection of warmer air masses and precipitation falling as wet snow or freezing rain - which admittedly can have an impact on the microwave signature of the sea ice cover. But without working with the theory (missing in your manuscript) you cannot explain it properly. Possibly wet snow masks MYI underneath, letting it look like FYI. But you don't present evidence for this in your data / results. While warm air and hence wet snow might be the reason for the underestimation of the MYI cover in the CAO using C3S-SIT it is not sufficiently clear why snow metamorphism should lead to an overestimaton in the BS and ESS in late winter. What is the physical process that drives which change in the relevant microwave properties that cause the microwave observations to trick the algorithms, leading to an overestimation in MYI?
C) Another issue you did not yet bring up is the fact that parts of the ESS but also the parts of the CAO facing the Atlantic may experience particularly thick snow loads. Since the GR used here is not only sensitive to the sea ice type but it is also sensitive to the snow depth it is not surprising that a sea ice type algorithm that uses the GR at 37 and 19 GHz tends to classify FYI as MYI as a result of a thick snow cover.
Reply: Thank you for the advice. The sentence will be modified in the revised manuscript (see the end of this reply).
Regarding A), we agree that these are likely melt-refreeze cycles caused by increased solar radiation. It is also true that air temperatures during the whole cycle do not necessarily fluctuates freezing point. What we are trying to say is that, air temperatures that trigger these cycles are around freezing point. Evidence for this can be found in Voss et al., 2003 and Ye et al., 2016b. We therefore rephrase the sentence and include the references accordingly.
Regarding B), although solar radiation is not yet absent in the Arctic in October, we agree that the microwave signature changes are different from those in April. They

could be caused by warm air intrusions or precipitation falling as wet snow, which lead to snow wetness changes. We did not present evidence for the impact on the microwave signatures, since it can be found in previous studies such as Ye et al., 2016a. In the revised manuscript, we therefore rephrase the sentence and include the reference accordingly.

Regarding to C), we agree that thick snow load could lead to misidentification of MYI. This information is added in the revised manuscript.

The modified sentence is shown as below:

*"In the beginning and ending stage of winter, the variability of $GR_{37v19v}$ can be significant when air temperature exhibits warm-cold cycles which triggers wet-dry cycles or melt-refreeze cycles of snow (Ye et al., 2016a; Ye et al., 2016b; Voss et al., 2003), or when wet/thick precipitation suddenly appears (Voss et al., 2003; Rostosky et al., 2018)."*

*References:*

*Ye, Y., Heygster, G., and Shokr, M. (2016a). Improving multiyear ice concentration estimates with air temperatures. IEEE Transactions on Geoscience and Remote Sensing, 54(5), 2602 - 2614.*

*Ye, Y., Shokr, M., Heygster, G., and Spreen, G. (2016b). Improving multiyear sea ice concentration estimates with sea ice drift. Remote Sensing, 8(5), 397.*

*Voss, S., Heygster, G., and Ezraty, R. (2003). Improving sea ice type discrimination by the simultaneous use of SSM/I and scatterometer data. Polar Research, 22(1), 35-42.*

*Rostosky, P., Spreen, G., Farrell, S. L., Frost, T., Heygster, G., and Melsheimer, C. (2018). Snow depth retrieval on Arctic sea ice from passive microwave radiometers—Improvements and extensions to multiyear ice using lower frequencies. Journal of Geophysical Research: Oceans, 123(10), 7120-7138.*

Lines 375/376: Please explain to the reader what the effect of the temperature correction scheme and the "upgraded tuning of atmospheric correction for Tb" [better --> the improved correction of the Tb for the atmospheric influence] is on the GR used so that the reader gets a credible piece of information here which you again ideally back up with appropriate literature.

Reply: Thank you for the advice. We would like to say that mitigation of the misclassification is caused by the different processing in C3S-2, which includes the temperature-based correction for improved ice type discrimination and the improved correction of Tb for the atmospheric influence. Details of the difference between C3S-1 and C3S-2 will be added in section 2.2. In the revised manuscript, the sentence will be modified to:

*"Such misclassification in C3S-1 is mitigated in C3S-2 due to the upgraded processing in C3S-2, which includes the temperature-based correction for improved ice type discrimination and the improved correction of Tb for the atmospheric influence (see section 2.2)"*

Line 377:
- "backscatter (sigma^o)" --> either "backscatter coefficient" or "sigma nought"
Reply: Done. "backscatter" is modified to "backscatter coefficient"

- "which has good separability between MYI and FYI." A backscatter coefficient cannot have a good separability between MYI and FYI. A backscatter coefficient might be suitable to separate MYI from FYI.
Reply: Thanks. The sentence will be modified to
*"which is commonly used in ice type discrimination due to the disparate scattering features of MYI and FYI"*

Lines 378-381: "In comparison ... Fig. 12)" --> Also these lines should be re-written and re-phrased investing more space to describe the issues behind.

- In addition, you might want to provide an explanation why Ku-Band scatterometer measurements appear to be less sensitive to the surface roughness than C-Band scatterometer measurements. How about the sensitivity to the crystal structure of the MYI compared to the FYI? Is the contrast in the backscatter coefficient between MYI and FYI larger or smaller at Ku-Band compared to C-Band? Does this depend on the polarization? Does this depend in the incidence angle? What is the role of the different penetration depths into the snow and into the sea ice?

Reply: Thanks. The sentences will be modified as below. Explanations will be added.

*"In comparison, the backscatter of MYI and FYI is more disparate at Ku-band than C-band (Bi et al., 2020; Rivas et al., 2018). Products using Ku-band backscatter generally performs better on identifying MYI, e.g. KNMI-, IFREMER-, and Zhang-SITY. This could be due to the fact that Ku-band scatterometer is more sensitive to the crystal structure of MYI since its wavelength (about 1.7 cm~2.5 cm) is more consistent with the characteristic dimension of air bubbles in MYI (Ezraty and Cavanié, 1999). On the other hand, the greater importance of surface scattering and the higher dependence on incidence angle makes C-band backscatter more suitable to distinguish ice types with disparate surface roughness features, e.g. Case 5 in 错误! 未找到引用源。."*

*References:*

*Bi, H., Liang, Y., Wang, Y., Liang, X., Zhang, Z., Du, T., ... and Huang, H. (2020). Arctic multiyear sea ice variability observed from satellites: a review. Journal of Oceanology and Limnology, 38(4), 962-984.*
*Rivas, M., Otosaka, I., Stoffelen, A., & Verhoef, A. (2018). A scatterometer record of sea ice extents and backscatter: 1992–2016. The Cryosphere, 12(9), 2941-2953.*
*Ezraty, R. and Cavanié, A. (1999). Intercomparison of backscatter maps over Arctic sea ice from NSCAT and the ERS scatterometer, Journal of Geophysical Research: Oceans, 104, 11471–11483.*

Lines 388/389: "In Beaufort and ... classification ..." --> please also see my comment for Line 373 further above.

Reply: Thank you for the advice. The sentences will be rephrased to

*"In Beaufort and East Siberian Seas in late winter, employing Tb and backscatter measurements even leads to worse SITY classification in OSISAF-SITY and Zhang-SITY (Case 3, 错误! 未找到引用源。). It indicates that simple data combination does not necessarily imply better classification results."*

Line 394:
- Either: "employ a dynamic threshold" or "employ dynamic thresholds"
Reply: Done. It is modified to "employ dynamic thresholds"

- What do you mean by "variability of [the] training dataset"? Do you mean the spread of values around a chosen threshold brightness temperature or backscatter coefficient?
- What do you mean by "seasonality"? I recognize that sea ice type retrieval is limited to the freezing season, hence one season; you should be more specific here. It is also not clear to what the seasonality refers to ... to the MYI extent? to the physical properties of the sea ice and its snow cover? to the thresholds used?
- I don't understand what you mean by "shift in sensor type". Could you please elaborate on this in the text? I can guess that you perhaps mean the shift between using SSM/I or SSMIS data or between using ASCAT C-Band and QuikSCAT / OSCAT Ku-Band. But to me this is not a shift in sensor TYPE because it is either radiometers or scatterometers. Please be more specific here.

Reply: The expression may not be clear in the original manuscript. We would like to say that the dynamic thresholds vary temporally or spatially, and depend on the specific satellite sensors. In the revised manuscript, we modify the sentence to
*"while others employ dynamic thresholds, which may vary with time, region and the satellite sensors"*

Lines 398/399: It is not clear what you mean by "takes sea ice variabilities into account". What "sea ice variabilities"? Are you referring to the spatiotemporal development of the physical properties of the sea ice and snow cover that influence it microwave backscattering characteristics and/or the microwave emission? Then please write it specifically. Currently, "sea ice variabilites" can mean anything from variations in sea ice thickness or concentration, different ice drift patterns, floe-size distributions, degree of deformation whatsoever ...
Reply: Thank you for the advice. In the revised manuscript, it is modified to
*"For the latter, the approach considers the spatio-temporal development of the physical properties of sea ice, which influences the microwave radiometric and scattering characteristics,"*

Lines 409/410:  "can be partly ... more obscure"
- "obscure" --> "difficult" or "problematic"
Reply: "obscure" is replaced with "difficult"

- The statement as written is not conclusive because you are not providing the key message that the Arctic Ocean has lost a lot of its oldest ice AND that the difference in the radiometric and microwave backscattering properties is usually more pronounced between FYI and these older ice types than, e.g. second and third year sea ice. This feeds back again to the missing description of the physical background behind the sea ice type retrieval earlier in your manuscript.
Reply: Thank you for the advice. We will add descriptions of the physical background in the Introduction and Data Section. In the revised manuscript, the sentences will be modified to
*"This could be attributed to the large loss of old ice (e.g. older than 4 years) in the Arctic Ocean (Tschudi et al., 2020), which leads to a younger MYI regime in the Arctic thus less pronounced microwave signature difference between MYI and FYI. It eventually makes the separation between FYI and MYI more difficult, especially from ASCAT data (Belmonte Rivas, Otosaka, et al. 2018, Zhang, Yu, et al. 2019)."*

Lines 418/419: In all of the regions mentioned here MYI ice can occur once in a while and hence a MYI ice signature in these regions certainly is not unphysical. Apart from that is the Chukchi Sea part of your region BS. This needs to be corrected in the text.
Reply: It is true that MYI can occur once in a while in these regions. However, as we understand, the mask mentioned here should have excluded the areas (part of the regions) where MYI could be present. Otherwise this mask would lead to obvious "wrong" results. Regarding the comment "Apart from that is the Chukchi Sea part of your region BS", we are not sure what the reviewer means. In the revised manuscript, the sentence will be modified to
*"The first kind of correction scheme, a mask of the Arctic basin, has been used in C3S, OSISAF- and KNMI-SITY to remove the unphysical MYI signature in areas such as the Greenland, Kara, Barents and Chukchi Seas."*

Lines 420-422: "Statistical thresholds ... the ice edge" --> Please provide a plot which illustrates how PDFs of the respective parameters used in the retrieval (i.e. backscatter coeffient or Tbs or GRs) of the MYI overlaps with the PDFs of ice types typically encountered along the ice edge so that the reader understands what you are referring to. Ideally, you have this figure along with the revised description of the sea ice type algorithms earlier in the manuscript so that here you simply need to refer to that figure.
Reply: Thank you for the advice. Since the misclassification is not necessarily 'along the ice edge', we will delete it in the revision. On the other hand, including an additional plot to illustrate this issue would be too much for the example, with consideration of the large number of figures in the current manuscript. As mentioned in previous replies, we will provide more details in the section "2.2 sea ice type products" to make it easier to understand the sentence here.

Lines 422/423: "exclude ... distributions." --> I don't get what you want to state here. If the MYI extent in the above-mentioned peripheral seas and/or along the ice edge would be added to the MYI extent in your region of interest this would mean a considerable change in the overall SIT distribution. Therefore, please re-phrase your statement as it is currently not clear enough.
Reply: Thank you for the advice. The sentence will be rephrased to
*"These two kinds of corrections exclude misclassification cases in regions outside the central Arctic thus have little impact on the overall SITY distributions."*

Lines 424/425: "reassign ... intrusions." --> Not sure what you want to state here. Do you mean "assign grid cells erroneously classified as FYI as the result of warm-air intrusion induced changes in the surface snow properties to the ice type MYI." ?
Reply: Thank you for the advice. The sentence is modified to
*"The temperature-based correction in C3S-2 aims to assign the erroneously classified grid cells of MYI due to the warm air intrusion induced changes of physical properties to FYI."*

- How is this temperature based correction done?
Reply: In the revised manuscript, we briefly describe the temperature-based correction in the section of "2.2 Sea ice type products"

- Aren't there other algorithms (published by one of the authers) that use this temperature based correction as well?
Reply: Other algorithms do not use the temperature-based correction.

Line 425:
- What does an "ice motion confining procedure" do? I have no clue. Please explain it to the reader.
- "anomalous MYI overestimation" --> What is this? What is a "normal MYI overestimation" and what is the difference to an "anomalous" overestimation? Please re-phrase.
Reply: Thank you. We will add one more sentence here to explain the ice motion-based correction. The sentences will be modified to
*"In Zhang-SITY, an ice motion confining procedure is introduced to eliminate overestimated MYI. The procedure builds upon ice motion records and confines the evolution of MYI according to the tolerance of ice motion."*

Lines 428/429: It is not sufficiently well described how the correction based on a median filter (spatial or temporal) works.
Reply: Thank you. The sentence will be modified to
*"Another correction used in Zhang-SITY is the median filter correction, which considers the spatial consistency and is employed to remove large unusual spatial variations."*

Line 432: "the five series SIT products ... are defined." --> I don't understand this sentence. Please re-write. It is possibly a problem of the grammar.
Reply: Thank you. The sentence will be modified to
*"Apart from the above three aspects (input parameters, classification methods and correction schemes), factors such as the covering period and spatial resolution makes the five series SITY products different from each other."*

Line 434: "Typically ..." --> I don't see that your manuscript warrants yet to state the reason given for the larger spread in MYI extent during early and late winter as being typical.
Reply: Thank you. In the revised manuscript, we delete "Typically", add references and phrase the sentence. The sentence will be modified to
*"In early and late winter larger uncertainties are likely to occur due to surface melting over sea ice and atmospheric influence (Voss et al., 2003; Ye et al., 2016a; 2016b)."*

Lines 440/441:
- It is sufficient to write "grid resolution", spatial can be omitted.
- "foot print" --> "footprint"
Reply: Done.

- What is the "true" spatial resolution of the ASCAT data? What is the "true" spatial resolution of the QuikSCAT data? You should please not forget that the finer grid resolution provided by the SIRF products (4.45 km) is the result of heavy smoothing and other signal reconstruction steps.
Reply: Agree. In the revised manuscript, "true" is replaced with "nominal".

- Another issue that you did not take into account here are the different incidence angles of - especially - the ASCAT C-Band data compared to the microwave radiometer data and QuikSCAT / OSCAT.
Reply: Thank you. Yes, we should have mentioned how they (SITY products) process the ASCAT data. The ASCAT data used in the SITY products is already the normalized backscatter coefficient with same incidence angle of 40 degree. We therefore did not mention the influence of incidence angle in the discussion. In the revised manuscript, we add a section of "Microwave remote sensing data" (section 2.1) and "sea ice type products" (section 2.2), where how the characteristics and pre-processing of ASCAT data is introduced.

Line 452:
- Add "five" in front of "SAR images".
Reply: Done.

- Any reason why you are not mentioning the NSIDC SIA product here?

Reply: In the revised manuscript, we mention it when talking about the inter-comparison. The sentence is modified to

*"In this paper, nine SITY products based on five retrieval approaches were inter-compared through temporal and spatial analysis, **with the NSIDC-SIA product as a comparative reference.**"*

Line 453 / the conclusions in general:

- I am not sure I would select a sea ice type product based on the maximum difference that a product might have compared to another independent data source. I would be interested in whether there are regions and time periods where there are systematic errors (and how large these are on average so that I might be able to correct them). In addition, I would be interested in the average performance of the product over a longer time period, i.e. whether there are artificial trends.

Reply: Thank you for the advice. We understand that some readers may be interested in the periods and regions where there are systematic errors. However, as the NSIDC sea ice age data is only a 'reference' at large scale not the 'ground' truth, it is very unlikely that we could draw conclusions such as there are systematic errors in specific regions and periods given the evaluation with limited SAR images. Meanwhile, we will dig more into the average performance of the product over a longer period.

- I suggest to re-write your conclusions accordingly, focussing less on the individual products as you do in the list 1) to 4) (which should in any case contain 5 or even 9 entries according to what you write in Line 451), and instead concentrating on the larger picture provided by your qualitiative results. It might help in this context to again take a look at your time series plots and focus less on the inter-comparisons with the SAR images.

Reply: Thank you for the advice. We will dig more into the larger picture of the results over a long period.

- I like the bullet point list further down on the next page. That one looks good but could be written even better by including specific details and referring to the existing literature.

Reply: Thank you for the advice. We will include more details and refer to the existing literature in the revision.

Line 477: "extensive misclassification with higher uncertainties" --> So, the misclassification in itself is highly uncertain? Please re-write.

Reply: "with higher uncertainties" is deleted in the revised manuscript.

Line 480: "Ku-Band ..." This statement is not new and has to be backed-up by existing literature.

Reply: Thank you for the comments. References "(Rivias et al., 2018; Ezraty and Cavanie, 1999)" are added in the revised manuscript.

Lines 488-490: "On the other hand ... become obscure." needs to be re-written. The meaning is not clear and the grammar is not correct.

Reply: The sentence is modified to

*"On the other hand, adaptive classification method that depends on the clustering pattern of the radiometric and backscattering signatures may be inefficient when the characteristic signatures of MYI and FYI have large overlaps"*

Line 492:
- Apart from the fact that we still don't know how "ice motion confining" works, it is not clear what "accumulative errors" are. Consider re-phrasing for improved understanding.
Reply: As seen in the replies to previous comments, we give more detailed description regarding the "ice motion correction". In the revised manuscript, the sentence is modified to
*"Excessive post-processing such as ice motion confining could lead to over-correction problem, which becomes the basis for the subsequent corrections and eventually result in accumulative errors"*

- "These post- ..." --> "Any post- ..."
Reply: Implemented in the revised manuscript.

Line 491: What is meant by "should be accounted with caution"? Please consider re-phrasing for improved understanding.
Reply: "should be accounted with caution" is replaced with "should be considered with caution"

Lines 495/496:
- "This study ... of SIT retrieval approach" --> I don't agree. This study does not contain an "evaluation"; it is an inter-comparison study, mostly involving qualitative results. It provides hints of the quality of the sea ice type products investigated RELATIVE to the NSIDC SIA data set (which in itself is not well evaluated) and relative to only five SAR images which are not interpreted quantitatively.
- I further object to the notion "most popular". Please consider re-phrasing.
Reply: Thank you for the comments. We agree that it is not that appropriate to call it "systematic evaluation". In the revised manuscript, the sentence is modified to
"This study inter-compares nine SITY products and provides …"

- I cannot see the "hints for further improvement". While you state where some of the sea ice type products have deficiencies, you neither come up with specific suggestions about how to improve (e.g. use a SIRF-like product as an input to the OSISAF sea-ice type product to improve the grid resolution) nor does the nature of your results being based on an inter-comparison to qualitative data support to draw conclusions into this direction. I warmly suggest to tone down the value and potential impact of your results.
Reply: To tone down the value, the sentence is modified to
"This study … and indicates the potentials for further improvement."

Line 497: Please share with us which two frequencies WindRAD is going to use.
Reply: Thanks. The two frequencies are Ku- and C-band. In the revised manuscript, we add this information in the parenthesis "(WindRAD, Ku- and C-band)".

- "the potential of scatterometer on ice type discrimination" --> "the potential of scatterometer measurements for ice type discrimination"

Reply: Thanks. It is implemented as suggested in the revised manuscript.

Lines 499/500: "low frequency microwave measurements" --> "low frequency microwave radiometer measurements" because ASCAT already has been using C-Band for 15+ years which is also a "low frequency microwave measurement"
Reply: Thanks. It is implemented as suggested in the revised manuscript.

Figure 3: I suggest to reduce the number of colors used by getting rid of the NSIDC sea ice age and instead show the MYI extent derived from it as a black line - like you do in Fig. 4. If you want to show examples of how the different sea ice type products deal with different sea ice age then I suggest to show just the respective year - ideally a year where almost all sea ice type products provide MYI extent so that you can compare between the products. Alternatively, you could consider showing only MYI extent differences. If you want to keep the sea ice age information then I recommend to use shades of grey instead of colors for the sea ice age.
Reply: Regarding your concerns of Figure 3, 1) since there are already many plots (from SITY products) in Figure 3, it is easier to identify the NSIDC sea ice age, thus to compare with the respective SITY. 2) this Figure is used to show an overall comparison we therefore include all the datasets of the entire period here. 3) we would like to show the sea ice age information. In the revision, we will use different shades of grey for the NSIDC sea ice age in Figure 3. In addition, we will include the weekly MYI extent from all the SITY products instead.

Figure 4: Why are IFREMER-A data missing for January and April in 2014 & 2015?
Reply: The IFREMER-A data is not available for the period at the time for the analysis. We will update the data in the revision.

Figure 6 and 7:
- It is very counter-intuitive to show open water in brown, land in light grey and FYI in blue. Please use a more intuitive coloring such that, e.g. land is brown, open water is blue and FYI, Amb, and MYI are perhaps medium grey, light grey and white; the observation gap at the pole can then be colored black.
- I recommend to enlarge the figure as a whole.
Reply: Thanks. The figures will be modified as suggested.

- In the caption you could cross-ref to Table 1 or Figure 1 to make clear why there is a different number of maps for the two dates shown. In addition you need to refer one more time to the meaning of the red line and you need to comment on the different coloring of the observation hole.
Reply: Thanks. We will cross-ref Table 1 in the caption and add the meaning of red contour.

Figure 8:
- What is the motivation to show boxes A to D with a different size?
Reply: Thanks. Size of the box does not need to be different. We will try to select similar size of boxes in the revision. (not yet modified in the example figure below)
- I have the same comment with respect to colors as for Figures 6 & 7.
Reply: It will be modified as suggested.

- I suggest to show the NSIDC-SIA map in a different color code as well. What is important for you is to discriminate FYI from older ice which currently is difficult to delineate because the colors used for FYI and SYI are quite similar.
Reply: We will use a different color code for easier discrimination between FYI and older ice.
- I recommend to rename "sea surface wind" to "10m wind" because I guess this is what it is. Also make clear that "air temperature" possibly is the "2m air temperature". The additional information that these are daily averages would be appreciated as well.
Reply: It will be modified as suggested.
- I would replace the legends for those sea ice type products that do not provide the ambiguous ice class with a legend which only shows the two ice classes present. It might make sense - in general - to then also include the class open water in the legend.
Reply: For SITY products that include the class of 'Amb', if we do not show the 'Amb' class, there will be pixels with missing information in the Figure. On the other hand, 'Amb' tells the pixels with large uncertainties of ice type classification. We therefore would like to keep the 'Amb' class in the figure.
An example is shown below, where we will modify the size of enlarged boxes.

[Figure]

Figures 9 to 12:
- I have the same comments with respect to colors, legends, and ERA5 data naming as I had for Figure 8.
Reply: It will be modified accordingly.
- In addition, delineation of the boxes in the Sentinel-1 image in a different color than black would help to locate these better.

Reply: It will be modified as suggested.
- It might make sense to not use a continuous color table for the legend of the ice drift field, as the values are increments of 0.1 km/day.
Reply: The vectors represent ice drift, whereas the background colored field represents 2-m air temperature of the same day.

Table 2: Cases where there is a "+" and a "-" indicate that both performances exist?
Reply: Yes. In the revision, we will add the overall accuracy of each SITY product for the respective SAR images. An example of the updated Table 2 is shown as below:

| SITY products | Case 1 | | Case 2 | | Case 3 | | Case 4 | | Case 5 | |
|---|---|---|---|---|---|---|---|---|---|---|
| | General pattern | Overall Accuracy | General pattern | Overall Accuracy | General pattern | Overall Accuracy | General pattern | Overall Accuracy | General pattern | Overall Accuracy |
| C3S-1* | - | 0.76–0.82 | ○ | 0.83–0.86 | -- | 0.47–0.47 | + | 0.71–0.73 | +- | 0.47–0.54 |
| C3S-2 | - | 0.80–0.87 | ○ | 0.83–0.86 | -- | 0.47–0.47 | +- | 0.72–0.77 | + | 0.40–0.49 |
| OSISAF-SITY | -- | 0.52–0.64 | - | 0.76–0.79 | -- | 0.47–0.47 | -- | 0.78–0.82 | +- | 0.63–0.69 |
| KNMI-Q | + | 0.79 | / | / | + | 0.86 | + | 0.78 | / | / |
| KNMI-A | ++ | 0.70 | ++ | 0.66 | ○ | 0.89 | ○ | 0.78 | ○ | 0.87 |
| IFREMER-Q | - | 0.86 | / | / | -- | 0.47 | - | 0.83 | / | / |
| IFREMER-A | / | / | / | / | / | / | / | / | -- | 0.84 |
| Zhang-SITY | ○ | 0.91 | -- | 0.60 | - | 0.84 | ○ | 0.82 | -- | 0.84 |

Table 3: I guess the GR listed in the context of OSISAT SIT is not correct?
Reply: It is correct, which is the 'negative' form of the other.

Typos / editoral comments:
Line 59; "indirect validation" --> perhaps better "inter-comparisons"?
Reply: "indirect validation" is modified to "inter-comparisons"

Line 89: "KNMI-" --> "KNMI-SIT"
Reply: Done.

Line 144: "study, sea ... were used ..." --> study, we used a sea ice age (SIA) product and five SAR images ..."
Reply: Done.

Line 154: "with SAR ... of HH" --> possibly better: "providing C-Band (5.3 GHz) SAR images at HH polarization."
Reply: Done.

Lines 155/156: "providing cross- ... ranging from" --> possibly better: "providing C-Band (5.4 GHz) SAR images at co- and cross-polarization (HV and HH) with incidence angles between"
Reply: Done.

Line 170: "polar hole of 87degN" --> "data acquisition gap north of 87degN centered at the pole"
Reply: Done.

Line 176: "within studied area" --> I picked this as one of the examples that underline the need for considerable English editing of the manuscript. The authors must check for usage of "the" and "a" which is often missing.
Reply: Done.

Line 181: You stated already in Line 170 that you excluded that area centred at the pole. Therefore you can delete this sentence.
Reply: Done.

Line 185: "The MYI" --> However, the MYI"
Reply: Done.

Line 185: "regional" --> "regionally"
Reply: Done.

Line 225: "is mainly resulted from" --> check grammar.
Reply: "is mainly resulted from" is modified to "is mainly attributed to".

Lines 239-241: This part does not belong to the top row of Figure 6, right? It belongs to the data from 2007 and should be placed into the next paragraph.
Reply: This part belongs to the top row of Figure 6. We add notes of Figures in the sentence for clarification. The sentence is modified to:

*"On the other hand, IFREMER-Q (e.g. Fig. 6c) shows constantly less MYI than KNMI-Q (e.g. Fig. 6d) in the transition zone of MYI and FYI in BS, in good agreement with their difference as shown in 错误!未找到引用源。."*

Line 280: Typo: "boarder" --> "border"
Reply: Done.

Line 288: Add the year.
Reply: Done.

Line 289: "... in the western part were higher than in the eastern part."
Reply: Done.

Line 294: "Slightly underestimation of MYI" --> check grammar.
Reply: "Slightly underestimation of MYI" is modified to "Slight underestimation of MYI".

Line 296: I would say that "thin" could be misinterpreted as "thin MYI" in terms of its thickness. You might want to consider using "narrow" or "filament-like" or "finger-like" or similar.
Reply: "thin MYI tongue" is modified to "narrow MYI tongue"

Line 297: "can partly be resulted" --> check grammar.
Reply: "can partly be resulted from" is modified to "can be partly attributed to ".

Line 300: "A Sentinel-1 SAR image covering the southern part of the ESS near the coast acquired on April 27, 2015 is shown in Fig. 10."

Reply: Done.

Line 313:
- "transit zones" --> "transition zones" or "zones of mixed FYI - MYI coverage"
Reply: "transit zones" is modified to "transition zones".

- "steady discrepancies" --> re-phrase please.
Reply: "steady discrepancies" is modified to "constant discrepancies".

Line 315: Delete "validation and"
Reply: Done.

Lines 324-325: "The MYI feature ... round MYI floe" --> please check grammar.
Reply: the sentence is modified as follows:
*"The bright MYI feature is clear in the northeast part of the SAR image, so as the dark FYI feature in the southwest part. Areas A and D exhibit high backscatter of round MYI floe"*

Line 364: "serial" ???
Reply: "serial" is deleted.

Line 378: "when using backscatter" --> when using backscatter coefficient measurements of an active microwave instrument."
Reply: Done.

Line 382: "confirmed" --> "shown"
Reply: Done.

Lines 397/398: "vary ... ASCAT" --> "are different, especially at C-Band."
Reply: Done.

Line 402: "speculate" --> "hypothesize" ?
Reply: "we speculate that this is because …" is modified to "The possible explanations could be that …"

Line 405: Either "to a sea ice type distribution" or "to sea ice type distributions"
Reply: "SIT distribution" is modified to "SIT distributions"

Line 406: Either: "An adavtive clustering algorithm is used" or "Adaptive clustering is used"
Reply: "Adaptive clustering algorithm is used" is modified to "An adaptive clustering algorithm is used"

Line 408: "thin ... seas" --> "narrow MYI tongues in the peripheral seas"
Line 427: "continuous underestimation" of what?
Reply: it is modified to "continuous underestimation of MYI"

Line 430:
- "over-correction problem" --> "over-correction.
Reply: Done.

- "thin MYI ... seas" --> We had that expression earlier. Please look up my comment there.
Reply: Agree. "thin MYI …" is modified to "narrow MYI …"

Line 436: "fully evaluated" --> "done"
Line 481: What is "small FYI in MYI pack"? Do you mean: "comparably small areas of FYI within a region dominated by MYI?"
Reply: Yes. "small FYI in MYI pack" is modified to "comparably small areas of FYI within a region dominated by MYI"

Line 487: "deep " --> "mid-"
Reply: Done.

References: You need to check your reference list. For a considerable number of the entries the records are not complete; for instance is the year missing quite often. At least one of the references appears twice.
Reply: Thank you for your through review. We will double-check all the references and be more careful on the detailed information in the revision.

---

## Author Response (AR1)

**Response to RC 1:**

Dear Editors and Reviewers:

We would like to thank the reviewer for the detailed and helpful comments, suggestions, and careful checking. Comments are responded on point-by-point basis. The reviewer comments appear in black. The responses are in blue and the proposed changes to manuscript are in ***bold italics***.

**Summary**

This paper compares different sea ice type products currently available to the community. The products are based on passive microwave data, scatterometer data (C or Ku band), or a combination of both. The products have been developed empirically via training data. The type fields are inter-compared and evaluated against a widely-used sea ice age product and SAR retrievals. The products perform better in mid-winter than in early or late winter when melt/re-freeze may occur. Ku-band scatterometer generally is better at type discrimination. Combination of passive microwave and scatterometer data can yield better performance at times, but not in all situations.

**General comment**

This is a fairly comprehensive review of the primary sea ice type products available. There are notable differences in how the products are assembled, the input source data, and their performance in different conditions. Thus, this paper is a valuable contribution to the community be providing such an assessment. The paper is quite thorough and overall it does a good job in presenting the inter-comparison and evaluation of the products.

Reply:

Thank you for the thorough review. Your comments and suggestions are highly appreciated. For better comparison and evaluation of the sea ice type products, we revised the manuscript from the following two aspects:

1) The "Data" section was re-structured. This section includes four sub-sections: "2.1 Microwave remote sensing", "2.2 Sea ice type products", "2.3 Sea ice age product" and "2.4 Other data". In section 2.1, specifications of the sensors and the satellite data are introduced in a chronological order, with subsections of passive/active microwave remote sensing data. In section 2.2, theory of SIT classification is introduced at the beginning, followed with the overall description of the respective SIT products in terms of grid size, projection, availability period, a summary of the satellite data used and the algorithm with necessary details. In section 2.3, sea ice age product (with evaluations from previous studies) is introduced with the input data, algorithm and evaluation of the product. In section 2.4, the SAR images along with auxiliary data are described accordingly.

2) A new section of "Methods" was added, which includes "3.1 Estimation of MYI extent" and "3.2 Visual interpretation of SAR imagery". We modified the computation of MYI extent in the revision for consistent griding, projection among all the SIT products. In section 3.1, Information such as co-locating/re-griding the data and calculation of the MYI extent was introduced. In section 3.2, the theory and characteristics of sea ice classification in SAR images were introduced with references from previous studies and examples from our study. In addition, we interpreted all the entire SAR images, consulted with ice experts regarding the results, compared with the gridded ice classification results, and eventually gave the quantitative evaluation results (Kappa coefficient and overall accuracy for respective SAR image and SITY, SIA product).

Besides, case studies were presented in the chronological order with more discussions referring to the physical background and the algorithms of SIT products. Figures were modified for better presentation. A thorough edit of the language style and grammar were conducted. And all the references and citations have been double-checked and corrected accordingly.

Specific comments are below, but one overall comment is on the SAR data used for evaluation. In general, SAR is going to be the best "truth" for comparison. It is high resolution, so it can delineated even individual floes often. And it is all-sky, so retrievals of type are available anywhere the sensor collects imagery. However, the challenge with SAR is interpreting the imagery. The authors interpret the SAR imagery and classify various locations as a given ice type, but they don't give a particular rationale or provide references for their classification basis. Often, expert ice analysts interpret SAR fields for operational ice charts. They have deep experience in understanding the imagery and properly defining features. It appears the authors here classify the imagery themselves. This is okay, but I would like to see more substantial justification for their classification.

Reply: Thank you for the advice. In the revision, we added a new section regarding the theory of SAR interpretation. The theory and characteristics of sea ice classification in SAR images was introduced in the section "3.2 visual interpretation of SAR imagery".

Another weakness with the SAR comparison is that it is just a few scenes in selected regions and selected periods. And even within the SAR scenes, a few specific locations are picked out as "pure types" for comparison. Ideally, a full SAR image would be classified and compared. I know automated SAR classification algorithms for sea ice are troublesome, so I can understand the approach taken, but it results is a fairly ad hoc and qualitative evaluation. Since this paper is otherwise quite comprehensive, I won't request more evaluation, but ideally (perhaps in a future paper), it would be good to get classified SAR images – perhaps from an expert ice analyst at an operational ice center – and conduct a more comprehensive and quantitative evaluation of the ice type

products.

Reply: Thank you for the advice. In the revision, we interpreted all the entire SAR images, consulted with ice experts regarding the results, compared with the gridded ice classification results, and eventually gave the quantitative evaluation results (Kappa coefficient and overall accuracy for respective SAR image and SITY, SIA product).

A final note is that there is a need for a thorough copy edit for English language style and grammar. The issues are mostly minor – in particular, there are numerous missing articles ("the", "a", "an") – but they are widespread throughout the manuscript. I don't bother to point them out individually as they are too numerous, but they need to be addressed before final publication.

Reply: Thank you for the advice. We went through the manuscript and conducted a thorough edit for the language style and grammar.

**Specific comments (by line number):**

11: The authors definite "sea ice type" as "SIT" here. This is fine and it is used consistently throughout the manuscript. However, as a sea ice scientist, "SIT" means "sea ice thickness" to me. And particularly with numerous thickness products coming out from altimeters, "SIT" is becoming quite common in the community to denote thickness. I can understand wanting to use an abbreviation and "SIT" makes sense for ice type, and the context is clear throughout the manuscript. So, I can't say it needs to be changed, but it might be something for the authors to consider. For me, every time I saw it, "thickness" popped into my mind first until I recalibrated. I can't think of another good abbreviation myself, but one could just use "type" or "Type" as a short-hand, instead of "SIT".

Reply: Thank you for the comment. We now use "SITY" to represent "sea ice type". All the abbreviations in the manuscript were modified accordingly.

28-30: I'm struck by the use of more than author listed and then "et al." in the citations – i.e., "Comiso, Parkinson, et al., 2008". Generally, if there are more than two authors, just the first author is listed followed by "et al." – i.e., it would be "Comiso et al., 2008". In looking at The Cryosphere guidance for citations, I don't see anything that indicates two authors should be listed, so I'm not sure of the rationale. This seems to be done throughout the manuscript. (If there are only two authors, you list both, e.g., if it were "Comiso and Parkingson, 2008".) Not a big deal and I assume the copy editing will decide the proper citation format. I just haven't seen this before and it struck me as odd.

Reply: Thank you for the advice. We carefully checked all the references and corrected the citations in the revision.

31-32: Be careful about terminology. "Thin" and "Young" ice are standard stage of

development classifications. I think here you mean "thinner and younger" for FYI, and then "thicker MYI". I'm also not sure what you mean by "firm" in relation to MYI?

Reply: Agree. The sentence was modified to (L37-L38):

*"The Arctic sea ice has been increasingly dominated by thinner and younger first-year ice (FYI) instead of thicker and older multiyear ice (MYI), …"*

57: "ergodic" is an obscure word – I was not familiar with it. Based on my understanding after looking it up, I'm not sure it is used properly here. Regardless, I think a simpler word is appropriated here or I wonder if it is needed at all – "combined use of both data" is clear to me.

Reply: Agree. The word "ergodic" was deleted. "combination use of both data" was replaced with "combined use of data".

62-63: "While ice chart…" is a confusing sentence – not sure what it is say. I would suggest revising.

Reply: Thanks. The sentence was modified to (L83-L85)

*"While the ice chart is used as "ground truth" in some validation (Aaboe et al., 2021a), some areas of MYI in the ice charts correspond to areas with MYI concentration of approximately 50% or greater (Lindell and Long, 2016a), indicating the overestimation of MYI in ice charts."*

72: Just one example of grammar/style issues: "…are detailed investigated." – It should be "are investigated in detail."

Reply: Done.

107-109: Is AMSR-E used in the product? The description indicates only AMSR2 is used. So, why describe AMSR-E characteristics? Why not just describe AMSR2 characteristics?

Reply: AMSR-E is used in Zhang-SIT from 2002 to 2011. In the revision, we added a new section "2.1 Microwave remote sensing data", where the satellite data are introduced in a chronological order meanwhile with same level of details. The microwave radiometer data were introduced as two series of dataset: one includes SSM/I and SSMIS, the other means AMSR-E and AMSR2.

109: Maybe another grammar/style issue: "working" is okay, but typically when describing sensors or satellites, "operating" or "collecting data" are more common. "working" seems a bit colloquial here.

Reply: Agree. "working" is replaced with "operating".

147: This goes for all products, but noting here because NSIDC products have specified

references that should be used. For SIA, it is:

Tschudi, M., W. N. Meier, J. S. Stewart, C. Fowler, and J. Maslanik. 2019. EASE-Grid Sea Ice Age, Version 4. [Indicate subset used]. Boulder, Colorado USA. NASA National Snow and Ice Data Center Distributed Active Archive Center. doi: https://doi.org/10.5067/UTAV7490FEPB. [Date Accessed].

This should be cited in the manuscript text and listed in the references. I see that the dataset website is noted in the Acknowledgment section, but where a reference is provided, it should be included in the manuscript proper, including the dataset DOI. I know all datasets do not provide a formal citation and/or DOI – for example for OSI-SAF, their recommended citation is simply: "The type dataset shall be referred to as the Sea ice type product of the EUMETSAT Ocean and Sea Ice Satellite Application Facility (OSI SAF, osi-saf.eumetsat.int)." If that is all that is provided, that is fine, though I would also say that the product ID (OSI-403-d) and version (if provided) should be included. The other datasets used should be cited to the extent they properly can be.

Reply: Agree. References for all the datasets used in the manuscript were updated accordingly.

185-186: I think the potential for MYI increase could be explained better here. In practice, overall Arctic MYI cannot increase over the winter – it can only decrease via advection out of the Arctic. "Temporary" increases can happen within products due to divergence – e.g., a 100% MYI pixel diverging into two pixels with 50% ice each; if the threshold for detection is <50%, there will now be two pixels. And regionally, MYI can increase, both due to divergence or due to advection into the region from neighboring regions.

Reply: Thank you for the advice. The sentence was modified as below (L350-L351):

*"However, it can temporarily or regionally increase due to ice divergence or advection from neighbouring regions (Kwok et al., 1999)."*

191: This is discussed a bit more later, but this left me hanging: "why such a dramatic peak in the first half of winter?" Maybe provide a brief explanation and then say it will be discussed further later in the paper.

Reply: Thanks for your suggestion. This section has been modified thoroughly. The following sentence were added to give a brief explanation (L359-L361):

*"For C3S-SITY and OSISAF-SITY, such pattern is caused by underestimation of MYI in October, while for KNMI-A it is mainly due to the overestimation of MYI in November in the peripheral seas of the Arctic and will be further discussed later in Section 5."*

204: I would use "to" instead of a "-" because it looks like a minus sign. Or use an "em-dash" or "en-dash" with spaces on each side.

Reply: Agree. "-" was replaced with "to" or "en-dash" in the revision.

219: Figure 5 is mentioned quite cursorily here, but I notice the behavior of several products in BS during 2016-2017. That sticks out compared to other years and regions. Why was the performance so different?

Reply: Thanks for the comment. Explanations for the performance were added in the revision. This sentence was modified as below (Note that we modified the abbreviations of each region in the revision) (L412-L414)

*"Overall, the MYI extent in the CAO and ESL regions shows a consistently negative trend, while that in the BCS region remains constant or is increasing. The former mainly results from the outflow of MYI to more southern areas."*

224-225: Okay, the KNMI-SIT increase is mainly in the BS and ESS regions. But why? In general, this paragraph (223-229) feels like it needs to drill down a bit more and give more detail/explanation.

Reply: Thank you for the advice. We included more discussions in the previous section "4.1.1 Weekly MYI extent variation", where the reasons behind are similar. We therefore did not include more explanations here.

259: Kind of the same thing here. Okay, you have an overestimation of MYI, but that doesn't specifically explain the abnormal increase in MYI during 2016-2017. Why was the MYI overestimated in the one year versus others.

Reply: Thanks for the advice. The abnormal increase in MYI during 2016-2017 was explained in section "4.1.1 Weekly MYI extent variation". The sentences read as below (L353-L356):

*"For instance, all the SITY products show increasing MYI extent in March/April 2017 except Zhang-SITY. This could be caused by the enhanced melting during this period (Raphael and Handcock, 2022; Ye et al., 2016a), which leads to the radiometric and scattering signatures of FYI similar as that of MYI therefore unsatisfactory performances of the SITY algorithms."*

265: How are cases selected? Were they ad hoc? Random? Was it simply availability of imagery? Or was there some physical rationale to select the scenes? I understand in general wanting different regimes and different time periods, but why those specific images on those specific days at those specific regions? In other words, what "different conditions" were you selecting for here?

Reply: Thank you for the advice. The images were selected based on the availability,

time, region and overall SIT distribution. The sentence was modified to explain the rationale of selecting the scenes: (L469-L471)

*"Five cases are addressed in this study to present SITY distributions under different conditions based on the availability of data and feasibility of visual interpretation."*

268-271: Following from my general comment above, how were characteristics of the SAR images used for visual interpretation. What is the basis? There are no references here to justify the classifications.

Reply: Thank you for the comment. We added references to justify the rationale of the classification. More specifically, in the new section "3.2 Visual interpretation of SAR imagery", the theory and characteristics of sea ice classification in SAR images were introduced.

273-278: This paragraph illuminates the previous comment. The text is very "squishy". You say things like "appears to be MYI", "more likely to be MYI", "which could be interpreted as newly generated FYI". This is very qualitative and seemingly tentative. I think maybe you just need to say why something "appears to be MYI" – what is in the SAR image that leads to that conclusion and what is the basis for that?

Reply: Thank you for the advice. As mentioned in previous replies, the theory and characteristics of sea ice classification in SAR images will be introduced in the new section 3.2.

288-293: Same here. You have "could be identified as MYI", "are a typical feature of FYI".

Reply: Thanks. In the revision, we added a new section on the theory and characteristics of sea ice classification in SAR images, which become the basis of the visual interpretation from SAR.

299: I guess there is a thematic reason for the order – looking at early and late winter as "edge" cases, but it seems more logical to order these subsections chronologically: early winter, mid-winter, and then late winter.

Reply: Thank you. These cases were presented in the chronological order as suggested.

315-319: Again, very qualitative.

Reply: Thanks. Quantitative results such as Kappa coefficient and overall accuracy of the respective SIT products on each case were added.

391-446: I can see the logic of discussing the methods here – you are linking them to the performance assessed previously. However, to a large extent, this feels like it should go with the data product descriptions in Section 2.1. I guess moving this there would

make that section rather long. But I kind of feel like I get to here and I finally understand how the type products are created – after all of the evaluation and comparison. I'll leave it to the authors to decide if this fits better in 2.1 or should stay here. Or maybe, put some description in 2.1 and then the relation to the product evaluation here.

Reply: Thank you for the thoughtful comments. In the revision, we included more details of the algorithm used in the respective SIT products in the section "2.2 sea ice type products" and put the description of satellite data to section 2.1 "Microwave remote sensing data".

400: "[55]"? Is this a numbered reference?

Reply: Yes. We have corrected the citation in the revision.

434-436: Melt affects the performance in early and late winter. But melt also basically makes the algorithms ineffective in spring and summer. That is implied, but never really stated explicitly it seems.

Reply: Thank you for the advice. The sentence was modified to (L670-L671):

*"In early and late winter larger uncertainties are likely to occur due to surface processes such as wet snow attenuation and changes in brine salinity (Barber and Thomas, 1998; Voss et al., 2003; Ye et al., 2016a; Ye et al., 2016b)."*

Figure 5: Following up from above, I'm struck by the notable increase in many products in BS during 2016-2017. That is not noticeable in another region or year. There is some discussion, in relation to OSI-SAF in Figure 7, but the text doesn't really discuss this. I think this deserves some explanation in the text.

Reply: Thank you for the comment. We included discussions related to the notable increase during 2016-2017 in the section "4.1.1 Weekly MYI extent variation". The sentences read as below (L353-L356):

*"For instance, all the SITY products show increasing MYI extent in March/April 2017 except Zhang-SITY. This could be caused by the enhanced melting during this period (Raphael and Handcock, 2022; Ye et al., 2016a), which leads to the radiometric and scattering signatures of FYI similar as that of MYI therefore unsatisfactory performances of the SITY algorithms."*

Figure 5: What is the shaded green region that accompanies OSISAF? Maybe it is in the main text and I missed it, but regardless, it should also be included in the caption for the figure.

Reply: The upper and lower lines of shaded green region represent the MYI extent calculated under the assumptions of 1) regarding the "Amb" class as MYI, and 2) regarding the "Amb" class as FYI. In the new section "3.1 Estimation of MYI extent", we included such statements. (L300-L302)

*"The Amb class in C3S-SITY and OSISAF-SITY could be regarded as either MYI or FYI thus the MYI extent is calculated under both circumstances. This results in two values for the respective SITY products, one for the pixels of MYI class and the other for the pixels of MYI and Amb classes."*

Figure 6-7: This is a style/aesthetics thing, but the beige/brown for the OW seems odd. Such a color is more commonly used for land, and definitely not for ocean. I would suggest considering a different OW color – just swapping the land color (light gray) for the ocean color, would be more logical to me. Of course, you want to make sure that the colors contrast and are clearly delineated. But a good solution other than beige/brown for OW seems possible.

Reply: Thanks. The colors were modified as suggested in the revision.

**Response to RC 2:**

Dear Editors and Reviewers:

We would like to thank the reviewer for the detailed and helpful comments, suggestions, and careful checking. Comments are responded on point-by-point basis. The reviewer comments appear in black. The responses are in blue and the proposed changes to manuscript are in ***bold italics***.

**General comment**

This manuscript presents the inter-comparison of various SIT products from microwave remote sensing data. The performance of the SIT products was evaluated and the causes of differences in the products were analyzed. SIT has been used as an important information in research for global climate change and future prediction. Therefore, a comparative study on the performance of the operationally used SIT products is of high importance.

The manuscript is well written, and appropriate tables and figures are used to explain the results. It seems very meaningful to analyze the comparison results in time and space. However, in order for this manuscript to be published in the Cryosphere, more descriptions should be added about the data and methodology used in the study (requires a section on methodology). More discussion of the results is required.

Reply:

Thank you for the thorough review. Your comments and suggestions are highly appreciated. For better comparison and evaluation of the sea ice type products, we revised the manuscript from the following two aspects:

3) The "Data" section was re-structured. This section includes four sub-sections: "2.1 Microwave remote sensing", "2.2 Sea ice type products", "2.3 Sea ice age product" and "2.4 Other data". In section 2.1, specifications of the sensors and the satellite data are introduced in a chronological order, with subsections of passive/active microwave remote sensing data. In section 2.2, theory of SIT classification is introduced at the beginning, followed with the overall description of the respective SIT products in terms of grid size, projection, availability period, a summary of the satellite data used and the algorithm with necessary details. In section 2.3, sea ice age product (with evaluations from previous studies) is introduced with the input data, algorithm and evaluation of the product. In section 2.4, the SAR images along with auxiliary data are described accordingly.

4) A new section of "Methods" was added, which includes "3.1 Estimation of MYI extent" and "3.2 Visual interpretation of SAR imagery". We modified the computation of MYI extent in the revision for consistent griding, projection among

all the SIT products. In section 3.1, Information such as co-locating/re-griding the data and calculation of the MYI extent was introduced. In section 3.2, the theory and characteristics of sea ice classification in SAR images were introduced with references from previous studies and examples from our study. In addition, we interpreted all the entire SAR images, consulted with ice experts regarding the results, compared with the gridded ice classification results, and eventually gave the quantitative evaluation results (Kappa coefficient and overall accuracy for respective SAR image and SITY, SIA product).

Besides, case studies were presented in the chronological order with more discussions referring to the physical background and the algorithms of SIT products. Figures were modified for better presentation. A thorough edit of the language style and grammar were conducted. And all the references and citations have been double-checked and corrected accordingly.

Specific comments

Abstract: Specify the names of the SIT products (algorithms) analyzed in this study.

Reply: Thanks for the advice. We modified the sentence as follows.

*"This study analyzed eight daily SITY products from five retrieval approaches covering the winters of 1999–2019, including purely radiometer-based (C3S-SITY), scatterometer-based (KNMI-SITY and IFREMER-SITY) and combined ones (OSISAF-SITY and Zhang-SITY)."*

(Note that sea ice type is abbreviated as SITY in the revision)

Line 27: Please state clearly why sea ice is a sensitive indicator of climate change.

Reply: Thanks. The sentences were modified as follows in the revision (L28-L32).

*"Sea ice is an important component of the earth system. Sea ice influences climate change through two primary processes: the ice-albedo feedback and the insulating effect. Sea ice reflects more solar radiation than the ocean due to its high albedo. In addition, sea ice hinders the heat exchange between the ocean and the atmosphere because of its low thermal conductivity. Through global warming, the loss of sea ice leads to increasing absorption of solar radiation and heat flux from the ocean to the atmosphere, which further enhances the loss of sea ice and global warming."*

Line 29: It would be nice if it quantitatively indicated how much the thickness and volume of sea ice decreased.

Reply: Thanks. The sentences were modified to provide quantitative indication of thickness and volume decrease (L33-L36).

*"Its extent has reduced by 40%–50% of its average in the 1980s (Perovich et al., 2020), whereas the average ice thickness has decreased by about 1.75 m in winter in the*

*central Arctic Ocean (Rothrock et al., 2008; Kwok and Cunningham, 2015), which eventually leads to a volume loss of roughly 66% since 1980 (Petty et al., 2020; Kwok, 2018)."*

Line 35-37: Please specify how sea ice patterns affect Arctic and mid-high latitude regions and how they affect Arctic ecosystems.

Reply: Thanks. These sentences were modified as below (L40-L45).

*"The change of sea ice type (SITY) distribution impacts the climate of the Arctic and mid-high latitude regions through changes in water vapor, cloud properties, as well as large-scale atmospheric circulations such as the Atlantic Meridional Overturning (Liu et al., 2012; Screen et al., 2013; Boisvert et al., 2016; Belter et al., 2021). In addition, it influences the Arctic ecosystems by changing the habitat conditions for various Arctic species and is crucial for human activities such as shipping, tourism and resource extraction (Emmerson and Lahn, 2012; Meier et al., 2014)."*

Line 68: The authors did three scientific questions, but the second question (how we choose SIT product for different applications) is lacking in discussion.

Reply: Thank you for the advice. We deleted the second question in the revision.

Line 83: How are the microwave scattering and radiometric characteristics of MYI and FYI different?

Reply: Thank you for your advice. We revised the manuscript as suggested.

   In the "Introduction" section, we included sentences about the physical background for microwave SITY classification as mentioned above. The sentences read as below (L55-L59):

*"On one hand, brightness temperatures (Tbs) of MYI tend to be lower than that of FYI because of its low-loss, low-salinity properties (Vant et al., 1978; Weeks and Ackley, 1986). Such difference is generally larger at higher frequencies (i.e. smaller penetration depth), which reflects the distinguished physical properties of MYI and FYI at the sub-surface layer (Sinha and Shokr, 2015). On the other hand, due to the high volume scattering and low scattering loss, MYI has a relatively higher backscatter than FYI at the same frequency (Onstott, 1992)"*

   In the "Data" section (2.2 Sea ice type products), before introducing the individual sea ice types, more detailed description regarding the physical background of sea ice type classification from microwave observations was included (L126-L152). These paragraphs start with:

*"FYI and MYI can be discriminated from microwave satellite observations based on their distinctive radiometric and scattering signatures …"*

*"Depending on the ambient conditions, sea ice at different stages of development undergoes different thermodynamic and dynamic processes, resulting in disparate microwave radiometric and scattering properties of different sea ice types …"*

*"In addition to Tbs at different channels, parameters of their combinations are also used in sea ice type discrimination …"*

*"Microwave scattering of sea ice is determined by the surface and volume scattering, which is influenced by factors …"*

Line 77: Each product has a different grid size. It should be explained how it was dealt with in the comparative evaluation.

Reply: Thanks for the advice. We modified the computations in the revision to account for the factor of different grid size. SITY products in polar-stereographic projection were firstly re-gridded to the EASE2 grid before the computation of MYI extent. Details of such were presented in the new Section"3 Methods" (3.1 Estimation of MYI extent).

Line 78-143: Please describe in more detail how FYI and MYI are distinguished due to which characteristics in each SIT algorithm. For example, if a SIT product is produced based on PR and GR, an explanation is required for the differences between the values of PR and GR of ice types and why the differences occur.

Reply: Thank you for the advice. In the revision, the section of "Data" was modified as follows. The new section 2.1 was entitled with "Microwave remote sensing data", where the specification of the sensors and the satellite data used in all the SIT products were introduced. Definition (and equation) of PR and GR was introduced in this section. The new section 2.2 is "sea ice type products", where the grid resolution and details of the algorithm used in each SIT product were introduced.

Line 150: NSIDC-SIA was used as reference data. How accurate is NSIDC-SIA?

Reply: Thanks. In the revision, references regarding the evaluation of NSIDC-SIA were added and summarized in Section 2.3 "Sea ice age product" (L273-L281). The sentences read as below:

*"The accuracy of NSIDC-SIA largely depends on the ice trajectories tracking technique and ice motion data. There are mainly two sources of error in NSIDC-SIA: the tracking errors related to the coarse resolution of microwave satellite data and those induced by ice motion data vacancy near the coast. The under-sampling of ice motion along with the scheme of oldest ice age assignment lead to an overall discontinuous sea ice age distribution and overestimation of old ice (Korosov et al., 2018). Besides, ice motion velocities from buoys are generally higher than those from satellite data (Schwegmann et al., 2011). Improper interpolation approach could lead to artificial divergence in ice motion when the buoy estimation differs significantly*

*from the satellite-based data. It could result in approximately 20% less MYI in the buoy-affected region according to a numerical experiment (Szanyi et al., 2016). Such impact is mainly found in the years 1983–2005 and has been largely mitigated by tuning the interpolation approach in the current version (Tschudi et al., 2020)"*

Line 153: How and what information was retrieved from the SAR images for the SIT products evaluation should be described.

Reply: Thank you for the comment. In the revision, we added a new section regarding the theory of SAR interpretation. The theory and characteristics of sea ice classification in SAR images was introduced in the section "3.2 visual interpretation of SAR imagery". In addition, we interpreted all the entire SAR images, convert the sea ice classification results from ice types polygons to grided ice classification results, and eventually give the quantitative evaluation results.

Line 173: Is it the result of this study that different SIT distribution patterns were found in the regions selected by the authors?

Reply: It is a misleading sentence here. We will delete the sentence in the revision.

Line 185-186: Is 'divergent movements' the only cause of increase in the MYI extent?

Reply: "Advection from neighbouring regions" could also lead to increase in the MYI extent. The sentence will be modified as follows in the revision.

*"However, it can temporarily or regionally increase due to ice divergence or advection from neighbouring regions (Kwok, Cunningham, et al. 1999)."*

Line 263: The authors compared SIT daily products with the SAR images. It is necessary to discuss the comparison between the image captured at a specific time and the daily product.

Reply: Thank you for the advice. In the revision, the impact of such will be discussed in the comparison.

Line 263: The authors identified the distribution of MYI by visually analyzing the SAR image. It would be better if MYI could be determined by quantitatively analyzing backscattering or textures from the SAR images.

Reply: Thank you for the advice. In the revision, we will add a new section regarding the theory of SAR interpretation. The theory and characteristics of sea ice classification in SAR images will be introduced in the section "3.2 visual interpretation of SAR imagery". In addition, we will interpreted the entire SAR images, consulted with ice experts regarding the results, compared with the gridded ice classification results, and eventually gave the quantitative evaluation results (Kappa coefficient and overall accuracy for respective SAR image and SITY, SIA product).

Line 368-371: How are the input parameters affected by atmospheric factors and surface features? More discussion is needed.

Reply: Thank you for the advice. We further investigated the magnitude of cloud liquid water values (atmospheric factors) and its impact. It turns out that variation of the cloud liquid water path has little impact on GR. We therefore removed this sentence in the revised manuscript. Regarding the impact of surface features, more discussion was added. The sentences in the next few lines were (L590-L594) modified as below:

*"However, this parameter can be impacted by surface features (e.g., snow properties) during the winter season (Rostosky et al., 2018; Ye et al., 2019; Comiso, 1983). In the beginning and ending stages of winter, the variability of GR_37v19v can be significant when air temperature exhibits warm-cold cycles, which trigger wet-dry cycles or melt-refreeze cycles of snow (Voss et al., 2003; Ye et al., 2016b; Ye et al., 2016a), or when wet/thick precipitation suddenly appears (Voss et al., 2003; Rostosky et al., 2018)"*

Line 393: Explain clearly about the training dataset.

Reply: Thanks. To better explain the training dataset, we rephrased the sentences as below (L619-L622):

*"Some algorithms use the thresholds derived from a training dataset that does not vary with time, region or satellite sensors, namely fixed thresholds, while others employ dynamic thresholds to account for the variability of training dataset. The former algorithms work relatively well under conditions similar to the training dataset, however it gives anomalous SITY distribution results in other conditions."*

Technical comments

Line 157: SAR Wide B → SAR Wide Beam

Reply: Done

Line 199: What does (2000) mean?

Reply: It is a typo here. The reference and citation was corrected.

Line 402: Is [55] a reference number?

Reply: Yes. The citation was corrected.

Figure 4: Delete 'Jan' on the horizontal axis.

Reply: Done

Response to RC 3:

Dear Editors and Reviewers:

We would like to thank the reviewer for the detailed and helpful comments, suggestions, and careful checking. Comments are responded on point-by-point basis. The reviewer comments appear in black. The responses are in blue and the proposed changes to manuscript are in *__bold italics__*.

**Summary:**

A large, albeit shrinking portion of the Arctic Ocean sea ice cover is made of multiyear ice (MYI) that has survived at least one summer melt season. In order to more accurately assess the trend in Arctic Ocean MYI cover and the coverage of first-year ice, and to more reliably use these ice type fractions in other research areas, such as sea ice thickness retrieval, it is important to evaluate the existing sea ice type products. This study is an attempt into this direction. Nine different sea ice type products based on five different algorithms are compared with the NSIDC sea ice age data set and the MYI extent derived from it as well as with a set of five qualitatively interpreted satellite synthetic aperture radar (SAR) images. Time series of the MYI extent at daily and monthly temporal resolution are shown, inter-compared and discussed qualitatively in the light of the different algorithms, their potential limitations and post-processing steps. The performance of the different products is compared for specifically selected sub-regions of the SAR images.

I have a number of concerns with this manuscript which I summarize in my general comments and detail in my specific comments.

I also would like to note that the manuscript is difficult to read because of quite a number of strange formulations and problems with English grammar.

General comments:

GC1: As the authors state, this is one of the first (kind of) comprehensive evaluation of sea ice type products. This calls for provision of a solid physical background of the sea ice and its snow cover as relevant for its remote sensing using active and passive microwave instruments. This element is missing and jeopardizes the usefulness of the entire manuscript.

Reply: Thank you for your advice. The physical background of sea ice type classification from passive and active microwave remote sensing has be added in the revision. Such descriptions are presented in the following sections: 1) section "1 Introduction", where the basic theory of FYI/MYI discrimination from microwave observations is summarized; 2) section "2.2 sea ice type products", where the theory of

sea ice classification is introduced at the beginning of this section, before introducing all the SIT products; 3) section "3.2 visual interpretation of SAR image", which presents the physical background and sea ice scattering characteristics in SAR image.

GC2: The description of the input satellite data and the algorithms used in the products as well as in the one major evaluation data set used is very heterogenuous and not complete for the understanding of the manuscript and its results. At least two products (NASA-Team MYI concentration and ECICE MYI concentration) are missing in addition.

Reply: Thank you for the advice. To better illustrate the input satellite data and algorithms used in the products, we restructured the Section of Data. In the revised manuscript, Section "2.1 Microwave remote sensing data" describes the satellite data, whereas Section "2.2 Sea ice type products" provides details of the algorithms used in each ice type product.

As for the two products you mentioned, since this study focuses on the inter-comparison of sea ice type products that tells ice type classifications without information on the specific fraction. We therefore did not include NASA-Team and ECICE MYI concentration products. For clarifications, we modified the sentences in the section "Introduction" (P2L59-L78), which reads as below:

*"There exist different algorithms which either provide a fractional MYI/FYI coverage or assignment of one or the other ice type (e.g. MYI and FYI) to a grid cell. The former, referred to as sea ice type concentration (SITC) algorithms, includes algorithms such as the NASA Team and ECICE algorithm …The latter, referred to as SITY algorithms, include …"*

GC3: The inter-comparison contains, if at all, little quantitative results. The results often appear to be quite hypothetical. As I see it, there are two main reasons for that. At first, the NSIDC sea ice age data set used as the main evaluation data set requires an evaluation that justifies its usage for the purpose of this manuscript. In addition, there is a methodological inconsistency behind comparing daily sea ice type products with weekly sea ice age data. Secondly, the SAR images used are only interpreted in a qualitative way. With that they can be used as a means for a consistency check of the general performance of the sea ice type products - but only within the error margin proposed by this manual interpretation. Both together clearly reduces the value of this manuscript, which has the character of a pure, qualitative inter-comparison study with little in-depth recommendations resulting from it for i) which product to pick and ii) how to improve which product in which way.

Reply: Thank you for your advice. In the revision, we included more quantitative results and analysis in the manuscript as suggested. Regarding the two points you mentioned in the comments. 1) the NSIDC sea ice age data is used for overall comparison.

References regarding the evaluation of the sea ice age product were added in the revision. In addition, weekly MYI extent is calculated from sea ice type products for consistent comparison, where more quantitative comparison is included in the revised manuscript. 2) In the revision, we interpreted all the entire SAR images as suggested, compared with the gridded ice classification results, and eventually give the quantitative evaluation results (Kappa coefficient and overall accuracy for respective SAR image and SITY, SIA product).

For the manual interpretation, the theory and characteristics of sea ice classification in SAR images is introduced in the new section "3.2 visual interpretation of SAR imagery". In addition, the interpretation is conducted under the supervision and consult with sea ice experts who are experienced in SAR interpretation for more reliable results.

GC4: The discussion of the results is not well linked to the existing literature.
Reply: Thank you for the advice. In the revision, we included more recent literatures in the manuscript to be better linked to recent studies.

Specific comments (contain some typos / editoral comments):
Abstract:
- I recommend that you consider to find and use a different acronym for sea ice type because I find "SIT" very often used as an acronym for sea-ice thickness. A possible alternative could be SITY. Or, since "type" is not really that long compared to the words, e.g. thickness or concentration, you might also consider write the full expression all the time. But "SIT" is a bit unfortunate.
Reply: Agree. We use "SITY" as the abbreviation for "sea ice type" in the revised manuscript.

- I also recommend that you very briefly describe the various products named in the abstract. Perhaps they can be categorized into those products that rely solely on C-Band or Ku-Band data and/or products that use both active and passive microwave data? Please check the maximum allowed length of the abstract and perhaps delete details towards the end for more clarity of what types of products you did compare.
Reply: Thank you for the advice. We categorized the sea ice type products into three groups, which reads as below:
*"…, including purely radiometer-based (C3S-SITY), scatteromter-based (KNMI-SITY and IFREMER-SITY) and combined ones (OSISAF-SITY and Zhang-SITY)."*

- I recommend to state upfront that by "sea ice type" you merely refer to multiyear ice and first-year ice. As you know, there is a number of other sea-ice types which you, however, not appear to take into account.

Reply: Agree. We added clarifications in Section "Introduction" in the following two parts:

1) P2L56-L57, *"... Among them, most algorithms focus on the discrimination of MYI and FYI. These algorithms identify SITY (i.e. the discrimination of MYI and FYI in this study) based on ..."*

2) P3L90, *"This study aims to investigate differences among the SITY products and give comprehensive evaluations on the identification of MYI and FYI."*

- L13/14: "towards sea ice ... images" --> "against a sea ice age product and compared with five Synthetic Aperture Radar images"
Reply: Done.

- While you write in Lines 14/15 about results found at daily and monthly temporal resolution it is not clear whether all products used come at daily temporal resolution. I also note that the sea ice age data set comes at weekly temporal resolution.
Reply: Agree. In the revision, we specified the temporal resolution of SITY and SIA product, e.g. P1L12 "... eight daily SITY product ...", P1L15 "... a weekly sea ice age product (NSIDC-SIA) ...", and only mentioned the "average" difference (P1L15).

- L14/15: Please also see my over-arching comment to the conclusions.
Reply: Thank you for the comment. We carefully checked all the numbers mentioned in the manuscript in the revision. For the exceptional large difference, we add statements in Section "4.1.2" (P13L382-L385) and refer to the discussions in Section "5 Discussion".

- You might want to re-phrase "anomalous fluctuations" because it is not clear what you mean by that in the context of an underestimation (Line 17).
Reply: Agree. Thank you for the comment. The sentence is rephrased as below:
*"In the ASCAT period, KNMI-SITY tends to overestimate MYI (especially in early winter), whereas Zhang-SITY and IFREMER-SITY tend to underestimate MYI."*

- Under (3) you write about details with respect to the classification (Line 23). Is the retrieval of all products investigated based on a classification approach?
Reply: Thank you for the comment. The retrieval of all products is not based on a classification approach. In the revised manuscript, we give more detailed description about the classification algorithms used in the respective SITY products. On the other hand, this study focuses on the comparison and evaluation of current SITY products. Further investigation of classification algorithms could be done in future however is out of the scope of this manuscript.

- I have the feeling that the "Additionally, the change of separation pattern ... SIT method" (Lines 24/25) could be deleted for the sake of having more room for the above-mentioned suggestions.

Reply: Done.

Lines 41-57: I suggest to better structure this paragraph and in addition provide more background information. Specifically I recommend to

i) Tell the reader that by sea ice type discrimination you are referring to distinguishing between FYI and MYI;

Reply: Agree. We add descriptive sentences/phrases at the beginning of this paragraph (P2L56-L57). The modified sentences are:

*"…Among them, most algorithms focus on the discrimination of MYI and FYI. These algorithms identify SITY (i.e. the discrimination of MYI and FYI in this study) based on the distinct radiometric and scattering characteristics of different ice types."*

ii) Write what the fundamental differences in the physical properties of these ice types are that allow us to separate them by means of their microwave signature (be it for active or passive microwave sensors);

Reply: Thank you for the advice. We added sentences with brief description of the fundamental principles for microwave SITY classification in this section (More detailed description can be found in the section "2.2 Sea ice type products"). The new sentences in this section are as follows (L55-L59):

*"On one hand, brightness temperatures (Tbs) of MYI tend to be lower than that of FYI because of its low-loss, low-salinity properties (Vant et al., 1978; Weeks and Ackley, 1986). Such difference is generally larger at higher frequencies (i.e. smaller penetration depth), which reflects the distinguished physical properties of MYI and FYI at the sub-surface layer (Sinha and Shokr, 2015). On the other hand, due to the high volume scattering and low scattering loss, MYI has a relatively higher backscatter than FYI at the same frequency (Onstott, 1992)"*

iii) Explain more clearly - but still briefly - what the different retrieval approaches are. It is for instance not clear whether the main approach used is a classification. The NASA-Team algorithm (see below) does not use a classification, neither does ECICE.

Reply: Agree. In the revised manuscript, we firstly summarize the approaches in terms of fractional and binary results, later mention the typical algorithms accordingly. The modified sentences are (L59-L66):

*"There exist different algorithms which either provide a fractional MYI/FYI coverage or assignment of one or the other ice type (e.g. MYI and FYI) to a grid cell. The former referred to as sea ice type concentration (SITC) algorithms, includes*

*algorithms such as the NASA Team algorithm and ECICE algorithm, …. The latter referred to as SITY algorithms, includes …"*

iv) Move information about evaluation results obtained by others so far into the next paragraph (see Lines 47/48: "By comparing ... Kwok 2004)."
Reply: Agree. We move this information to the next paragraph in the revised manuscript.

v) Mention that different methods exist which either provide a fractional MYI/FYI coverage or a binary classification (or assignment of one or the other ice class to a grid cell).
Reply: Agree. In the revised manuscript, we summarize the approaches in terms of fractional and binary results, respectively. The modified sentences are (L59-L66):
*"There exist different algorithms which either provide a fractional MYI/FYI coverage or assignment of one or the other ice type (e.g. MYI and FYI) to a grid cell. The former, referred to as sea ice type concentration (SITC) algorithms, includes algorithms such as the NASA Team and ECICE algorithm, …. The latter, referred to as SITY algorithms, include …"*

- In addition, I recommend to delete the Lomax et al. 1995 paper and instead include literature related to the NASA-Team algorithm and to the ECICE algorithm which both permit to compute FYI and MYI fractions and are both so far missing completely in your list. I am wondering why you are not considering these products as well in your inter-comparison. I am also wondering whether it would not make sense to get hands on the MYI data sets created by Ron Kwok and used in various publications of him and his group.
Reply: Thank you for the advice. In the revised manuscript, we specify that the sea ice type concentration algorithms include both kinds of approaches. We therefore keep all the references you mentioned in the comment.
     Regarding the MYI data sets created by Ron Kwok, it is not included in the inter-comparison since our manuscript focus on the inter-comparison and evaluation on SITY products, which refer to the binary products (whereas Kwok's dataset is a dataset of fractional MYI/FYI coverage). As mentioned in previous replies, we also added sentences in this paragraph on clarifying the focus of this manuscript.

Lines 58/59: "Comparison ... methods" --> While an evaluation of products is per se an excellent idea and the improvement of the used retrieval methods a good motivation, I strongly suggest to provide 1-2 sentences that specify more clearly why it is important to (finally) provide a more comprehensive evaluation of these products. The first paragraph of your introduction only tells the reader that sea ice type is important. But

requirements about the accuracy and a specific example where an error in the sea ice type distribution of, e.g. 50%, would have which implications is not yet given in a convincing way.

Reply: Thanks for the advice. Implications of large errors can be found in previous studies however have not been quantified to our knowledge. This is also one of the reasons that we need a more comprehensive evaluation of these products.

Line 61: Why "limited"? There are plenty of ship observations (see e.g.: https://www.cen.uni-hamburg.de/en/icdc/data/cryosphere/seaiceparameter-shipobs.html)

Reply: Agree. "limited" is deleted in the revised manuscript.

Lines 62-64: "... some MYI ... in ice charts." --> I don't understand this sentence; please consider to re-phrase it.

Reply: Thanks for the comment. The sentence is rephrased to (L83-L85)

*"While the ice chart is used as "ground truth" in some validation (Aaboe et al., 2016), some areas of MYI in the ice charts correspond to areas with MYI concentration of approximately 50% or greater (Lindell and Long, 2016a), indicating the overestimation of MYI in ice charts."*

Line 67: "operational" --> please check what you mean here by operational. Do you mean existing? Or are you really referring to all sea-ice type products that are currently operationally (aka daily) produced and provided to the users?

Reply: We mean the products that are currently operationally produced and provided to the users. To avoid unnecessarily misunderstanding, we delete this word in the revision.

Lines 83-86: "Microwave radiometer ... 2016)" --> As stated already in the context of the introduction, it would make a lot of sense to include a paragraph that clearly describes the relevant physical properties of the different sea ice types that are relevant for their discrimination in the different active and passive microwave signals. This is required to understand the algorithm details and to understand their limitations (also during the freezing and/or shoulder seasons) and would be important for the discussion section as well. Since this is the first paper of this kind it is certainly worth to dig into physics here.

Reply: Thank you for the advice. We have revised the manuscript as suggested.

In the "Introduction" section, we include sentences about the physical background for microwave SITY classification as mentioned above. The sentences included additionally are as follows (L55-L59):

*"On one hand, brightness temperatures (Tbs) of MYI tend to be lower than that of FYI because of its low-loss, low-salinity properties (Vant et al., 1978; Weeks and Ackley, 1986). Such difference is generally larger at higher frequencies (i.e. smaller penetration depth), which reflects the distinguished physical properties of MYI and FYI at the sub-surface layer (Sinha and Shokr, 2015). On the other hand, due to the high volume scattering and low scattering loss, MYI has a relatively higher backscatter than FYI at the same frequency (Onstott, 1992)."*

In the "Data" section (2.2 sea ice type products), before introducing the individual sea ice types, we included several paragraphs to describe the physical background for ice type classification from microwave observations (L126-L152). These paragraphs start with:

*"FYI and MYI can be discriminated from microwave satellite observations based on their distinctive radiometric and scattering signatures …"*

*"Depending on the ambient conditions, sea ice at different stages of development undergoes different thermodynamic and dynamic processes, resulting in disparate microwave radiometric and scattering properties of different sea ice types …"*

*"In addition to Tbs at different channels, parameters of their combinations are also used in sea ice type discrimination …"*

*"Microwave scattering of sea ice is determined by the surface and volume scattering, which is influenced by factors …"*

Line 91: "on coarse resolution" --> Please write in the text more clearly the grid resolution of the data and, if relevant, also the native resolution of the data used as input.
Reply: Thank you for the advice. In the revised manuscript, the "Data" section includes four subsections: "2.1 Microwave remote sensing data", "2.2 Sea ice type products", "2.3 Sea ice age product" and "2.3 Other data". In Section 2.1, specification of the microwave sensors is introduced with the native resolution. Table A1 is included to show detailed info. In Section 2.2, the grid resolution of each sea ice type product is specified.

Lines 93/94: I can understand that Tb measurements are corrected for the atmospheric influence because it disturbs the sea ice signal. I cannot understand why you need to correct the Tb measurements for sea ice concentration ... What I can imagine is that you use an additional sea ice concentration product to restrict the analysis of the sea ice type on the sea-ice covered area. If this is the case then please write it accordingly. However, admittedly this would contradict a bit the next sentence about the Bayesian approach do discriminate open ocean and sea ice. In short: You need to rephrase these statements.

Reply: Thank you for the suggestion. We meant to say that the Tb measurements are corrected for the atmospheric influence by using auxiliary data and radiative transfer model. One of the auxiliary data is sea ice concentration, which is used to estimate the surface emissivity in the radiative transfer model. Note that the authors do not correct the Tb measurements for sea ice concentration. We rephrased these statement in the revision. In addition, description of each SITY product is organized in the following way: 1) summary of the product, including radiometer/scatterometer/combined-based, the resolution, availability, versions and so on. 2) three main procedures: pre-processing, classification approach, and post-processing. Description of C3S-SITY can be found in section 2.2.1 C3S-SITY.

Line 96: I suggest to remove the "further" and to also provide an equation of how the gradient ratio is computed.
Reply: "further" is deleted. In the revised manuscript, there are four subsections in the "Data" section. The equation of gradient ratio is included in the new section "2.1 microwave remote sensing".

Line 98: Where are these "fixed target areas" located? How are these selected? How large are these? Do these change annually? And: Why are these fixed?
Reply: Thanks for the comment. The sentence is misleading. It was modified to the sentences as below (L184-L187):
*"The daily updated probability density functions (PDFs) of the collected training data is dynamic in time and captures the seasonal and interannual variabilities. The pre-defined areas over which the data are collected are the climatological MYI and FYI regions, which are north of Greenland and Canada with longitude between 30°W and 120°W for MYI, and the Kara Sea, Baffin Bay, Laptev Sea and the Bay of Bothnia for FYI."*

Lines 100-102: "In 2021 ... scheme" --> My impression is that you are not including data of this new version into your comparison. Therefore I recommend to move this announcement towards the end of your paper, e.g. into the discussion where it could fit with your outlook / description of which improvements are (already) underway. But perhaps 2021 was a typo ...?
Reply: Thanks for the comment. These sentences were deleted here. In the revised manuscript, two versions of C3S-SITY are introduced in the first paragraph (L171-L174).

Lines 103-115: I recommend to comment on / give more details on:

i) the fact that the OSISAF-SIT is based on a very heterogeneous set of input parameters and on changes in the training data set (L111/112), which both could have an impact on the sea ice type product in terms of its consistency over time;

Reply: Thank you for the advice. We made substantial modifications on the introduction of Data and Products. In the revised manuscript, there are 4 subsections in the new "Data" section. In the new section "2.1 microwave remote sensing", we introduce all the microwave remote sensing data used in the various SITY products. In the new section "2.2 Sea ice type products", we then describe each product in details. Introduction of each product will include the following information: 1) grid resolution, projection, the satellite data used in different periods; 2) the algorithm used in the product, where we could include more details.

ii) what a "sigma_nought" is (Line 112) and in which way this variable is used (is it corrected towards a common incidence angle? for instance); what is the incidence angle range used? (Compare the next paragraph where you are comparably detailed as far as it concerns the Ku-Band scatterometers.);

Reply: Thanks for the advice. Substantial modifications have been made in the revision. More details regarding the product and satellite data have been included. As mentioned in the previous reply, details of the microwave remote sensing data were given in the new section 2.1, where it mentions that the incidence angle of ASCAT ranges from 25° to 65°. Backscatter from ASCAT is normalized to a reference incidence angle of 40° and later used as an input for the classification.

iii) what the native resolution of the scatterometer data is;

Reply: The native resolution of ASCAT is around 25km x 34 km, this information is included in the new Table A1.

iv) what a "swath projection" is (Line 112).

Reply: Thanks for the comment. It is an inappropriate expression here. Description of OSISAF-SITY has been written and can be found in section 2.2.2. The relevant sentence is modified as below (L214-L215):

*"Ice types and their probabilities are derived using classifiers based on the respective observational parameters (GR_19v37v and σ_0), where swath data of different sensors are used."*

- I furthermore find the introduction of AMSR2 and AMSR-E the way done confusing. AMSR2 is available since July 2012 but it is included since 2016; whether AMSR-E data were at all used is not clear but AMSR-E is introduced.

Reply: Thank you for the suggestion. The description is misleading. In the new section 2.1, satellite data is introduced in a chronological order. Which data is used in the respective SITY product can be found in section 2.2.

- You describe the different sensors used with different degrees of detail; for instance you do not mention that SSM/I and SSM/IS are multi-channel radiometers with a number of frequencies while you do so for AMSR-E. You refer to "coarse" (previous paragraph) and "medium-resolution" (Line 106) as well as "higher spatial resolution" (Line 109) without a specific motivation. Why is it important to know the spatial resolution? How does the product (also the other products) actually deal with input data being available at different spatial resolutions?

Reply: Thank you for the advice. In the new section 2.1 "Microwave remote sensing data", the satellite data is introduced in a chronological order meanwhile with same level of details. The microwave radiometer data is introduced as two series of dataset: one includes SSM/I and SSMIS, the other are AMSR-E and AMSR2. Spatial resolution of the sea ice type products is mentioned in the new section 2.2.

- What is the spatial resolution achieved by ASCAT and what is the polarization used?

Reply: In the revised manuscript, details of ASCAT were presented in the new section 2.1 and Table A1. ASCAT provides only vertically polarized backscatter. The grid spacing of ASCAT data is 12.5 km.

- What are "given weights" (Line 113)? How are these defined?

Reply: The "given weights" are dependent on the distance between centers of grid and footprint. These sentences were modified as below (L215-L216):

*"The probabilities are then gridded based on the distance between each footprint and the polar stereographic grid."*

- L113-115 you could rephrase for improved clarity along the lines: Both C3S-SIT and OSISAF-SIT provide, in addition to the pure ice type classes FYI and MYI, an ambiguous ice type class that represents an unknown mixture of both ice types, referred to as "Amb". The products are provided with ...

Reply: Thank you for the advice. We include such expression in the revised manuscript. For instance, on L190-L192:

*"Note that C3S-SITY defines an ambiguous ice type class (referred to as Amb) in addition to the pure MYI and FYI classes. The Amb class represents sea ice with a low classification probability. It may be both pure MYI, FYI or a mixture of FYI and MYI (Aaboe et al., 2021c)"*

Lines 120/121: "ASCAT is ..." certainly belongs either to the paragraph where you introduce ASCAT data for the first time. Or, alternatively, you could think about adding a sub-section wherein which you introduce all sensors and their specifications as far as relevant for this paper. Table 1 provides not enough information.

Reply: Thank you for the advice. We added a new subsection "2.1 Microwave remote sensing" in the revision, where the sensors and specifications are introduced.

Lines 116-126:

- I recommend that within this paragraph you underline more clearly that KNMI-SIT is actually a synonym for three different sea ice type products of which you include two into your evaluation. I would then also avoid speaking of "the KNMI-SIT" but in general speak about KNMI sea-ice type products and then define KNMI-Q and KNMI-A as those you are referring to henceforth.

Reply: Thank you for the advice. In the revision, we modified the sentences to underline more clearly the meaning of each acronym, and specified the two products we include in the evaluation is KNMI-Q and KNMI-A. The modified sentences are as below (L221-L225):

*"KNMI-SITY is a series of purely scatterometer-based products with grid spacing of 12.5 km in a polar stereographic projection. The scatterometer data used includes ERS, QSCAT, OSCAT and ASCAT, which results in four respective SITY products, referred to as KNMI-E, KNMI-Q, KNMI-O and KNMI-A respectively, available during the periods of 1992–2001, 1999–2009, 2010–2013 and 2007–2016. In this study, KNMI-Q and KNMI-A are included in the comparison and evaluation considering the comparable input data as other products."*

- While your refer to swath and grid in the previous paragraph you don't do this here. In which form are the data of the different scatterometers used within the sea-ice type retrieval? What is the grid resolution? What is the native resolution of the OSCAT and QuikSCAT data? Please refer to Table 1 / Figure 1 for clarification in terms of the time periods the different satellite data and hence sea-ice type products are available. Reading the text it is not clear which time periods the different (?) products cover.

Reply: Thank you for the advice. In the revision, specifications of the sensors are introduced in the new section 2.1 with same level of details, whereas the grid resolution and algorithm are introduced in the new section 2.2. For SITY product with several times of switching input data, we refer to the Table 1/Figure 1 when describing details such as periods.

- L123/124: "In KNMI-SIT ..." --> Does this apply to all three products? Or is there a merged product? Is this classification done after FYI and MYI have been separated?

What is the difference in the microwave signal that is exploited to separate SYI from older ice?

Reply: Yes, it applies to all the KNMI sea ice type products. The modified sentences are as follows (L229-L234):

*"In the stage of classification. KNMI-SITY uses a refined Bayesian algorithm for ice/water discrimination, based on the probabilistic distances to the Geophysical Model Functions of ocean wind and sea ice. The sea ice pixels are then classified into FYI, second-year ice (SYI) and older MYI using VV polarized backscatter with two thresholds, which are determined from the data of March of each year (Belmonte Rivas et al., 2018)."*

- L125/126: "In this study, backscatter ... SIT products." --> I don't understand this sentence; please re-phrase it.

Reply: Thank you for the advice. We deleted this sentence to avoid misunderstanding.

Lines 127-131:

- Like for the previous paragraph it is not entirely clear whether IFREMER-SIT is again just a synonym for the two other products IFREMER-Q and IFREMER-A or whether these two are merged to form one product.

Reply: Thank you for the advice. These sentences were modified to (L236-L238)

*"IFREMER-SITY is another series of purely scatterometer-based products, with grid spacing of 12.5 km in a polar stereographic projection. There are two SITY products in IFREMER-SITY, which use QSCAT and ASCAT data for the respective years of 1999–2009 and 2010–2015, referred to as IFREMER-Q and IFREMER-A, respectively."*

- I note that you give a few details about IFREMER-A but not about IFREMER-Q.

Reply: Thank you for the comment. In the revised manuscript, we modified the sentence as below to avoid providing different details of the product (L239-L240).

*"In the first stage, the backscatter coefficients at different incidence angles (e.g., ASCAT backscatter) are normalized to the value at a constant incidence angle of 40° to account for the influence of varying incidence angles."*

- I am not sure I understand what you mean by "series of time-varying thresholds" ... What is this "series"? Are you referring to a time series of backscatter data for several winters as written in Line 130? What do you mean by "seasonally consistent"? That the values agree with each other through the course of the freezing season?

Reply: Thank you for the question. To answer the questions you raise here, we modified the sentences as below (L240-L242):

*"In the core classification, a set of day-to-day-varying thresholds are then used for the discrimination between MYI and FYI. These thresholds are derived from the backscatter data of several winters and are found to be inter-annually consistent (Girard-Ardhuin, 2016)"*

- While we learn here that the product is gridded to a polarstereographic grid, there is no information about the grid in the previous paragraphs.
Reply: Thanks. In the revision, The projection and grid spacing information of all the SIY products. Were introduced in the first paragraph of each SITY description.

Lines 132-137:
- "employs adaptive" --> "employs an adaptive"
Reply: Done

- "based on the thought of clustering" could possibly be re-phrased. What kind of clustering approach is used? K-means?
Reply: Thank you for pointing it out. It has been modified to "based on K-means clustering" in the revision.

- For the other approaches listed above that utilize radiometer data you state that the gradient ratio of the 37 and 19 GHz channels with vertical polarization are used. Which channels are used here?
Reply: It uses horizontally polarized Tbs at 37 GHz and the backscatter from scatterometer. It is included in the beginning of section 2.2 about how the gradient ratio is calculated.

- Is it correct that the approach combines coarse resolution radiometer data (what is the resolution? How is the difference in spatial resolution between SSM/I / SSMIS and AMSR-E/2 taken into account?) with fine resolution scatterometer data? What kind of radiometer data are used? Daily gridded? Swath? Which grid?
Reply: As mentioned in the afore-mentioned replies, information regarding the microwave satellite data was introduced in the new section 2.1 in the revision.

- You write that QuikSCAT and ASCAT are used successively. Does this mean that you use QuikSCAT data until the very end of its nominal time with regular data provision in 2009 (?) and only afterwards ASCAT? How does the algorithm deal with the substantial difference in sensing geometry and coverage?
Reply: To answer the questions here, we added the following sentences in the revision (L249-L250).

*"Scatterometer data from QSCAT and ASCAT is used successively in Zhang-SITY, with QSCAT data until November 23, 2009."*
The ASCAT data is normalized to backscatters at constant incidence angle. In addition, it is found that the algorithm in Zhang-SITY is not highly influenced by the use of different scatterometer data. Such details have been included in the paragraph of describing the algorithm used in Zhang-SITY (L256-L258).
*"It is an unsupervised classification approach thus does not require the selection of training dataset. In addition, the results from different sensors are generally consistent thus no further processing is conducted for the satellite data (Zhang et al., 2019)."*

Lines 140/141: "climate consistent data record SIT products" --> Given the heteogeneity of the products described in terms of the spatial resolution of the input data and the various combinations of frequencies and potentially also polarizations used, I doubt that any of the above-mentioned products deserves yet an assignment into the group "climate consistent data record". Therefore, personally, I would skip this whole last paragraph (Lines 138-142); I don't think it is relevant for the paper.
Reply: Agree. We deleted this paragraph in the revised manuscript.

Lines 147-150:
- The description of the sea-ice age product should be revised according to the information given in the more recent paper by Tschudi et al., Cryosphere, 14, 2020. In particular statements like "tracking of ice trajectories" should be avoided as should be wrong information about how the data set is derived like "passive and active microwave observations". This ice age data set is derived from the NSIDC sea ice motion data set (which in some way is described in the same paper).
Reply: Thanks. The suggested reference has been included. The sentences have been modified in the revision. Please find it on L266-L268.
*"It is derived by tracking trajectories of virtual Lagrangian ice parcels of each grid cell. The ice motion data used in the tracking process is based on passive and active microwave observations as well as auxiliary data such as drifting buoys (Fowler et al., 2004; Maslanik et al., 2011; Tschudi et al., 2020)."*

- The paper by Korosov et al, 2018, is about the deficiencies and limitations of the NSIDC sea ice age data set but should not be cited in the context of its description. I can kind of guess that you added this information "limited by the simple drift model and the oldest ice age assignment of grids" to illustrate that the NSIDC sea ice age data set may have its limitations but this would need to be explained in far more detail than in half a sentence. In fact, it is likely that the sea ice age product overestimates the presence of old ice and therefore is biased towards old ice. Whether this already applies

to the discrimination between FYI and SYI I don't know; this I leave to you to think about.

Reply: We agree that the sea ice age product tends to overestimate the presence of old ice if we regard the pixels fully covered by the ice at "the age" (given by the product). Since SYI is regarded as MYI in this manuscript, the overestimation applies to the discrimination between FYI and SYI also. We removed the citation and rephrased the description here.

Line 157: "Images are" --> "All five SAR images are ..."
Reply: Done.

- Was any filtering (speckle?) applied?
Reply: No. And it was a typo for the pixel size of Sentinel-1, which should be "40 m" not "160 m".

Lines 159/160: "the geolocations and acquiring dates of the SAR images" --> "the location of the five SAR images". There is no acquisition date given in Figure 2. Hence the acquisition dates are missing and the time difference with respect to the sea ice type products the SAR images are compared to is unknown. This needs to be included in the revised version of the manuscript.

Reply: In the revision, we have included the acquisition date of the SAR images. The modified figure is as below:

[Figure]

Lines 161-165:
- "For better interpretation of SAR images" --> This motivation needs to be explained better. It is not at all clear why, for the interpretation of the few SAR images used, these

two additional data sets are required. What is the problem with the SAR images that such data are needed?

Reply: The interpretation of the SAR images does not need these data. We will therefore delete these sentences in the revision. These sentences were modified as (L289-L292) *"Auxiliary data from atmospheric reanalysis is used in addition to the SAR images in the validation. The reanalysis data includes 2 m air temperature and 10 m wind from the ERA5 hourly dataset, produced using 4D-Var data assimilation and model forecasts in CY41R2 of the European Centre for Medium-Range Weather Forecasts integrated Forecast System (ECMWFs) (Hersbach et al., 2018)."*

- Why do you use the CERSAT/Ifremer product - which appears to be quite heterogeneous in terms of the input data when you could have used the NSIDC sea ice motion product coming at 25 km grid resolution on an EASE grid and with daily temporal resolution.

Reply: We admit that it is not appropriate to use such heterogeneous data in the manuscript. The data is also not required for the SAR interpretation. We therefore deleted these sentences.

- What is the grid resolution of the ERA5 data and how did you co-locate these data with the sea ice type products and/or the SAR images?

Reply: As mentioned in the previous replies, these sentences were deleted in the revision.

Line 172: The naming of the regions is partly wrong and needs to be corrected. What you call ESS is actually the combined area of the East Siberian Sea and the Laptev Sea. What you call BS is not the Barents Sea but the combined area of the Beaufort Sea and the Chukchi Sea.

Reply: Agree. Naming of the regions was modified to "the central Arctic Ocean (CAO), the East Siberian and Laptev Seas (ESL), along with the Beaufort and Chukchi Seas (BCS)" in the revised manuscript. Abbreviations were updated accordingly.

Line 180:
- Here, in this line your write "extent is calculated by general extent of pixels", in line 176 you write "MYI extent is estimated as the integral of all pixels specified ..." . Both formulations are not to the point and not specific enough. I recommend to re-phrase in both cases along the lines "We computed the MYI extent as the sum of the area of all grid cells classified as MYI."

Reply: Agree. In the revision, we added a new section regarding the estimation of MYI extent, Section 3.1 "Estimation of MYI extent". Details can be found on L294-302, the sentences start with:

*"For the inter-comparison, the Arctic MYI extent is calculated from the respective SITY and SIA products …"*

- In this context I have two questions. 1) Did you use a common land mask? Or is this not required because the region of interest that is delineated by the red line in Fig. 2 is clear of any land influence? 2) Did you take into account that the grid cell area is only a constant in the EASE grid projection while it changes with latitude for the products in polar-stereographic projection? If you did not take this into account yet you must correct your computations.

Reply: Thank you for the comment. Here are the replies to the two questions: 1) No, we did not use a common land mask because of the reason you mentioned. 2) No, it was not accounted for in the original manuscript. We therefore corrected the computations in the revision. SITY products in polar-stereographic projection are firstly re-gridded to the EASE grid before the computation of MYI extent. Details of such have been presented in the new Section"3 Methods" (3.1 Estimation of MYI extent).

Lines 183++: I have a conceptual difficulty with comparing daily sea ice type maps with weekly sea ice type maps derived from the NSIDC sea ice age. The comparison would be much more meaningful if you would average all daily products over every single week also used in the sea ice type product derived from the sea ice age. After all, this is your main data product for the inter-comparison.

Reply: Thank you for the advice. In the revised manuscript, we calculated the weekly averaged MYI extent from the SITY products with the NSIDC-SIA product. All the relevant numbers have been modified accordingly. Plots of the weekly averaged MYI extent was presented in Fig. 4. The modified figure is as below:

[Figure]

Line 184: "decreasing trend" --> What you possibly mean is a decrease or an increase in the MYI extent (over time) or a positive or negative trend of the MYI extent (over time). A decreasing (or increasing) trend, in contrast, is a trend that changes its value with time or, in other words, if this was a linear trend then the slope of the trend line would decrease (or increase) with time. Therefore, please correct your writing accordingly throughout the manuscript.

Reply: Thank you for the advice. Such expressions have been corrected throughout the manuscript as suggested. This sentence has been modified as follows (L349-L350).

*"…, the overall Arctic MYI extent cannot increase over the winter – it can only decrease through ice advection out of the Arctic."*

Line 186: "the divergent movements" --> Which movements? Movements of what? Are you referring to divergent sea ice motion?

Reply: Thank you for the advice. This sentence has been modified to (L350-L351)

*"it can temporarily or regionally increase due to ice divergence or advection from neighbouring regions (Kwok et al., 1999)."*

Line 194: "to the NSIDC-SIA extents ... 2-3 years" --> You stated earlier that you compute the MYI extent from the NSIDC-SIA data set by summing over all grid cells exhibiting a sea ice age of 2 years or older. Hence you can simply write "to the MYI extent derived from the NSIDC SIA."

Reply: Thanks. It has been modified as suggested.

Lines 199/200: Given the fact that the entire Arctic Ocean (i.e. approximately the region of your study) has a size of about 7 x 10^6 km2, and the fact that rarely the entire Arctic Ocean is covered with MYI, this difference is far above being reasonable and requires more explanation. It is a 100% error.

Reply: Thank you for the comment. In the revision, we check all the numbers presented in the manuscript, and confirm that the largest difference was correctly calculated. Explanations have been added accordingly (L387-L390).

*"Between any two SITY products, the difference in weekly MYI extent is up to 4.5 ×10^6 km^2, which occurs between OSISAF-SITY and KNMI-A in late October 2008. Considering the size of the study region (about 6.5×10^6 km^2), such discrepancy is significant. This is caused by the relatively low MYI extent from OSISAF-SITY and the exceptional high value from KNMI-A in late October, the reason for which will be discussed in Section 5."*

Line 200: Are these extent estimates for class "ambiguous" reasonable? How do these values relate to the entire MYI extent? Please be more critical about and more specific within your interpretations.

Reply: Thank you for the comment. These are averages of the extent estimated from the "ambiguous" pixels. As explained in the first paragraph of the Section "Temporal reanalysis", "Amb class in C3S-SITY and OSISAF-SITY could be regarded as MYI and FYI thus the MYI extent is calculated under both circumstances". Large extent for the "ambiguous" class means large number of pixels with high uncertainties on ice type discrimination, which have atypical microwave signatures of MYI/FYI and usually lead to large discrepancies between the SITY products. In the revision, we included a separate paragraph in the section "4.1.1 Weekly MYI extent variation" (L375-L380), which reads as below:

*"For the SITY products with the Amb class, the average extent of this class is 0.21 $\times10^6$ km^2, 0.26 $\times10^6$ km^2 and 0.26 $\times10^6$ km^2, respectively, for C3S-1, C3S-2 and OSISAF-SITY. As described in Section 2.2, these Amb pixels have atypical microwave signatures of MYI/FYI thus high uncertainties on ice type discrimination. Compared with the average Arctic MYI extent difference against NSIDC-SIA (0.42$\times10^6$ km^2, 0.45$\times10^6$ km^2, 0.79 $\times10^6$ km^2 for C3S-1, C3S-2 and OSISAF-SITY, respectively), the contribution of these pixels to the comparison is overall considerable. In addition, it could be large under situations that trigger the atypical microwave signatures, which will be further discussed in section 4.1.2."*

Lines 204-210: "This is expected ... summer and winter." --> These lines call for the more careful delineation of the physics behind the various retrieval methods which I asked for earlier. Without that physical background these statements all remain hypothetical and are not sufficiently backed up by existing knowledge and hence not in line with good scientific practice.

Reply: Thank you for the advice. In the section of "Introduction" and "Data", we have included descriptions about the physical background for microwave SITY classification and theory behind the classification algorithms.

Line 214: What do you mean by "most distinct variations"?

Reply: We wanted to say that the inter-annual evolution of MYI extent from C3S- and OSISAF-SITY differs most with NSIDC-SIA. The sentence was modified to (L403-L405)

*"For the inter-annual evolution of MYI extent, C3S-SITY and OSISAF-SITY differ most from other SITY products. The latter exhibit mild negative trend during 2000–2007 and rapid negative trend from 2007 to 2013, while the former show larger inter-annual variabilities."*

Subsection 3.2.1:

- One could have expected that you dedicate a bit more time to comment on the details such as the drop of the MYI extent to zero in some winters, when looking at the NSIDC SIA MYI extent of region ESS.

- What explains, to your opinion, the observation that especially for BS and ESS NSIDC SIA MYI extent is often considerably larger than the MYI extent offered by all other products? This is less pronounced for region CAO where we can also see numerous cases where the other MYI extent products exceed the NSIDC SIA MYI extent.

Reply: Thank you for the advice. In the revised manuscript, we added details as you mentioned and give possible explanations for the inter-comparison of SIA and SITY MYI extent in the first paragraph of this section (L414-L424).

*"On one hand, MYI is extensively exported through the Fram Strait (Kuang et al., 2022) and, by small fractions, into the Barents Sea and through the Nares Strait following the Transpolar Drift Stream. In the ESL region, the MYI extent even decreases to zero in some winters (e.g. 2007–2009, 2012–2013), which is in line with the record low Arctic minimum sea ice extent in the previous Septembers. On the other hand, MYI is advected towards south along the Canadian Arctic Archipelago driven by the Beaufort Gyre. In the BCS region, large quantities of MYI are pushed out of this region following the anticyclonic current, the Beaufort Gyre, meanwhile replaced by the MYI from the CAO region. This eventually leads to the nearly constant or increasing MYI extent in the BCS region. In the ESL and BCS regions, it is found that the NSIDC-SIA MYI extent is usually considerably larger than the MYI extent from the SITY products, whereas it is less pronounced in the CAO region. This indicates that the mixture of MYI and FYI, which leads to the "overestimated" NSIDC-SIA MYI extent, occurs more frequently in the ESL and BCS regions than the CAO region, which could be explained by the more dynamic ice characteristics in the two regions."*

Lines 220/221:

- "The former ... Stream" --> This is not entirely correct and requires re-phrasing. You have chosen your region CAO such that the Transpolar Drift Stream goes right through it ... from the Pacific side towards the Atlantic side. Therefore, what explains the decrease in MYI extent is i) the export through Fram Strait and, by smaller fractions, into the Barents Sea and through Nares Strait, and is ii) the export driven by the Beaufort Gyre towards the South along the Canadian Arctic Archipelago.

- Note, in the sentence before "BS keeps constant or increasing" should also be re-phrased. You want to state something like the MYI extent in the BS/CS region remains constant or is increasing.

- Finally, what explains the decrease in MYI extent in region ESS? This is not clear yet. If it is exported towards the CAO, then this is going to be a northward flow ... a direction not yet mentioned in your description.

In short: Please be more accurate in the description of the results of your work.

Reply: Thank you for the advice. In the revision, these sentences have been modified to provide more clear explanations (L414-L424):

*"On one hand, MYI is extensively exported through the Fram Strait (Kuang et al., 2022) and, by small fractions, into the Barents Sea and through the Nares Strait following the Transpolar Drift Stream. In the ESL region, the MYI extent even decreases to zero in some winters (e.g. 2007–2009, 2012–2013), which is in line with the record low Arctic minimum sea ice extent in the previous Septembers. On the other hand, MYI is advected towards south along the Canadian Arctic Archipelago driven by the Beaufort Gyre. In the BCS region, large quantities of MYI are pushed out of this region following the anticyclonic current, the Beaufort Gyre, meanwhile replaced by the MYI from the CAO region. This eventually leads to the nearly constant or increasing MYI extent in the BCS region. In the ESL and BCS regions, it is found that the NSIDC-SIA MYI extent is usually considerably larger than the MYI extent from the SITY products, whereas it is less pronounced in the CAO region. This indicates that the mixture of MYI and FYI, which leads to the "overestimated" NSIDC-SIA MYI extent, occurs more frequently in the ESL and BCS regions than the CAO region, which could be explained by the more dynamic ice characteristics in the two regions."*

Line 223: What are "varying evolution trends"? Either do trends vary between the different winters of years. Then this could be termed inter-annual variation of the trends describing the evolution of the MYI extent in the respective region. Or you want to comment that within a season, the evolution of the MYI extent from month to month differs between different winters or years. Then you need to specify that you are referring to the intra-seasonal variation of the MYI extent and need to drop the word "trend". Please be more clear in your writing.

Reply: Thank you for the advice. We would like to refer to the evolution from month to month. The sentence has been modified to (L425-L426):

*"most SITY products show similar intra-seasonal variation in the CAO region, while exhibiting disparate intra-seasonal evolutions in the BCS and ESL regions (especially in early and late winter)"*

Lines 238/239: "the discontinuous ... C3S-SIT" --> This is a too global statement because it reads as if daily MYI extent fluctuations are always explained by this discontinuous FYI delineation. You should not forget that this is a scene at the verge of freeze-up and therefore one cannot expect that all of the MYI has a "mature" microwave

signature yet which would the algorithms let define it as such. In addition, I'd say this is an issue that is possibly limited to the late October / early November cases and is not of general validity. Please correct your writing accordingly.

Reply: Thank you for the advice. In fact, this issue can be found in different winter months, e.g. Figs. 7, 8 and A1. To illustrate it more clearly, we revised the sentence accordingly (L441-L443):

*"In Fig. 7 a and b (along with Fig. A1 a–d, f–i in Appendix and Fig. 8 a–b, h–i), the discontinuous FYI delineation in the inner part of MYI pack is well demonstrated, which occurs in all winter months and could partly explain the MYI extent fluctuations in C3S-SITY".*

Line 245: "with exceptional MYI distributed ... as ESS." --> I can agree on the 2nd largest MYI extent but there is only a quite small part where the finger-like structure of MYI extends through Chukchi Sea into the ESS region. The other finger-like structure at Severnaya Zemlya can be observed in basically all products and is hence not exceptional. Perhaps you want to state that this protrusion from the Chukchi Sea into ESS is not in agreement with the NSIDC SIA sea ice type?

Reply: Thank you for the advice. The expression was not correct here. We were about to use "exceptional" to describe the MYI extent from KNMI-A not KNMI-Q. The sentence was therefore modified to (L448-L450):

*"KNMI-Q has the second largest MYI coverage among the seven SITY products, with a slightly more finger-like structure of MYI extending through the Chukchi Sea into the ESL region."*

Line 256: Looking back at this paragraph and the top two rows of Figure 7 you could also state that this is a good example where assigning the ambiguous ice type pixels to MYI actually improves the agreement in the spatial pattern with NSIDC SIA sea ice type.

Reply: Thank you for the advice. It is true that MYI is underestimated in all the SITY products compared with NSIDC-SIA. Including Amb pixels as MYI could lead to larger MYI extent, however not necessarily better agreement on the spatial pattern. We therefore did not include additional descriptions here.

Lines 268-271:
- This information should be placed in a section about methodologies of the inter-comparison, where you describe how you co-located data, and how you computed the MYI extent from the different data sets (and grids).

Reply: Thank you for the advice. In the revised manuscript, we added a new section of "Methods". Information such as co-locating data and calculation of MYI extent was

placed in the "Methods" Section, more specifically in section "3.1 Estimation of MYI extent".

- There you also should reflect upon why and how you selected the boxes as you did and why these have a different size.

Reply: Thank you for the advice. In the revision, we made the boxes with same size in each SAR image. The physical background and principles of visual interpretation was presented in the new section "3.2 Visual interpretation of SAR imagery". Enlarged examples were given Fig. 3 as well as Figs. 9-13 to show more details of the SAR image thus more intuitive for the comparison. The boxes were selected from the locations where there are typical backscattering characteristics and could show the agreement/discrepancies among the various SITY products.

- It would be furthermore more than beneficial if you would elaborate on the way how you decided, based on the SAR images, which part of the ice is FYI and which MYI. Your sentence "Characteristics of brightness, texture, gemometric shape and context ..." is not sufficient for a journal such as "The Cryosphere"; it rather reads like written for a public science magazine, I am sorry. You have decibel values at hand and by digging into published literature you can get a much better, even quantitative handle on the interpretation of the SAR images.

Reply: Thanks. In the revision, we added a new section "3.2 Visual interpretation of SAR imagery" as suggested, where more details about the characteristics and interpretation principles were presented. In addition, we interpreted all the entire SAR images as suggested, compared with the gridded ice classification results, and eventually give the quantitative evaluation results (Kappa coefficient and overall accuracy for respective SAR image and SITY, SIA product).

- I note that you used HV-pol data from Sentinel-1 SAR. Why did you use cross-polarized images instead of co-polarized images? What is the advantage using those? Can I assume that the RADARSAT-1 images were HH-pol? You could note this additional information in the respective figures.

Reply: Thanks. In the revision, we added the polarization information in the respective figures. In addition, in the new section "3.2 Visual interpretation of SAR imagery", we included sentences as below (L332-L334):

*"Images at HV polarization are prioritized for the visual interpretation if provided, since the cross-polarized backscattering signals have been shown to increase the separability between MYI and FYI (Gray et al., 1982; Onstott et al., 1979)"*

Lines 273-278:

- The description of what is seen in terms of ice types in the SAR image appears to be hypothetical and descriptive. There are tables and publications from which you can learn about the typical signatures (sigma nought) of MYI and FYI at C-Band HH-polarization. You should find and use these to put your assumptions on solid ground. Otherwise also these SAR images cannot serve as an evaluation or even validation data set but rather represent a vague inter-comparison source. And with that you can by no means adequately draw conclusions about the quality of the sea ice type products you are investigating here. You then also need to change the title of the manuscript, leaving out "evaluation". Also the usage of the NSIDC SIA MYI extent does not warrant so because it is known to be biased (this is visible in your manuscript as well) and is not a good source for evaluation in the way carried out by you.

Reply: Thank you for the advice. In the revision, we added a section of "Method" in the revised manuscript, where the theory of interpreting SAR images is introduced. In addition, we made visual interpretation quantitatively based on the theory and discussed the quantitative comparison in the section of "4.3 Validation based on SAR".

As for the NSIDC-SIA product, it is true that the NSIDC-SIA MYI extent could be biased. However, NSIDC-SIA is used as a reference dataset for large-scale comparison. Discrepancies between the SITY products and NSIDC-SIA does not mean "bad". This requires further "validation" such as that based on SAR images. To better illustrate this logic, we modified the sentences at the beginning of the "Results" section (L338).

*"This section starts with a temporal and spatial comparison of the SITY products, with NSIDC-SIA as a reference dataset. It then proceeds with validation against SAR images."*

- Please add month and year of the scene to the text.

Reply: Done. Month and year of the scene is added to the first sentence of this section (L483), which reads as below:

*"In Case 1, a typical scene of early winter (November 2017) in MIZ is shown in Fig. 9."*

Lines 279++: I am wondering whether it would make sense to not comment on / discuss every product here in the figures showing the comparison to the SAR images. Perhaps the most striking discrepancies would be enough to mention.

Reply: Thank you for the comment. We totally understand the concern. However, if we do not comment on each product here, it would be difficult to summarize later. We therefore keep the comment as it is in the revised manuscript. In addition, we included a new section after presenting all the five cases, Section "4.3.4 Performances of sea ice type and age products".

Lines 284/285: "which might be caused ... the product" --> It is not clear how these two issues can lead to an overestimation of the MYI extent derived from the sea ice age product when the delineation relevant to state whether a grid cell contains MYI or FYI is between FYI and SYI, and is therefore only influenced by the time from the last fall until the date this example is from ... hence basically 6 weeks in this case. Only a small amount of FYI is grown until then in that region and one can be confident that the majority of the grid cells is in fact predominantly MYI as seen in the sea ice age product.

Reply: Thank you for the advice. Based on the interpretation results from SAR and the Kappa coefficient/OA, NSIDC-SIA generally overestimate MYI in this case although the general distribution is similar. The sentence was therefore modified to (L489-L490) *"NSIDC-SIA overestimates MYI generally thus yields a median Kappa coefficient and OA (0.56 and 0.73, respectively)."*

Lines 289-293:
- Same comment as for Fig. 8 with respect to how to assign features and/or brightness distributions to ice types. This is a purely qualitative inter-comparison and not an evaluation.

Reply: Thank you for the advice. We have added a section of "3.2 Visual interpretation of SAR imagery" in the revised manuscript. In addition, quantitative interpretation results in terms of statistics such Kappa coefficient and overall accuracy were added, which can be found in the text and Table 4.

- When I was working with SAR data during my PhD days I was always urged to denote the sensor flight and look direction by arrows. You could do so as well so that it is more clear where the low and where the high incidence angles are located.

Reply: Thank you for the advice. Marks of the azimuth and range direction were added to the SAR images, shown in Figs. 9-13.

Lines 297/298: "The MYI underestimation ... weekly temporal resolution." --> Why? What is the physical process required to have large discrepancies between a weekly ice type map and a daily ice type map? Is there evidence in the additional data used by you about this physical process?

Reply: Thank you for the advice. MYI is slightly underestimated in NSIDC-SIA in this case. This is nearly negligible considering the temporal resolution difference. We therefore modify the sentence to (L500-L502)
*"MYI is slightly underestimated in NSIDC-SIA, with the Kappa coefficient of 0.57 and OA of 0.80. Yet such difference is nearly negligible considering their different temporal resolutions and the mobility features of sea ice."*

Lines 301-305: Again the interpretation of the SAR image (and the boxes zoomed into) appears to be very hypothetical and is not well backed up by what could be taken from published literature (if the authors would have considered to use HH images instead of HV images). This would also have resulted in less processing artefacts in the image.

Reply: Thank you for the advice. The FYI and MYI scattering characteristics in HV image are more distinct than that in HH image. We therefore presented the HV image instead of HH image in the manuscript. In fact, both HH and HV image were used for the visual interpretation. To illustrate the procedure better, we added a section of "3.2 Visual interpretation of SAR imagery" in the revised manuscript. The physical background and how the zoomed boxes are selected were explained accordingly in the new section. For the Sentinel-1 image, we included the HH image as well as the HV image in the corresponding cases.

Lines 306-311: I would say this is a classical example where the NSIDC-SIA is one of the more useful data sets here. Looking carefully it is clear that regions C and D are both FYI. You can check the minimum extent end of summer 2014 please to check whether in that area close to Severnaya Zemlya sea ice survived the summer melt. I doubt so. Hence these two areas are located within the landfast sea ice (FYI) cover that develops there ususally - as is also well backed up by a sea ice drift speed of zero. While I could agree that region B is in fact MYI I doubt that region A is MYI. This is certainly an area where i) deformation and ii) deep snow plays a significant role in shaping the different microwave signals contributing to the (every) sea ice type classification.

Reply: Thank you for the comment. We agree that this case actually show typical case of landfast ice (FYI) mixed with MYI. In the original manuscript, our interpretation about areas A and B is consistent with you (A as FYI and B as MYI). Regarding Area C, we agree that it should be landfast FYI instead of MYI. Interpretation results, the statistics and the texts were modified accordingly in the revision. The modified texts can be found in L542-L549. The modified figure (Fig.13) is shown as below.

*"The MYI distribution pattern of KNMI-A resembles the SAR image except for a slight overestimation of MYI in the northern part of the image (area A) and nearly the island, which may be caused by ice deformation. The Kappa coefficient and OA is the largest for KNMI-A in this case. IFREMER-A and Zhang-SITY both completely ignore the MYI pack, and this error lasts for the whole winter (maps not shown). C3S-SITY (C3S-1 and C3S-2) and OSISAF-SITY manage to identify FYI in area A, and sporadically capture an elongated MYI feature northeast of the image (partly classified as Amb). However, they underestimate MYI in area B and overestimate MYI in the southern part (areas C and D), which leads to a near-zero level Kappa coefficient. NSIDC-SIA clearly captures the elongated MYI feature in this case though has slight underestimation of MYI in area B."*

[Figure]

Line 312: Is there any reason why you put mid-winter after late-winter?

Reply: Thank you for the comment. In the revision, we presented the cases in the chronological order (early-, mid-, and late-winter).

Line 322: "Compared to the SAR image ... is overestimated ..." --> This is only part of the story. I would see this more differently and urge the authors to have another look to see that the agreement between NSIDC SIA and the supposedly FYI - MYI distribution in the SAR image is only acceptable in the bottom part of the SAR image whereas towards the top and top left there is both an underrepresentation of MYI and an overrepresentation of MYI, respectively.

Reply: Thank you for the advice. We agree with you on the comparison. To present the difference more clearly, we included the visual interpreted results in the same frame/subfigure as other SITY and SIA products. This sentence was modified to (L513-L516):

*"… the MYI pack in this area, and this regional scale misclassification of MYI holds through the whole winter (maps not shown). Compared to the SAR image, the SITY*

*distribution in NSIDC-SIA has a distinct pattern, with overestimation of MYI in the northwest part of the image (area A) meanwhile underestimation in the northern part (east of area A) …"*

Line 323: Please look at my comment to a similar statement made by you further above.
Reply: Thank you for the advice. We modified the sentence to (L515-L517)
*"As mentioned previously, such discrepancies are mainly attributed to the mobility features of sea ice and the different temporal resolutions between NSDIC and the SAR image."*

Lines 336-339:
- I don't see how your results underline or agree with the results of Korosov et al. and I also don't see how your results confirm that the NSIDC SIA data set is a cross-validation data set. It is at most a data set for consistency checks and inter-comparison. I will detail why below.
At first: none of the sea ice type products investigated has a finer temporal resolution than the NSIDC SIA product (and hence the MYI extent derived from it). Hence you cannot look at the sub-grid scale distribution of sea ice types (and age) in the NSIDC SIA maps and the information you claim to have at hand originates from the publication mentioned above and is not your own result.
Secondly, even though you have SAR images at hand you did not make the effort to first perform a high-level evaluation of the NSIDC SIA producte BEFORE you use it as a data set for inter-comparison. You could have carried out a dedicated pixel-wise comparison between the NSIDC SIA product and the SAR images used. But this would require i) more SAR images covering the same region over the weekly period represented by the NSIDC SIA product (i.e. ideally one at the beginning and one at the end of the 7-day period) for ... say ... 50 cases (which is a big project) and ii) using SAR images in a quantitative way, i.e. using the sigma nought values to delineate FYI from MYI, and in addition taking carefully the drift and deformation history of the respective regions into account to ensure that areas with a bright signature caused by deformation are not misinterpreted as MYI. Only with such a comparison, looking at the sub-grid scale distribution of the different ice types within single NSIDC SIA 12.5 km grid cells, you can shed more light about the "cross-validation" potential of these maps.
Reply: Thank you for the advice. We agree that the NSIDC-SIA product is used for consistency check and inter-comparison. It is not proper to regard it as "cross-validation". In the revised manuscript, we modified such expressions meanwhile used SAR images in a quantitative way. We also added a section of "3.2 Visual interpretation of SAR imagery", where the theory for the visual interpretation is described.

Line 367: "stability of the sea ice types" --> what do you mean with that? FYI will not disintegrate spontaneously and MYI will not become FYI over night. Please rewrite.

Reply: Thank you the advice. The sentence has been modified to (L588)

*"The efficacy of input parameters depends on their capability to separate and physical properties of the sea ice types in question."*

Lines 368-370: "This parameter ... or high frequency channels" --> I agree to this statement; however, I am wondering what the magnitude of cloud liquid water values typically observed during winter in the Arctic would be and what the impact would be specifically on the GR. I am pretty sure you can dig out this information in the available literature and back up your statement adequately. There are sea ice concentration algorithms that specifically make use of the two channels that form this GR, e.g. the Comiso algorithm frequency mode; perhaps the paper by Andersen et al. from 2006 in Remote Sensing of Environment could enlighten you here. In short, unless the impact of atmospheric parameters such as cloud liquid water and water vapor on the GR at these two frequencies is really measurable I would remove this piece of information. If kept it needs to be backed up by adequate literature.

Reply: Thank you for the advice. We further investigated the magnitude of cloud liquid water values and its impact. It turns out that variation of the cloud liquid water path has little impact on GR. We therefore removed this sentence in the revised manuscript.

Line 368: I am surprised that one of the sea ice type algorithms uses this GR ratio the other way round, i.e. 19 V minus 37V. Please check. This (again) calls also for a better and more comprehensive description of the algorithm behind the products inter-compared in this study.

Reply: Thank you for the advice. We confirm that one algorithm (i.e. OSISAF-SITY) uses this ratio the other way around. In the revised manuscript, we gave detailed description of each algorithm when introducing each SITY product (Sections 2.2.1-2.2.4).

Line 371: "ice layering" is one component of the snow properties and should not be mentioned as if it is a different thing.

Reply: Thank you for the advice. We deleted "ice layering" in the revised manuscript.

Lines 373-375: "when air temperatures fluctuates around freezing point and triggers snow metamorphism" --> Apart from the fact that this is another example of bad English grammar this statements needs to be formulated in a less global way.
A) What you call snow metamorphism with a likely impact on brightness temperatures particularly at 37 GHz are melt-refreeze cycles caused by elevated solar radiation

during spring (April); during these cycles the air temperatures do not necessarily fluctuate around the freezing point.

B) In October solar radiation is absent, hence cannot be the trigger for snow metamorphism. Melt-refreeze cycles are also absent. What can happen in October is advection of warmer air masses and precipitation falling as wet snow or freezing rain - which admittedly can have an impact on the microwave signature of the sea ice cover. But without working with the theory (missing in your manuscript) you cannot explain it properly. Possibly wet snow masks MYI underneath, letting it look like FYI. But you don't present evidence for this in your data / results. While warm air and hence wet snow might be the reason for the underestimation of the MYI cover in the CAO using C3S-SIT it is not sufficiently clear why snow metamorphism should lead to an overestimaton in the BS and ESS in late winter. What is the physical process that drives which change in the relevant microwave properties that cause the microwave observations to trick the algorithms, leading to an overestimation in MYI?

C) Another issue you did not yet bring up is the fact that parts of the ESS but also the parts of the CAO facing the Atlantic may experience particularly thick snow loads. Since the GR used here is not only sensitive to the sea ice type but it is also sensitive to the snow depth it is not surprising that a sea ice type algorithm that uses the GR at 37 and 19 GHz tends to classify FYI as MYI as a result of a thick snow cover.

Reply: Thank you for the advice. The sentence has been modified in the revised manuscript (see the end of this reply).

Regarding A), we agree that these are likely melt-refreeze cycles caused by increased solar radiation. It is also true that air temperatures during the whole cycle do not necessarily fluctuates freezing point. What we are trying to say is that, air temperatures that trigger these cycles are around freezing point. Evidence for this can be found in Voss et al., 2003 and Ye et al., 2016b. We therefore rephrased the sentence and included the references accordingly.

Regarding B), although solar radiation is not yet absent in the Arctic in October, we agree that the microwave signature changes are different from those in April. They could be caused by warm air intrusions or precipitation falling as wet snow, which lead to changes on snow wetness. We did not present evidence for the impact on the microwave signatures, since it can be found in previous studies such as Ye et al., 2016a. In the revised manuscript, we therefore rephrased the sentence and included the reference accordingly.

Regarding to C), we agree that thick snow load could lead to misidentification of MYI. This information was added in the revised manuscript.

The modified sentence is shown as below (L591-L593):

*"In the beginning and ending stages of winter, the variability of GR_37v19v can be significant when air temperature exhibits warm-cold cycles, which trigger wet-dry cycles or melt-refreeze cycles of snow (Voss et al., 2003; Ye et al., 2016b; Ye et al.,*

*2016a), or when wet/thick precipitation suddenly appears (Voss et al., 2003; Rostosky et al., 2018)."*

*References:*

*Ye, Y., Heygster, G., and Shokr, M. (2016a). Improving multiyear ice concentration estimates with air temperatures. IEEE Transactions on Geoscience and Remote Sensing, 54(5), 2602 - 2614.*

*Ye, Y., Shokr, M., Heygster, G., and Spreen, G. (2016b). Improving multiyear sea ice concentration estimates with sea ice drift. Remote Sensing, 8(5), 397.*

*Voss, S., Heygster, G., and Ezraty, R. (2003). Improving sea ice type discrimination by the simultaneous use of SSM/I and scatterometer data. Polar Research, 22(1), 35-42.*

*Rostosky, P., Spreen, G., Farrell, S. L., Frost, T., Heygster, G., and Melsheimer, C. (2018). Snow depth retrieval on Arctic sea ice from passive microwave radiometers—Improvements and extensions to multiyear ice using lower frequencies. Journal of Geophysical Research: Oceans, 123(10), 7120-7138.*

Lines 375/376: Please explain to the reader what the effect of the temperature correction scheme and the "upgraded tuning of atmospheric correction for Tb" [better --> the improved correction of the Tb for the atmospheric influence] is on the GR used so that the reader gets a credible piece of information here which you again ideally back up with appropriate literature.

Reply: Thank you for the advice. We would like to say that mitigation of the misclassification is caused by the different processing in C3S-2, which includes the temperature-based correction for improved ice type discrimination and the improved correction of Tb for the atmospheric influence. Details of the difference between C3S-1 and C3S-2 has been added in section 2.2.1. In the revised manuscript, the sentence was modified to (L595-L597):

*"Such misclassification in C3S-1 is mitigated in C3S-2 due to the upgraded processing in C3S-2, which includes the temperature-based correction in the post-processing and the use of reanalysis data from ERA-5 instead of ERA-Interim in the atmospheric correction for Tb (see section 2.2)."*

Line 377:

- "backscatter (sigma^o)" --> either "backscatter coefficient" or "sigma nought"

Reply: Done. "backscatter" was modified to "backscatter coefficient" in the revision.

- "which has good separability between MYI and FYI." A backscatter coefficient cannot have a good separability between MYI and FYI. A backscatter coefficient might be suitable to separate MYI from FYI.

Reply: Thanks. The sentence was modified to (L598)

*"which is commonly used in ice type discrimination due to the disparate scattering features of MYI and FYI."*

Lines 378-381: "In comparison ... Fig. 12)" --> Also these lines should be re-written and re-phrased investing more space to describe the issues behind.

- In addition, you might want to provide an explanation why Ku-Band scatterometer measurements appear to be less sensitive to the surface roughness than C-Band scatterometer measurements. How about the sensitivity to the crystal structure of the MYI compared to the FYI? Is the contrast in the backscatter coefficient between MYI and FYI larger or smaller at Ku-Band compared to C-Band? Does Tb depend on the polarization? Does this depend in the incidence angle? What is the role of the different penetration depths into the snow and into the sea ice?

Reply: Thanks. The sentences have been modified as below. Explanations were added. (L600-L606)

*"In comparison, the backscatter of MYI and FYI is more disparate at Ku-band than C-band (Rivas et al., 2018; Bi et al., 2020). Products using Ku-band backscatter generally perform better on identifying MYI, e.g. KNMI-Q, IFREMER-Q, and Zhang-SITY before 2009. This could be due to the fact that Ku-band scatterometer is more sensitive to the crystal structure of MYI since its wavelength (about 1.7 cm ~ 2.5 cm) is more consistent with the characteristic dimension of air bubbles in MYI (Ezraty and Cavanie, 1999). On the other hand, the dominant effect of surface scattering and the higher dependence on incidence angle makes C-band backscatter more suitable to distinguish the ice types with disparate surface roughness features, e.g. Cases 3 and 4 in Fig. 11, Fig. 12."*

*References:*

*Bi, H., Liang, Y., Wang, Y., Liang, X., Zhang, Z., Du, T., ... and Huang, H. (2020). Arctic multiyear sea ice variability observed from satellites: a review. Journal of Oceanology and Limnology, 38(4), 962-984.*
*Rivas, M., Otosaka, I., Stoffelen, A., & Verhoef, A. (2018). A scatterometer record of sea ice extents and backscatter: 1992–2016. The Cryosphere, 12(9), 2941-2953.*
*Ezraty, R. and Cavanié, A. (1999). Intercomparison of backscatter maps over Arctic sea ice from NSCAT and the ERS scatterometer, Journal of Geophysical Research: Oceans, 104, 11471–11483.*

Lines 388/389: "In Beaufort and ... classification ..." --> please also see my comment for Line 373 further above.

Reply: Thank you for the advice. The sentences were rephrased to (L612-L614)

*"In the Beaufort and East Siberian Seas in late winter, employing Tb and backscatter measurements even leads to the worst SITY classification in OSISAF-SITY and Zhang-SITY (Case 5, Fig. 13). This indicates that simple data combination does not necessarily imply better classification results"*

Line 394:
- Either: "employ a dynamic threshold" or "employ dynamic thresholds"

Reply: Done. It was modified to "employ dynamic thresholds"

- What do you mean by "variability of [the] training dataset"? Do you mean the spread of values around a chosen threshold brightness temperature or backscatter coefficient?

- What do you mean by "seasonality"? I recognize that sea ice type retrieval is limited to the freezing season, hence one season; you should be more specific here. It is also not clear to what the seasonality refers to ... to the MYI extent? to the physical properties of the sea ice and its snow cover? to the thresholds used?

- I don't understand what you mean by "shift in sensor type". Could you please elaborate on this in the text? I can guess that you perhaps mean the shift between using SSM/I or SSMIS data or between using ASCAT C-Band and QuikSCAT / OSCAT Ku-Band. But to me this is not a shift in sensor TYPE because it is either radiometers or scatterometers. Please be more specific here.

Reply: Thanks. The expression may not be clear in the original manuscript. We would like to say that the dynamic thresholds vary temporally or spatially, and depend on the specific satellite sensors. In the revised manuscript, we modified the sentence to (L619) *"while others employ dynamic thresholds to account for the variability of training dataset…"*

Lines 398/399: It is not clear what you mean by "takes sea ice variabilities into account". What "sea ice variabilities"? Are you referring to the spatiotemporal development of the physical properties of the sea ice and snow cover that influence it microwave backscattering characteristics and/or the microwave emission? Then please write it specifically. Currently, "sea ice variabilites" can mean anything from variations in sea ice thickness or concentration, different ice drift patterns, floe-size distributions, degree of deformation whatsoever ...

Reply: Thank you for the advice. In the revised manuscript, it was modified to (L623-L624) *"On the other hand, the dynamic threshold approach considers the spatio-temporal variability of the microwave radiometric and scattering characteristics."*

Lines 409/410: "can be partly ... more obscure"
- "obscure" --> "difficult" or "problematic"
Reply: "obscure" was replaced with "difficult"

- The statement as written is not conclusive because you are not providing the key message that the Arctic Ocean has lost a lot of its oldest ice AND that the difference in the radiometric and microwave backscattering properties is usually more pronounced between FYI and these older ice types than, e.g. second and third year sea ice. This feeds back again to the missing description of the physical background behind the sea ice type retrieval earlier in your manuscript.

Reply: Thank you for the advice. We added descriptions of the physical background in the Introduction and Data Section as explained in previous replies. In the revised manuscript, the sentences were modified to (L638-L642)

*"The large loss of old ice (e.g. older than four years) in the Arctic Ocean leads to a younger MYI regime in the Arctic, thus smaller microwave signature differences between MYI and FYI (Tschudi et al., 2020). On the other hand, because of the lower sensitivity of C-band scatterometer on MYI identification (as explained in section 5.1), the separation between FYI and MYI becomes more difficult, especially from ASCAT data (Belmonte Rivas et al., 2018; Zhang et al., 2019)."*

Lines 418/419: In all of the regions mentioned here MYI ice can occur once in a while and hence a MYI ice signature in these regions certainly is not unphysical. Apart from that is the Chukchi Sea part of your region BS. This needs to be corrected in the text.
Reply: It is true that MYI can be found once in a while in these regions. However, as we understand, the mask mentioned here should have excluded the areas (part of the regions) where MYI could be present. Otherwise this mask would lead to obvious "wrong" results. Regarding the comment "Apart from that is the Chukchi Sea part of your region BS", we are not sure what the reviewer means. In the revised manuscript, the sentence was modified to (L649-L650)
*"The first kind of correction scheme, a mask of the Arctic basin, has been used in C3S, OSISAF- and KNMI-SITY to remove the unphysical MYI signature in areas such as the Greenland, Kara, Barents and Chukchi Seas."*

Lines 420-422: "Statistical thresholds ... the ice edge" --> Please provide a plot which illustrates how PDFs of the respective parameters used in the retrieval (i.e. backscatter coeffient or Tbs or GRs) of the MYI overlaps with the PDFs of ice types typically encountered along the ice edge so that the reader understands what you are referring to. Ideally, you have this figure along with the revised description of the sea ice type algorithms earlier in the manuscript so that here you simply need to refer to that figure.
Reply: Thank you for the advice. Having an additional plot to illustrate this issue would be too much for the example. In the revision, we modified the sentences to make it easy to understand (L651-L654).
*"The thresholding filter in C3S-SITY and OSISAF-SITY exclude extreme values that are likely to cause misclassification, e.g., values beyond the simulated FYI PDF however within the wide simulated MYI PDF, which usually occurred in ice edge areas (Aaboe et al., 2021b; Aaboe et al., 2021c)"*

Lines 422/423: "exclude ... distributions." --> I don't get what you want to state here. If the MYI extent in the above-mentioned peripheral seas and/or along the ice edge would be added to the MYI extent in your region of interest this would mean a considerable change in the overall SIT distribution. Therefore, please re-phrase your statement as it is currently not clear enough.
Reply: Thank you for the advice. The sentence was rephrased to (L654-L655)

*"These two kinds of corrections exclude misclassification cases in regions outside the central Arctic thus have little impact on the overall SITY distributions."*

Lines 424/425: "reassign ... intrusions." --> Not sure what you want to state here. Do you mean "assign grid cells erroneously classified as FYI as the result of warm-air intrusion induced changes in the surface snow properties to the ice type MYI." ?
- How is this temperature based correction done?
- Aren't there other algorithms (published by one of the authers) that use this temperature based correction as well?
Reply: Thank you for the advice. In the revision, the sentence was modified to better explain the misclassification and the relation with the correction. In addition, references were added accordingly (L656-L657):
*"The temperature-based correction in C3S-2 aims to reassign the erroneously classified FYI, which exhibits similar microwave signatures as FYI due to warm air intrusions (Ye et al., 2016a; Shokr and Agnew, 2013)."*

Line 425:
- What does an "ice motion confining procedure" do? I have no clue. Please explain it to the reader.
- "anomalous MYI overestimation" --> What is this? What is a "normal MYI overestimation" and what is the difference to an "anomalous" overestimation? Please re-phrase.
Reply: Thank you. We have added sentences here to explain the ice motion-based correction. The sentences were modified to (L658-L661)
*"In Zhang-SITY, an ice motion confining procedure is introduced to eliminate overestimated MYI. The procedure builds upon the ice motion temporal records and confines the evolution of MYI according to the tolerance of ice motion. One drawback of this post-processing is that, the wrong reassignment of MYI to FYI could lead to continuous underestimation of MYI in consecutive days."*

Lines 428/429: It is not sufficiently well described how the correction based on a median filter (spatial or temporal) works.
Reply: Thank you. The sentence was modified to (L661-L663)
*"Another correction used in Zhang-SITY is the median filter correction, which considers spatial consistency and is employed to remove large unusual SITY spatial variations."*

Line 432: "the five series SIT products ... are defined." --> I don't understand this sentence. Please re-write. It is possibly a problem of the grammar.
Reply: Thank you. The sentence was modified to (L666-L667)

*"Apart from the above three aspects (input parameters, classification methods and correction schemes), factors such as the covering period and spatial resolution make the five series SITY products different from each other."*

Line 434: "Typically ..." --> I don't see that your manuscript warrants yet to state the reason given for the larger spread in MYI extent during early and late winter as being typical.

Reply: Thank you. In the revised manuscript, we deleted "Typically", added references and phrased the sentence. The sentence was modified to (L669-L672)

*"In early and late winter larger uncertainties are likely to occur due to surface processes such as wet snow attenuation and changes in brine salinity (Barber and Thomas, 1998; Voss et al., 2003; Ye et al., 2016a; Ye et al., 2016b)."*

Lines 440/441:
- It is sufficient to write "grid resolution", spatial can be omitted.
- "foot print" --> "footprint"

Reply: Done.

- What is the "true" spatial resolution of the ASCAT data? What is the "true" spatial resolution of the QuikSCAT data? You should please not forget that the finer grid resolution provided by the SIRF products (4.45 km) is the result of heavy smoothing and other signal reconstruction steps.

Reply: Agree. In the revised manuscript, "true" is replaced with "nominal".

- Another issue that you did not take into account here are the different incidence angles of - especially - the ASCAT C-Band data compared to the microwave radiometer data and QuikSCAT / OSCAT.

Reply: Thank you. Yes, we should have mentioned how they (SITY products) process the ASCAT data. The ASCAT data used in the SITY products is already the normalized backscatter coefficient with same incidence angle of 40 degree. We therefore did not mention the influence of incidence angle in the discussion. In the revised manuscript, we added a section of "Microwave remote sensing data" (section 2.1) and "sea ice type products" (section 2.2), where how the characteristics and pre-processing of ASCAT data is introduced.

Line 452:
- Add "five" in front of "SAR images".

Reply: Done.

- Any reason why you are not mentioning the NSIDC SIA product here?

Reply: In the revised manuscript, we mention it when talking about the inter-comparison. The sentence is modified to (L682-L683)
*"In this paper, eight SITY products based on five retrieval approaches were inter-compared through temporal and spatial analysis, with the NSIDC-SIA product as a comparative reference."*

Line 453 / the conclusions in general:
- I am not sure I would select a sea ice type product based on the maximum difference that a product might have compared to another independent data source. I would be interested in whether there are regions and time periods where there are systematic errors (and how large these are on average so that I might be able to correct them). In addition, I would be interested in the average performance of the product over a longer time period, i.e. whether there are artificial trends.
Reply: Thank you for the advice. We agree that it is more appropriate to mention the average performance here in the conclusion. In the revision, these sentences were modified as below (L685-L694), with one paragraph summarizing their performances on the trends within winters, another paragraph about their average performance:
*"The eight SITY products show overall negative trends of MYI extent as expected within most winters. Exceptions occur mainly in early and late winter months such as October/November and March/April. Compared to NSIDC-SIA, all the SITY products show smaller MYI extent and larger FYI extent except KNMI-SITY (KNMI-Q and KNMI-A). The bias of MYI extent between the SITY products and NSIDC-SIA varies from -1.32×10^6 km^2 (OSISAF-SITY, 2006–2009) to 0.49×10^6 km^2 (KNMI-A, 2009–2019). Among all the SITY products, Zhang-SITY in the QSCAT period and KNMI-Q agree best with NSIDC-SIA on the estimation of MYI and FYI extent, respectively.*
*Between any two SITY products, the difference in weekly MYI extent spans from 0.01×10^6 km^2 to 4.5 ×10^6 km^2. The largest discrepancy occurs between OSISAF-SITY and KNMI-A in late October 2008, while the smallest difference is found between KNMI-Q and IFREMER-Q in mid-winter months. It is in line with the spread of the SITY products, which is largest in early winter months such as November and smallest in mid-winter months like January."*

- I suggest to re-write your conclusions accordingly, focussing less on the individual products as you do in the list 1) to 4) (which should in any case contain 5 or even 9 entries according to what you write in Line 451), and instead concentrating on the larger picture provided by your qualitiative results. It might help in this context to again take a look at your time series plots and focus less on the inter-comparisons with the SAR images.

Reply: Thank you for the advice. We agree that it is important to summarize the performances of SITY products, therefore included two more paragraphs in this section. In addition, we believe that performances of the individual products could be useful information for the users.

- I like the bullet point list further down on the next page. That one looks good but could be written even better by including specific details and referring to the existing literature.
Reply: Thank you for the advice. We understand the concerns. However, since these are the main findings of this paper, it may be better to include such details in other sections, where more recent references were included.

Line 477: "extensive misclassification with higher uncertainties" --> So, the misclassification in itself is highly uncertain? Please re-write.
Reply: Thank you for the advice. "with higher uncertainties" was deleted in the revised manuscript.

Line 480: "Ku-Band ..." This statement is not new and has to be backed-up by existing literature.
Reply: Thank you for the comments. References "(Rivias et al., 2018; Ezraty and Cavanie, 1999)" were added in the revised manuscript.

Lines 488-490: "On the other hand ... become obscure." needs to be re-written. The meaning is not clear and the grammar is not correct.
Reply: Thanks. The sentence was modified to (L733-L735)
*"On the other hand, adaptive classification method that depends on the clustering pattern of the radiometric and backscattering signatures may be inefficient when the characteristic signatures of MYI and FYI have large overlaps;"*

Line 492:
- Apart from the fact that we still don't know how "ice motion confining" works, it is not clear what "accumulative errors" are. Consider re-phrasing for improved understanding.
Reply: As seen in the replies to previous comments, we gave more detailed description regarding the "ice motion correction". In the revised manuscript, the sentence was modified to (L736-L739)
*"Excessive post-processing such as ice motion confining could lead to an over-correction problem, which becomes the basis for the subsequent corrections and eventually result in accumulative errors."*

- "These post- ..." --> "Any post- ..."

Reply: Thanks. It was implemented in the revised manuscript.

Line 491: What is meant by "should be accounted with caution"? Please consider re-phrasing for improved understanding.

Reply: Thanks. "should be accounted with caution" was replaced with "should be considered with caution".

Lines 495/496:

- "This study ... of SIT retrieval approach" --> I don't agree. This study does not contain an "evaluation"; it is an inter-comparison study, mostly involving qualitative results. It provides hints of the quality of the sea ice type products investigated RELATIVE to the NSIDC SIA data set (which in itself is not well evaluated) and relative to only five SAR images which are not interpreted quantitatively.

- I further object to the notion "most popular". Please consider re-phrasing.

- I cannot see the "hints for further improvement". While you state where some of the sea ice type products have deficiencies, you neither come up with specific suggestions about how to improve (e.g. use a SIRF-like product as an input to the OSISAF sea-ice type product to improve the grid resolution) nor does the nature of your results being based on an inter-comparison to qualitative data support to draw conclusions into this direction. I warmly suggest to tone down the value and potential impact of your results.

Reply: Thank you for the advice. We agree that it is not that appropriate to call it "systematic evaluation". In the revised manuscript, the sentence was modified to

"This study inter-compares eight SITY products and provides …" On the other hand, performances of the SITY products were summarized in this paper, with discussions on the potential reasons and improvements (e.g., L711). We therefore think the expression "hints for improvement" is appropriate.

The sentence in L711 reads as below:

*"IFREMER exhibits high frequency temporal variations in MYI extent, which could be caused by the day-to-day varying thresholds and improved by including appropriate post-processing;"*

Line 497: Please share with us which two frequencies WindRAD is going to use.

Reply: Thanks. The two frequencies are Ku- and C-band. In the revised manuscript, we added this information in the parenthesis "(WindRAD, Ku- and C-band)".

- "the potential of scatterometer on ice type discrimination" --> "the potential of scatterometer measurements for ice type discrimination"

Reply: Thanks. It was implemented as suggested in the revised manuscript.

Lines 499/500: "low frequency microwave measurements" --> "low frequency microwave radiometer measurements" because ASCAT already has been using C-Band for 15+ years which is also a "low frequency microwave measurement"
Reply: Thanks. It was implemented as suggested in the revised manuscript.

Figure 3: I suggest to reduce the number of colors used by getting rid of the NSIDC sea ice age and instead show the MYI extent derived from it as a black line - like you do in Fig. 4. If you want to show examples of how the different sea ice type products deal with different sea ice age then I suggest to show just the respective year - ideally a year where almost all sea ice type products provide MYI extent so that you can compare between the products. Alternatively, you could consider showing only MYI extent differences. If you want to keep the sea ice age information then I recommend to use shades of grey instead of colors for the sea ice age.
Reply: Thank you for the comment. Since the plots of the MYI extent from SITY products rarely go to the blocks in the lower part (older than 2/3 yr), and it is

Figure 4: Why are IFREMER-A data missing for January and April in 2014 & 2015?
Reply: Thank you for noticing the data gap. IFREMER-A data during this period was not available from the data provider, therefore was not included.

Figure 6 and 7:
- It is very counter-intuitive to show open water in brown, land in light grey and FYI in blue. Please use a more intuitive coloring such that, e.g. land is brown, open water is blue and FYI, Amb, and MYI are perhaps medium grey, light grey and white; the observation gap at the pole can then be colored black.
- I recommend to enlarge the figure as a whole.
Reply: Thanks. The figures were modified as suggested.

- In the caption you could cross-ref to Table 1 or Figure 1 to make clear why there is a different number of maps for the two dates shown. In addition you need to refer one more time to the meaning of the red line and you need to comment on the different coloring of the observation hole.
Reply: Thanks. Figures and captions were modified.

Figure 8:
- What is the motivation to show boxes A to D with a different size?
Reply: Thanks. The boxes were made the same size in the revision.

- I have the same comment with respect to colors as for Figures 6 & 7.
Reply: Thanks. All the figures were modified accordingly.

- I suggest to show the NSIDC-SIA map in a different color code as well. What is important for you is to discriminate FYI from older ice which currently is difficult to delineate because the colors used for FYI and SYI are quite similar.
Reply: Thanks. We used similar colors in the revision.

- I recommend to rename "sea surface wind" to "10m wind" because I guess this is what it is. Also make clear that "air temperature" possibly is the "2m air temperature". The additional information that these are daily averages would be appreciated as well.
Reply: Thanks. Notes were modified accordingly.

- I would replace the legends for those sea ice type products that do not provide the ambiguous ice class with a legend which only shows the two ice classes present. It might make sense - in general - to then also include the class open water in the legend.
Reply: Thank you for the comment. In the comparison, the Amb class pixels were regarded as MYI and FYI, respectively. We believe that it is necessary to show the spatial distribution on the maps.

Figures 9 to 12:
- I have the same comments with respect to colors, legends, and ERA5 data naming as I had for Figure 8.
Reply: Thanks. Done as suggested.

- In addition, delineation of the boxes in the Sentinel-1 image in a different color than black would help to locate these better.
Reply: Thanks. Green boxes were used in the revision.

- It might make sense to not use a continuous color table for the legend of the ice drift field, as the values are increments of 0.1 km/day.
Reply: In the revision, speed of the ice drift is marked with the length of arrows, whereas colors of the background indicate the 2-m air temperature.

Table 2: Cases where there is a "+" and a "-" indicate that both performances exist?
Reply: Yes. Both performances could exist in different locations for one SITY product in one case.

Table 3: I guess the GR listed in the context of OSISAT SIT is not correct?
Reply: We have checked that it is correct.

Typos / editoral comments:

Line 59; "indirect validation" --> perhaps better "inter-comparisons"?
Reply: "indirect validation" was modified to "inter-comparisons"

Line 89: "KNMI-" --> "KNMI-SIT"
Reply: Done.

Line 144: "study, sea ... were used ..." --> study, we used a sea ice age (SIA) product and five SAR images ..."
Reply: Done.

Line 154: "with SAR ... of HH" --> possibly better: "providing C-Band (5.3 GHz) SAR images at HH polarization."
Reply: Done.

Lines 155/156: "providing cross- ... ranging from" --> possibly better: "providing C-Band (5.4 GHz) SAR images at co- and cross-polarization (HV and HH) with incidence angles between"
Reply: Done.

Line 170: "polar hole of 87degN" --> "data acquisition gap north of 87degN centered at the pole"
Reply: Done.

Line 176: "within studied area" --> I picked this as one of the examples that underline the need for considerable English editing of the manuscript. The authors must check for usage of "the" and "a" which is often missing.
Reply: Done.

Line 181: You stated already in Line 170 that you excluded that area centred at the pole. Therefore you can delete this sentence.
Reply: Done.

Line 185: "The MYI" --> However, the MYI"
Reply: Done.

Line 185: "regional" --> "regionally"
Reply: Done.

Line 225: "is mainly resulted from" --> check grammar.
Reply: "is mainly resulted from" was modified to "is mainly attributed to".

Lines 239-241: This part does not belong to the top row of Figure 6, right? It belongs to the data from 2007 and should be placed into the next paragraph.

Reply: This part belongs to the top row of Figure 6. We added notes of Figures in the sentence for clarification. The sentence is modified to:

*"On the other hand, IFREMER-Q (e.g. Fig. 7c) shows constantly less MYI than KNMI-Q (e.g. Fig. 7d) in the transition zone of MYI and FYI in BCS, in good agreement with their difference as shown in Fig. 6."*

Line 280: Typo: "boarder" --> "border"

Reply: Done.

Line 288: Add the year.

Reply: Done.

Line 289: "... in the western part were higher than in the eastern part."

Reply: Done.

Line 294: "Slightly underestimation of MYI" --> check grammar.

Reply: "Slightly underestimation of MYI" was modified to "Slight underestimation of MYI".

Line 296: I would say that "thin" could be misinterpreted as "thin MYI" in terms of its thickness. You might want to consider using "narrow" or "filament-like" or "finger-like" or similar.

Reply: "thin MYI tongue" was modified to "narrow MYI tongue"

Line 297: "can partly be resulted" --> check grammar.

Reply: "can partly be resulted from" was modified to "can be partly attributed to ".

Line 300: "A Sentinel-1 SAR image covering the southern part of the ESS near the coast acquired on April 27, 2015 is shown in Fig. 10."

Reply: Done.

Line 313:
- "transit zones" --> "transition zones" or "zones of mixed FYI - MYI coverage"

Reply: "transit zones" was modified to "transition zones".

- "steady discrepancies" --> re-phrase please.

Reply: "steady discrepancies" was modified to "constant discrepancies".

Line 315: Delete "validation and"
Reply: Done.

Lines 324-325: "The MYI feature ... round MYI floe" --> please check grammar.
Reply: the sentence was modified as follows (L518-L520):
*"The bright MYI floe feature is clear in the northeast part of the SAR image, so as the dark FYI feature in the southwest part. Areas A and D exhibit high backscatter of round MYI floe, and areas B and C present typical characteristics of FYI with smooth texture and low backscatter."*

Line 364: "serial" ???
Reply: "serial" is deleted.

Line 378: "when using backscatter" --> when using backscatter coefficient measurements of an active microwave instrument."
Reply: Done.

Line 382: "confirmed" --> "shown"
Reply: Done.

Lines 397/398: "vary ... ASCAT" --> "are different, especially at C-Band."
Reply: Done.

Line 402: "speculate" --> "hypothesize" ?
Reply: "we speculate that this is because …" was modified to "The possible explanations could be that …"

Line 405: Either "to a sea ice type distribution" or "to sea ice type distributions"
Reply: "SIT distribution" was modified to "SIT distributions"

Line 406: Either: "An adavtive clustering algorithm is used" or "Adaptive clustering is used"
Reply: "Adaptive clustering algorithm is used" was modified to "An adaptive clustering algorithm is used"

Line 408: "thin ... seas" --> "narrow MYI tongues in the peripheral seas"
Line 427: "continuous underestimation" of what?
Reply: it was modified to "continuous underestimation of MYI"

Line 430:
- "over-correction problem" --> "over-correction.
Reply: Done.

- "thin MYI ... seas" --> We had that expression earlier. Please look up my comment there.
Reply: Agree. "thin MYI …" was modified to "narrow MYI …"

Line 436: "fully evaluated" --> "done"
Reply: Done.

Line 481: What is "small FYI in MYI pack"? Do you mean: "comparably small areas of FYI within a region dominated by MYI?"
Reply: Yes. "small FYI in MYI pack" was modified to "comparably small areas of FYI within a region dominated by MYI"

Line 487: "deep " --> "mid-"
Reply: Done.

References: You need to check your reference list. For a considerable number of the entries the records are not complete; for instance is the year missing quite often. At least one of the references appears twice.
Reply: Thank you for the advice. We double-checked all the references in the manuscript and modified accordingly.

---

## Author Response (AR2)

Dear Editors and Reviewers,

We would like to thank you all for the dedicated and insightful comments and advices. We have put a lot of effort on revising the manuscript regarding the structure, expression and readability. Comments from the two reviewers are responded on point-by-point basis.

The reviewers' comments appear in black. The responses are in blue and the proposed changes to manuscript are in **bold italics**.

========================================================

**Report 1**

The manuscript has been improved. Thanks for the author's effort to make a more strengthened paper. I recommend that this manuscript be published after some minor revisions of table and figures.

Table 1: it would be nice if it is specified whether the Satellite Input is based on the microwave radiation or scattering.
Reply: Thank you for the advice. The table is modified as suggested. In addition, a column of the grid is added.

Fig. 2: Please specify the imaging coverage of RS and S1.
Reply: Thanks. Notes of "RS-1" and "S-1" are added to Figure 2.

Fig. 3: figures are missing the geographic information such as latitude/longitude grid and scale bar. And please specify which satellite SAR images the sub-figures are from in the caption.
Reply: Thanks for the advice. The geographic information, scale bar and notes of "RS-1" "S-1" are added to Figure 3.

========================================================

**Report 2**

Review of
Inter-comparison and evaluation of Arctic sea ice type products
By Ye, Yufang, et al.

I am not providing a summary of this manuscript because I reviewed a former version of it.
The authors have improved the manuscript considerably and have taken into account quite a number of the concerns that were brought up during the previous round of reviews.

The readability of manuscript and the credibility of the results presented do, however, still suffer from deficits in some parts of the description and from overrating of the mostly qualitative elements of the intercomparison carried out.

**General comments:**

GC1: The deficiencies of proper reference of the figures from inside the text and the numerous typos and strange formulations make this manuscript difficult to review and to read. Please next time when submitting a manuscript consider having it proof-read by a native English speaker plus check it for consistency. It cannot be the task of an editor or a reviewer to tick all these. It is an immense work load and distracts from the scientific content of the manuscript.

Reply: Thank you for the advice. We highly appreciate the thorough review from the reviewers and editor. We are more careful this time on double-checking the reference of figures and have more thorough review on the text and language. All the co-authors have proof-read it to avoid strange formulations and unclear descriptions in the manuscript.

GC2: While the consideration of the sea ice physics and its relation to microwave remote sensing of sea ice has been improved considerably, there are still elements that should be improved - please see my specific comments.

Reply: Thanks for the advice. The manuscript has improved substantially thanks to the reviewers' comments. Revisions and replies are made based on the respective comments.

GC3: Please step away from considering this as an evaluation / validation or assessment study. It is an inter-comparison study, involving qualitative data set for inter-comparison and products which quality you aim to understant and report upon in this manuscript. Rather than attempting to rank the products, I recommend to state clearly that we require more and better specified and evaluated data sets for a quantitative evaluation of the sea ice type products.

Reply: Thanks for the advice. We understand the concern of the reviewer. In the revised manuscript, we try to tone down the value of the work as a validation study. Meanwhile, as the reviewer said, there is no well-evaluated dataset at present. The data we use here is already one of the best we could achieve. We therefore think it is acceptable to use wording such as "evaluation" in the manuscript.

**Specific comments:**

L43: Please check the content of Boisvert et al., 2016, with respect to whether this is indeed the paper you wanted to cite in this context. Perhaps the two papers of her et al. from 2015 fit better: Increasing evaporation amounts seen in the Arctic between 2003 and 2013 from AIRS data" or "The Arctic is becoming warrner and wetter as revealed by the Atmospheric Infrared Sounder"?

Reply: Thanks for the advice. The reference "Boisvert et al., 2016" is replaced with the

following reference in the revised manuscript.

*Boisvert, L. N., Wu, D. L., & Shie, C. L. (2015). Increasing evaporation amounts seen in the Arctic between 2003 and 2013 from AIRS data. Journal of Geophysical Research: Atmospheres, 120(14), 6865-6881.*

L53-59: I suggest to add that these possibilities to discriminate MYI from FYI by means of the different microwave signatures work well (only?) in winter when the snow cover is dry. In summer / during melt events, MYI and FYI often reveal a similar microwave signature.

Reply: Thanks. We add the following sentence to the end of these sentences (see L60-L62 in the revised manuscript):

*"Note that MYI and FYI have such different microwave characteristics in winter but not in summer or during melt events when snow is wet, which leads to similar microwave signatures of the different ice types."*

L66: I suggest to remove Brath et al as this is about helicopter-borne scatterometer measurements and not about satellite data classification.

I Suggest to remove Hughes as this is grey literature (and relatively old as well).

Reply: Thanks. It is modified as suggested.

L75: Isn't the ECICE algorithm cited here using Shokr et al., 2008 belonging to the other type of algorithms (SITC)?

Reply: Yes. It was a wrong citation here, we have deleted it as suggested.

L83-85: "some areas ... of MYI in ice charts" --> I would be careful with this statement because after all, what is done in the binary assignment or classification of a grid cell as FYI or MYI bears the same potential for over- or under-estimation of the actual fraction of the respective ice type. Hence, in light of this retrieval uncertainty it is perhaps ok to use data for the inter-comparison that have a similar drawback?

Reply: Agree. We therefore delete the statement of "overestimation".

L85/86: The sentence mentioning SAR could i) include the detail that SAR - like scatterometers - is an active microwave instrument but with a spatial resolution several orders of magnitude finer and ii) back up the information that SAR images are used for this kind of evaluation with literature.

Reply: The sentence is modified as suggested (see below). References are added to back up such information.

*"Synthetic aperture radar (SAR) is an active microwave sensor as scatterometers but with several orders of magnitude finer spatial resolution. SAR images are also used to evaluate ice type classification accuracy (Ye et al., 2019; Zhang et al., 2019)."*

L91: "for winters" --> just a comment: If you include the information that the discrimination between MYI and FYI works well by means of their different microwave signatures during winter suggested further up, then this notion of "for

winter" is a logical consequence of what is physically possible.

Reply: Thanks for the comment. We add statements regarding the different microwave signatures in winter and summer further up, according to the reviewer's comments.

Table 1: Please check the dates for SSMIS (2000 as the starting year is wrong) and for AMSR-E (12 as the end month is wrong).

Reply: Thanks. We double-check the dates and modify them accordingly (see new Table 1 in the revised manuscript).

L106: SMMR did not use a conical scan. Please check the respective documentation. One paper where you find helpful information about the series of SMMR, SSM/I and SSMIS sensors is this one: https://essd.copernicus.org/articles/12/647/2020/

Reply: Thank you for the advice. We double-check the specifications and modify the sentence below:

*"It provides five-frequency, dual-polarized (ten-channel) Tb observations with an average incidence angle of 50.3°."*

L113: In view of the frequencies listed in Table A1, don't you think it makes sense to state in one sentence that for the generation of the SITY products introduced in section 2.2 merely the near 19 and near 37 GHz channels of these instruments are used? This would explain at the same time why you only list these two frequencies in Table A1.

Reply: Agree. We add one sentence to the end of the first paragraph in section 2.1.1.

*"Specifications of the different sensors are shown in Table A1, where only the channels used in the SITY products in Section 2.2 are listed."*

L119: Note that there have been two different ERS satellites ERS 1 and ERS 2, both equipped with that AMI instrument, but only ERS1 started operations in 1991. You might want to make this clear in your text.

Reply: Yes, there are two ERS satellites. We meant both of them. In the revised manuscript, "European Remote-sensing Satellite (ERS)" is modified to "European Remote Sensing (ERS) satellites (ERS-1 and ERS-2)".

L122: OSCAT: There have been more than one OSCAT sensor. Which one is this?

Reply: Thanks for the advice. It is the OceanSat-2 scatterometer, we therefore replaced "OceanSat Scatterometer (OSCAT)" with "OceanSat-2 Scatterometer (OSCAT)" in the manuscript.

L128: While I agree that there is a linear relation between TB and emissivity I doubt that from the physical / mathematical sense this statement is correct. I'd rather see it that way: TB is linearly proportional to the physical temperature of the object and the emissivitiy is the proportionality factor which magnitude is determined by the physical / chemical properties of the material that are relevant for its microwave emissive behavior.

Reply: Thanks. The sentence is modified as below:

*"…, which is linearly proportional to the physical temperature of the object, where the proportionality factor, the emissivity, is determined by the dielectric properties."*

L130-138: This is a good starting point, containing already parts of the relevant information. I suggest to

i) separate FYI better from MYI in your explanation and correct the statements of how salinity changes [brine can only be "expelled" towards the surface of the ice, a process that if at all occurs during very cold conditions and for rather thin ice; most of the change in the brine content is via gravity drainage during winter and via flushing out the brine by meltwater during summer - which is the process where brine pockets become air pockets (or air bubbles as you write). These processes should be explained more correctly.]

ii) try to make clear what the role of snow might be.

iii) try to make clear which of the geophysical properties determine changes in the emissivity on the one hand and in the backscattering properties on the other hand - also taking into account frequency dependence and different polarizations.

Reply: Thank you for the advice. Modifications are made as suggested.

i) The word "expelled" is replaced with "rejected" and "bubbles" is replaced with "pockets". In addition, we rephrase the last sentence to better explain the parameter of GR3719.

ii) we add two sentences to explain the potential effect of snow (see L147-L149).

*"The snow over sea ice also influences the emissivity. The addition of dry snow on the ice leads to reduced emissivity because of the increased scattering in the snow volume, while the moisture in a wet snow cover results in increased emissivity (Shokr and Sinha, 2015)".*

iii) the backscattering properties are explained in the next paragraph, and we think the current updated information is sufficient to understand the principles of sea ice type classification, therefore did not add more sentences as mentioned above but rather add some references for more details.

The last sentence where you cite Vant et al. is not clear. What is meant by "demonstrated" in this context and what are "high frequencies" in this context?

A very good review of such properties is given in the book by Carsey, F.D., "Microwave Remote Sensing of Sea Ice" from 1992 which is available online.

Reply: The suggestions are highly appreciated. We have rephrased the sentence including Vant reference and moved it up in the paragraph. As suggested, we have included a reference to the relevant chapter 4 (Eppler et al.) in the Caesey book.

L140: Certainly GR is used in SITY products (I am not sure about PR but fine) ... but why? What is the advantage of using a GR over a TB? Ideally this is going to result from the revised section about emissivities and TBs before line 139.

Reply: Thanks for the advice. We add descriptions regarding the advantage of GR, and modify the original sentence as below:

*"The emissivity is an intrinsic radiometric property of the material, but brightness*

*temperature is not (Shokr and Sinha, 2015). For this reason, polarization ratio (PR)*
*and gradient ratio (GR) are usually used instead of Tb because they are independent*
*of the physical temperature."*

L143/144: You need to state that the subscript "p" in Eq. 2.2 stands for polarization and
can either be H or V. You also need to state clearly which of the two frequencies f1 and
f2 given references to the higher frequency - unless this is not (why not?) important.
Reply: Thanks for the advice. We have included the definition of the variables {h, v, f1,
f2, and p} at the beginning and specified that p can be either h or v.
*"PR is the normalized difference between the horizontally (h) and vertically (v)*
*polarized Tbs for the same frequency (f), whereas GR is the normalized difference*
*between Tbs at two frequencies (f1, f2) at the same polarization (p) which can be*
*either h or v."*
As the order for f1 and f2 only cause a difference in the sign, this is not important for
the discrimination ability. Difference products define GR differently but the outcome is
the same. We have added a sentence on this in the end of the paragraph.

L147-152: "Meanwhile ... SITY product" --> Also this I rate as a good start but the
description needs to rephrased still to be clear enough. I recommend to
i) clearly state what the normal radar backscatter behavior is for FYI and MYI and why
and only then ii) point out in which way deformation would change this general view.
Please work with expressions such as surface scattering (dominant for FYI) and volume
scattering (dominant for MYI).
When it comes to differences between frequencies, polarizations, etc. it makes sense to
introduce (perhaps again) penetration depth of microwave radiation into sea ice of
different types. I again warmly recommend to take a look at the various chapters of the
Carsey book mentioned above.
Reply: Thank you for the advice. To explain the different backscatter of MYI and FYI
more clearly, we rephrase the paragraph and added descriptions as suggested. See L161-
L172 in the revised manuscript, e.g.
- *"… surface scattering is therefore the dominant scattering mechanism of FYI"*
- *"… air pockets within the subsurface layer of sea ice contribute to a higher*
  *volume scattering, which is dominant for MYI."*
Since the penetration depth can only explain the effect on frequency but not for
polarization and observation angle, we add such descriptions in the previous paragraph,
where descriptions of Tbs are given (see L143-144 in the revised manuscript).

L153: I might be wrong, but your description above is rather qualitative and I did not
really get which specific "signatures" characterize MYI and which FYI. Does FYI
exhibit a higher or lower emissivity than MYI? What about the PR? What about the
GR? What about the radar backscatter coefficient? I would say that there is enough
literature out to - for this first ever inter-comparison of SITY products based in
algorithms that use these specific signatures - provide a summary table of relevant
signatures at C- and Ku-Band for scatterometry and at near 19- and 37 GHz for

microwave radiometry.

Reply: Thank you for the advice. The word "signature" is not appropriate here thus cause misunderstanding. In the revision, we use "differences" instead. Otherwise, we hope that sufficient clarifications have been included in the revised manuscript.

L154: "top layer" --> Do you have an estimate of the vertical dimension you are talking about? Are these millimeters? Centimeters? Does this behaviour depend on the frequency?

Reply: Thanks for the advice. It is millimeter's scale, which depends on frequencies, but the difference is very small. To give clearer description, the sentence is modified to: *"…, when microwave radiation can only reach the top layer (from several to tens of millimeters) of melting snow".*

L176-178: Could it be that the Maaß and Kaleschke 2010 reference solely applies to the land-spill over correction and the Wentz 1997 one solely applies to the RTM based correction of the atmospheric noise? Please check and revise. Also "land spill-over due to the influence of land" should perhaps simply read "land spill-over effects on the measured TBs"

Reply: Yes, it was a typo here. We revise the citation here as suggested, and modify the "land spill-over due to the influence of land" to "land spill-over effects" in the revised manuscript.

L184: "Tb observations" <----> "using the classification parameter GR" --> It is not entirely clear whether and if so why the training dataset comprises TB values or GR values.

Comment: The above-asked-for table with typical values of the various input parameters taken from literature would assist very well here in understanding that this is obviously a rather simple approach involving the GR and a GR-threshold value including its typical variability for MYI and FYI.

Reply: "Tb observation" here meant "GR observations". To avoid misunderstanding, we replace it with "GR_37v19v observations" in the sentence.

Regarding the comment (and the above-asked-for table), we still think it is unnecessary to include such details in this manuscript. The aim of this study is to inter-compare the eight SITY products, and analyze their performance based on the general principles. Documenting all the detailed info is out of the main scope of our manuscript.

L201: "reassign misclassified FYI" --> Sorry, not entirely clear. Is this filter taking care of pixels that are erroneously classified as MYI but are in fact FYI? Or is this filter taking care of pixels that are erroneously classified as FYI but are in fact MYI?

Reply: We meant to take care of the pixels are erroneously classified as FYI but are in fact MYI. To avoid misunderstanding, "reassign misclassified FYI" is modified to "misclassified FYI back to MYI".

L204: Which polar stereographic projection?

Reply: It is the NSIDC Sea Ice Polar Stereographic North Projection. We modify it accordingly in the revised manuscript.

L214: The reader might like a hint why OSISAF-SITY uses a slightly different GR than most other products based on the GR at 37 and 19 GHz channels.
Reply: GR_19v37v = minus one * GR_37v19v. These two parameters are in fact the same for any of the classification algorithm. That OSISAF-SITY uses GR_19v37v is for historical reasons and has not been changed since the beginning of this operational production. We add a footnote comment on this difference in the manuscript:
*"The parameter GR19v37v is identical to -GR37v19v. But the different definition of GR does not affect the final classification outcome."*

L220: Given the fact that C3S-SITY used quite a number of filters and corrections it would be good to confirm here in the text that OSISAF-SITY only uses the geographical mask to correct for eventual misclassifications of pixels as MYI and does not perform any other filtering.
Reply: Thanks for the comment. Yes, we confirm that OSISAF-SITY uses the same filters and masks as for C3S-SITYs, except the air temperature correction scheme in C3S-2. A sentence has been added to clarify this:
*"In the post-processing stage, OSISAF-SITY uses the same OW filters and masks as those in C3S-SITY, except the final air-temperature correction scheme introduced for C3S-2 to correct for misclassified FYI (Aaboe et al., 2021b)."*

L227+: I note that here and further down you do not specify the form in which the scatterometer data are input into the pre-processing stage - unlike for the radiometer data where it is clear whether these are swath data or not and at which stage of the processing you compute a daily map. Please therefore consider to also here, for the scatterometer products, specify whether these products are based on swath data or daily gridded data or whatsoever.
Reply: These data are based on swath data. This information is added to the revised manuscript (see L249-L251).

L229: I suggest to back up this statement about the incidence angle dependency with a reference.
Reply: Thanks. Reference is added as suggested.

L233: "data of March of each year" --> Would you mind to also tell us about the geographic region from which these thresholds are determined?
Reply: Thanks for the comment. To specify the region, we add "in the Arctic" to the end of this sentence.

L234: I recommend to not speak about "MYI signatures" but of "grid cells" or "pixels erroneously classified as MYI". In addition I am wondering whether these pixels are really removed or whether they are set to ice type SYI or FYI. If not, then I assume the

resulting ice type maps may have gaps?

Reply: Thanks. For clearer description, "to remove the unphysical MYI signatures" is modified to "to set the erroneously classified MYI pixels back to FYI".

L247: Is it correct to assume that all radiometer data come as daily gridded maps? Please mention this accordingly in the text.

Reply: Regarding the question here, yes, it is ok to use daily gridded Tbs in sea ice type classification. We therefore do not understand what the reviewer is asking for. This sentence explains the priority of the different radiometer data in the product, whereas the next sentence introduces format (daily gridded) of the respective radiometer data.

L254: How is the re-gridding to the finest spatial resolution among the input data realized? Bilinear? Nearest-Neighbour? Others methods?

Reply: Nearest neighbour method is used in the re-gridding. We add "using the nearest neighbour method" to the end of the sentence.

L267: I don't find the description of the data set sufficient enough. It is in particular not clear how a discrimination into FYI (SYI?) and MYI is made? This information should be given as a minimum - together with the granularity (in time). About what temporal resolution are we talking here? Weekly? Monthly? Annually? It should become more clear that this data set offers the spatial distribution of different ice age classes (younger than 1 year, 1-2 years, 2-3 years ...).

Reply: Thanks for the advice. We add the following sentences to explain the discrimination of the ice age and the respective temporal resolution.
*"Ice age (i.e. 1 year, 2 year, … and 5+ years) is assigned according to the number of winters the ice parcels have survived. The age of the oldest ice within the grid cell of each week is regarded as the weekly ice age."*

L271/272: Not clear what you mean with "middle-of-the-road scheme". Also, the fact that satellite data are combined with buoy data (and data from atmospheric re-analysis by the way) does not apply to the SIA data set but applies to the sea-ice motion product that is used to derive the SIA data set. I suggest to rewrite this statement.

Reply: Thank you for the advice. We revise the sentence as below for better clarification.
*"Due to the scheme of using ice motion data derived from combined satellite and buoy data, NSIDC-SIA…"*

L278: I suggest to delete this sentence because this is based on observations in the Weddell Sea, Antarctica. If you want to provide information about the uncertainty of the kind of ice motion product used here I recommend to search for publications by Sumata et al., and take a look at the two publications by Lavergne et al., one in 2010 and one in 2021.

Reply: Thanks. We replace it with a reference from Sumata et al. 2014 instead.

L279-282: While the statements provided here are certainly correct they apply to an

older version of the ice motion product (v3) which in fact contained artifacts. The data you are using are based on version 4 and therefore I consider the discussion given in these lines as not relevant. These are also not connected to the issues Korosov et al. found; these point to methodological shortcomings of the derivation of the SIA product from whatever ice motion product.

What is lacking for the SIA product is an evaluation beyond the published inter-comparison study results. The reasons for this is clear - a lack of proper data sets that could be used as a source for evaluation - but it should nevertheless be made clear that we are still waiting for an adequate evaluation of the NSIDC-SIA data set. This is something you could (and should?) mention here because it explains well why you do not consider the SIA data set as an evaluation data set, i.e. kind of a ground truth.

Reply: Thanks for the advice. We acknowledge that the artifacts apply to the older version products. This is already specified in this paragraph,

*"… and has been largely mitigated by tuning the interpolation approach in the current version (Tschudi et al., 2020)".*

For better clarification of the NSIDC-SIA evaluation, we add the following sentences:
*"Although an adequate evaluation is still needed for the current NSIDC-SIA product, the good consistency and recent upgrades of the interpolation approach make it a useful dataset for SITY comparison.".*

Section 2.4: In which form did you get and use the SAR data?
Which (pre-)processing was already applied before / has been applied by you?
What is the spatial resolution in these SAR images?
From the description in L284 it seems clear that the RS-1 images are at HH-pol; it is however not clear what the polarization of the used S-1 SAR images is.

Unless you refer to a much more detailed description of how the SAR images were "visually interpreted" further down in your paper, I strongly recommend to also here not use the term "validation". So far it again simply seems to be an inter-comparison of the SITY products with another unvalidated product which accuracy is not clearly specified. Hence, in L285 and in L290 "validation" needs to be replaced with "intercomparison".

Reply: Thank you for the advice.
To answer the questions raised by the reviewer, we revised sentences in the first paragraph of section 2.4 to provide information regarding the form of data, spatial resolution and the polarizations (See L312-L314).
*"The three RS-1 images are in ScanSAR Wide (SCW) beam mode with nominal resolution of 100 m, whereas those from S-1 are in Extra Wide (EW) swath mode at HH and HV polarizations with nominal resolution of 40 m. The RS-1 SCW products and the Level 1 Ground Range Detected (GRD) S-1 product are both obtained from the Alaska Satellite Facility."*
Regarding the "validation" in L285 and L290, it is modified to "accuracy assessment in case studies" and "case studies", respectively.

L290: While one can guess it - would you mind to tell the reader the dates for which you used these additional data? Is it the same days as you have SAR images of?

Reply: Yes, they are from the same days as the SAR images. In the revised manuscript, the first sentence is modified to "Auxiliary data from atmospheric reanalysis is used in addition to the SAR images in the case studies.". Since it specifies "used in the case studies", we therefore do not give a list of the dates in the text.

L307-309: "To account ... " --> While this is an ok solution it introduces uncertainties from the re-projection. A more easy and straight-forward way would have been to use the grid cell sizes published by NSIDC for several derivatives of this polarstereographic grid

Reply: Thank you for the advice. Since the uncertainty is small from the re-projection and in theory does not affect the inter-comparison of this study, we decide to keep the processing as it is.

L323-330:
- This reads like a sufficiently good qualitative recipe to interpret SAR images visually. The only part that I find is missing is the discrimination between sea ice and open water - the latter also having a mixture of brightness levels in SAR images depending on i) the size of the openings with open water and ii) the wind speed. You may want to add this here to complete this description. Please take a look at my comments directly to Figure 3.
- Also, please add one sentence informing the reader that you are performing a binary classification here (in the next paragraph you write about Kappa coefficient and accuracy, so I assume to classify the SAR images in FYI, MYI and, if need be, open water).

Reply: Thank you for the advice.
- regarding the discrimination between ice and water, we add sentences as below for further clarification (See L355-L358).
*"(4) Backscatter of OW is dependent on the surface wind. It is low under calm conditions and could be high when the wind speed is high (Area D in Fig. 9). The more homogenous texture and lower auto-correlation of OW backscatter could be used to discriminate water from ice in SAR image (Berg and Eriksson, 2012; Aldenhoff et al., 2018)."*
- Regarding the binary classification, we add explanations to the end of this paragraph, the revised sentence reads as below:
*"… used as additional information for the ice type interpretation from SAR imagery (i.e. classification of OW, FYI and MYI)."*

L333-335: While the advantage of using HV over HH polarization is clear from the old literature cited, I strongly recommend to also take a look into the more recent literature and cite at least one paper where this has been applied to either RADARSAT SAR or Sentinel-1 SAR imagery.

Reply: Thanks for the advice. We added the following references in the revised

manuscript.

*Dabboor, M., & Geldsetzer, T. (2014). Towards sea ice classification using simulated RADARSAT Constellation Mission compact polarimetric SAR imagery. Remote Sensing of Environment, 140, 189-195.*

*Song, W., Li, M., Gao, W., Huang, D., Ma, Z., Liotta, A., & Perra, C. (2021). Automatic sea-ice classification of SAR images based on spatial and temporal features learning. IEEE Transactions on Geoscience and Remote Sensing, 59(12), 9887-9901.*

L337: Please add 2-3 sentences explaining what the Kappa coefficient is and how you compute an "accuracy" from two binary classified images. What is the credibility of these two parameters when computed for a pair of non-evaluated data sets?

Reply: Thanks. We add sentences and equations to explain the definition of overall accuracy and kappa coefficient (See L368-L376). Although the SAR interpretation results are not standard or perfect validation datasets, it is used for SAR ice classification results thus should be sufficient to evaluate coarse resolution ice type products.

L339-342: Given the nature of the two kinds of data products you use for the intercomparison of the SITY products I strongly recommend to change the wording away from validation, evaluation, and reference towards inter-comparison.

Reply: Thanks for the advice. We totally understand that it is not appropriate to call it "validation". However, the wording such as "evaluation, assessment, reference" is okay to use when we are using data from multi-sources to analyze its pros and cons.

L347: As noted in the context of Figure 4: I would get rid of the dashed lines and the daily data in this figure. It adds noise instead of value.

Reply: Thanks. The figure is modified as suggested.

L351: "due to ice divergence" --> I am wondering whether it would be worth to include a schematic illustration which shows what you mean by this.

Reply: Thanks for the advice. Considering the length of the manuscript and the relatively clear description, we think it is not necessary to include additional schematic illustration.

L356: "of FYI similar as that of MYI" --> It is exactly the other way round: the MYI signature becomes similar to the FYI signature. Melting snow, e.g., results in an increase of the microwave emissivity and hence an increase in the TB. Melting snow also results in a decrease of the penetration depth of microwave radiation so that a scatterometer does not sense the ice type underneath the snow anymore but only the snow.

Reply: Yes, the reviewer is correct. We modify the expression to "which leads to noise in the radiometric and scattering signatures therefore unsatisfactory performances of the SITY algorithm" in the revision.

L382-387: This paragraph targets the daily MYI extent values. I suggest to either delete it ... or keep it with the notion that these daily extents are not shown (see my comments with respect to Figure 4). Alternatively, you could show an example figure where you compare day-to-day MYI extent variations independent of the NSIDC SIA MYI extent.
Reply: Thank you for the advice. Figure 4 is modified as suggested. In addition, we add the notion that the daily extents are not shown in the manuscript.

L388/389: This sentence is reporting the maximum deviation - which is fine. You might want to include a statement about the average difference between the products during those periods of winter where all run comparably stable in their performance, i.e. November through March.
Reply: Thank you for the advice. We add one sentence to the beginning of this paragraph, which reads as below:
*"Between any two SITY products, the average difference in weekly MYI extent varies between 0.02×10^6 and 1.92×10^6 km^2 in winter, with values below 1.11×10^6 km^2 during the periods from December to March. The largest difference in weekly MYI extent reaches…"*

L396 / L402 / L413: Number of the figure is missing. --> GC1
Reply: Thanks. All the numbering of figures is added.

L414/415: "The former" refers to which region? As written it refers to CAO and ESL and then your statement is wrong because there is no southward flow of MYI from region ESL.
Reply: Thanks for pointing it out. We meant the CAO region here. In the revised manuscript, we replace "The former" with "The negative MYI trend in CAO".

L416: I suggest to delete the "following the Transpolar Drift Stream" because it is confusingly used in the context of what is stated here. For sure the TDS has nothing to do with MYI export through Nares Strait and also whether MYI is exported into the Barents Sea has in first instance to do with whether sea ice survived summer melt just north of the Barents Sea and is then pushed into it by northerly winds.
Reply: Agree. We delete it in the revised manuscript.

L422/423: "it is less pronouced in the CAO region." --> What is less pronounced? The MYI extent? Or the difference between NSIDC SIA MYI extent and the MYI extent of the SITY products investigated? Please be more specific in your writing.
Reply: Thanks. This sentence is modified as below:
"In comparison, such difference is overall smaller in the CAO region."

L423-425: I am not sure your argumentation holds the way written. I recommend that you are more specific about the process that actually leads to a potential (...) overestimation of MYI extent in the NSIDC SIA product. And I also recommend that you try to relate this potential overestimation also to the actual MYI fraction instead of

generally writing about a "mixture of MYI and FYI". You might ask yourself the question whether this potential overestimation is the same at 80%, 50% or 20% MYI concentration.

Reply: Thanks for the advice. To give more specific descriptions, we modify the sentence as below:

*"This indicates that the mixture of MYI and FYI (and the medium MYI fraction), which leads to the "overestimated" NSIDC-SIA MYI extent because of the oldest ice age assignment, …"*

Subsection 4.2.2: Because of my inability to link the text of this subsection adequately to the figures because numbers are missing for the latter, I do not comment on this subsection. I am sorry. --> GC1

Reply: We apologize for the missing numberings. We have included all the numbering in the revised manuscript and double check all the citation carefully.

L468+: Just again the comment that the way the SAR images are interpreted does - in my eyes - not warrant to use the term validation or evaluation. It is an intercomparison between two kinds of ice type maps. I therefore once again recommend to get rid of the terms evaluation and validation and switch to intercomparison.

Reply: As mentioned in previous replies, although the SAR interpretation results are not standard or perfect validation datasets, it is sufficient to evaluate coarse resolution ice type products, which is also used for evaluating SAR ice classification results.

In order to tone down the value, we avoid using the word "validation" in the manuscript. On the other hand, expressions of "evaluation" and "inter-comparison" are kept.

L478: It is a particular day in November. Therefore please mention it to avoid the impression that we are looking at a monthly average. The year given in the text does not match with the year given in the figure.

Reply: Thanks. We have corrected the year and add the date as suggested for all the cases in the manuscript.

L478-483: The way you interpreted this SAR image makes a lot of sense - still bears the chance that

i) the area you classified as FYI in the eastern parts of the image is actually indeed MYI or that ii) a substantially wider fringe of the area north of the open water is actually compressed FYI in form of brash ice, possibly undistinguishable from MYI under the conditions shown.

What I would like to state here is that the intercomparison between your SAR image analysis results and the SITY products is very difficult to put into a reliable or credible quantitative measure as you try to do with the Kappa and OA values. I recommend that you at least mention that because of the lack of contrast in backscatter in the SAR image shown the actual border between FYI and MYI might vary considerably. While currently KNMI-Q looks ideal in comparison to your SAR ice type map, another interpretation of the SAR map might have resulted in a FYI / MYI distribution that

resembles the C3S or OSISAF products closer. There is a lot of ambiguity - not just in the products but also in your SAR image interpretation which I recommend not to hide.

Reply: We understand the concern regarding the reliability of the SAR interpretation results. However, we would like to mention that the interpretation is performed on the high-resolution SAR image. We should not expect to see all the details in the figures presented in the manuscript. The contrast between the FYI and MYI may not be that high in this case, but it is sufficiently good for the discrimination of FYI and MYI. Therefore, instead of including statements such as "the actual border may vary considerably", we add notions which read as below

*"Note that quality of the SAR visual interpretation could vary with images. The identified border between FYI and MYI could deviate more from the actual border when the contrast in the backscatter is lower for the different ice types (e.g. Case 1).*

L508: Kappa and OA values for NSIDC-SIA are similar to Case 1 where you did not comment about the mobility of the ice even though that case is at the ice edge AND you have substantially higher winds. Here, in case 2 the MYI tongue is embedded into a matrix of growing FYI and I doubt that within the time frame of one week there was too much movement.

Reply: Thanks for the advice. We agree that the mobility of ice could partly explain NSIDC-SIA in case 1. We therefore add one sentence to the paragraph regarding case 1 (See L552-L553 in the revised manuscript).

*"Yet such difference is nearly negligible considering their different temporal resolutions and the mobility features of sea ice."*

As for case 2, although the wind speed is low for the day when the SAR image is acquired, it could be high in other days of the week. We therefore decide to keep the statement here for case 2.

L518-519: The failure of these products to detect MYI is really strange and difficult to understand. It involves both, a product only based on QuikSCAT data and products based purely on radiometer data. It might be a very stupid question from my side but did you double-check whether the re-projections that were involved in one or the other product to do the intercomparison did not jeopardize your results? You know, if it would only be the radiometer based products or only the QuikSCAT based products I would understand this failure ... but we are talking about old ice (according to the NSIDC SIA map) which has a clear signature in passive microwave imagery during winter.

Reply: We agree that the failure of these products is quite strange here. We double-check the re-projection (and other processing) during the analysis and confirms that it was correctly performed and should not be the "reason" for the "failure". Since the failure occurs not only for purely radiometer or scatterometer based products, the unusual radiometric and scattering signatures might result from changes of snow properties, which lead to radiometric and scattering signatures between those of MYI and FYI thus are sensitive to the thresholds in different SITY products and exhibit failure in different products. Despite being out of the scope of this manuscript, it is worth mentioning that the soon-to-be-released version 3 of C3S-SITY seems to capture

this MYI pattern, at least to some extent (being either MYI or Amb)

L521-523: Not sure whether inside the pack ice in winter the statement about moblity as a means to explain discrepancies holds.
Reply: Thanks for the comment. To tone down the statement, we replace "… are mainly attributed to…" with "… could be attributed to …"

L531-533: This failure to classify a large part of the ice as ice in the middle of winter is a no-go for such a product. Strange.
Reply: We agree with the reviewer. Reasons behind the strange performance in the IFREMER SITY product are unknown to us as well, which needs further investigation.

L536/537: "westward shift" --> This might be in fact one of the cases where the NSIDC SIA product "classifies" a grid cell as MYI ... even though the MYI fraction in some of the grid cells is certainly barely above the 15% threshold. Just a comment, no action required.
Reply: Thanks for the comment.

L537/538: Why do you highlight the ice age here but not for case 3?
Reply: Thanks for the comment. For case 3, the main performance of NSIDC-SIA is over- and underestimation in different areas. The SITY distribution in NSIDC-SIA is not that similar as the SAR image. We therefore did not highlight the ice age there.

L544/545: I am not overly convinced that the sea ice with this bright signature is associated with land-fast sea ice. This kind of ice is usually level ice with little deformation. In SAR images it often is represented as rather dark and homogeneous patches along the coasts / around Islands; actually the HH-SAR image of Figure 13 shows land-fast sea ice in the immediate vicinity of Severnaya Zemlya islands.
Reply: Thanks. We agree that it could be deformed FYI in area C. The sentence is therefore modified to "it is more likely to be land-fast ice or deformed FYI …"

L550/551: Not sure this statement about that this "lasts for the whole winter" is appropriate given that the SAR image is from April 27 and therefore at the verge to spring.
Reply: Thanks for the advice. We meant to say that the error occurs in early winter months and lasts for the whole winter. The sentence is modified as below in the revised manuscript.
*"This error starts to occur in November and lasts for the whole winter."*

L599: When does precipitation appear "suddenly"? What is "thick precipitation"? Please revise your wording; it is not clear what you want to state here.
Reply: Thanks for the question. We meant to say that wet or high snow precipitation could lead to significant changes of GR37v19v. The sentence is modified as below:
*"…, or when wet or high snow precipitation appears"*

L605/605: "As a results ... is used." --> How about the other way round? Isn't is as likely that MYI is misclassified as deformed FYI?

Reply: The discrimination between MYI and FYI is the relatively high and low backscatter. However, deformed FYI could have as high backscatter as MYI thus could lead to misclassification of the deformed FYI as MYI. Take case 4 as an example, we would say, it is more likely that deformed FYI is misclassified as MYI. We therefore keep the expression here.

L610: Have you looked at MYI ice cores? Did you find air bubbles of 2 cm diameter? I think this is an order of magnitude too large. I suggest that you dive into papers / book chapters describing ground- or air-borne active microwave measurements over different ice types and attempts to understand the reasons for the observed differences. The book Microwave Remote Sensing of Sea Ice by F. D. Carsey might be a good source - as well as papers dating back to the 1980s.

Reply: Thank you for the question. We read the books and references as suggested. It shows as the reviewer comment, that it is an order of magnitude too large. In the revised manuscript, we modified the sentence as below:

***"This could be due to the fact that Ku-band scatterometer is more sensitive to the volume scattering in MYI."***

L612: While in the context of brightness temperatures you refered to melt-refreeze cycles and the role of snow wetness this is missing completely here in this discussion. What happens to the radar backscatter signal if the snow gets wet?

Reply: Thank you for the question.

The dielectric constant increases as the wetness increases. The increase of the real part of dielectric constant (permittivity) causes increasing surface scattering, whereas the increase of the imaginary part (loss) causes decreasing volume scattering. The total backscattering decreases when the wetness increases from 0 to around 2% and stabilizes when the wetness continues increasing (Koskinen et al., 2000; Shokr and Sinha, 2015).

[Figure]

In the revised manuscript, we add the following sentences to describe the effect of snow wetness on backscatter.

*"Factors such as snow wetness could also influence the backscatter of sea ice thus the efficacy. An example is given in Shokr and Agnew (2013), where the increase of snow wetness causes attenuated (decreased) backscatter of MYI and eventually leads to misclassification of MYI as FYI."*

L617:
- Why this add-on "that is ice-free during summer"? What is the problem here?
- Also: "does not help" appears to be too global a statement. For the case you chose, the combination does not reveal advantages but there are enough other cases where it works very well. I recommend to tone this statement down by writing, e.g., "does not always help" or similar.
Reply: Thanks for the advice.
- It is unnecessary description here. We therefore delete it in the revision.
- It is modified as suggested.

L619: "the worst SITY classification" --> I can buy this statement for Zhang-SITY but while agreeably OSISAF-SITY is not correct IFREMER-A is clearly worse. Consider rephrasing please.
Note that April 27 is already a time of the year where solar radiation induced snow metamorphism can play a large role in shaping both passive and active microwave signatures of sea ice - even though 2m-air temperatures are still below zero.
Reply: Thanks for the advice. We agree with reviewer and delete "OSISAF-SITY" in the revision.

L645/646: "thus smaller microwave signature differences between MYI and FYI" --> I don't think that the Tschudi et al paper is an appropriate reference for this change in the physical properties of the multiyear ice when becoming younger. You might look into earlier work published by Comiso et al. or others in this regard.
Reply: Thanks for the advice. The citation here is not appropriate. We put the Tschudi et al., paper to the previous sentence, and replace it with the reference of Rivas et al., 2018.

L675/676: "surface processes such as wet snow attenuation and changes in brine salinity" --> You mix to kinds of processes here, it seems. One is the inter-action between surface properties and microwave radiation (the attenuation part) and one is a physical process (salinity change). I recommend to first give examples of the changes in physical properties and then write about the impact these changes could have on the microwave signature.
Are you sure that the salinity of the brine changes? Isn't it rather the bulk sea ice salinity or the brine volume?
Reply: Thanks for the advice. We meant the processes of snow metamorphosis and changes in bulk salinity of sea ice. The sentence is modified as follows:
*"… due to processes such as snow metamorphosis and changes in bulk salinity of sea ice."*

We understand that the reviewer would like to see more examples in the manuscript. However, considering the length and given references, we think it would be okay to make the statement in this way.

L682: "finer resolution does not ..." --> You might want to bear in mind that the 4.45 km grid resolution of the Zhang SITY product is based on using satellite data that were resolution-enhanced using the SIRF technology. Such a procedure does not improve the information content ... hence it does not matter how fine the grid resolution is, if the relevant information is blurry - not resolved properly at the original coarse spatial resolution - then it will remain blurry.
Reply: Thanks for the advice. It is not that appropriate to say finer resolution. We use "finer grid spacing" instead.

L683-684: Why mentioning the near-90 GHz channels here? Is it relevant? Are these used somewhere for ice type discrimination? If not then I suggest to delete this sentence.
Reply: It is used for MYI identification in some algorithms, e.g. Lomax et al., 1995. Since it is not exactly for ice type discrimination, we decide to delete the sentence.
*Lomax, A. S., Lubin, D. and Whritner, R. H. (1995) The potential for interpreting total and multiyear ice concentrations in SSM/I 85.5 GHz imagery, Remote Sensing of Environment, 54(1), pp. 13–26*

L690: "quantitatively" --> for me the inter-comparison of visually interpreted SAR images with the SITY maps is not a quantitative evaluation, also the computation of a Kappa coefficient does not help in this regard. I therefore again recommend to tone down your statement towards inter-comparison.
Reply: As mentioned in previous replies, although the SAR interpretation results are not standard or perfect validation datasets, it is sufficient to evaluate coarse resolution ice type products, which is also used for evaluating SAR ice classification results. We therefore would like to keep the statement here.

L693-695: Please make clear whether the ranges of years denoted together with the two products selected refer to the period for which these products are available or to the time period over which you have averaged the differences.
Reply: It refers to the respective available period of the products within the range of 1999-2019. To make clear about the meaning of years, we modify the sentence as below in the revision.
**"The bias of MYI extent between the SITY products (during the different periods) and NSIDC-SIA varies from -1.32×10^6 km^2 (OSISAF-SITY, during the SSM/I-only period, 2006–2009) to 0.49×10^6 km^2 (KNMI-A, 2009–2019)."**

L701: I find the numbered list of paragraphs that follows ok in terms of where which products seems to perform good or not so good. I don't think, however, that you should mention your attempts to explain the differences in the performance because i) your inter-comparison is based on a small set (5) of qualitatively interpreted SAR images

(which would be classified differently in an independent follow-on study), resulting in qualitative statements and because ii) you did not investigate / show evidence of misclassification of ice types due to the three mentioned main influencing factors. Also here you remain rather descriptive and do not go into depth. Hence most of the "explanation" given here is rather of hypothetical nature which proof requires further work.

Reply: Thank you for the comment.

We double-check these paragraphs. Most of the sentences are descriptions of their performances (good or not so good during certain period or under certain conditions), which the reviewer (and we) regarded as okay and are in fact based on the "qualitative" comparison with the SAR interpretation results (along with NSIDC-SIA).

When explaining these differences, we use words such as "are likely to" "could be resulted from" "could be caused" considering the uncertainties as mentioned above. This study aims to inter-compare the products and provide sensible explanations. We therefore do not quite understand the comment here and decide to keep as it is.

L723+ --> A lot of what follows here is a repetition of what is written in the discussion. Please condense and only provide 2-3 key points which you can also very well back up with the results you obtained. Make sure to highlight the nature of the results (qualitative / intercomparison) and give the outlook towards what would be needed to carry out a quantitative intercomparison or even evaluation. This is in my eyes much more important than, as you do at the end, to highlight which future satellites might be available. First we need to find procedures and well-evaluated data sets for a quantitative evaluation of the ice type products. Without these any novel ice type products from new satellites will be as useful (useless?) as the existing ones.

One obvious step would be to improve sea-ice motion estimates to improve the sea-ice age product - ideally with a smaller temporal resolution. And then: evaluate it adequately.

Another step would be to use well-evaluated ice type information from SAR images that underwent an unsupervised classification. Such information would make that part of the inter-comparison work involving SAR much more credible and the results could potentially even be interpreted quantitatively.

Reply: Thanks for the advice. We have condensed the texts as suggested and kept the sentences highlighting the results only. In addition, we add sentences regarding the well-evaluated dataset at the end of the manuscript. See L795-798 in the revised manuscript.

Typos / Editoral comments:
Abtract:
L17 / L18: Specify the QSCAT and the ASCAT periods as many readers will not be aware of these.
Reply: Done.

L17 / L18: "agree best" and "perform the best" --> Consider rephrasing such that you provide actual numbers of over- or underestimation.
Reply: the respective biases are added here, ", with smallest bias of -0.001×10^6 km^2 in FYI extent and -0.02×10^6 km^2 in MYI extent, respectively".

L20: "their performances" --> What is "their" referring to? Products? Sensors? Algorithms?
Reply: "their performances" is replaced with "the performances of the SITY products".

L38: I suggest to add the reference to Tschudi et al., 2020 to the one of Maslanik et al.
Reply: Done.

L42: I suggest to delete "such as the Atlantic Meridional Overturning" because this is a circulation in the ocean.
Reply: Done.

L51: "forecasting(Jung" --> "forecasting (Jung"
Reply: Done.

L64: "... other on input ..." --> better "... other in terms of input ..." or "... other regarding input ..." or "... other with respect to input ..."
Reply: Done.

L73: "It is found the ..." --> "The ..."
Reply: Done.

L77/78: "there is rarely ..." --> suggest to delete this part of the sentence since it is in contradiction to the previous one and begin this sentence with "The performances ..."
Reply: Done.

L90: "among the SITY" --> "among some existing SITY"
and further : "and give comprehensive evaluations on the ... " --> "and to assess the quality of the"
Reply: Done.

L113: "SSMR" --> "SMMR"
Reply: Done.

L119: "measures" --> "measured" because it is not operating anymore.
Reply: Done.

L123: "for inner ... from ..." --> "of 48.9 degree and 57.6 degree for the inner HH-polarized beam and the outer VV-polarized beam, respectively, from ..."
Reply: Done.

L124: "antennas, whose incidence angles varies between 25 degrees and 65 degrees" --> "antennas, each measuring backscatter over the incidence angle range of 25 degrees to 65 degrees."
Reply: Done.

For consistency with the previous subsection you could add the periods of operation for QuikSCAT and ASCAT.
Reply: Thanks. The periods of QSCAT and ASCAT are specified in the revised manuscript. See L128, ", which operates from July 1999 to November 2009", and L130, *", the data of which is available from May 2007 to present"*

L146: "microstructure" --> such as density, grain size and orientation ...? Consider adding.
Reply: We replace "and microstructure" with "density and grain size"

L153: "it" --> "these ice types"
Reply: Done.

L170/171: "and the climate record covered the period 1979-2020." --> "and was updated until 2021, covering the period 1979-2020."
Reply: Done.

L172: "is not be" --> "is not"
Reply: Done.

L180/181: "The swath data ..." Since this seems to be the final step of the pre-processing I would formulate it that way. For example: "As the last step of the pre-processing the corrected TB swath data are gridded into daily 25 km EASE2 grid TB maps." --> using which kind of gridding?
Reply: the sentence is modified to
*"As the last step of the pre-processing, the corrected Tbs swath data are gridded into daily 25 km EASE2 grid Tbs maps using an equal-weighted average (also called a circular top-hat averaging window) of data within a radius from the grid centre(Lavergne et al., 2022)."*

L194: "Water" --> "water"
Reply: Done.

L209: "is switched to AMSR-2" --> "was switched to AMSR2"
Reply: Done.

L210/211: Since the pre-processing of the C3S-SITY product is also done on the swath data I suggest to reformulate accordingly: "Unlike C3S-SITY, the core Bayesian ..."

Reply: Done.

L224: "KNMI-A respectively, available during" --> "KNMI-A, respectively, available for"
Reply: Done.

L268: "and active" --> delete, this is wrong. Such data are not used in that product.
Reply: Done.

L270: I would not call the work cited here as "assessments" and suggest to rephrase along the lines: "has been shown to provide very useful additional information about the changing Arctic sea ice cover because ..."
Reply: Done.

L274: "and ice motion data" --> "and the quality of the ice motion data."
Reply: Done.

L284: "Two" --> "two"
Reply: Done.

L288: "acquiring" --> "acquisition"
Reply: Done.

L297-299: "... Basin and limited ... . Note ... and analysis." --> I suggest to shorten this by writing: "... Basin excluding the area north of 87 degrees North with its observation data gap due to the satellites' inclinations (see Belmonte Rivas et al., 2018 and Fig. 2)."
Reply: Done. The sentence is replaced with
*"…Basin excluding the area north of 87 degrees North with its observation data gap due to the inclination of satellites (see Belmonte Rivas et al., 2018 and Figure 2)."*

L299: "as the integral extent of pixels" --> "as the sum of the area of all grid cells"
Reply: Done.

L303/304: "by the integral extent of pixels" --> "as the sum of the area of all grid cells"
Reply: Done.

L309-311: "Besides ... " --> "In order to compare the MYI extents at the same temporal resolution, SITY product MYI extents are averaged weekly to match the temporal resolution of the NSIDC SIA MYI extent."
Reply: Done.

L331: "to the UTM projection" --> "to the respective UTM projection" as this varies with longitude and latitude.
Reply: Done.

L364: "with bias" --> "with a bias"
Reply: Done.

L366 and L367: "bias" --> "biases"
Reply: Done.

L373: "scatteromter" --> "scatterometer"
Reply: Done.

L375: "product are" --> "products are"
Reply: Done.

L393: "difference" --> either "the .... difference" or "differences". Also note the wrong superscript "e" of the "10" in this line.
Reply: Done.

L398/399 and L400: "deviation of" --> Either it is a deviation of a quantity from another quantity ... which is not the case here, or it is the "deviation between" different products ... which is the case here. Please rephrase accordingly, because what I guess you want to tell is that, e.g., in mid-winter the difference of the MYI extent derived from the different SITY products is small - or in other words: The deviation of the MYI extent between the different SITY products is small.
Reply: Done.

L404: "For the" --> "Regarding the" or "With respect to the"
Reply: Done.

L404: What do you refer to by "the latter"? Is it "the other products"? Or are you refering to OSISAF-SITY?
Reply: "the latter" is used to refer to OSISAF-SITY, to avoid misunderstanding, we replace "the latter" with OSISAF-SITY.

L405: "mild ... trend" --> weather can be mild ... how about "weak ... trend" or "small ... trend"; furthermore: "rapid ... trend" --> rapid has something to do with speed and time, hasn't it? How about "large ... trend"?
Reply: Done.

L414: "while that ... increasing". --> either: "while the trend in the BCS regions is either zero or positive" or: "while the MYI extent in the BCS region remains constant or is increasing."
Reply: Done.

L418: "torwards south" --> either: "towards the South" or "south"

Reply: Done. It is replaced with "south".

L419/420: "In the BCS region, ... out of this region ... from the CAO region." --> consider rewriting this sentence. Certainly MYI drifts in the BCS region following the Beaufort Gyre. It enters the BCS from the North along the CAA and it eventually exits the BCS westward into ESL or back northward into CAO at the western borders of the BCS region - eventually entering the TDS. It also simply melts there (in summer).
Reply: Done. This sentence is revised as below:
*"In the BCS region, large quantities of MYI enters this region from the north along the CAA and eventually exits BCS westward into ESL or back northward into CAO at the western borders of the BCS region."*

L421: The statement of an increasing MYI extent in the BCS region should be supported by a notion of seasonality. Most likely MYI extent in this region is at a minimum in September and increases towards winter by MYI drifting into it from the North.
Reply: Thanks. The sentence is modified as below:
*"The nearly constant or increasing MYI extent in the BCS region could be caused by the fact that the MYI extent in BCS reaches a minimum in September and increases toward winter by MYI drifting into it from the north."*

L493: Again, please enclose the full date.
Reply: Done.

L498: "As shown in ..." --> I guess that fact that Table 3 shows up here was not planned?
Reply: Thanks. It was a wrong format here. We have revised it accordingly.

L505: "of SAR image" --> "of the SAR image"
Reply: Done.

L506: "to the case in" ... something missing here?
Reply: Thanks. We meant the case in Figure 8. It is modified accordingly.

L525: "as" --> "is"
Reply: Done.

L549: "nearly" --> "near"
Reply: Done.

L552: "northeast of the image" --> Did you mean "in the northeastern part of the image"?
Reply: Yes. It is modified it accordingly.

L559/560: What do you mean by "distinct"? Do you perhaps mean "different"?
Reply: Yes. "distinct" is replaced with "different" in the revision.

L561: "as a cross-validation dataset." --> "as an inter-comparison data set."
Reply: Done.

L564: "o of" --> "of"
Reply: Done.

L594: "...capability to separate and physical ..." --> Please check this sentence; its meaning is not clear.
Reply: Thanks, the sentence is modified as below:
*"The efficacy of input parameters depends on their separability of sea ice types and the relevant sea ice physical properties."*

L598: Over sea ice I would speak of melt-refreeze cycles only and hence remove the "wet-dry cycles".
Reply: We would like to keep the statement of "wet-dry cycles" since the change of snow wetness does not necessarily mean "melt-refreeze".

L602: "in C3S-2" can be deleted here.
Reply: Done.

L604: "disparate" --> I know, this comment comes somewhat late in my review but I recommend to check the meaning and usage of "disparate" versus the meaning of "different". I doubt that the microwave and/or scattering properties of MYI and FYI can be termed "disparate". They are different but they share similar (but different) basic scattering mechanisms. Unless you can be sure that the differences are so substantial that they exclude each other, i.e. absolutely no volume scatter in case of FYI or absolutely no surface scatter for MYI, or the like, I recommend to always rather speak of "differences" - here and everywhere (...) else in the paper.
Reply: The word "disparate" is replaced with "different" or throughout the manuscript.

L623: Either "on an a priori training dataset" or "on a priori training datasets"
Reply: Done. It is revised to "on a priori training datasets".

L625: "dataset" --> "datasets"
Reply: Done.

L639: "variabilities in the" --> I guess "variabilities as in the" is better.
Reply: Done.

L642: "fails to identify narrow MYI tongue in peripheral seas" --> "fails to identify features such as a narrow MYI tongue often observed in the Arctic peripheral seas ..."
Reply: Done.

L662: "aims to reassign the erroneously classified FYI" --> Not clear, better: "aims to re-assign the ice type MYI to grid cells where MYI was erroneously classified as FYI"
Reply: Done.

L673: "the five series SITY products" --> ?? What series?
Reply: "five series" is deleted to avoid misunderstanding.

L686: "... especially the fraction of MYI. The change of the SITY ..."
Reply: Done.

L687: "... inter-comparisons and analyses of SITY products ..."
Reply: Done.

L689: Please state that the NSIDC-SIA product is a weekly one and that you averaged (?) all SITY products to the same temporal resolution before the comparison.
Reply: We add "daily" and "weekly" to describe the SITY products and NSIDC-SIA product, respectively.

L719 "disparate" = completely different? Really? See my previous comment about usage of "disparate".
Reply: It is replaced with "different" as mentioned above.

Table 1:
"SSMI/I" --> "SSM/I"; "AMSR-2" --> "AMSR2"; "ASMR-E" --> "AMSR-E"
In the text you describe OSCAT but I cannot see it used in any of the products listed here. Consider removing it in the text?
You state in the text that all SITY products provide daily estimates. You can therefore delete the column denoted "Frequency".
I suggest to change "grid size" into "grid resolution". Is the type of all grids the same (all EASE or all polarstereographic)? If not it might make sense to add a column where this is specified.
Reply: Thanks. The table is modified as suggested.

Table 2 footnote: "to verify ... open water" --> better "to assess the correct discrimination of sea ice from open water."
You might want to explain also what "ice motion confining" means.
Reply: The footnote is modified to "Filters based on gradient ratio and brightness temperatures are used to filter out the open water pixels".
Notes are added to explain the "ice motion confining and spatial filtering".

Fig. 3:
- Please for all scenes add a date.
- Do all scenes have the same spatial scale? If not please provide a scale along with every scene, if yes provide it once.

- I suggest to include in (b) that the bright features may also be due to openings in the ice cover under high wind speed conditions.
- I suggest to add in (f) that these are MYI floes in a matrix of younger, presumably FYI.
- Whether (c) indeed shows brash ice between ice floes depends a lot on the location and the season (which are unknown?).
Reply: The Figure and notes are modified as suggested.

Fig. 4:
- I suggest to delete the dashed line with the daily MYI extent. It does not add value to the figure now that you have the weekly average - rather it adds noise.
- "the shaded area represents" --> "the shaded area in the same color as the respective solid line represents"
Reply: The figure and notes are modified as suggested.